mathematical modelling

Kala-azar, post kala-azar dermal leishmaniasis, asymptomatic transmission, parameter estimation, vector-borne diseases

**Author for correspondence:**
Jan Rychtář
e-mail: rychtarj@vcu.edu

# Mathematical modelling of the use of insecticide-treated nets for elimination of visceral leishmaniasis in Bihar, India

Anna K. Fortunato[1], Casey P. Glasser[2], Joy A. Watson[3], Yongjin Lu[3], Jan Rychtář[4] and Dewey Taylor[4]

[1]Department of Mathematics, University of Richmond, Richmond, VA 23173, USA
[2]Department of Mathematics, Virginia Tech, Blacksburg, VA 24061-1026, USA
[3]Department of Mathematics and Economics, Virginia State University, Petersburg, VA 23806, USA
[4]Department of Mathematics and Applied Mathematics, Virginia Commonwealth University, Richmond, VA 23284-2014, USA

AKF, 0000-0002-9563-6049; CPG, 0000-0002-5848-8382;
JAW, 0000-0003-1142-7977; YL, 0000-0001-8520-1212;
JR, 0000-0001-6600-2939; DT, 0000-0003-3012-8104

Visceral leishmaniasis (VL) is a deadly neglected tropical disease caused by a parasite *Leishmania donovani* and spread by female sand flies *Phlebotomus argentipes*. There is conflicting evidence regarding the role of insecticide-treated nets (ITNs) on the prevention of VL. Numerous studies demonstrated the effectiveness of ITNs. However, KalaNet, a large trial in Nepal and India did not support those findings. The purpose of this paper is to gain insight into the situation by mathematical modelling. We expand a mathematical model of VL transmission based on the KalaNet trial and incorporate the use of ITNs explicitly into the model. One of the major contributions of this work is that we calibrate the model based on the available epidemiological data, generally independent of the KalaNet trial. We validate the model on data collected during the KalaNet trial. We conclude that in order to eliminate VL, the ITN usage would have to stay above 96%. This is higher than the 91% ITNs use at the end of the trial which may explain why the trial did not show a positive effect from ITNs. At the same time, our model indicates that asymptomatic individuals play a crucial role in VL transmission.

## 1. Introduction

### 1.1. Epidemiology of visceral leishmaniasis

Leishmaniasis is a vector-borne disease caused by protozoan parasites of genus *Leishmania* and transmitted by phlebotomine

sand flies [1]. Visceral leishmaniasis (VL) is the most serious form of the disease and can be fatal in 95% of cases if left untreated [2,3]. VL is responsible for around 500 000 infections and 51 000 deaths per year, worldwide, deeming it second only to malaria in numbers of fatalities [4]. Leishmaniasis is endemic and presents a global health problem in 98 countries [5]. Over 94% of new cases occur in India, Ethiopia, Kenya, Somalia, Sudan, South Sudan and Brazil [6]. The Indian subcontinent accounts for two-thirds of the total global cases, of which more than 50% occur in the state of Bihar, India [7] where VL is known as 'Kala-azar' (Hindi for 'black fever'). It has been targeted for elimination with the goal of reducing the incidence of VL to below 1/10 000 by the year 2020 [8]. The elimination efforts are working [9] although the targets were reformulated for 2030 [10]. WHO's goals for 2030 are (1) to validate 64 countries for VL elimination as a public health problem—defined as less than 1% case fatality rate due to primary visceral leishmaniasis [11]; and (2) to detect and treat 100% of post kala-azar dermal leishmaniasis (PKDL) cases (by VL post-treatment follow-up for 3 years) [4].

## 1.2. Cause of visceral leishmaniasis

In the Indian subcontinent, VL is caused by parasites belonging to *Leishmania donovani* complex [8]. It is transmitted from human to human by female sand fly *Phlebotomous argentipes*, without any known animal reservoirs [12]. The sand flies are active and feed during the night with the female activity peaking just before midnight [13]. They normally seek shelter in animal burrows, or other protected areas [14] and thrive in poor housing conditions [15,16]. They are generally weak flyers and usually fly close to the ground in short hops [17]. At the same time, there are indications that they are capable of longer and more sustained flight and may be more exophilic and exophagic than previously reported [18].

## 1.3. Signs and symptoms of visceral leishmaniasis

In humans, the parasite infects the reticuloendothelial system, causing persistent fever and anaemia and affecting several internal organs, usually the spleen, liver and bone marrow [6,15]. Because the symptoms persist, the individuals typically seek treatment, especially in Bihar where treatment is available [19]. However, the social and cultural stigma linked to VL results in a large percentage of individuals seeking treatment at private rather than public health facilities [20], which results in under-reporting true incidence and prevalence of the disease [2]. After recovery of acute illness, about 5–10% of patients develop a chronic cutaneous form called PKDL [12]. Moreover, a few PKDL patients have had no history of VL [21]. Since PKDL is not a life-threatening condition and the treatment used to be very burdening and unpleasant, many PKDL patients remained untreated [22]. Because of the anthroponotic nature of the transmission of *L. donovani* in the Indian subcontinent [12], the PKDL patients are considered reservoirs of infection, albeit there is a range of opinions on this [23]. At the same time, the role of asymptomatic individuals in transmission is still unclear [24]. Furthermore, HIV-VL co-infection could be a growing concern in Bihar [25]. HIV-VL patients often relapse and their treatment lasts longer [26]. Overall, HIV reduces the sustainability of a successful VL elimination programme [27].

## 1.4. Control of visceral leishmaniasis

The control of VL depends on chemotherapy treatment [12], vector control [28], bite prevention [29] and active case detection [30]. Human VL vaccines are not yet fully developed although several trials are underway [31,32] and their impact is already modelled [33]. Preventative measures in high-risk areas include: (i) avoiding sleeping in mud and thatched housing [1], (ii) environmental management [34], (iii) using insect repellent [35], (iv) indoor residual spraying (IRS) [7] and, most recently, (v) treated wall lining [36].

## 1.5. Insecticide-treated nets and the KalaNet trial

KalaNet, a cluster randomized controlled trial in Nepal and India evaluated the efficacy of long-lasting insecticide-treated nets (ITNs) in the prevention of VL [37]. Before the trial, ITNs were used as a control measure [38–40]. It was known that human behaviour such as inconsistent use due to hot weather or inadequate education diminishes the effectiveness of ITNs [40,41]. Also, the ITN ownership varied significantly with wealth; almost all of the wealthiest households owned an ITN while many of the

poorest did not [42]. At the same time, even in relatively poor areas, the use of ITNs was already demonstrated to be very cost-effective [43,44].

A cluster-wide distribution of ITNs during the KalaNet trial reduced the vector density [45] but did not reduce the risk of *L. donovani* infection or clinical VL [46]. As a consequence, since 2010, ITNs are not part of the standard government VL control programme in India.

Other trials in Bangladesh showed that ITNs may reduce the VL incidence rate [36,47]. The use of ITNs was also recommended for PKDL and VL-HIV patients [9]. This apparent contradiction raises the question about the role that ITNs may play in controlling VL [46]. The purpose of this paper is to gain insight into the situation by mathematical modelling.

## 1.6. Mathematical models of visceral leishmaniasis

There are many mathematical models of VL dynamics, see for example [48–51] for recent reviews. A model of VL transmission at the district-level of Bihar to estimate the basic reproduction number was created in [2]. Different vector control measures for VL elimination were considered in [52]. A multi-state Markov model of VL was developed in [53]; an individual-based, stochastic, compartmental model of a temperature-driven sand fly population was developed in [54], which also simulated the effects of the use of drugs administered to cattle on the vector control. Chapman *et al.* [55] developed methods to analyse longitudinal spatial incidence data on VL and PKDL. A set of three age-structured model variants based on [56], each with individuals from a different disease stage being the main contributors to transmission: asymptomatic individuals, previously immune individuals in whom infection has reactivated, and individuals with PKDL, was created in [57]. The cost-effectiveness of different drug treatments was studied in [58,59].

A comprehensive model of VL for the Indian subcontinent to fit data collected from the KalaNet trial was developed in [56]. Their model extends the Susceptible-Infected-Recovered structure for the human population by segmenting it into five distinct stages according to an individual's infection status determined by the results of three diagnostic markers: (i) a polymerase chain reaction (PCR) test, the earliest infection marker able to pick up the presence of antigens [60], (ii) a direct agglutination test (DAT) [39] which measures the antibody response, and (iii) the leishmanin skin test (LST), also called Montenegro test [61] which detects the cellular immunity [62]. The model incorporates the role and significance of asymptomatic cases on VL transmission, which remains uncertain to this day [63]. The model also includes two lines of treatment of symptomatic VL cases, a possible treatment failure, relapse into PKDL and PKDL treatment. Most of the model parameters were found by fitting to data from the KalaNet trial using maximum-likelihood optimization method.

## 1.7. Game theory and disease prevention

From the behavioural perspective, disease prevention, such as ITN usage, produces public goods (such as herd protection and potentially the elimination—achieving zero cases—of the disease) that are non-rivalrous and non-exclusive [64]. When a sufficient proportion of the population is protected, then the slightest cost associated with using protection will outweigh the risk from infection [65]. Since the individuals often act in a way that maximizes their self-interests rather than the interests of the entire group [66,67], disease prevention is prone to free-riding. The 'free-riders', incorporated in our model as individuals that do not use ITNs, avoid the costs of prevention while they benefit from the preventive actions of others.

This social dilemma is captured by the game theory framework [65]. The framework has now been applied to help model the prevention of many diseases such as African trypanosomiases [68], chikungunya [69], cholera [70], dengue [71], Ebola [72], hepatitis B [73], hepatitis C [74], meningitis [75], monkeypox [76], polio [77], toxoplasmosis [78] and many others, see for example [67,79] for recent reviews. It has already been demonstrated theoretically [65,80] as well as empirically [66] that individuals behave rationally and that the high cost of vaccination or ITNs [81] is often the reason why the protection is not adopted.

Residents in VL endemic areas seem to be reasonably aware of the role of ITNs in the prevention of VL and other vector-borne diseases [44]. We will assume that all individuals are provided with the same information such as VL incidence rates, treatment costs and ITN coverage, and use the information in the same and rational way to assess costs and risks [65]. In our setting, the game theoretical framework assumes that people will sleep under ITNs not to protect others, but to protect themselves. Specifically, people will not start sleeping under ITNs once they learn they have VL, but they may start sleeping under ITNs once they learn that someone in their community has VL or PKDL.

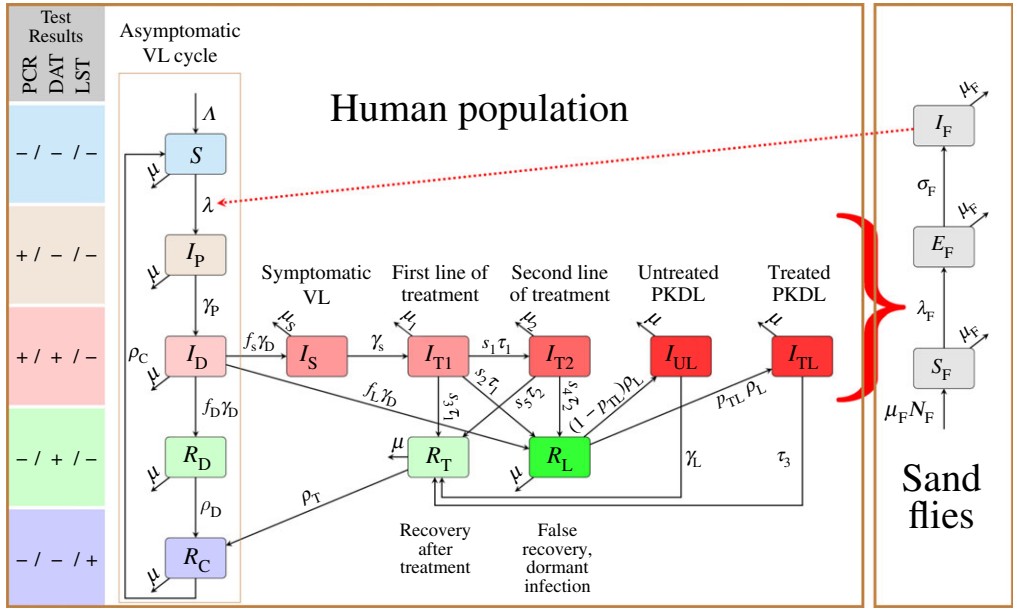

**Figure 1.** Scheme of the ODE model for VL transmission; based on [56]. Majority of the human population is in the asymptomatic VL cycle: individuals are born as susceptible ($S$) and if bitten by an infected sand fly ($I_F$)—signified by the red dotted arrow—they progress through early asymptomatic ($I_P$), late asymptomatic ($I_D$), recovering asymptomatic ($R_D$) and recovered ($R_C$). Recovered individuals lose immunity and eventually become susceptible again. Sand flies are born susceptible ($S_F$). If they bite an individual in any of the red or brown compartments—signified by the red curly brace—they become exposed ($E_F$) and eventually infectious ($I_F$). All human compartments are colour-coded based on their PCR, DAT and LST tests as shown on the left; informally, the intensity of the colour corresponds to the intensity of the infection which increases from left to right. The solid black arrows represent transitions between compartments.

## 1.8. Main objectives

In this paper, we will study an individual's choice to use ITNs to protect themselves against sand fly bites. We are interested to see why the KalaNet trial found that the ITNs were not effective in reducing VL incidence. We would also like to see if ITNs, possibly together with other control measures, could assist with VL elimination.

This paper is organized as follows. We will adapt a compartmental ODE model of VL originally introduced in [56] and extend it by adding a compartment for untreated PKDL cases and by a parameter describing the ITN use. We then derive explicit formulae for the disease-free and endemic equilibria of the ODE system. We calibrate the model based on data found in literature. We then use maximum-likelihood method to estimate the probability of VL transmission from asymptomatic cases to sand flies. We validate our model by demonstrating it fits the KalaNet trial data. We show that asymptomatic individuals play a crucial role in VL transmission and without the asymptomatic transmissions, VL would be eliminated. We derive conditions for the VL elimination (no cases of VL), perform the game theory analysis and compare the levels of ITN use needed for the elimination with the Nash equilibria levels. We show that, even if ITNs are not provided for free, it is in individuals' self-interests to use ITNs more than the current coverage. Moreover, if all individuals used ITNs optimally, VL would be very close to being eliminated; in fact, the disease would become eliminated as a public health problem. We include detailed mathematical analysis as well as the Policy-Relevant Items for Reporting Models in Epidemiology of Neglected Tropical Diseases (PRIME-NTD) Summary Table [82] as table 7 in appendix A.

# 2. Mathematical model

## 2.1. Description of model and disease dynamics

We slightly modify the compartmental model of VL dynamics developed by Stauch *et al*. [56]. The model is shown in figure 1.

Humans are born susceptible ($S$) at rate $\Lambda$. They test negative in all PCR, DAT and LST tests. At some point, they may be bitten and infected by an infectious sand fly. For simplicity, we assume that ITNs offer a perfect protection and thus the force of infection is

$$\lambda = (1 - p)\beta i_F I_F, \tag{2.1}$$

where $p$ is the percentage of the population that uses ITNs, $\beta^{-1}$ is the duration of the sand fly's feeding cycle, $i_F$ is the probability a human gets infected by a bite of an infectious sand fly, and $I_F$ is the number of infected sand flies.

The infection starts by an early asymptomatic stage ($I_P$). Individuals in $I_P$ are potentially infectious to sand flies, but do not exhibit symptoms. They are PCR-positive, DAT-negative and LST-negative. The stage lasts for a period $\gamma_P^{-1}$ and patients then progress to a late asymptomatic stage ($I_D$). They are still potentially infectious to sand flies, and PCR-positive, DAT-positive and LST-negative. The stage lasts for a period $\gamma_D^{-1}$. The vast majority (a fraction of $f_D$) recovers without ever showing any symptoms. At first, the recovering individuals are PCR-negative, DAT-positive and LST-negative. We will denote those $R_D$. Eventually, after time $\rho_D^{-1}$, the recovering individuals become DAT-negative and LST-positive. We will denote those by $R_C$. Both $R_D$ and $R_C$ individuals are immune to re-infection. However, the individuals in $R_C$ lose immunity at rate $\rho_C$ and become susceptible once again.

Only a small fraction, $f_S$, of late asymptomatic patients (in $I_D$) develop symptoms of an acute VL and progress to a compartment $I_S$. Since the symptoms are persistent, individuals typically seek diagnosis and treatment. The time from the onset of the symptoms to receiving a proper diagnosis and the beginning of the treatment is denoted $\gamma_S^{-1}$. The first-line treatment ($I_{T1}$) lasts for a time $\tau_1^{-1}$. However, with probability $s_1$, the treatment fails and individuals have to go through a second line of treatment ($I_{T2}$) which lasts for a time $\tau_2^{-1}$. The most common first-line drug used in Bihar, India is miltefosine and the most common second-line drug is amphotericin B [83].

The first and the second line of treatments are successful with probabilities $s_3$ and $s_5$, respectively. The individuals recover and move to the $R_T$ stage. The $R_T$ individuals are immune to re-infection, PCR-negative, DAT-positive and LST-negative. After time $\rho_T^{-1}$, they will move to $R_C$. With probability $s_2$ ($s_4$ for the second line of the treatment), the treated individuals appear to be recovering, but their infection becomes dormant. These patients, denoted $R_L$, are PCR-negative, DAT-positive, LST-negative. They are not infectious to the sand flies, but unless they die of natural causes, they will relapse and develop PKDL in time $\rho_L^{-1}$. A fraction $p_{TL}$ of PKDL cases seek a relatively long and expensive treatment that lasts a time $\tau_3^{-1}$. We will denote those patients by $I_{TL}$. The remaining PKDL cases, denoted by $I_{UL}$, remain untreated. They spontaneously recover at rate $\gamma_L$. The cases that recover move to $R_T$.

All individuals in $I_D$, $I_S$, $I_{T1}$, $I_{T2}$, $I_{TL}$ and $I_{UL}$ are PCR-positive, DAT-positive and LST-negative. All individuals in $I_X$ compartments for $X \in \{P, D, S, T1, T2, TL, UL\}$ are potentially infectious to sand flies. The probability that a fly gets the infection while biting such an individual is denoted by $i_X$.

All individuals die naturally at rate $\mu$. The symptomatic VL cases without treatment, ($I_S$), have a higher mortality and die at additional rate $\mu_K$, i.e. at the rate $\mu_S = \mu + \mu_K$. The VL treatments are aggressive, and the drug toxicity may cause permanent, irreversible damage (although a newer drug miltefosine is far less toxic than previous SSG [84]). A fraction $d_{T1}$ (or $d_{T2}$) of the treated patients die from the first (or second) line of treatment. The mortality rates in $I_{T1}$ and $I_{T2}$ are thus $\mu_1 = \mu + d_{T1}\tau_1$ and $\mu_2 = \mu + d_{T2}\tau_2$.

The sand flies follow the standard Susceptible-Exposed-Infected dynamics. All sand flies are born susceptible ($S_F$) at rate of $\mu_F N_F$. They become exposed ($E_F$) after feeding on an infectious human (in $I_P$, $I_D$, $I_S$, $I_{T1}$, $I_{T2}$, $I_{TL}$, $I_{UL}$) at rate

$$\lambda_F = \beta(i_P I_P + i_D I_D + i_S I_S + i_{T1} I_{T1} + i_{T2} I_{T2} + i_{TL} I_{TL} + i_{UL} I_{UL}), \tag{2.2}$$

where $\beta$ is the bite rate and $i_X$ is the probability that an individual in stage $I_X$ infects a sand fly. Note that, unlike for the force of infection $\lambda$ in (2.1), we do not consider the factor $(1 - p)$ because we assume that only individuals who do not use any protection can get infected (and become infectious to sand flies).

The duration of the latent stage is denoted by $\sigma_F^{-1}$. After that period, the sand flies become infectious ($I_F$). Since the mortality rate of sand flies, $\mu_F$, is relatively high, we do not consider any recovered stage.

## 2.2. Summary of the notation

The scheme of the VL dynamics is shown in figure 1.

**Table 1.** Notation—compartments.

| notation | meaning | PCR | DAT | LST |
|---|---|---|---|---|
| $N$ | total number of humans | | | |
| $S$ | susceptible humans | − | − | − |
| $I_P$ | early asymptomatic cases | + | − | − |
| $I_D$ | late asymptomatic cases | + | + | − |
| $I_S$ | symptomatic Kala-azar cases | + | + | − |
| $I_{T1}$ | patients in the first line of treatment | + | + | − |
| $I_{T2}$ | patients in the second line of treatment | + | + | − |
| $I_{TL}$ | treated patients with PKDL | + | + | − |
| $I_{UL}$ | untreated cases with PKDL | + | + | − |
| $R_D$ | recovering asymptomatic cases | − | + | − |
| $R_T$ | recovering patients after treatment | − | + | − |
| $R_C$ | recovered cases | − | − | + |
| $R_L$ | cases with dormant infection | − | + | − |
| $N_F$ | total number of sand flies ($N_F = n_F \frac{\Lambda}{\mu}$) | | | |
| $S_F$ | susceptible sand flies | | | |
| $E_F$ | exposed sand flies | | | |
| $I_F$ | infected sand flies | | | |

**Table 2.** Parameter values—sand flies. Times are in days.

| symbol | description | value | range | reference |
|---|---|---|---|---|
| $n_F$ | number of sand flies per human | 3 | [2.1, 3.5] | ([7], electronic supplementary material, table S15) |
| $\beta^{-1}$ | duration of feeding cycle | 4 | [2, 8] | [85,86] |
| $\sigma_F^{-1}$ | sojourn time in latent stage $E_F$ | 5 | [3, 7] | [87] |
| $\mu_F^{-1}$ | expected lifespan of sand flies | 14 | [6, 31] | [88] |
| $l_F$ | probability that a bite by an infected sand fly infects a susceptible human | 1 | [0, 1] | [56] |

The compartments are denoted $S$ (susceptible), $I$ (infected and potentially infectious), $R$ (recovering) and $E$ (exposed). Subscript F denotes sand flies. For the human compartments, we use the subscript P for PCR-positive, D for DAT-positive, C for cellular immunity (LST-positive), S for symptomatic, T for treated, U for untreated and L for PKDL. The notation is summarized in table 1. Parameters are summarized in tables 2–5. The parameter values are estimated in §4 based on empirical evidence and data from the literature. Greek letters stand for the rates. Lower case roman letters stand for probabilities/proportions of the populations.

## 2.3. Differences between our model and the original model in [56]

Aside from slight differences in the notation, we made the following changes and additions to the original model from [56].

- (i) We explicitly added a parameter $p$ signifying the level of protection against sand fly's bites by using ITNs.
- (ii) We separated PKDL cases ($I_{HL}$ in [56]) into treated ($I_{TL}$) and untreated ($I_{UL}$) cases to better account for the different duration of the stages and infectivity of the cases.

**Table 3.** Parameter values—humans. Times are in months. Rates are *per capita* per month.

| symbol | description | value | range | reference |
|---|---|---|---|---|
| $\Lambda$ | recruitment rate | $\frac{0.0277}{12}$ | $\left[\frac{0.0105}{12}, \frac{0.0333}{12}\right]$ | [89] |
| $\mu^{-1}$ | life expectancy | $67.8 \cdot 12$ | $[50, 90] \cdot 12$ | [90] |
| $\mu_{K}^{-1}$ | life expectancy with untreated symptomatic VL | 30 | [5, 36] | [2] |
| $\gamma_{P}^{-1}$ | sojourn time in $I_P$ | 5 | [4, 6] | [91] |
| $\gamma_{D}^{-1}$ | sojourn time in $I_D$ | 1.13 | [0.5, 4] | [91] |
| $\gamma_{S}^{-1}$ | time between onset of symptoms to the start of treatment | 1 | [0.5, 2.5] | [92] |
| $\gamma_{L}^{-1}$ | mean duration of the stage $I_{UL}$ | 55.5 | $[32, 16 \cdot 12]$ | [22] |
| $\rho_{C}^{-1}$ | duration of LST-positivity in $R_C$ | 33 | [10, 38] | [62] |
| $\rho_{D}^{-1}$ | duration of DAT-positivity in $R_D$ | 6 | [4, 8] | [93] |
| $\rho_{T}^{-1}$ | duration of DAT-positivity in $R_T$ | 6 | | $\rho_{D}^{-1}$ |
| $p$ | fraction of the population using ITNs | 0.7 | [0,1] | [38] |
| $f_L$ | fraction of $I_D$ who become dormant | $5.5 \cdot 10^{-4}$ | $[10^{-4}, 10^{-3}]$ | [21,22] |
| $f_S$ | fraction of $I_D$ who develop symptomatic VL | 0.035 | [0.01, 0.15] | [22] |
| $f_D$ | fraction of $I_D$ who recover without showing symptoms | 0.96445 | | $1 - (f_S + f_L)$ |
| $i_P$ | probability that an individual in $I_P$ infects a feeding sand fly | 0.0111 | [0,1] | estimated |
| $i_D$ | probability that an individual in $I_D$ infects a feeding sand fly | 0.0481 | [0,1] | estimated |
| $i_S$ | probability that an individual in $I_S$ infects a feeding sand fly | 0.1 | [0,1] | [94] |
| $i_{T1}$ | probability that an individual in $I_{T1}$ infects a feeding sand fly | 0 | [0,1] | [24] |
| $i_{T2}$ | probability that an individual in $I_{T2}$ infects a feeding sand fly | 0 | [0,1] | [24] |
| $i_{TL}$ | probability that an individual in $I_{TL}$ infects a feeding sand fly | 0 | [0,1] | [24] |
| $i_{UL}$ | probability that an individual in $I_{UL}$ infects a feeding sand fly | 0.1 | [0,1] | [94,95] |

**Table 4.** Parameter values—treatment. Times are in months. Costs are in USD (2010). 2010 Exchange rate: 1 USD = 40 INR.

| symbol | description | value | range | reference |
|---|---|---|---|---|
| $\tau_{1}^{-1}$ | duration of the first-line VL treatment | 1 | [0.75, 1] | [96] |
| $\tau_{2}^{-1}$ | duration of the second-line VL treatment | 1 | [0.75, 1] | [96] |
| $\tau_{3}^{-1}$ | duration of the PKDL treatment | 6 | [2.47, 7] | [21,97,98] |
| $s_1$ | probability of not responding to the first-line treatment | 0.06 | [0.01, 0.15] | [96] |
| $s_2$ | fraction of $I_{T1}$ cases that appear to be recovering but became dormant | 0.063 | [0.02, 0.1] | [56] |
| $s_3$ | probability that the first line of treatment is successful | 0.877 | | $1 - (s_1 + s_2)$ |
| $s_4$ | fraction of $I_{T2}$ cases that appear to be recovering but became dormant | 0.063 | [0.02, 0.1] | $s_2$ |
| $s_5$ | probability that the second line of treatment is successful | 0.937 | | $1 - s_4$ |
| $d_{T1}$ | fraction of patients dying due to the first line of treatment | 0.04 | [0.02, 0.13] | [99] |
| $d_{T2}$ | fraction of patients dying due to the second line of treatment | 0.04 | [0.02, 0.13] | [99] |
| $\rho_{L}^{-1}$ | duration until relapse to PKDL | 21 | [16, 26] | [21] |
| $p_{TL}$ | fraction of PKDL patients that seek treatment | 0.5 | [0, 1] | [22] |
| $C_{I_S}$ | cost of being in stage $I_S$ | 19 | [11, 26] | [100] |
| $C_{I_{T1}}$ | cost of first-line treatment | 146 | [110, 170] | [83,100] |
| $C_{I_{T2}}$ | cost of second-line treatment | 146 | [110, 170] | [83,100] |
| $C_{I_{TL}}$ | cost of PKDL treatment | 349 | [290, 410] | [101] |
| $C_{R_T}$ | cost of being in $R_T$ after treatment | 57 | [30, 100] | [100] |
| $C_{R_L}$ | cost of being in $R_L$ after treatment | 57 | | $C_{R_T}$ |
| $C_{ITN}$ | cost of ITN | 3.62 | [1.75, 5.50] | [36] |

**Table 5.** Auxiliary notation.

| symbol | description | equation |
|---|---|---|
| $\lambda_F$ | force of infection (humans infecting vectors) | (2.2) |
| $\lambda$ | force of infection (vectors infecting humans) | (2.1) |
| $\mu_S^{-1}$ | life expectancy in $I_S$ | $(\mu + \mu_K)^{-1}$ |
| $\mu_1^{-1}$ | life expectancy in $I_{T1}$ | $(\mu + d_{T1}\tau_1)^{-1}$ |
| $\mu_2^{-1}$ | life expectancy in $I_{T2}$ | $(\mu + d_{T2}\tau_2)^{-1}$ |
| $C_{VL}$ | expected cost of getting sick | (4.4) |
| $T_{Comp}$ | expected time an individual spends in a compartment Comp $\in \{I_P, \ldots, R_C\}$ given it started in compartment $I_P$ | (A 33)–(A 43) |
| $T_{Cycle}$ | expected time it takes an individual to become susceptible again (given it started in $I_P$ and conditional on surviving) | (A 47) |
| $T_I$ | average time an individual spends as infectious to sand fly (weighted by the infectivity) | (3.3) |

(iii) We assume that the human population size is constant with a birth rate $\Lambda$ (as opposed to $\alpha_H N_H$ considered in [56]). This change makes the ODE system less sensitive to changes in the birth rate and death rate. In the original model of [56], even a small change in $\alpha_H$, $\mu$ or $\mu_K$ would destabilize the system and result in either exponential growth or exponential extinction of the population. In our model, a (reasonable) change in $\Lambda$, $\mu$ or $\mu_K$ will cause the system to converge to an equilibrium with a potentially different population size, but the population will not go extinct nor grow above any bound. We can also solve for the equilibria of the dynamics.

(iv) We keep the exponential growth of sand flies, denoted $\alpha_F N_F$ in [56], but we stipulate that $\alpha_F = \mu_F$ in order to keep the population size constant.

(v) We consider the death rate of treated individuals to be $\mu_1 = \mu + d_{T1}\tau_1$ and $\mu_2 = \mu + d_{T2}\tau_2$, instead of $\mu_1 = \mu_2 = \mu + \mu_K + f_T\tau_1$. Specifically, we do not include the death rate of untreated individuals ($\mu_K$) and we consider the rates different with different treatment.

## 2.4. Numerical implementation

We coded the model in Matlab (v. 2020a with optimization toolbox) and made the code available in the electronic supplementary material. Following the best practices highlighted in [102], we included several tests to ensure correctness of the code. Specifically, we checked that analytical and graphical solutions (which were coded independently) yield the same results. We closely tracked the code execution to make sure the code runs as expected and values are passed correctly from function to function. Throughout the work on this manuscript, the formal analysis and numerical code were developed side by side and checked against each other. Independent cross-checks on a graphical calculator were also performed.

## 3. Analysis

The detailed analysis is shown in appendix A. Here we present only the summary. There are two possible equilibria: (1) the disease-free equilibrium with all humans and sand flies susceptible in $S^0 = \Lambda/\mu$ and $S_F^0 = n_F(\Lambda/\mu)$, and (2) the endemic equilibrium given by

$$E^* = (S^*, I_P^*, I_D^*, R_D^*, I_S^*, I_{T1}^*, I_{T2}^*, R_L^*, I_{TL}^*, I_{UL}^*, R_T^*, R_C^*, S_F^*, E_F^*, I_F^*). \tag{3.1}$$

The explicit formulae are given in appendix A.3.

When the ITNs usage is $p$, the effective reproduction number, $\mathcal{R}_e(p)$, is the average number of new infections caused by a single-infected individual [103,104]. The formula for $\mathcal{R}_e(p)$ can be derived using the next-generation matrix method [105] but also directly as shown below.

Assume that there is a single-infected person in compartment $I_P$. As derived above, that person spends an expected time $T_{Comp}$ in each of the compartments Comp $\in \{I_P, I_D, I_S, I_{T1}, I_{T2}, I_{TL}, I_{UL}\}$; the

formulae for $T_{\text{Comp}}$ are given in (A 33)–(A 43). During the time $T_{\text{Comp}}$, the infectious individuals expose sand flies at rate $\beta i_X N_F = \beta i_X n_F(\Lambda/\mu)$ where $X \in \{\text{P, D, S, T1, T2, TL, UL}\}$. Each of the exposed sand flies becomes infectious with the probability $\sigma_F/(\mu_F + \sigma_F)$. Each sand fly stays infectious for the time $\mu_F^{-1}$ and during that time it infects humans at rate $(1 - p)\beta i_F N = (1 - p)\beta i_F(\Lambda/\mu)$. Putting it all together yields

$$\mathcal{R}_e(p) = (1 - p)\beta^2 i_F n_F \left(\frac{\Lambda}{\mu}\right)^2 \left(\frac{\sigma_F}{\mu_F + \sigma_F}\right)\left(\frac{1}{\mu_F}\right) T_I, \tag{3.2}$$

where

$$T_I = i_P T_{I_P} + i_D T_{I_D} + i_S T_{I_S} + i_{T1} T_{I_{T1}} + i_{T2} T_{I_{T2}} + i_{TL} T_{I_{TL}} + i_{UL} T_{I_{UL}}. \tag{3.3}$$

It follows from (A 59) that the endemic equilibrium exists only if $\mathcal{R}_e(p) > 1$. While we did not perform a formal stability analysis, the ODE system—although large—is quite standard and similar to one considered in [106]. Consequently, the disease-free equilibrium is globally asymptotically stable if $\mathcal{R}_e(p) \leq 1$, and unstable if $\mathcal{R}_e(p) > 1$. The endemic equilibrium is locally asymptotically stable if $\mathcal{R}_e(p) > 1$.

# 4. Model calibration

## 4.1. Sand fly parameters

The number of sand flies per human is $n_F = 3$ with range 2.1–3.5 ([7], electronic supplementary material, table S15). This is in line with the India KalaNet site: 938 sand flies (94.2% of which were *P. argentipes*) from 325 households [45]. We note that this is different to the estimated value $n_F = 5.27$ with the range of 3.45–9.90 used by Stauch *et al.* [56].

The duration of the feeding cycle is $\beta^{-1} = 4$ days as in [56] with the range 2–8 days [85]. Sand flies normally take one blood meal per oviposition cycle [86,107,108] and the cycle usually takes 4 days [85] although infectious sand flies seem to feed more often [109,110]. We note that other models such as [3,111,112] used $\beta^{-1} = 10$ days based on [86], which estimated $\beta^{-1}$ between 6 and 30 days based on experimental results of [107], which seemed to let the flies *P. ariasi* oviposit for six or more days. While the literature varies on the actual length, it all agrees on the fact that the flies bite only once per oviposition cycle.

The sojourn time in the latent stage $E_F$ is $\sigma_F^{-1} = 5$ days, with the range 3–7 days [87].

The life expectancy of sand flies is $\mu_F^{-1} = 14$ days with the range 6–31 days [88].

## 4.2. Human parameters

The birth rate in rural Bihar, India is 27.7 births per year per 1000 people [89]. For the simulation purposes, we will assume the range of $\Lambda$ to be [0.0105/12, 0.0333/12] per month.

The life expectancy in Bihar is about 67.8 years [90]. For simulation purposes, we will use the range 50–90 years.

If the symptomatic VL patients remain untreated, their life expectancy, $\mu_K^{-1}$ is between 2 and 3 years [2]. We note that we did not find any data confirming this estimate. In [56], the authors assumed $\mu_K^{-1} = 5$ months. In ([4], p. 14, table 6), it is reported that a significant number of patients die within a month from the diagnosis; yet it is not quite clear how long it took to be diagnosed. The report itself states that the mortality is associated with many factors, including late health care seeking. Consequently, while we did not find data supporting [2], we will use $\mu_K^{-1} = 30$ months with the range 5–36 months.

The sojourn time of early asymptomatic stage, $\gamma_P^{-1}$, is between four and six months [91]. This value is based on unpublished laboratory experiments on *L. donovani*-infected grivet monkeys (*Cercopithecus aethiops*) in which the seroconversion (DAT− to DAT+) takes place four to six months after infection [91]. The value $\gamma_P^{-1} = 5$ months differs significantly from 60 days used in [56]. However, since [56] derived their values by fitting data to their model, we opted to use the value directly measured in the experiments. Also, the models considered in [113] use 150–202 days. Finally, a multi-state Markov model of VL developed by Chapman *et al.* [53] predicts the duration to be 147 days (95% CI 130–166).

The sojourn time of the late asymptomatic stage, $\gamma_D^{-1}$, is about one month. In [91], the authors were able to determine the time from the seroconversion to the development of active VL in 28 incidents of VL during their study. While they do not provide specific times, they report that most patients (23, i.e. 82.0%)

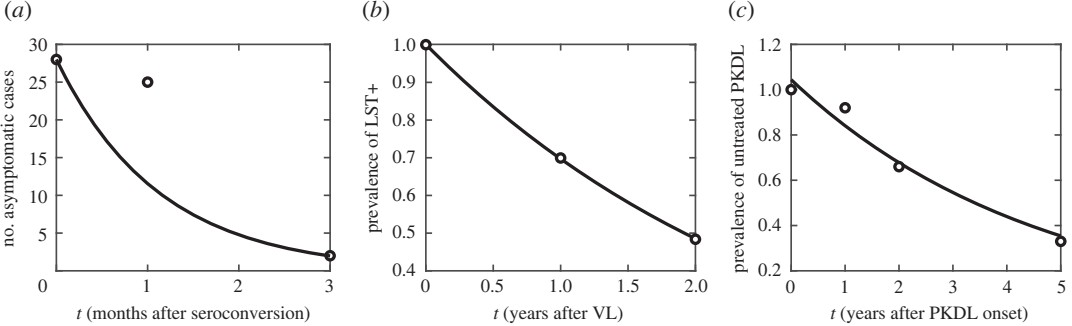

**Figure 2.** (a) Data from [91] fitted to $y = 28\,e^{-0.8845t}$ which yields $\rho_D^{-1} = 1.13$ months. (b) Data from [62] fitted to $y = 1.0008\,e^{-0.36176t}$ which yields an estimate for $\rho_C^{-1}$ as $0.36176^{-1}$ years, i.e. 33 months. (c) Data from [22] fitted to $y = 1.0441\,e^{-0.21625t}$ which yields $\gamma_L^{-1} = 0.21625^{-1}$ years, i.e. 55.5 months.

developed active VL in two to three months after seroconversion, while three (11.0%) took less than two months and the remaining two (7.0%) took more than three months. This means that within three months, all but two patients developed the symptoms, i.e. $\gamma_D = -\ln(2/28)/3$, i.e. $\gamma_D^{-1} \approx 1.13$ months (figure 2). We will consider the range to be 0.5–4 months.

The time between onset of symptoms to the start of treatment, $\gamma_S^{-1}$, is one month, with the range 0.5–2.5 months [92]. According to [92], there are 30 days (with the range 17–73 days) between the onset of symptoms and the diagnosis, and then 1 day (with the range 0.5–3 days) between the diagnosis and the start of treatment. Also, Sundar *et al.* [100] reports the median diagnosis time (symptoms to diagnosis) as five weeks with the range three to eight weeks. Finally, WHO [4] reports the median delay from onset of fever till diagnosis and then diagnosis to treatment is 59 days. In [56], $\gamma_S^{-1}$ is 1 day because, as a condition of the KalaNet trial, it took the patients 1 day to get treated once diagnosed (as in [92]). However, the proper meaning of $\gamma_S$ is the rate at which individuals leave the compartment $I_S$ (conditional on not dying). Since it takes a long time to get diagnosed [92,100], $\gamma_S^{-1}$ is more in the order of a month than in the order of a day.

We estimate that the immunity in $R_C$ lasts for $\rho_C^{-1} = 33$ months, ranging from 10 to 38 months. An unexpected loss of LST reactivity in 150 (30%) of 499 cases over 1 year and in 127 (51%) of 246 cases over 2 years was observed in [62]. This indicates an exponential decay at rate $\rho_C = -\ln((499 - 150)/499) = 0.3575$ per year. At this rate, there should be $120 = 246\,\exp(-0.3575 \times 2)$ of LST-positive cases after 2 years which is in close agreement with observed data from [62]. We also used Matlab to fit an exponential decay function to data from [62]. It resulted in $\rho_C^{-1} = 0.36176^{-1}$ years with the range from $0.321^{-1}$ to $0.4025^{-1}$ years, see figure 2. We note that [91] also observed a loss of LST positivity in the range of $l = 15$–40% within several six-month-long periods corresponding to $\rho_C^{-1} = -6/\ln(1 - l)$ between about 12 and 37 months. They attributed such a great variability to the variability of VL incidence. Data from [62] seem more reliable and more suited for the estimate as they track the same individuals for a longer period. Finally, our estimates are significantly more than 307 days that [56] used to fit the KalaNet trial data. However, as seen later in the validation, our model with these parameters still fits the KalaNet reasonably well, and since this parameter value fits data from [62], we will use $\rho_C^{-1} = 33$ months.

We will estimate $\rho_D^{-1}$ by six months based on data from [93]. Individuals in 26 villages were followed up in November–December 2006, 2007 and 2008. In 2006, there were 9034 DAT-negative cases. From that cohort, 375 cases were DAT-positive in 2007 and only 50 of those 375 were still DAT-positive in 2008. This means that the DAT-positivity is vanishing at the rate $-\ln(50/375) = 2.015$ per year, i.e. $\rho_D^{-1} = 5.96 \approx 6$ months. We did not find data for the range, but estimate it between four and eight months.

Since there is no significant difference between compartments $R_T$ and $R_D$, we assume, as in [56], that $\rho_T^{-1} = \rho_D^{-1}$.

We will assume that the baseline value of $p$, the fraction of population using ITNs, is 0.7 as in [38]. However, as $p$ is variable, we will be interested in what values of $p$ yield the elimination of VL.

We set $f_S$, the fraction of $I_D$ that develop symptoms to be 0.035 with the range from 0.01 to 0.15. There is a wide range of values found in the literature for this parameter. For example, 0.0142 in [113], 0.013 in [114] and 0.147 in [53]. Out of 375 DAT-positive individuals in Bihar and Nepal followed by Ostyn *et al.* [93], seven developed an acute case of VL, meaning that $f_S = 7/375 = 0.019$. Rahman *et al.* [21] surveyed $N = 22\,699$ individuals for KA and PKDL in the past 6 years and determined that 813 respondents had

KA and eight had PKDL with no history of KA. From these findings, we calculate $f_S$ as follows. If we disregard any symptomatic infections and assume that individuals only undergo asymptomatic cycles $S \rightarrow I_P \rightarrow I_D \rightarrow R_D \rightarrow R_C \rightarrow S$, we will get that $\gamma_P I_P = \rho_C R_C$ (the influx to and the outflow from a compartment is the same within and across all compartments). Since $\gamma_P I_P$ is the (monthly) incidence rate of late asymptomatic cases, we know that in $t$ years there would be $12 t \rho_C R_C$ late asymptomatic cases. While [21] does not provide the value of $R_C$, we can use $R_C/N = 0.35$ for the prevalence of LST-positive cases from [62]. This gives us the estimate that over the period of 6 years, the cohort of $N = 22\,699$ individuals should experience $72 \cdot (1/33) \cdot 0.35 \cdot 22\,699 \approx 17\,334$ late asymptomatic cases. Since 813 of those developed KA, we get $f_S = 813/17\,334 = 0.0469$. Similarly, Islam *et al.* [22] surveyed 24 814 individuals for KA and PKDL in the past 9 years. Of those, 1002 reported KA and 17 had PKDL with no history of KA. This gives an estimate

$$f_S = \frac{1002}{9 \cdot 12 \cdot 24\,814 \cdot 1/33 \cdot 0.35} = 0.0353. \tag{4.1}$$

Because [21,22] used a much larger sample size than [93] also notes that incidences were larger in India than in Nepal, we will adopt $f_S = 0.035$ and consider a range from 0.01 to 0.15.

We estimate the fraction of asymptomatic patients who become dormant to be $f_L = 0.00055$ with the range [0.0001, 0.001]. This will be done in a similar fashion as the estimate for $f_S$ based on data from [21,22]. A total of $N = 22\,699$ individuals were surveyed by Rahman *et al.* [21] for KA and PKDL in the past 6 years and determined that 813 respondents had KA and eight had PKDL with no history of KA. This yields

$$f_L = \frac{8}{6 \cdot 12 \cdot 22\,699 \cdot \rho_C \cdot 0.35 - 813} = 0.000484. \tag{4.2}$$

Similarly, Islam *et al.* [22] surveyed 24 814 individuals for KA and PKDL in the past 9 years. Of those, 1002 reported KA and 17 had PKDL with no history of KA. This gives an estimate

$$f_L = \frac{17}{9 \cdot 12 \cdot 24\,814 \cdot \rho_C \cdot 0.35 - 1002} = 0.00062. \tag{4.3}$$

We note that authors of [56] provide an estimate $f_L = 0.0001$ based on [21] although it seems that they wanted to use $f_L = 8/22\,699 = 0.00035$.

## 4.3. Estimates for visceral leishmaniasis transmission probabilities

Unlike [56], we will assume that individuals receiving treatment cannot infect the sand flies, i.e. $i_{T1} = 0$, $i_{T2} = 0$ and $i_{TL} = 0$. The treated individuals are not in their regular environments but rather at the treatment facility. Consequently, they are probably better protected from sand fly bites. Moreover, even if they are bitten and infect the fly, they are too far from their home for this event to meaningfully contribute to VL transmission in their home/village. This assumption is in agreement with [24], which reported that after receiving treatment, the VL patients are probably less infectious to the sand flies resulting in less transmission to family members during their 18-month follow-up. Similar findings were presented in [94].

To estimate the remaining transmission probabilities from humans to sand flies, we use the recent findings from [94], see also [115]. In [94], the authors found that 42 (54.5%) or 60 (77.9%) of 77 patients with active visceral leishmaniasis (as assessed by microscopy or quantitative PCR, respectively) and 11 (42.3%) or 23 (88.4%) of 26 patients with active PKDL transmitted parasites to sand flies. The results for PKDL transmission are slightly higher than those observed in [95] where only 27 out of 47 PKDL patients transmitted the disease. The difference can probably be attributed to the facts that [94] used more sand flies per patient (30–35 females and 10–12 males versus 20–25 females and 5–10 males) and for a longer time (30 min versus 15 min) than [95]. While the number 88.4% is large and it may seem that the parasite transmission from untreated PKDL patient to the vector is almost certain, it does not automatically mean that every bite results in the transmission. Unfortunately, neither [94] nor [95] report exact numbers of infected sand flies per patient and categorize patients only to those that did not transmit the parasite at all or those that transmitted it to at least one fly. Based on ([94], electronic supplementary material, figure 1B), the average number of infected sand flies was around 1.5 while the average number of dissected blood-fed sand flies was around 15 per patient. This would give $i_S \approx 0.1$. For the lack of better data, we adopt $i_{UL} \approx 0.1$ as

well. We demonstrate in §5 that the exact values of $i_S$ and $i_{UL}$ have only a small effect on the overall results of the model.

The role of asymptomatic individuals in VL transmission was also studied in [94]. They found that none of 187 asymptomatic VL patients transmitted the parasite to a sand fly. However, based on ([94], table 2), the vast majority of their patients tested DAT-positive and PCR-negative, i.e. they are categorized as recovering asymptomatic ($R_D$) by Stauch et al. [56] and our model, and as such already assumed to not be infectious. Despite an unfortunate difference in terminology, there is thus no factual difference between results of [94] and the assumptions of our model. None of the patients studied in [94] were PCR-positive and DAT-negative (early asymptomatic, $I_P$). Only 11 patients were DAT-positive and PCR-positive (late asymptomatic, $I_D$). Given the results presented in §5, this sample size may still be too small to draw conclusions about the roles of these early and late asymptomatic individuals in VL transmission. Consequently, in §5 we estimated the values of $i_P$ and $i_D$ differently by fitting the model to KalaNet data.

## 4.4. Treatment parameters

We will assume that the first line of treatment fails with the probability $s_1 = 0.06$ [96] and that the range is [0.01, 0.12]. Miltefosine is a common and effective first-line treatment in Bihar, India, with a clinical efficacy of 94% [96]. Studies like [116] give $s_1 = 0.03$ and lower, while other studies such as [117] show that the treatment failures by common treatments can be as high as $s_1 = 0.12$.

The fraction of KA patients who appear to recover under KA first-line treatment but will develop PKDL (conditional on surviving treatment, $1 - d_{T1}$) is $s_2 = 0.063$ [117]. This is based on the presentation 'Cohort observational study to estimate the cumulative incidence of PKDL in VL patients treated with three regimens in Bihar (Presenter: Suman Rijal)' that reported on a cohort of 1622 KA patients treated between 2012 and 2015 and followed up in 2016 and 2017 to determine the occurrence of PKDL. The cumulative incidence of PKDL was 6.3%, with a PKDL rate of 4.8%, 5.7% and 9.2% depending on the treatment. Since Stauch et al. [56] reported $s_2 = 0.03$, we will use [0.02, 0.1] for the range.

We will assume the probability to die during the treatment (as a result of the treatment) to be about $d_{T1} = d_{T2} = 0.04$. This is based on ([99], table 6) which reported the death incidence of (presumably treated) KA cases as 45.75 per 1000 (compared with a regular mortality rate of 2.9 per 1000 in the state of Bihar for the age group 15–59). The mortality varies greatly by age group from about 0.02 (5–14 years) to about 0.13 (60+ years). We will thus assume the range to be [0.02, 0.13]. We note that [56] assumed $f_T = 0.05$. Overall, the numbers seem in line with other studies; Ahasan et al. [118] reported 27 deaths out of 553 patients (4.8%) treated with sodium antimony gluconate and, cumulatively, nine studies discussed in [118] report 51 deaths out of 1819 patients (2.8%).

The time from the apparent cure from KA until relapse to PKDL is assumed to be $\rho_L^{-1} = 21$ months, with the range 16–26 months [21]. This is in line with 19 months reported by Islam et al. [22].

The duration of the first line of treatment is $\tau_1^{-1} = 1$ month, with the range [0.6, 1] months [96]. The duration varies by the treatment type, but the most common first-line drug used in Bihar, India is miltefosine [83].

Similarly, the duration of the second line of treatment is $\tau_2^{-1} = 1$ month, with the range [0.6, 1] months [96]. The most common second-line drug used in Bihar, India is amphotericin B [83].

The duration of the PKDL treatment is $\tau_3^{-1} = 6$ months with range one to seven months. There are three different kinds of treatments: sodium stibogluconate (SSG), miltefosine (MF) and amphotericin B. The SSG treatment takes six months (six 30-days-long cycles [21]), the MF treatment takes approximately three to four months [97,98], and amphotericin B treatment takes three weeks [22]. While amphotericin B appears to be the fastest treatment, it does not guarantee that PKDL will be cured within that time period. In fact Islam et al. [22] followed 30 PKDL cases treated with amphotericin B, and while 27 reported an improvement in four months, only one case was completely cured. We do not have additional data but it seems that even in the case of this treatment, the duration of PKDL is six months.

The duration of untreated PKDL, $\gamma_L^{-1}$, will be estimated as follows. In [22], the authors followed 98 PKDL patients that never received treatment and provides estimated resolution rates: 8% within 1 year of onset, 34% within 2 years and 67% within 5 years. Fitting the exponential decay to these data by Matlab (see figure 2) yields $\gamma_L^{-1} = 55.5$ months with the range of 32 months to 16 years.

The fraction of PKDL patients that receive treatment is estimated as $p_{TL} = 0.5$. This estimate is based on [22], which reports that out of 185 PKDL cases, 98 did not seek the treatment.

## 4.5. The cost parameters

Once infected, all individuals have to go through $I_P$ and $I_D$. Those stages are asymptomatic and consequently with no associated costs. The costs appear only if and when an individual experiences symptoms (in $I_S$), is being treated ($I_{T1}$, $I_{T2}$ and $I_{TL}$) or is recovering after the treatment (in $R_T$ and $R_L$). The costs are denoted by $C_{I_S}$, $C_{I_{T1}}$, $C_{I_{T2}}$, $C_{I_{TL}}$, $C_{R_T}$ and $C_{R_L}$ and the actual values are estimated below.

The Indian government provides free care to VL patients; however, hoping for a quick cure of seemingly minor illness, patients prefer to access private providers, which contributes to high out-of-pocket expenditures [119]. Even though VL is often misdiagnosed in private hospitals, patients continue to access private care until they can no longer afford it [119].

The cost of being in compartment $I_S$ is estimated as $C_{I_S} = 19$ USD [100], and we will assume the range [11, 26]. The average monthly income in Bihar, India in 2010 was 38 USD per month [100]. It takes about a month to get a diagnosis [92]. During that time, the patient loses 2.14 weeks (about half a month) of work, i.e. the cost is about 19 USD.

The cost of getting the first line of treatment is $C_{I_{T1}} = 146$ USD [100], and we will assume the range [120, 170]. This includes 127 USD of direct medical costs for one month of treatment [100], and the indirect cost of lost work. For every month of illness, half a month of work is lost (19 USD loss) [100]. This is in line with 100 USD direct medical costs used by Hasker *et al.* [83].

The cost of getting the second line of treatment is $C_{I_{T2}} = C_{I_{T1}} = 146$ USD [100]. Both lines of treatment last approximately one month, so the cost of the individual second treatment will be about the same.

The cost of getting treatment for PKDL is assumed to be $C_{I_{TL}} = 349.00$ USD [101]. We will assume the range [290, 410]. The direct cost of SSG treatment for PKDL costs 179 USD [101]. The treatment lasts about six months and so the loss of productivity during this illness is 170 USD.

The cost of recovering from treatment without a dormant infection is $C_{R_T} = 57$ USD [100], and we will assume the range [60, 110]. During recovery, a patient can only work 3.21 weeks/month on average, instead of the normal 4.29, i.e. losing 25% of the wage per month [100]. Since the average monthly wage is 38 USD and $R_T$ lasts approximately $\rho_T^{-1} = 6$ months, the total loss is about $0.25 \cdot 38 \cdot 6 = 57$ USD.

The cost of recovering from treatment, but with a dormant infection is $C_{R_L} = C_{R_T} = 57$ USD [100]. We assume that even though individuals in this stage do have a dormant infection, the time and the cost needed to recover from treatment will be about the same.

The overall cost of VL infection is thus given by

$$C_{VL} = \mathbb{P}(I_P \to I_S)C_{I_S} + \mathbb{P}(I_P \to I_{T1})C_{I_{T1}} + \mathbb{P}(I_P \to I_{T2})C_{I_{T2}} + \mathbb{P}(I_P \to I_{TL})C_{I_{TL}}$$
$$+ \mathbb{P}(I_P \to R_T | I_{T1} \text{ or } I_{T2} \text{ or } I_{TL})C_{R_T} + \mathbb{P}(I_P \to R_L | I_{T1} \text{ or } I_{T2})C_{R_L}, \tag{4.4}$$

where the probabilities $\mathbb{P}(I_P \to C)$ (and $\mathbb{P}(I_P \to C|C')$) of getting to a compartment $C$ (through a compartment $C'$) when currently at a compartment $I_P$ are given by

$$\mathbb{P}(I_P \to I_S) = T_{I_D} f_S \gamma_D, \tag{4.5}$$
$$\mathbb{P}(I_P \to I_{T1}) = T_{I_S} \gamma_S, \tag{4.6}$$
$$\mathbb{P}(I_P \to I_{T2}) = T_{I_{T1}} s_1 \tau_1, \tag{4.7}$$
$$\mathbb{P}(I_P \to I_{TL}) = T_{R_L} p_{TL} \rho_L, \tag{4.8}$$
$$\mathbb{P}(I_P \to R_T | I_{T1} \text{ or } I_{T2} \text{ or } I_{TL}) = T_{I_{T1}} s_3 \tau_1 + T_{I_{T2}} s_5 \tau_2 + T_{I_{TL}} \tau_3 \tag{4.9}$$

and
$$\mathbb{P}(I_P \to R_L | I_{T1} \text{ or } I_{T2}) = T_{I_{T1}} s_2 \tau_1 + T_{I_{T2}} s_4 \tau_2. \tag{4.10}$$

The cost of ITN is $C_{ITN} = 3.62$ USD [36].

# 5. Results

## 5.1. The role of asymptomatic individuals in visceral leishmaniasis transmissions

The transmission probabilities $i_P$, $i_D$ for asymptomatic individuals, and $i_F$ for the transmission from sand flies to humans were estimated using the built-in Matlab function to fit the model to KalaNet trial data ([56], table 1) as described in appendix B.

The probability that a susceptible human becomes infected by an infected sand fly was estimated as $i_F \approx 1$ (figure 3). This high transmission probability is consistent with the fact that the parasite can modify the sand fly's feeding apparatus so that the fly feeds more persistently and releases more parasites [109,120].

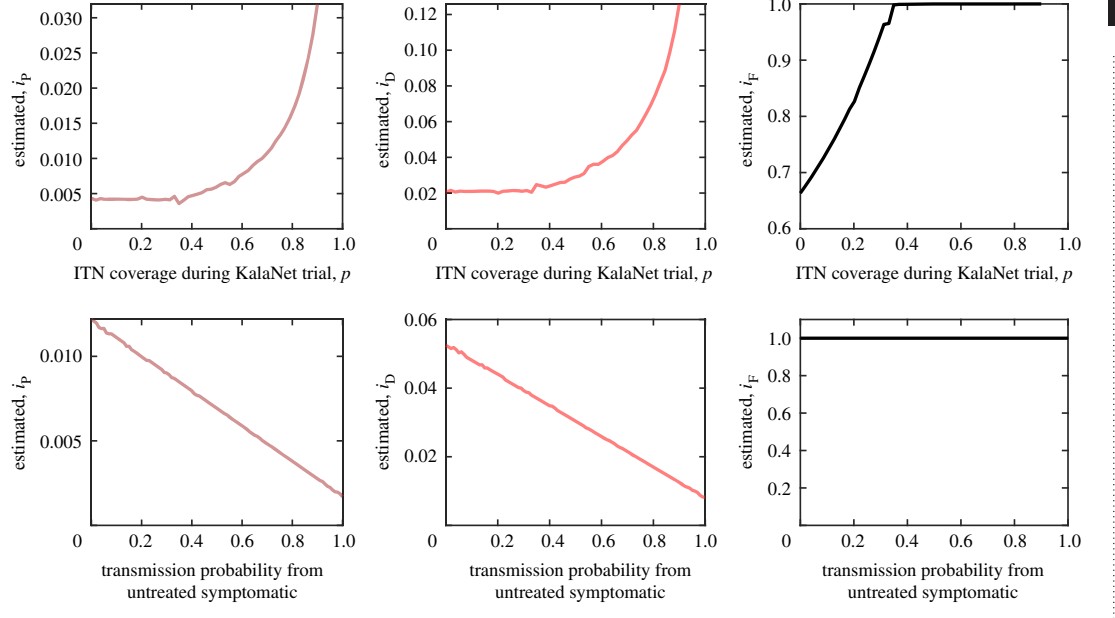

**Figure 3.** Maximum-likelihood estimates for $i_P$ (left), $i_D$ (centre) and $i_F$ (right) as a function of $p$, $i_S$ and $i_{UL}$. Top row: $i_S = i_{UL} = 0.1$ while the ITN coverage $p$ varies. The transmission probabilities increase (as the protection level increases, the disease must be more transmittable to have the same disease prevalence). Bottom row: $p = 0.7$ while $i_S = i_{UL}$ varies. As the probability of symptomatic transmission increases, the role of asymptomatic transmissions decreases while the role of sand flies remains constant.

The estimates for $i_P$ and $i_D$ depend on $p$, the ITN coverage during KalaNet trial as well as the transmission probabilities from untreated symptomatic individuals, $i_S$ and $i_{UL}$. However, figure 3 demonstrates that, for reasonable values of $p$, $i_S$ and $i_{UL}$, the estimates for $i_P$ and $i_{UL}$ are fairly stable.

For small $p \approx 0$, we get $i_D \approx 0.02$ and $i_P \approx 0.004$ which is consistent with [56]. However, for $p \approx 0.7$, we get $i_D \approx 0.05$ and $i_P \approx 0.01$. If $p$ grows even more, the values of $i_D$ and $i_P$ grow as well. Note that the estimates for $i_P$ and $i_D$ are indeed increasing in $p$—to achieve the same disease prevalence as measured in KalaNet trial when the population uses higher level of protection $p$, the disease must be more transmittable.

At the same time, as the probability of transmission from an untreated symptomatic individuals increases, both $i_P$ and $i_D$ decrease, i.e. the asymptomatic individuals become less important if the symptomatic individuals are more likely to transmit the disease. Nevertheless, even if symptomatic individuals transmit the disease with 100% probability, the role of late asymptomatic individuals is still not negligible.

When $i_S$ and $i_{UL}$ are around 0.1 as recently measured by Singh *et al.* [94], we get $i_P = 0.01$ and $i_D = 0.05$. This means that late asymptomatic individuals (PCR-positive, DAT-positive and LST-negative) are roughly 50% as important to VL transmission as untreated symptomatic VL and PKDL cases.

## 5.2. Model validation

The model is validated using KalaNet trial data ([56], table 1). Our model gives the following prevalences (the KalaNet data are in the parenthesis): 0.7599 (0.76) for $S + R_C$, 0.0979 (0.1) for $I_P$, 0.0221 (0.02) for $i_D$, 0.1173 (0.12) for $R_D$ and 0.0108 (0.005) for $I_F$.

To validate the model on another dataset, one would have to potentially update the parameter values to properly reflect the time and location of the experiment where the data came from. Then, we can use formulae for endemic equilibrium from appendix A.3 to obtain distribution of population across different model compartments.

We note that while not impossible, it is hard to make KalaNet trial data, our model and the model from [56] consistent without asymptomatic transmissions. Specifically, if the PCR-positive, DAT-positive and LST-negative asymptomatic individuals in compartment $I_D$ cannot infect the sand flies, the model can still predict prevalences of 0.0952 for $I_P$, 0.0215 for $I_D$, 0.114 for $R_D$ and 0.009 for $I_F$ which is in reasonable agreement with KalaNet data; however, all of this can be achieved only under

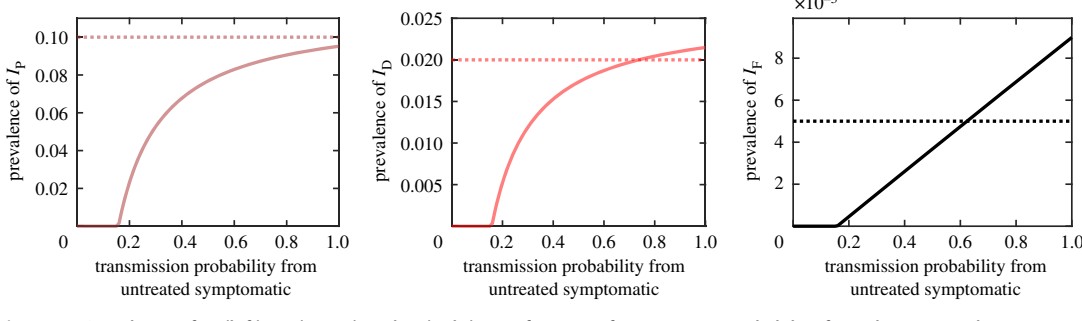

**Figure 4.** Prevalence of $I_P$ (left), $I_D$ (centre) and $I_F$ (right) as a function of transmission probability from the untreated symptomatic individuals if the asymptomatic individuals do not transmit parasites at all. The dotted lines corresponds to data from KalaNet trial.

the very unrealistic assumption that symptomatic individuals transmit the parasite 100% of the time. In fact, with much more realistic values of $i_S \approx i_{UL} \approx 0.1$ as estimated in §4 from [94], the population would be in disease-free equilibrium. Without asymptomatic transmissions, one would need $i_S > 0.18$ and $i_{UL} > 0.18$ for VL to become endemic. This is illustrated in figure 4.

## 5.3. Minimal insecticide-treated net coverage needed for visceral leishmaniasis elimination

To find the ITN usage level necessary to achieve the complete elimination of VL, we need to find the smallest $p_0 \in [0, 1]$ such that when $p \geq p_0$, then $\mathcal{R}_e(p) \leq 1$. It follows from (3.2) that $\mathcal{R}_e(p) = (1 - p)\mathcal{R}_e(0)$ where

$$\mathcal{R}_e(0) = \beta^2 i_F n_F \left(\frac{\Lambda}{\mu}\right)^2 \left(\frac{\sigma_F}{\mu_F + \sigma_F}\right)\left(\frac{1}{\mu_F}\right)T_I \tag{5.1}$$

is the effective reproduction number when nobody is using the protection. Consequently,

$$p_0 = \begin{cases} 1 - \dfrac{1}{\mathcal{R}_e(0)}, & \text{if } \mathcal{R}_e(0) > 1, \\ 0, & \text{otherwise.} \end{cases} \tag{5.2}$$

For the parameter values as specified in tables 2–4, $p_0 = 0.95963$, i.e. one needs just under 96% ITN coverage for a complete VL elimination.

## 5.4. Optimal voluntary use of insecticide-treated nets

In this section, we will find the optimal proportion of the use of ITNs. We are looking for Nash equilibrium, i.e. a proportion that, when adopted by the population, no individual has an incentive to deviate from their choice. To find $p_{NE}$, a Nash equilibrium value of $p$, we have to solve

$$C_{ITN} = \frac{\beta i_F I_F^*}{\mu + \beta i_F I_F^*}C_{VL}, \tag{5.3}$$

where $C_{ITN}$ is the cost protection, $\beta i_F I_F^*/(\mu + \beta i_F I_F^*)$ is the probability of getting infected by an infected sand fly, and $C_{VL}$ is the expected cost one pays after such an event. Note that (5.3) is an equation for $p$ because $I_F^*$ is a function of $p$. Figure 5 illustrates a graphical solution of (5.3). After algebraic manipulations shown in appendix C, we get

$$p_{NE} = 1 - \frac{1}{\mathcal{R}_e(0)[1 - (I_F^*/N_F^*)(\beta i_F T_{Cycle}(N_F^*/\mathcal{R}_e(0)) + (\mu_F/\sigma_F) + 1)]}, \tag{5.4}$$

where $I_F^*$ is given by (C5).

It follows that $p_{NE} \approx p_0$. For our parameter values, we have $p_0 = 0.95963$, $p_{NE} = 0.95956$ with $\mathcal{R}_e(p_{NE}) = 1.0018$. It means that the disease can be almost eliminated by the optimal voluntary use of ITNs alone (provided broken ITNs are replenished and people get perfect information about incidence rates, ITNs usage and various costs). In fact, with the voluntary use of ITNs, the disease would become eliminated as a public health problem.

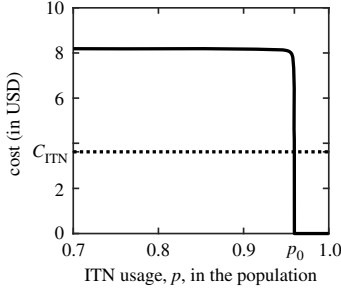

**Figure 5.** The expected cost of getting sick if not using ITNs when the population usage is $p$ (solid line) versus the cost of ITNs (dotted line). The optimal use (a Nash equilibrium) occurs when the lines intersect. VL is eliminated when the solid line reaches the $x$-axis. The parameter values are as specified in tables 2–4.

## 5.5. Sensitivity analysis

The sensitivity of the outcomes (ITN use for disease elimination, $p_0$, and optimal ITN use, $p_{NE}$) on different parameter values is displayed in table 6 and shown in figures 8 and 9. Since $p_{NE} \approx p_0$, only the sensitivity of $p_0$ is shown.

It follows that $p_0$ (and $p_{NE}$) are not overly sensitive to any parameter. The sensitivity index is at most 0.0846 (for $\mu$ or $\Lambda$) or $-0.0849$ (for $\beta^{-1}$). It is the second highest for $\mu_F^{-1}$ and closely followed by $n_F$ and $i_F$. Sensitivity to other parameters is between $-0.011$ and 0.02.

It follows that increasing the time between the bites and the reduction of the lifespan or the number of sand flies are the most promising control measure apart from using ITNs.

## 6. Discussion

In this section, we will mostly discuss the model limitations and possible extensions.

The model of sand fly dynamics could be improved in at least four possible ways. First, one could incorporate a seasonal dynamics by allowing for a variable birth rate. With this change, the analysis would become more involved as we could not focus on equilibria any more. Second, it is known that infectious sand flies tend to feed more often than non-infectious sand flies [109,110]. One may thus need to consider two different bite rates: one for susceptible sand flies (this will play a role in $\lambda_F$, i.e. humans infecting sand flies) and another one for sand flies infecting humans (this will play a role in $\lambda$, i.e. sand flies infecting humans).

Third, we did not consider any ITN-related mortality of the sand flies. This is in agreement with [122], although slight reduction of sand fly prevalence was observed by Picado *et al.* [45]. Fourth, we also do not consider increased mortality of infected sand flies [110,123]. To incorporate any of these last two improvements, one would have to consider $\mu_F = \mu_F(p)$ for an appropriate increasing function $\mu_F(p)$. Furthermore, the total mortality of the infected sand flies would be $\mu_F(p) + \mu_I(p)$ where $\mu_I(p)$ represents the additional mortality caused by the infection and the fact that infected flies bite more often, i.e. could suffer from an increased ITN-related mortality as well. To make these additions work, we would have to assume the birth rate to be $\Lambda_F$; otherwise, the sand fly population would not have a non-trivial finite equilibrium.

We also did not consider that ITNs offer only imperfect protection. First, smaller mesh size and insecticide treatment increase the net efficacy, but a small number of sand flies were still found inside ITNs [124]. Furthermore, as noted in [46], recent entomological findings in India indicate that *L. donovani* vectors are more exophilic and exophagic than previously reported [18,122]. If *P. argentipes* bite people outdoors (e.g. in the early evening when and where ITNs are not deployed), ITNs will have a limited impact on *L. donovani* transmission. To account for these, one would have to introduce a parameter $e$, the entomological efficacy of ITNs against the vector. The factor $(1 - p)$ in (2.1) for the force of infection in humans would then change to $(1 - pe) = (1 - p) + p(1 - e)$. Here $1 - p$ corresponds to an individual not using an ITN and $p(1 - e)$ corresponds to the fact that ITN was used but did not protect (because the fly got inside anyway, or the bite occurred outside). Further changes would have to be made in (2.2) in the calculation of the force of infection for the sand flies. We would have to properly account for two distinct scenarios depending whether the fly bites an infected individual

**Table 6.** The sensitivity index $SI_y$ of a variable $y$ on a parameter $x$ was calculated as $(x/y) \cdot (\partial y/\partial x)$ [121]. The numbers were rounded to the three decimal places. Parameters are as specified in tables 2–4. The sensitivity index $-0.5$ means that a 1% increase of a parameter value $x$ will result in the 0.5% decrease of the variable $y$.

| parameter | $SI_{p_0}$ | $SI_{(p_0 - p_{NE})}$ |
|---|---|---|
| $\Lambda$ | 0.0846 | −3.9272 |
| $\mu^{-1}$ | 0.0846 | −5.0223 |
| $\mu_F^{-1}$ | 0.0620 | −3.1936 |
| $i_F$ | 0.0425 | −1.9916 |
| $n_F$ | 0.0424 | −1.9688 |
| $i_D$ | 0.0201 | −0.8375 |
| $\gamma_P^{-1}$ | 0.0192 | −0.7946 |
| $\gamma_D^{-1}$ | 0.0188 | −0.8742 |
| $i_P$ | 0.0148 | −0.5655 |
| $f_S$ | 0.0031 | −1.7764 |
| $i_{UL}$ | 0.0024 | −0.1131 |
| $\gamma_{UL}^{-1}$ | 0.0020 | −0.0938 |
| $s_2$ | 0.0018 | −0.1683 |
| $i_S$ | 0.0012 | −0.0557 |
| $\gamma_S^{-1}$ | 0.0011 | 0.0216 |
| $f_L$ | 0.0005 | −0.0460 |
| $\mu_K^{-1}$ | 0.0001 | −0.0708 |
| $s_1$ | 0.0001 | −0.0692 |
| $s_4$ | 0.0001 | −0.0099 |
| $\rho_C^{-1}$ | 0 | 0.6465 |
| $\rho_D^{-1}$ | 0 | 0.1220 |
| $i_{T1}$ | 0 | 0.0000 |
| $i_{T2}$ | 0 | 0.0000 |
| $i_L$ | 0 | 0.0000 |
| $\tau_1^{-1}$ | 0 | 0.0014 |
| $\tau_2^{-1}$ | 0 | 0.0001 |
| $\tau_3^{-1}$ | 0 | 0.0003 |
| $d_{T2}$ | 0 | 0.0025 |
| $C_{ITN}$ | 0 | 1.8081 |
| $C_{I_{T1}}$ | 0 | −1.0668 |
| $d_{T1}$ | −0.0001 | 0.0455 |
| $\rho_L^{-1}$ | −0.0001 | 0.0069 |
| $p_{TL}$ | −0.0024 | −0.0040 |
| $\sigma_F^{-1}$ | −0.0111 | 0.5252 |
| $\beta^{-1}$ | −0.0849 | 3.8384 |

who does not use ITNs, or whether it bites an individual despite them using ITNs. The first scenario accounts for $(1-p)/(1-pe)$ cases, the second one for the remaining $p(1-e)/(1-pe)$ cases. Moreover, in the second scenario, we would need to add one more factor $(1-e)$ to stand for a second 'failure' of the ITN.

Recent results of [55] suggest that incorporating spatial structure into the model would greatly increase its realism.

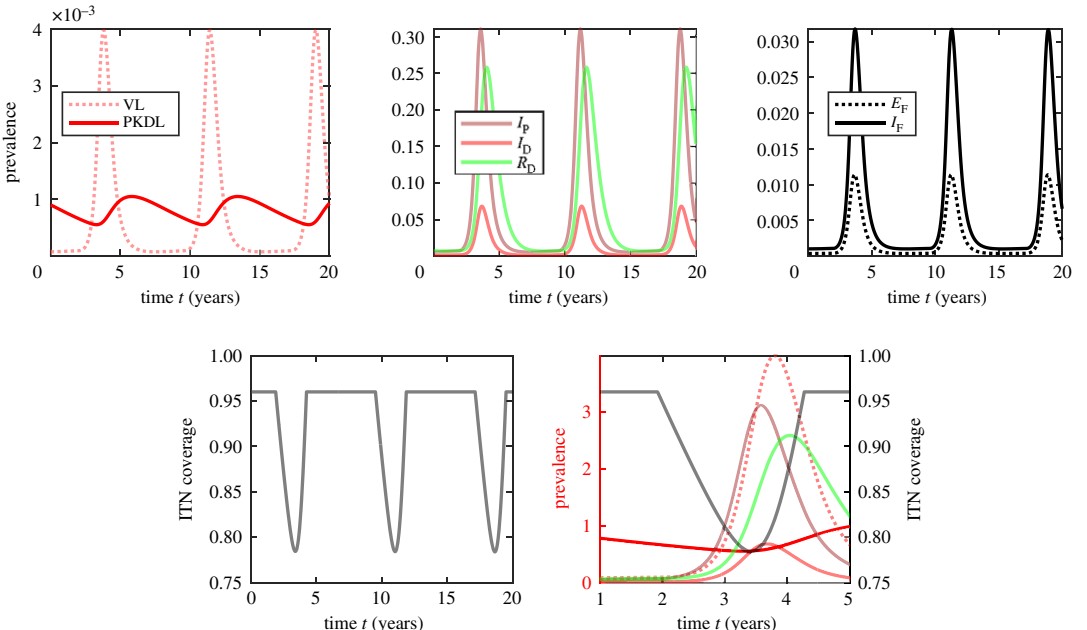

**Figure 6.** Adaptive behaviour when people use ITN based on the prevalence of untreated PKDL cases can lead to periodic spikes of VL and PKDL cases with periods of about 7.5 years. For illustration purposes, we assume that people use ITNs with a probability $\min\{0.96, 1500 \, (I_{UL}/N)\}$; this corresponds to using ITNs at the level proportional to the number of untreated PKDL cases per village (the average village size in Bihar, India is about 1500 people [128]), and capped at 0.96, the approximate coverage needed for VL elimination. The top row shows prevalence of VL and PKDL (left), $I_P$, $I_D$ and $R_D$ (centre) and $I_F$ and $E_F$ (right). The bottom row shows the ITN coverage (left) and the prevalences of human compartments together with protection levels (right). To fit better in one figure, the units for VL and PKDL are in $10^{-3}$ while the units for $I_P$, $I_D$ and $R_D$ are in $10^{-1}$.

Also, there are several limitations of the game-theoretical framework, particularly the assumption that all individuals are provided with the same information and use it in the same (and rational) way to assess costs and risks [65]. First, ITN ownership is associated with wealth, and the cost of ITNs is often prohibitive for the poor [42]. Moreover, VL susceptibility is bound to reduce with wealth, as malnutrition and certain interior wall types are VL risk factors [125,126]. One would have to account for this by considering a non-homogeneous population. Second, VL has a long incubation period during which individuals are asymptomatic yet may be contagious. This makes the risk assessment prone to errors as people may think that there is no or very low risk of contracting VL (and thus stop using ITNs) while in reality someone in their community may already be asymptomatic. The risk assessment is further complicated by the fact that even professionals often misdiagnose acute symptomatic VL cases [119] and by the social and cultural stigma associated with VL [20].

For simplicity, we focused our analysis on equilibria of the ODE system. This approach had several limitations and disadvantages. As already mentioned above, it precludes us from incorporating seasonal sand fly dynamics. Similarly, the model cannot properly capture dynamical changes in pricing or availability of ITNs or drugs as was recently happening due to COVID-19 [127]. Most importantly, to understand and model the final stages of VL elimination, one should consider not only the equilibria but also how long it takes to reach an equilibrium. One also needs to explicitly model the interactions of the potential dynamical ITNs coverage and the disease dynamics.

Unlike vaccines that offer a long-lasting protection after just one decision, ITNs have to be used repeatedly. In principle, individuals decide every night whether or not to use ITNs. This dynamic decision process is not captured in our model. However, we hypothesize that coupled with long incubation periods of VL and especially PKDL, the seemingly optimal behaviour of using ITNs only if acute VL and PKDL cases reach above certain threshold, could lead to periodic spikes and drops of VL and PKDL cases. This is illustrated in figure 6. We can see that soon after the ITN coverage is relaxed, the early asymptomatic, $I_P$, and symptomatic VL cases are on the rise, followed by a rise of asymptomatic recovered cases, $R_D$, and only a very small and slow rise of

late asymptomatic cases, $I_D$. Once the population starts to use more ITNs again, $I_P$ declines almost immediately, soon followed by a decline of $I_D$ and symptomatic VL cases and a trailing decline of asymptomatic recovered cases. At the same time, even as ITN coverage is on the rise, there is a slow but steady increase of PKDL cases that act as a reservoir of infections and can restart the cycle again. Overall, the importance of understanding of the consequences of dynamical decisions warrant future studies.

# 7. Conclusion

We expanded the mathematical model of VL transmission [56] to understand the role of ITN usage in VL dynamics and possible elimination. We calibrated the model based on the data found in the literature. We validated the model on KalaNet data. We concluded that in order to completely eliminate VL, the ITN usage would have to be about 96% or more. At such an ITN coverage, the effective reproduction number is less than 1 and the disease-free equilibrium is stable. Our work seems to be the first explicitly giving the ITN coverage threshold for VL elimination.

Our work, specifically figure 5, may also explain why there was no significant effect of ITN use on VL incidence during the KalaNet trial—the risk of VL infection (which correlates with the mean cost function) is essentially a step function with a jump just around the disease elimination threshold. At the end of the KalaNet trial, the use of ITNs was very high: 91% of the individuals slept more than 80% of the nights under treated nets [46]. Yet, even with such high ITN use, our model predicts that the disease is not eliminated and the risk of infection almost does not change. The difference between 91% and 96% does not seem large, but note that 91% coverage leaves 9% of the population unprotected. This is more than double the unprotected population if the ITNs use is 96% or more.

From the policy-making perspective, while our model demonstrates that the elimination of VL by ITN use is possible in theory, it is unlikely in practice; the ITN coverage of 96% required for complete VL elimination is unrealistically high. At the same time, as seen from figure 6, a decrease of ITNs coverage can lead to sharp and dramatic increases of VL cases. Instead of abandoning the use of ITNs completely after the unfortunate findings of KalaNet trial, we recommend that the ITN use should be combined with other intervention methods. Figure 6 also carries a warning that regardless of what method is used to bring down VL incidence, the methods should stay in place long after VL seems eliminated—otherwise the disease can quickly resurface back.

Our model, specifically the sensitivity analysis illustrated in figure 8, suggests that increasing $\beta^{-1}$, the interval between two sand fly bites, from 4 days to 6 or more days can reduce the ITN coverage needed for VL elimination from 96 to 90% or even lower (which falls below 91% observed during the KalaNet trial). The increase of $\beta^{-1}$ could be achieved by giving the sand flies other biting opportunities. While controversial, this intervention method has been studied for other vector-borne neglected tropical diseases such as Chagas disease [129]. The impact of the presence of cattle on VL incidence has already been studied separately. The cattle are associated with increased VL risk in some studies and decreased risk in others [38,130], reflecting the complexity of the effects on sand fly abundance, aggregation, feeding behaviour and VL incidence [131]. An increase in blood meal opportunities could lead to increase in the sand fly population size, possibly erasing the positive effect of increased biting interval. Consequently, an effective elimination strategy should include multiple interventions at the same time—(i) using ITNs to prevent insect bites as much as possible, (ii) using cattle and/or other animals (that cannot serve as hosts for *L. donovani*) to divert vector bites away from humans [132], and (iii) using insecticide residual spraying [7], or (iv) destroying breeding sites [52] to keep the vector population under control.

Another important finding of our model is that asymptomatic individuals, specifically those that test PCR-positive and DAT-positive, play a crucial role in VL transmission. Our model predicts that those individuals are about 50% as infectious as the untreated symptomatic individuals; although we admit that such an estimate seems about twice as high as assumptions made in other studies, such as [55,56]. Based on our model, without the asymptomatic transmissions, VL would already be eliminated. The apparent difference between our results and recent experimental results of [94] stems only from a different use of the word asymptomatic. Majority of cases from [94] tested DAT-positive but PCR negative, i.e. were considered as asymptomatic *recovered* and assumed non-infectious in our model. There is thus no factual difference between results of [94] and the assumptions of our model.

We believe that empirical research needs to be done to understand the role of asymptomatic patients in VL transmissions—specifically those that tested DAT-positive and PCR-positive and were infected roughly four to six months in the past. As shown in figure 6 as well as in the sample of cases studied in [94], these individuals may be a relatively elusive part of the population; yet they may be an important missing piece in understanding VL dynamics.

Data accessibility. The Matlab code supporting this article has been uploaded as part of the electronic supplementary material.

Authors' contributions. A.K.F., J.A.W.: conceptualization, formal analysis, investigation and writing: original draft. C.P.G.: conceptualization, formal analysis, investigation, software and writing: original draft. Y.L. supervision, writing: original draft. J.R. writing: original draft, writing: review and editing, software, methodology, supervision, conceptualization, resources. D.T. writing: original draft, writing: review and editing, methodology, supervision, conceptualization, formal analysis, validation, funding acquisition. The order of the authors was determined as follows: The first three authors are undergraduate students in alphabetical order. The last three authors are professors in alphabetical order.

Competing interests. There are no competing interests.

Funding. A.K.F., C.P.G. and J.A.W. were supported by the VCU REU program in mathematics funded by the National Security Agency (https://www.nsa.gov/what-we-do/research/math-sciences-program/proposal-guidelines/) grant no. H98230-20-1-0011 awarded to D.T. The funders had no role in study design, data collection and analysis, decision to publish or preparation of the manuscript. The publication fees were covered by the Virginia Tech's Open Access Subvention Fund.

Acknowledgements. We thank Dr Gideon Wasserberg, UNC Greensboro, for useful discussion on vector biology. We also thank the editor, a reviewer Timothy M. Pollington and an anonymous reviewer for their detailed feedback and comments that helped us to improve the manuscript.

# Appendix A. ODEs for the model

The model of the transmission dynamics described in §2 and shown in figure 1 yields the following system of ordinary differential equations.

$$\frac{dS}{dt} = \Lambda + \rho_C R_C - (\mu + \lambda)S, \tag{A 1}$$

$$\frac{dI_P}{dt} = \lambda S - (\mu + \gamma_P)I_P, \tag{A 2}$$

$$\frac{dI_D}{dt} = \gamma_P I_P - (\mu + \gamma_D)I_D, \tag{A 3}$$

$$\frac{dR_D}{dt} = f_D \gamma_D I_D - (\mu + \rho_D)R_D, \tag{A 4}$$

$$\frac{dI_S}{dt} = f_S \gamma_D I_D - (\mu_S + \gamma_S)I_S, \tag{A 5}$$

$$\frac{dI_{T1}}{dt} = \gamma_S I_S - (\mu_1 + \tau_1)I_{T1}, \tag{A 6}$$

$$\frac{dI_{T2}}{dt} = s_1 \tau_1 I_{T1} - (\mu_2 + \tau_2)I_{T2}, \tag{A 7}$$

$$\frac{dR_L}{dt} = s_2 \tau_1 I_{T1} + s_4 \tau_2 I_{T2} + f_L \gamma_D I_D - (\mu + \rho_L)R_L, \tag{A 8}$$

$$\frac{dI_{TL}}{dt} = p_{TL} \rho_L R_L - (\mu + \tau_3)I_{TL}, \tag{A 9}$$

$$\frac{dI_{UL}}{dt} = (1 - p_{TL})\rho_L R_L - (\mu + \gamma_L)I_{UL}, \tag{A 10}$$

$$\frac{dR_T}{dt} = s_3 \tau_1 I_{T1} + s_5 \tau_2 I_{T2} + \tau_3 I_{TL} + \gamma_L I_{UL} - (\mu + \rho_T)R_T, \tag{A 11}$$

$$\frac{dR_C}{dt} = \rho_D R_D + \rho_T R_T - (\mu + \rho_C)R_C, \tag{A 12}$$

$$\frac{dS_F}{dt} = \mu_F N_F - (\mu_F + \lambda_F)S_F, \tag{A 13}$$

$$\frac{dE_F}{dt} = \lambda_F S_F - (\mu_F + \sigma_F)E_F \tag{A 14}$$

and

$$\frac{dI_F}{dt} = \sigma_F E_F - \mu_F I_F. \tag{A 15}$$

## A.1. Equilibria of the ODE system

The equilibria of the dynamics (A 1)–(A 15) are found by solving the following system of algebraic equations:

$$0 = \Lambda + \rho_C R_C - (\mu + \lambda)S, \tag{A 16}$$

$$0 = \lambda S - (\mu + \gamma_P)I_P, \tag{A 17}$$

$$0 = \gamma_P I_P - (\mu + \gamma_D)I_D, \tag{A 18}$$

$$0 = f_D \gamma_D I_D - (\mu + \rho_D)R_D, \tag{A 19}$$

$$0 = f_S \gamma_D I_D - (\mu_S + \gamma_S)I_S, \tag{A 20}$$

$$0 = \gamma_S I_S - (\mu_1 + \tau_1)I_{T1}, \tag{A 21}$$

$$0 = s_1 \tau_1 I_{T1} - (\mu_2 + \tau_2)I_{T2}, \tag{A 22}$$

$$0 = s_2 \tau_1 I_{T1} + s_4 \tau_2 I_{T2} + f_L \gamma_D I_D - (\mu + \rho_L)R_L, \tag{A 23}$$

$$0 = p_{TL}\rho_L R_L - (\mu + \tau_3)I_{TL}, \tag{A 24}$$

$$0 = (1 - p_{TL})\rho_L R_L - (\mu + \gamma_L)I_{UL}, \tag{A 25}$$

$$0 = s_3 \tau_1 I_{T1} + s_5 \tau_2 I_{T2} + \tau_3 I_{TL} + \gamma_L I_{UL} - (\mu + \rho_T)R_T, \tag{A 26}$$

$$0 = \rho_D R_D + \rho_T R_T - (\mu + \rho_C)R_C, \tag{A 27}$$

$$0 = \mu_F N_F - (\mu_F + \lambda_F)S_F, \tag{A 28}$$

$$0 = \lambda_F S_F - (\mu_F + \sigma_F)E_F \tag{A 29}$$

and

$$0 = \sigma_F E_F - \mu_F I_F. \tag{A 30}$$

The system (A 16)–(A 30) will be solved as follows, corresponding to starting at compartment $S$ and going downstream. By (A 17), and since $\lambda = (1 - p)\beta i_F I_F$,

$$I_P^* = \frac{\lambda}{\mu + \gamma_P}S^* = (1 - p)\beta i_F \frac{1}{\mu + \gamma_P}S^* I_F^*. \tag{A 31}$$

Similarly, by (A 18)–(A 27), for every compartment Comp $\in \{I_P, I_D, R_D, R_C, I_S, I_{T1}, I_{T2}, I_{TL}, I_{UL}, R_T, R_C\}$ we have

$$\text{Comp}^* = (1 - p)\beta i_F T_{\text{Comp}} S^* I_F^* \tag{A 32}$$

where the quantities $T_{\text{Comp}}$ correspond to the expected time an individual spends in a compartment Comp given it started in compartment $I_P$ and are given by

$$T_{I_P} = \frac{1}{\mu + \gamma_P}, \tag{A 33}$$

$$T_{I_D} = \frac{\gamma_P}{\mu + \gamma_D}T_{I_P}, \tag{A 34}$$

$$T_{R_D} = \frac{f_D \gamma_D}{\mu + \rho_D}T_{I_D}, \tag{A 35}$$

$$T_{I_S} = \frac{f_S \gamma_D}{\mu_S + \gamma_S}T_{I_D}, \tag{A 36}$$

$$T_{I_{T1}} = \frac{\gamma_S}{\mu_1 + \tau_1}T_{I_S}, \tag{A 37}$$

$$T_{I_{T2}} = \frac{s_1 \tau_1}{\mu_2 + \tau_2}T_{I_{T1}}, \tag{A 38}$$

$$T_{R_L} = \frac{s_2 \tau_1 T_{I_{T1}} + s_4 \tau_2 T_{I_{T2}} + f_L \gamma_D T_{I_D}}{\mu + \rho_L}, \tag{A 39}$$

$$T_{I_{TL}} = \frac{p_{TL}\rho_L}{\mu + \tau_3}T_{R_L}, \tag{A 40}$$

$$T_{I_{UL}} = \frac{(1 - p_{TL})\rho_L}{\mu + \gamma_L}T_{R_L}, \tag{A 41}$$

$$T_{R_T} = \frac{s_3 \tau_1 T_{I_{T1}} + s_5 \tau_2 T_{I_{T2}} + \tau_3 T_{I_{TL}} + \gamma_L T_{I_{UL}}}{\mu + \rho_T} \tag{A 42}$$

and

$$T_{R_C} = \frac{\rho_D T_{R_D} + \rho_T T_{R_T}}{\mu + \rho_C}. \tag{A 43}$$

## A.2. Disease-free equilibrium

We assume that $I_P^0 = 0$. By (A 18), $I_D^0 = 0$. By (A 19), $R_D^0 = 0$. By (A 20), $I_S^0 = 0$. By (A 21), $I_{T1}^0 = 0$. By (A 22), $I_{T2}^0 = 0$. By (A 23), $R_L^0 = 0$. By (A 24), $I_{TL}^0 = 0$, and, by (A 25), $I_{UL}^0 = 0$. By (A 26), $R_T^0 = 0$. By (A 27), $R_C^0 = 0$. From (A 16), $S^0 = \Lambda/\mu$. By (2.2), $\lambda_F = 0$. Thus, by (A 28), $S_F^0 = N_F = n_F(\Lambda/\mu)$. Thus, $E_F^0 = 0$ and $I_F^0 = 0$. So our disease free equilibrium is given by

$$E^0 = \left(\frac{\Lambda}{\mu}, 0, 0, 0, 0, 0, 0, 0, 0, 0, 0, 0, n_F\frac{\Lambda}{\mu}, 0, 0\right). \tag{A 44}$$

## A.3. Endemic equilibrium

By looking at all human compartments and balancing what is coming in and out (i.e. adding all equations (A 16)–(A 27)), we get

$$\Lambda = \mu S^* + \mu I_P^* + \mu I_D^* + \mu R_D^* + \mu_S I_S^* + \mu_1 I_{T1}^* + \mu_2 I_{T2}^* + \cdots + \mu R_L^* + \mu I_{TL}^* + \mu I_{UL}^* + \mu R_T^* + \mu R_C^*. \tag{A 45}$$

By (A 33)–(A 43),

$$\Lambda = \mu[S^* + (1-p)\beta i_F T_{\text{Cycle}} S^* I_F^*], \tag{A 46}$$

where

$$T_{\text{Cycle}} = T_{I_P} + T_{I_D} + T_{R_D} + \frac{\mu_S}{\mu}T_{I_S} + \frac{\mu_1}{\mu}T_{I_{T1}} + \frac{\mu_2}{\mu}T_{I_{T2}} + \cdots + T_{R_L} + T_{I_{TL}} + T_{I_{UL}} + T_{R_T} + T_{R_C}. \tag{A 47}$$

Solving for $S^*$ yields

$$S^* = \left(\frac{\Lambda}{\mu}\right)\left(\frac{1}{1 + (1-p)\beta i_F T_{\text{Cycle}} I_F^*}\right). \tag{A 48}$$

By (2.2),

$$\lambda_F = \beta(i_P I_P^* + i_D I_D^* + i_S I_S^* + i_{T1} I_{T1}^* + i_{T2} I_{T2}^* + i_{TL} I_{TL}^* + i_{UL} I_{UL}) \tag{A 49}$$

$$= (1-p)\beta^2 i_F T_I S^* I_F^* \tag{A 50}$$

$$= (1-p)\beta^2 i_F T_I \left(\frac{\Lambda}{\mu}\right)\frac{I_F^*}{1 + (1-p)\beta i_F T_{\text{Cycle}} I_F^*}. \tag{A 51}$$

Thus,

$$\frac{I_F^*}{\lambda_F} = \frac{1 + (1-p)\beta i_F T_{\text{Cycle}} I_F^*}{(1-p)\beta^2 i_F T_I \left(\dfrac{\Lambda}{\mu}\right)}. \tag{A 52}$$

From (A30),

$$E_F^* = I_F^* \frac{\mu_F}{\sigma_F}. \tag{A 53}$$

From (A29),

$$S_F^* = \frac{\mu_F + \sigma_F}{\lambda_F} E_F^* = \frac{\mu_F + \sigma_F}{\lambda_F} \cdot \frac{\mu_F}{\sigma_F} \cdot I_F^* \tag{A 54}$$

$$= \left(\frac{\mu_F + \sigma_F}{\sigma_F}\right) \cdot \mu_F \cdot \left(\frac{1 + (1-p)\beta i_F T_{\text{Cycle}} I_F^*}{(1-p)\beta^2 i_F T_I \left(\dfrac{\Lambda}{\mu}\right)}\right) \tag{A 55}$$

$$= N_F^* \frac{1 + (1-p)\beta i_F T_{\text{Cycle}} I_F^*}{\mathcal{R}_e}. \tag{A 56}$$

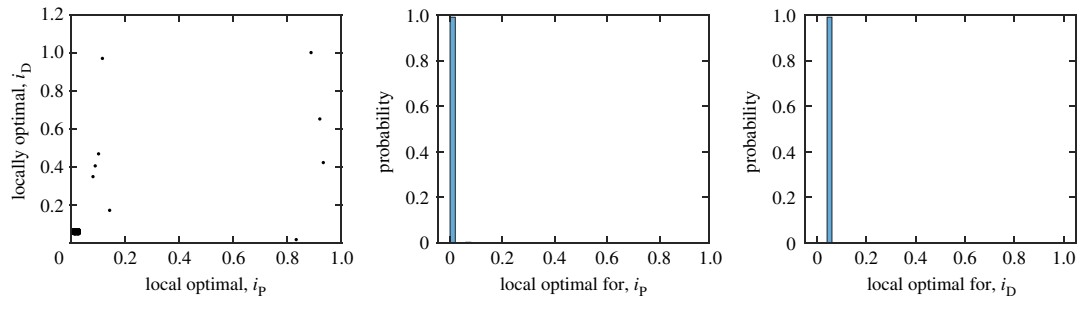

**Figure 7.** Graphical representation of the pairs of local optima (left) with histograms showing the frequencies of $i_P$ (centre) and $i_D$ (right). Other parameters as in tables 2–4.

Thus,

$$N_F^* = S_F^* + E_F^* + I_F^* \tag{A 57}$$

$$= N_F^*\left(\frac{1 + (1-p)\beta i_F T_{\text{Cycle}} I_F^*}{\mathcal{R}_e}\right) + \frac{\mu_F}{\sigma_F}I_F^* + I_F^* \tag{A 58}$$

and

$$I_F^* = \frac{N_F^*\left(1 - \dfrac{1}{\mathcal{R}_e}\right)}{(1-p)\beta i_F T_{\text{Cycle}}\dfrac{N_F^*}{\mathcal{R}_e} + \dfrac{\mu_F}{\sigma_F} + 1}. \tag{A 59}$$

Once $I_F^*$ is calculated from (A 59), we can calculate $E_F^*$, $S_F^*$ and $S^*$ as

$$E_F^* = I_F^* \frac{\mu_F}{\sigma_F}, \tag{A 60}$$

$$S_F^* = \frac{\mu_F + \sigma_F}{\lambda_F}E_F^* \tag{A 61}$$

and

$$S^* = \left(\frac{\Lambda}{\mu}\right)\left(\frac{1}{1 + (1-p)\beta i_F T_{\text{Cycle}} I_F^*}\right). \tag{A 62}$$

Finally, using above, we can also find all the human compartments as below

$$I_P^* = (1-p)\beta i_F T_{I_P} S^* I_F^*, \tag{A 63}$$

$$I_D^* = (1-p)\beta i_F T_{I_D} S^* I_F^*, \tag{A 64}$$

$$R_D^* = (1-p)\beta i_F T_{R_D} S^* I_F^*, \tag{A 65}$$

$$I_S^* = (1-p)\beta i_F T_{I_S} S^* I_F^*, \tag{A 66}$$

$$I_{T1}^* = (1-p)\beta i_F T_{I_{T1}} S^* I_F^*, \tag{A 67}$$

$$I_{T2}^* = (1-p)\beta i_F T_{I_{T2}} S^* I_F^*, \tag{A 68}$$

$$R_L^* = (1-p)\beta i_F T_{R_L} S^* I_F^*, \tag{A 69}$$

$$I_{TL}^* = (1-p)\beta i_F T_{I_{TL}} S^* I_F^*, \tag{A 70}$$

$$I_{UL}^* = (1-p)\beta i_F T_{I_{UL}} S^* I_F^*, \tag{A 71}$$

$$R_T^* = (1-p)\beta i_F T_{R_T} S^* I_F^* \tag{A 72}$$

and

$$R_C^* = (1-p)\beta i_F T_{R_C} S^* I_F^*. \tag{A 73}$$

# Appendix B. Numerical estimates of transmission probabilities

The transmission probabilities $i_P$, $i_D$ for asymptomatic individuals, and $i_F$ for sand flies were estimated using the built-in function *fmincon* from Matlab optimization toolbox. We minimized the difference between the model predicted prevalences and the prevalences measured during KalaNet trial ([56],

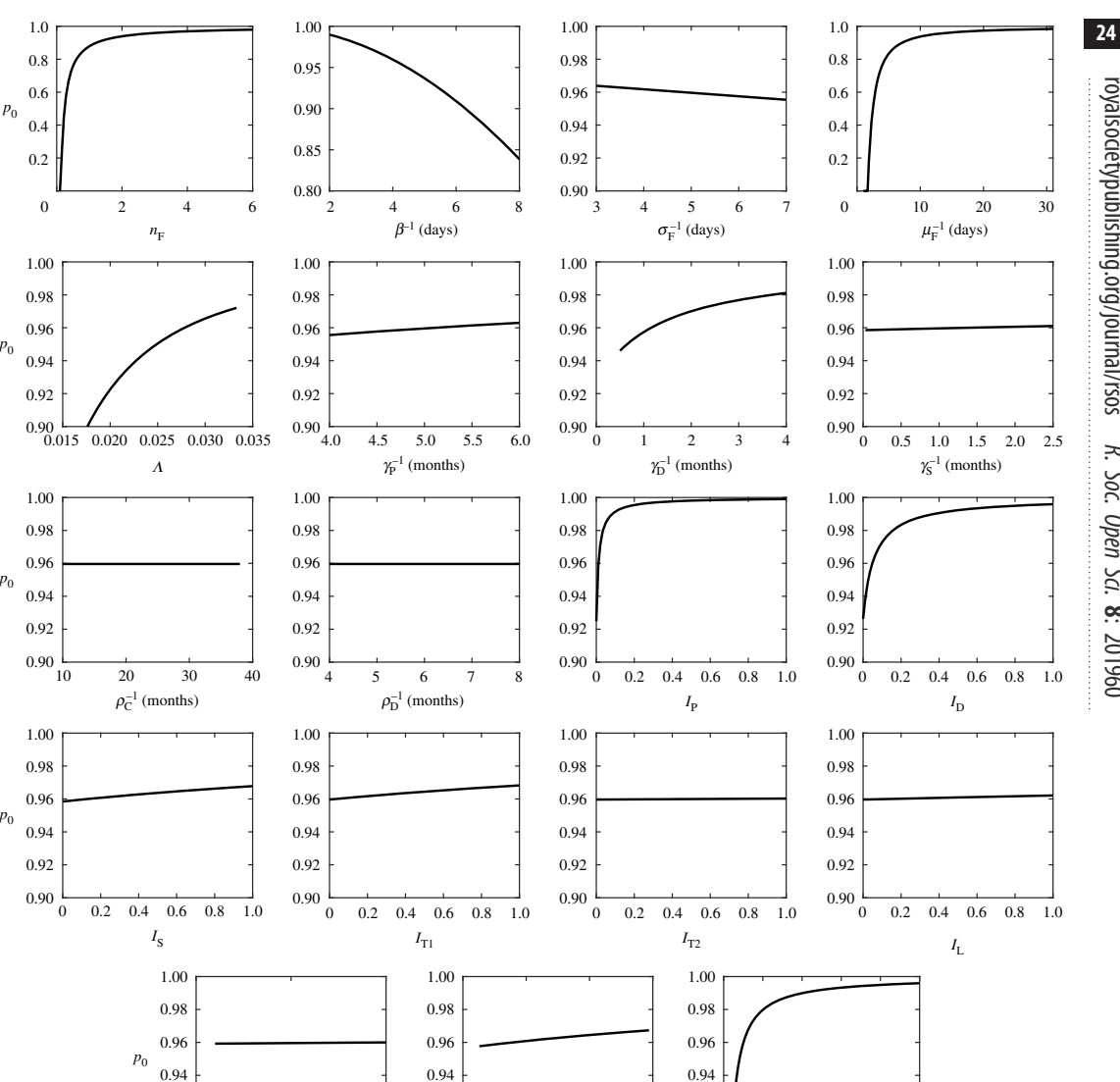

**Figure 8.** Dependence of $p_0$ on different parameter values. Unless varied, the parameter values are as specified in tables 2–4. For those parameters, $p_0 = 0.95963$, $p_{NE} = 0.95956$.

table 1). Depending on the initial guess for the optimum, the function *fmincon* sometimes returns a local optimum rather than global optimum. To make sure we seeded properly and indeed found global optima, we initialized the search with 1000 random seeds and collected all results. We collected all the local minima and searched for the global optima from those. The results are shown in figure 7. It follows that the global optima found by this complicated procedure, $i_P = 0.0110$, $i_D = 0.0487$ and $i_F = 1$ is close enough to local optima $i_P = 0.0111$, $i_D = 0.0481$ and $i_F = 1$ found by initiating the searches for $i_P$ and $i_D$ at 0 and for $i_F$ at 1 (figures 8 and 9),

## Appendix C. Finding Nash equilibrium

To find a Nash equilibrium value of $p$, we have to solve (5.3), i.e.

$$C_{ITN} = \frac{\beta i_F I_F^*}{\mu + \beta i_F I_F^*} C_{VL}.$$

$$(C 1)$$

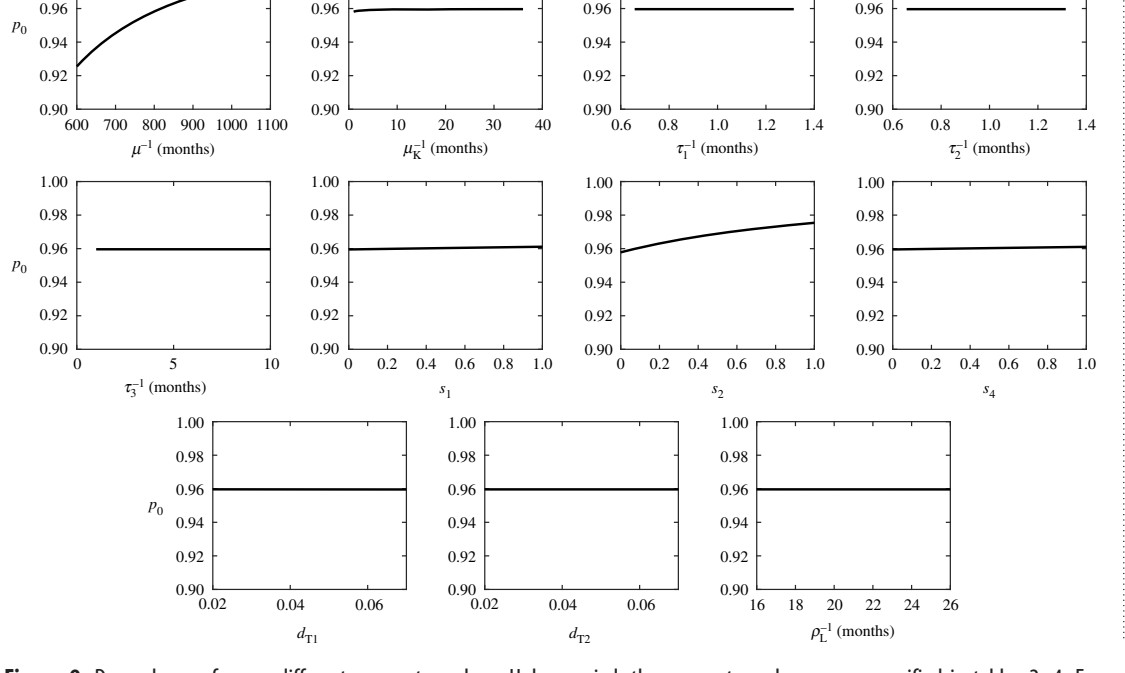

**Figure 9.** Dependence of $p_0$ on different parameter values. Unless varied, the parameter values are as specified in tables 2–4. For those parameters, $p_0 = 0.95963$, $p_{NE} = 0.95956$.

Recall that, by (A 59),

$$I_F^* = \frac{N_F^*\left(1 - \dfrac{1}{(1-p)\mathcal{R}_e(0)}\right)}{(1-p)\beta i_F T_{\text{Cycle}}\dfrac{N_F^*}{(1-p)\mathcal{R}_e(0)} + \dfrac{\mu_F}{\sigma_F} + 1} \tag{C 2}$$

and in particular $I_F^*$ is decreasing in $p$ and thus $\beta i_F I_F^*/(\mu + \beta i_F I_F^*)$ is decreasing in $p$. The maximum value of $I_F^*$ is thus

$$I_F^*(0) = \frac{N_F^*(1 - (1/\mathcal{R}_e(0)))}{\beta i_F T_{\text{Cycle}}(N_F^*/\mathcal{R}_e(0)) + (\mu_F/\sigma_F) + 1} \tag{C 3}$$

attained for $p = 0$, and consequently (5.3) has a solution only if

$$C_{\text{ITN}} \leq \frac{\beta i_F I_F^*(0)}{\mu + \beta i_F I_F^*(0)} C_{\text{VL}}. \tag{C 4}$$

It follows from (5.3) that (when (C 4) is true)

$$I_F^* = \frac{(C_{\text{ITN}}/C_{\text{VL}})\mu}{\beta i_F(1 - (C_{\text{ITN}}/C_{\text{VL}}))}. \tag{C 5}$$

Thus, after algebraic manipulations of (C 2), we get that $p_{NE}$ is given by

$$p_{NE} = 1 - \frac{1}{\mathcal{R}_e(0)[1 - (I_F^*/N_F^*)(\beta i_F T_{\text{Cycle}}(N_F^*/\mathcal{R}_e(0)) + (\mu_F/\sigma_F) + 1)]} \tag{C 6}$$

where $I_F^*$ is given by (C 5).

# Appendix D. Sensitivity analysis

(see table 7)

**Table 7.** Policy-Relevant Items for Reporting Models in Epidemiology of Neglected Tropical Diseases (PRIME-NTD) Summary Table as specified in [82].

| principle | What has been done to satisfy the principle? | Where in the manuscript is this described? |
|---|---|---|
| 1. stakeholder engagement | We did not directly engage the stakeholder. However, we based our model on [56] and extensive VL literature describing KalaNet trial. | §§1 and 2. |
| 2. complete model documentation | We provided a detailed model description. | §2. |
| | We implemented the model numerically in Matlab with ample comments. | electronic supplementary material. |
| 3. complete description of data used | We calibrated our model on data available in the literature. | §4. |
| | We detailed procedures of how we extracted parameter values that were not found directly. | §4. |
| | We derived VL transmission probabilities. | §5.1. |
| 4. communicating uncertainty | We performed sensitivity analysis based on [121]. | §5.5, table 6, figures 8 and 9 and the Matlab code. |
| 5. Testable model outcomes | We predicted that asymptomatic individuals (PCR+, DAT+) can transmit VL and their infectiousness is at about 50% of the level of untreated symptomatic individuals. | §5.1 and 5.2. |
| | We gave formula for the minimal ITN coverage needed to achieve complete elimination of VL and predicted that 96% ITN coverage is needed. | §5.3. |
| | We also predicted that voluntary use of ITN should yield elimination of VL as public concern. | §5.4 with mathematical details in appendix C. |

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
