## [Peer Review File · Royal Society Open Science]

Review History

RSOS-201960.R0 (Original submission)

Review form: Reviewer 1

Is the manuscript scientifically sound in its present form?

Yes

Are the interpretations and conclusions justified by the results?

Yes

Is the language acceptable?

Yes

Do you have any ethical concerns with this paper?

No

Have you any concerns about statistical analyses in this paper?

No

Recommendation?

Accept with minor revision (please list in comments)

Comments to the Author(s)

This is an interesting and very well written paper about mathematical modeling of individual behavior regarding the use of insecticide-treated nets (ITN) as preventive method against visceral leishmaniasis (VL). The authors present in detail a modification of the mathematical model describing the sand flies-humans dynamics, previously developed by Stauch et al. (2011, PLoS Neglected Tropical Diseases). The authors study the dynamical system within a game-theoretical framework, minimizing the expected cost of using ITN. The main outcome of the model is the proportion of people using ITN that may lead to VL elimination.

COMMENTS ON THE MODELING CHOICES

The manuscript may benefit from reformulating and/or explaining some modeling choices. For instance, in line 86, the authors mention free-riders but don't mention whether or not they model the free-rider behavior explicitly. In line 110, the authors say that equation (2.1) may be interpreted as if "ITNs provide a very effective protection", while their model considers perfect protection, instead. Also, the choice of the computation/definition of R_0 could be supported by a reference.

Assumption (iii) in line 176 could also be supported by a reference. In addition, the authors could contrast this assumption with available data on the population size. If not, they could discuss the limitations of such an assumption.

Please consider discussing the stability of the equilibria, which may give further insight about the system behavior and thus helping to interpret the results, as well as discussing the results of the KalaNet trial (see line 468).

COMMENTS ON NOTATION AND FORMATTING

The authors compute the reproduction number for the system subject to a control measure. Therefore, it is the effective reproduction number and not the basic reproduction number. The authors could use the notation R_e , instead of R_0 , where $R_e = R_e(p)$.

Please recall the parameters values used to produce figure 4, in the caption. Please expand the description of figures 5 and 6 in their captions, including a short description of the parameters set depicted in the x-axes.

Please ensure notation consistency for numbers of 4 or more digits.

Review form: Reviewer 2 (Tim Pollington)

Is the manuscript scientifically sound in its present form?

Yes

Are the interpretations and conclusions justified by the results?

Yes

Is the language acceptable?

Yes

Do you have any ethical concerns with this paper?

Yes

Have you any concerns about statistical analyses in this paper?

No

Recommendation?

Major revision is needed (please make suggestions in comments)

Comments to the Author(s)**SUMMARY**

My recommendation to the Editor is to accept this manuscript, pending major corrections. Please interpret 'major' in the sense that these suggestions will likely affect the results. I thought the quality of the research and its write-up were of a high standard for this modelling manuscript and would very much like to see its corrected version added to the VL research base.

The findings: The authors have developed a model by amending an existing one and parametrising it with mostly reasonable values and ranges. The two research impacts are:

A) They obtain a threshold estimate for the ITN coverage required for (zero) disease elimination. (labelling this as RI-A)

B) Considering that ITN use is voluntary, they make a sensible and (probably for VL) novel application of game theory to assess if voluntary ITN use alone, influenced by the costs and benefits at the individual level, is sufficient to meet this threshold. (RI-B)

Quality: It is of a high research standard: indicated by the mathematical foundations and clear exposition in the text, strongly informed in model structure and parameters by existing VL research, the rigour of the analysis including validation and sensitivity analysis. Moreover, it is the first manuscript I have reviewed which provides working code and reproducible results! It therefore ticks many best practices of 21st century modelling.

Potential impact: It also contributes to the ongoing discussion following the unfortunate findings of the Kalanet trial in the previous decade. This is the primary reason why ITNs do not form a part of today's VL control. WHO has launched the NTD 2021-2030 roadmap earlier in 2021 and so this could be an ideal time for research during this receptive policy window. It could reawaken minds and inform future epidemiological and entomological study designs to definitively answer if ITNs have a contributing role in VL control on the Indian subcontinent. The manuscript could work harder at highlighting this point for the VL policy maker audience.

To open up this research piece to further discussion and improve its publication impact I suggest releasing the initially submitted draft as a preprint and later interacting with the VL research community at forthcoming conferences such as WorldLeish 2022 which will probably start accepting abstracts later this year, but this of course is at the authors' discretion.

Scope of this review: I spent 3 days reviewing the manuscript, code and writing up this review. I did not check the majority of the mathematical formulae at an analytical level. Instead I focussed on code correctness from code lines 1-478, parameter suitability and if this modelling was appropriate for VL.

Conflict of interest: I declare that I have included citations within this review which either myself or my supervisors (T Déirdre Hollingsworth & Lloyd Chapman) are co-authors. The authors are not compelled to add these citations. I included them to provide justification and to encourage the authors why a suggestion is necessary or why it has been raised.

Current major limitations: I believe there are some major bugs in the code that would affect results but these should be quick to fix. Some coding practices may also help improve the confidence in the code correctness. Some parameters may not fit the setting or understanding of the disease the best and I have highlighted the ones most likely to impact results, partly from the authors' helpful sensitivity analysis; this may require a rapid review of the recent literature for

parameter updates but again this shouldn't take long. No change to the model structure is necessary. Finally, although the manuscript text itself was painless to read, it could focus better at the main audience—the VL research community. Some of these are detailed below and are also attached on commented files.

BEFORE THE AUTHORS READ THIS REVIEW (OR THE PUBLIC, AFTER PUBLICATION)

My overriding motivation for this peer review is to protect the field from errors in research methods or misinterpretation of results, that could make policy makers make suboptimal decisions in VL programmes that ultimately harm people. The annual research output for VL modelling is relatively small compared to other diseases, so you can realise that avoidable errors could have more impact in this context than others. I haven't seen the authors' names from previous VL modelling research, so if you are new to VL then welcome to the field! None of the following review is intended to 'put you in your place' nor be patronising, and your future modelling for this neglected tropical disease is very much welcomed. In the spirit of open science some of the reviewers may have declared their names to be made public and in return I hope the authors allow all reviews to be open (for reviewers who allow it).

REVIEW STRUCTURE

This review is structured by the manuscript then code run, and grouped by things essential to impacting the results' values or presentation (major) or affecting the understanding/model improvements. This review is long but I believe in thoroughness and once some of what I suggest is corrected, then I think what you have got here could be really valuable to the VL field.

MANUSCRIPT

MAJOR

Top, most important

- An extensive range of research has been drawn upon to inform parameters selected. Some suggestions have been made from empirical sources for the refinements of some parameters. See mns attachment for more details.
- "Multiple lab-based studies suggest that female sand flies have fairly short adult lifespans (<20 days) [16] with further reductions when infected [15]" (<https://journals.plos.org/plosntds/article?id=10.1371/journal.pntd.0009033>) therefore it is unrealistic to assume the mortality of I_F in Fig 1 is the same as S_F or E_F (=mu_F).
- Making it more policy-friendly: reduce the amount of mathematical formulae by moving formulae (lines 212-223 & 446-452) describing the intermediate mathematical logic to a Supplementary information as there's quite a few pages to flip through from line 185-225. Please do not understand what I mean, I just mean reduce the amount, the level at which you are communicating is fine and there is no need to dumb things down. Your short-hand notation like Comp on line 197 was clever and for instance Eqns 3.1-3.15 could be represented as a one-liner, something like $dX/dt = 0$ where $X=\{\text{etc}\}$.
- Although the Discussion covers limitations in the findings, model limitations are also necessary. Also despite the model being validated against previous trial data, what would be the external validity of these findings to say Nepal?
- Parts of the Introduction feel a bit encyclopaedic i.e. they demonstrate an excellent understanding of the natural history of disease progression and epidemiology however some of the detail may not be necessary if it is not part of the model. Therefore, streamlining the Introduction would save you several lines. On the other hand, the authors introduce game theory concepts that are likely not to be familiar to the VL audience and elaborating a little further would save the reader having to follow the citations to understand the basic concepts of game theory.
- Title may need changing depending on the corrected results. The current title for the presented results sounds a bit optimistic given line 478. It also gives the impression that RI-B is the primary finding, plus the use of 'game theory', 'nash equilibrium' keywords alongside,

however the abstract doesn't mention it and only mentions RI-A. I think the main body of the mns does give proper precedence to RI-A and then RI-B as a secondary and consequential finding. In fact, I think in the weight of scientific evidence, that RI-B is weaker than RI-A, as quite broad assumptions are applied to achieve RI-B, whereas RI-A is mostly informed from the published literature; this is elaborated on the attached manuscript lines 87-91. The authors may believe their main contribution to the field is the novel application of RI-B, when in fact I think it could be the robust model developed for RI-A which has the greatest effect. Have any other papers you are aware of estimated an ITN coverage threshold for elimination? I can't think of any. If not then this needs to be emphasised more in the mns as a key finding to highlight the contribution your work is making. Also, as this study is based in the IN/NP country setting, whereas VL also affects East Africa & South America, it may be worth including the regional setting eg "in Bihar, India" in the title.

MINOR

In no particular order:

- Other than suboptimal ITN coverage, other reasons have not been mentioned that could explain the lack of an effect of the Kalanet trial e.g. from [42] "In theory, LNs could be an effective tool to prevent *L. donovani* transmission, as *P. argentipes* are supposed to bite people indoors while they sleep [14]. However, recent entomological findings in India indicate that *L. donovani* vectors are more exophilic and exophagic than previously reported [15,16]. If *P. argentipes* bite people outdoors (e.g., in the early evening when and where bed nets are not deployed), LNs will have a limited impact on *L. donovani* transmission".
- On p3 line 40 you state that ITN ownership is associated with wealth however the assumptions on p4 line 78-79 collapses this to a single mean level for everyone. The heterogeneity of affordability for medical costs produces uncertainty that is hidden in the point estimate $p_{\{HI\}}$. Also, the VL susceptibility is bound to reduce with wealth, as malnutrition and certain interior wall types are VL risk factors.
- Consider adding a PRIME-NTD summary table to your appendix to describe how you have interacted with policymakers.
<https://journals.plos.org/plosntds/article?id=10.1371/journal.pntd.0008033>
- Some readers may interpret "elimination" as "elimination as a public health concern" rather than true/zero elimination. Clarifying this at first use would help.
- Note that eradication means zero disease globally so perhaps you mean elimination which is local to a particular region.
- I note that all the current authors are from US institutions. Although the authors did an excellent job in summarising the pertinent epidemiology and dynamics, I think the addition of authors with in-country experience would have greatly informed on the real-world conditions in Bihar, India. Please note I am basing the authors in-country experience on their institution location which could be false. Journals have rejected NTD manuscripts from my European colleagues last year because they didn't have authors from the affected continent and I have to agree with this policy. I encourage Editors to challenge this in future and the authors to collaborate with in-region researchers as otherwise research quality could suffer. Also it would deprive upcoming researchers in these regions from contributing to research that affects their home.

CODE RUN

Excellent use of commenting and code structuring in the MATLAB file. Believe it or not I did run it and also read through it to try and find bugs! So, thank you very much for spending the time in making the code understandable; it was not wasted time! Please note a commented PDF (Appendix A) of the code is also attached.

MAJOR

Top, most important:

- Code lines 60 & 115 incorrect parameter values. See attached. Editor: please don't mark the paper down for these errors. Think how many papers get published without ever releasing their code!
- Manuscript figs 2 & 3 were not produced when the code was run so not all the figures are reproducible.
- I am not entirely confident that the code runs as the authors may have intended. This is not just because of the initial errors found on code lines 60 & 115 but because I could not find any unit tests in the code to check the correctness of the code output. This could be improved by simple error checks inserted into functions to check function arguments (especially those called multiple times), and checks on the expected value ranges and data structures from each function return, which is essential in my opinion to be confident in the results. This is set out in further detail in Responsible modelling: Unit testing for infectious disease epidemiology (<https://www.sciencedirect.com/science/article/pii/S1755436520300451>). I encourage Prof Rychtář to involve the other co-authors in the collective task of pair code review and mentioning this in the Methods to add to the confidence in the code correctness.

MINOR

In no particular order:

- The file runs in one go and outputs all 41 MATLAB figs to screen and also saves as PDFs; this is good. It is unclear however from the screen, which of the figures relate to which part of the code, unless code chunks are run separately. Therefore, can the `disp()` and `saveFig()` also include a title to match it to the code please. This caused some inefficiency to the reviewing process.
 - Please could you make your publication, SI and code also available on Zenodo or similar archived repo as some journals have a time limit for archive removal.
 - Is MATLAB fig1 = manuscript fig 4? If yes then this has been successfully reproduced. The same goes for manuscript figures 5 {7-10,12,15-17,20-27,18-19,11} & 6 {13-14,29-31,33-38}? where the left-to-right order here refer to their top left to top-right then carriage return order in the sub figure arrays in the manuscript. There were some orphaned MATLAB figures generated {28,32,39-41}. Could these be described by a list of figures and caption for just these orphaned figures in a Supplementary information? Otherwise if they were for the authors' own checking then better to comment out their figure/PDF generation when the code is run as a whole.
- ENDS

Decision letter (RSOS-201960.R0)

Dear Dr Rychtar

The Editors assigned to your paper RSOS-201960 "A voluntary use of insecticide treated nets can help eliminate visceral leishmaniasis" have now received comments from reviewers and would like you to revise the paper in accordance with the reviewer comments and any comments from the Editors. Please note this decision does not guarantee eventual acceptance.

Please submit your revised manuscript and required files (see below) no later than 21 days from today's (ie 01-Mar-2021) date. Note: the ScholarOne system will 'lock' if submission of the revision is attempted 21 or more days after the deadline. If you do not think you will be able to meet this deadline please contact the editorial office immediately.

on behalf of Professor Tim Rogers (Associate Editor) and Mark Chaplain (Subject Editor)
openscience@royalsociety.org

Reviewer comments to Author:

Reviewer: 1

Comments to the Author(s)

This is an interesting and very well written paper about mathematical modeling of individual behavior regarding the use of insecticide-treated nets (ITN) as preventive method against visceral leishmaniasis (VL). The authors present in detail a modification of the mathematical model describing the sand flies-humans dynamics, previously developed by Stauch et al. (2011, PLoS Neglected Tropical Diseases). The authors study the dynamical system within a game-theoretical framework, minimizing the expected cost of using ITN. The main outcome of the model is the proportion of people using ITN that may lead to VL elimination.

COMMENTS ON THE MODELING CHOICES

The manuscript may benefit from reformulating and/or explaining some modeling choices. For instance, in line 86, the authors mention free-riders but don't mention whether or not they model the free-rider behavior explicitly. In line 110, the authors say that equation (2.1) may be interpreted as if "ITNs provide a very effective protection", while their model considers perfect protection, instead. Also, the choice of the computation/definition of R_0 could be supported by a reference.

Assumption (iii) in line 176 could also be supported by a reference. In addition, the authors could contrast this assumption with available data on the population size. If not, they could discuss the limitations of such an assumption.

Please consider discussing the stability of the equilibria, which may give further insight about the system behavior and thus helping to interpret the results, as well as discussing the results of the KalaNet trial (see line 468).

COMMENTS ON NOTATION AND FORMATTING

The authors compute the reproduction number for the system subject to a control measure. Therefore, it is the effective reproduction number and not the basic reproduction number. The authors could use the notation R_e , instead of R_0 , where $R_e = R_e(p)$.

Please recall the parameters values used to produce figure 4, in the caption. Please expand the description of figures 5 and 6 in their captions, including a short description of the parameters set depicted in the x-axes.

Please ensure notation consistency for numbers of 4 or more digits.

Reviewer: 2

Comments to the Author(s)

SUMMARY

My recommendation to the Editor is to accept this manuscript, pending major corrections. Please interpret ‘major’ in the sense that these suggestions will likely affect the results. I thought the quality of the research and its write-up were of a high standard for this modelling manuscript and would very much like to see its corrected version added to the VL research base.

The findings: The authors have developed a model by amending an existing one and parametrising it with mostly reasonable values and ranges. The two research impacts are:

A) They obtain a threshold estimate for the ITN coverage required for (zero) disease elimination. (labelling this as RI-A)

B) Considering that ITN use is voluntary, they make a sensible and (probably for VL) novel application of game theory to assess if voluntary ITN use alone, influenced by the costs and benefits at the individual level, is sufficient to meet this threshold. (RI-B)

Quality: It is of a high research standard: indicated by the mathematical foundations and clear exposition in the text, strongly informed in model structure and parameters by existing VL research, the rigour of the analysis including validation and sensitivity analysis. Moreover, it is the first manuscript I have reviewed which provides working code and reproducible results! It therefore ticks many best practices of 21st century modelling.

Potential impact: It also contributes to the ongoing discussion following the unfortunate findings of the Kalanet trial in the previous decade. This is the primary reason why ITNs do not form a part of today’s VL control. WHO has launched the NTD 2021-2030 roadmap earlier in 2021 and so this could be an ideal time for research during this receptive policy window. It could reawaken minds and inform future epidemiological and entomological study designs to definitively answer if ITNs have a contributing role in VL control on the Indian subcontinent. The manuscript could work harder at highlighting this point for the VL policy maker audience.

To open up this research piece to further discussion and improve its publication impact I suggest releasing the initially submitted draft as a preprint and later interacting with the VL research community at forthcoming conferences such as WorldLeish 2022 which will probably start accepting abstracts later this year, but this of course is at the authors’ discretion.

Scope of this review: I spent 3 days reviewing the manuscript, code and writing up this review. I did not check the majority of the mathematical formulae at an analytical level. Instead I focussed on code correctness from code lines 1-478, parameter suitability and if this modelling was appropriate for VL.

Conflict of interest: I declare that I have included citations within this review which either myself or my supervisors (T Déirdre Hollingsworth & Lloyd Chapman) are co-authors. The authors are not compelled to add these citations. I included them to provide justification and to encourage the authors why a suggestion is necessary or why it has been raised.

Current major limitations: I believe there are some major bugs in the code that would affect results but these should be quick to fix. Some coding practices may also help improve the confidence in the code correctness. Some parameters may not fit the setting or understanding of the disease the best and I have highlighted the ones most likely to impact results, partly from the authors' helpful sensitivity analysis; this may require a rapid review of the recent literature for parameter updates but again this shouldn't take long. No change to the model structure is necessary. Finally, although the manuscript text itself was painless to read, it could focus better at the main audience – the VL research community. Some of these are detailed below and are also attached on commented files.

BEFORE THE AUTHORS READ THIS REVIEW (OR THE PUBLIC, AFTER PUBLICATION)

My overriding motivation for this peer review is to protect the field from errors in research methods or misinterpretation of results, that could make policy makers make suboptimal decisions in VL programmes that ultimately harm people. The annual research output for VL modelling is relatively small compared to other diseases, so you can realise that avoidable errors could have more impact in this context than others. I haven't seen the authors' names from previous VL modelling research, so if you are new to VL then welcome to the field! None of the following review is intended to 'put you in your place' nor be patronising, and your future modelling for this neglected tropical disease is very much welcomed. In the spirit of open science some of the reviewers may have declared their names to be made public and in return I hope the authors allow all reviews to be open (for reviewers who allow it).

REVIEW STRUCTURE

This review is structured by the manuscript then code run, and grouped by things essential to impacting the results' values or presentation (major) or affecting the understanding/ model improvements. This review is long but I believe in thoroughness and once some of what I suggest is corrected, then I think what you have got here could be really valuable to the VL field.

MANUSCRIPT

MAJOR

Top, most important

- An extensive range of research has been drawn upon to inform parameters selected. Some suggestions have been made from empirical sources for the refinements of some parameters. See mns attachment for more details.
- "Multiple lab-based studies suggest that female sand flies have fairly short adult lifespans (<20 days) [16] with further reductions when infected [15]" (<https://journals.plos.org/plosntds/article?id=10.1371/journal.pntd.0009033>) therefore it is unrealistic to assume the mortality of I_F in Fig 1 is the same as S_F or E_F (=mu_F).
- Making it more policy-friendly: reduce the amount of mathematical formulae by moving formulae (lines 212-223 & 446-452) describing the intermediate mathematical logic to a Supplementary information as there's quite a few pages to flip through from line 185-225. Please do not understand what I mean, I just mean reduce the amount, the level at which you are communicating is fine and there is no need to dumb things down. Your short-hand notation like Comp on line 197 was clever and for instance Eqns 3.1-3.15 could be represented as a one-liner, something like $dX/dt = 0$ where $X=\{\text{etc}\}$.
- Although the Discussion covers limitations in the findings, model limitations are also necessary. Also despite the model being validated against previous trial data, what would be the external validity of these findings to say Nepal?

- Parts of the Introduction feel a bit encyclopaedic i.e. they demonstrate an excellent understanding of the natural history of disease progression and epidemiology however some of the detail may not be necessary if it is not part of the model. Therefore, streamlining the Introduction would save you several lines. On the other hand, the authors introduce game theory concepts that are likely not to be familiar to the VL audience and elaborating a little further would save the reader having to follow the citations to understand the basic concepts of game theory.
- Title may need changing depending on the corrected results. The current title for the presented results sounds a bit optimistic given line 478. It also gives the impression that RI-B is the primary finding, plus the use of 'game theory', 'nash equilibrium' keywords alongside, however the abstract doesn't mention it and only mentions RI-A. I think the main body of the mns does give proper precedence to RI-A and then RI-B as a secondary and consequential finding. In fact, I think in the weight of scientific evidence, that RI-B is weaker than RI-A, as quite broad assumptions are applied to achieve RI-B, whereas RI-A is mostly informed from the published literature; this is elaborated on the attached manuscript lines 87-91. The authors may believe their main contribution to the field is the novel application of RI-B, when in fact I think it could be the robust model developed for RI-A which has the greatest effect. Have any other papers you are aware of estimated an ITN coverage threshold for elimination? I can't think of any. If not then this needs to be emphasised more in the mns as a key finding to highlight the contribution your work is making. Also, as this study is based in the IN/NP country setting, whereas VL also affects East Africa & South America, it may be worth including the regional setting eg "in Bihar, India" in the title.

MINOR

In no particular order:

- Other than suboptimal ITN coverage, other reasons have not been mentioned that could explain the lack of an effect of the Kalanet trial e.g. from [42] "In theory, LNs could be an effective tool to prevent *L. donovani* transmission, as *P. argentipes* are supposed to bite people indoors while they sleep [14]. However, recent entomological findings in India indicate that *L. donovani* vectors are more exophilic and exophagic than previously reported [15,16]. If *P. argentipes* bite people outdoors (e.g., in the early evening when and where bed nets are not deployed), LNs will have a limited impact on *L. donovani* transmission".
- On p3 line 40 you state that ITN ownership is associated with wealth however the assumptions on p4 line 78-79 collapses this to a single mean level for everyone. The heterogeneity of affordability for medical costs produces uncertainty that is hidden in the point estimate $p_{\{HI\}}$. Also, the VL susceptibility is bound to reduce with wealth, as malnutrition and certain interior wall types are VL risk factors.
- Consider adding a PRIME-NTD summary table to your appendix to describe how you have interacted with policymakers.
<https://journals.plos.org/plosntds/article?id=10.1371/journal.pntd.0008033>
- Some readers may interpret "elimination" as "elimination as a public health concern" rather than true/zero elimination. Clarifying this at first use would help.
- Note that eradication means zero disease globally so perhaps you mean elimination which is local to a particular region.
- I note that all the current authors are from US institutions. Although the authors did an excellent job in summarising the pertinent epidemiology and dynamics, I think the addition of authors with in-country experience would have greatly informed on the real-world conditions in Bihar, India. Please note I am basing the authors in-country experience on their institution location which could be false. Journals have rejected NTD manuscripts from my European colleagues last year because they didn't have authors from the affected continent and I have to agree with this policy. I encourage Editors to challenge this in future and the authors to collaborate with in-region researchers as otherwise research quality could suffer. Also it would

deprive upcoming researchers in these regions from contributing to research that affects their home.

CODE RUN

Excellent use of commenting and code structuring in the MATLAB file. Believe it or not I did run it and also read through it to try and find bugs! So, thank you very much for spending the time in making the code understandable; it was not wasted time! Please note a commented PDF of the code is also attached.

MAJOR

Top, most important:

- Code lines 60 & 115 incorrect parameter values. See attached. Editor: please don't mark the paper down for these errors. Think how many papers get published without ever releasing their code!
- Manuscript figs 2 & 3 were not produced when the code was run so not all the figures are reproducible.
- I am not entirely confident that the code runs as the authors may have intended. This is not just because of the initial errors found on code lines 60 & 115 but because I could not find any unit tests in the code to check the correctness of the code output. This could be improved by simple error checks inserted into functions to check function arguments (especially those called multiple times), and checks on the expected value ranges and data structures from each function return, which is essential in my opinion to be confident in the results. This is set out in further detail in Responsible modelling: Unit testing for infectious disease epidemiology (<https://www.sciencedirect.com/science/article/pii/S1755436520300451>). I encourage Prof Rychtář to involve the other co-authors in the collective task of pair code review and mentioning this in the Methods to add to the confidence in the code correctness.

MINOR

In no particular order:

- The file runs in one go and outputs all 41 MATLAB figs to screen and also saves as PDFs; this is good. It is unclear however from the screen, which of the figures relate to which part of the code, unless code chunks are run separately. Therefore, can the disp() and saveFig() also include a title to match it to the code please. This caused some inefficiency to the reviewing process.
- Please could you make your publication, SI and code also available on Zenodo or similar archived repo as some journals have a time limit for archive removal.
- Is MATLAB fig1 = manuscript fig 4? If yes then this has been successfully reproduced. The same goes for manuscript figures 5 {7-10,12,15-17,20-27,18-19,11} & 6 {13-14,29-31,33-38}? where the left-to-right order here refer to their top left to top-right then carriage return order in the sub figure arrays in the manuscript. There were some orphaned MATLAB figures generated {28,32,39-41}. Could these be described by a list of figures and caption for just these orphaned figures in a Supplementary information? Otherwise if they were for the authors' own checking then better to comment out their figure/PDF generation when the code is run as a whole.

ENDS=====

===PREPARING YOUR MANUSCRIPT===

===PREPARING YOUR REVISION IN SCHOLARONE===

- If you are providing image files for potential cover images, please upload these at this step, and inform the editorial office you have done so. You must hold the copyright to any image provided.
- A copy of your point-by-point response to referees and Editors. This will expedite the preparation of your proof.

- Ensure that your data access statement meets the requirements at <https://royalsociety.org/journals/authors/author-guidelines/#data>. You should ensure that you cite the dataset in your reference list. If you have deposited data etc in the Dryad repository, please include both the 'For publication' link and 'For review' link at this stage.
- If you are requesting an article processing charge waiver, you must select the relevant waiver option (if requesting a discretionary waiver, the form should have been uploaded at Step 3 'File upload' above).
- If you have uploaded ESM files, please ensure you follow the guidance at <https://royalsociety.org/journals/authors/author-guidelines/#supplementary-material> to include a suitable title and informative caption. An example of appropriate titling and captioning may be found at https://figshare.com/articles/Table_S2_from_Is_there_a_trade-off_between_peak_performance_and_performance_breadth_across_temperatures_for_aerobic_scope_in_teleost_fishes_/3843624.

Author's Response to Decision Letter for (RSOS-201960.R0)

See Appendix B.

RSOS-201960.R1 (Revision)

Review form: Reviewer 1

Is the manuscript scientifically sound in its present form?

Yes

Are the interpretations and conclusions justified by the results?

Yes

Is the language acceptable?

Yes

Do you have any ethical concerns with this paper?

No

Have you any concerns about statistical analyses in this paper?

No

Recommendation?

Accept with minor revision (please list in comments)

Comments to the Author(s)

The manuscript, the supplementary material and the code have improved importantly with the revision. I believe that this article may still benefit from minor revisions regarding mostly display and notation, and that can be suitable for publication after discussing a few major points, mentioned below.

I would also like to acknowledge how the detailed comments of reviewer #2 have contributed importantly to the material.

MAJOR COMMENTS

Please consider commenting further the (in)stability of the system equilibria in the Discussion section. In lines 292-296, the authors mentioned explicitly the relation between the effective reproduction number and the endemic and disease-free states. While this concept may be well-known for the modeler readers, the remainder may be appreciated by the readers with less modeling background. However, I believe it still remains to be discussed how the stability of the system changes when the game interacts with the ODE system.

In that regard, the authors may consider discussing further sections 5c and 5d, and how their results interact. The game being coupled to the ODE system, the ITN usage becomes a function of the other parameters, notably the costs. However, the authors made the modeling choice of working with fixed costs to discuss the current epidemic situation. Hence, the outcomes of the model have not been read from a broader perspective of the dynamical system. This is not a problem, provided that is carefully discussed and mentioned as a limitation of the study.

For instance, the behavior around $R_e = 1$ and beyond (in the sense of $p > p_{NE}$) might be important to interpret to discuss the study results, and might also shed light into the results of the KalaNet trial. Indeed, the closer you get to $R_e = 1$, the longer it takes to get to the equilibrium (and so, the harder it takes to reach elimination), which seems to be the situation described in lines 584 and 585. Moreover, the disease-free state (reached when $p > p_{NE}$) may lose its stability when the ODE system is coupled to the game, which may impact the interpretation of elimination and thus, the discussion around elimination.

In agreement with comments from Reviewer 2, ensuring that the difference between elimination (as in $R_e < 1$) and elimination as a public health concern is clear is a priority; as well as ensuring that the purpose of the study is reflected in the title, the abstract and the discussion.

MINOR COMMENTS

Figure 6. Please add units (USD) for costs

Consider using a curly or \mathbb{P} for probabilities in section 4e

Review form: Reviewer 2 (Tim Pollington)

Is the manuscript scientifically sound in its present form?

Yes

Are the interpretations and conclusions justified by the results?

Yes

Is the language acceptable?

Yes

Do you have any ethical concerns with this paper?

No

Have you any concerns about statistical analyses in this paper?

No

Recommendation?

Accept with minor revision (please list in comments)

Comments to the Author(s)

Please see 2 attached PDFs (Appendices C & D) for these comments.

Decision letter (RSOS-201960.R1)

Dear Dr Rychtar

On behalf of the Editors, we are pleased to inform you that your Manuscript RSOS-201960.R1 "Mathematical modeling of the use of insecticide treated nets for elimination of visceral leishmaniasis in Bihar, India" has been accepted for publication in Royal Society Open Science subject to minor revision in accordance with the referees' reports. Please find the referees' comments along with any feedback from the Editors below my signature.

Please submit your revised manuscript and required files (see below) no later than 7 days from today's (ie 04-May-2021) date. Note: the ScholarOne system will 'lock' if submission of the revision is attempted 7 or more days after the deadline. If you do not think you will be able to meet this deadline please contact the editorial office immediately.

on behalf of Professor Tim Rogers (Associate Editor) and Mark Chaplain (Subject Editor)
openscience@royalsociety.org

Reviewer comments to Author:

Reviewer: 2

Comments to the Author(s)

Please see 2 attached PDFs for these comments.

Reviewer: 1

Comments to the Author(s)

The manuscript, the supplementary material and the code have improved importantly with the revision. I believe that this article may still benefit from minor revisions regarding mostly display and notation, and that can be suitable for publication after discussing a few major points, mentioned below.

I would also like to acknowledge how the detailed comments of reviewer #2 have contributed importantly to the material.

MAJOR COMMENTS

Please consider commenting further the (in)stability of the system equilibria in the Discussion section. In lines 292-296, the authors mentioned explicitly the relation between the effective reproduction number and the endemic and disease-free states. While this concept may be well-known for the modeler readers, the remainder may be appreciated by the readers with less modeling background. However, I believe it still remains to be discussed how the stability of the system changes when the game interacts with the ODE system.

In that regard, the authors may consider discussing further sections 5c and 5d, and how their results interact. The game being coupled to the ODE system, the ITN usage becomes a function of the other parameters, notably the costs. However, the authors made the modeling choice of working with fixed costs to discuss the current epidemic situation. Hence, the outcomes of the model have not been read from a broader perspective of the dynamical system. This is not a problem, provided that is carefully discussed and mentioned as a limitation of the study.

For instance, the behavior around $R_e = 1$ and beyond (in the sense of $p > p_{NE}$) might be important to interpret to discuss the study results, and might also shed light into the results of the KalaNet trial. Indeed, the closer you get to $R_e = 1$, the longer it takes to get to the equilibrium (and so, the harder it takes to reach elimination), which seems to be the situation described in lines 584 and 585. Moreover, the disease-free state (reached when $p > p_{NE}$) may lose its stability when the ODE system is coupled to the game, which may impact the interpretation of elimination and thus, the discussion around elimination.

In agreement with comments from Reviewer 2, ensuring that the difference between elimination (as in $R_e < 1$) and elimination as a public health concern is clear is a priority; as well as ensuring that the purpose of the study is reflected in the title, the abstract and the discussion.

MINOR COMMENTS

Figure 6. Please add units (USD) for costs

Consider using a curly or \mathbb{P} for probabilities in section 4e

===PREPARING YOUR MANUSCRIPT===

===PREPARING YOUR REVISION IN SCHOLARONE===

Author's Response to Decision Letter for (RSOS-201960.R1)

See Appendices E & F.

Decision letter (RSOS-201960.R2)

Dear Dr Rychtar,

I am pleased to inform you that your manuscript entitled "Mathematical modeling of the use of insecticide treated nets for elimination of visceral leishmaniasis in Bihar, India" is now accepted for publication in Royal Society Open Science.

on behalf of Professor Tim Rogers (Associate Editor) and Mark Chaplain (Subject Editor)
openscience@royalsociety.org

Appendix A**ROYAL SOCIETY
OPEN SCIENCE****A voluntary use of insecticide treated nets can help
eliminate visceral leishmaniasis**

Journal:	Royal Society Open Science
Manuscript ID	RSOS-201960
Article Type:	Research
Date Submitted by the Author:	03-Nov-2020
Complete List of Authors:	Fortunato, Anna; University of Richmond Glasser, Casey; Virginia Tech Watson, Joy; Virginia State University Lu, Yongjin; Virginia State University Rychtar, Jan; Virginia Commonwealth University, Mathematics and Applied Mathematics Taylor, Dewey; Virginia Commonwealth University, Mathematics and Applied Mathematics
Subject:	Mathematical modelling < MATHEMATICS
Keywords:	Kala-azar, PKDL, Nash equilibrium, Game theory, Vector-borne diseases
Subject Category:	Mathematics

Author-supplied statements

Relevant information will appear here if provided.

Ethics

Does your article include research that required ethical approval or permits?:

This article does not present research with ethical considerations

Statement (if applicable):

CUST_IF_YES_ETHICS :No data available.

Data

It is a condition of publication that data, code and materials supporting your paper are made publicly available. Does your paper present new data?:

My paper has no data

Statement (if applicable):

CUST_IF_YES_DATA :No data available.

Conflict of interest

I/We declare we have no competing interests

Statement (if applicable):

CUST_STATE_CONFLICT :No data available.

Authors' contributions

This paper has multiple authors and our individual contributions were as below

Statement (if applicable):

A.K.F.: conceptualization, formal analysis, investigation, and writing - original draft

C.P.G: conceptualization, formal analysis, investigation, software, and writing - original draft

J.A.W. - conceptualization, formal analysis, investigation, and writing - original draft

Y.L. - supervision, writing - original draft

J.R. - writing - original draft, writing - review and editing, software, methodology, supervision, conceptualization, resources

D.T. - writing - original draft, writing - review and editing, methodology, supervision, conceptualization, formal analysis, validation, funding acquisition

The order of the authors was determined as follows: The first three authors are undergraduate students in alphabetical order. The last three authors are professors in alphabetical order.

ROYAL SOCIETY OPEN SCIENCE

rsos.royalsocietypublishing.org

Research

Article submitted to journal

Subject Areas:

mathematics, biology, epidemiology

Keywords:

Kala-azar, PKDL, Nash equilibrium,
Game theory, Vector-borne diseases

Author for correspondence:

Jan Rychtář

e-mail: rychtarj@vcu.edu

A voluntary use of insecticide treated nets can help eliminate visceral leishmaniasis

Anna K Fortunato¹, Casey P Glasser², Joy A Watson³, Yongjin Lu³, Jan Rychtář⁴, Dewey Taylor⁴

¹ Department of Mathematics, University of Richmond, Richmond, VA 23173, USA

² Department of Mathematics, Virginia Tech, Blacksburg, VA 24061-1026, USA

³ Department of Mathematics and Economics, Virginia State University, Petersburg, VA 23806, USA

⁴ Department of Mathematics and Applied Mathematics, Virginia Commonwealth University, Richmond, VA 23284-2014, USA

Visceral leishmaniasis (VL) is a deadly neglected tropical disease caused by a parasite *Leishmania donovani* and spread by female sand flies *Phlebotomus argentipes*. There is conflicting evidence regarding the role of insecticide treated nets (ITNs) on the prevention of VL. Numerous studies demonstrated the effectiveness of ITNs. However, KalaNet, a large trial in Nepal and India did not support those findings. The purpose of this paper is to gain insight into the situation by mathematical modelling. We expand a mathematical model of VL transmission based on the KalaNet trial and incorporate the use of ITNs explicitly into the model. One of the major contributions of this work is that we calibrate the model based on the available epidemiological data, generally independent of the KalaNet trial. We validate the model on data collected during the KalaNet trial. We conclude that in order to eliminate VL, the ITN usage would have to be about 96%. This is higher than the 91% ITNs use at the end of the trial which may explain why the trial did not show a positive effect from ITNs. At the same time, our model indicates that coupling ITN use with other existing intervention methods should be an effective way to reduce VL incidence.

© 2014 The Authors. Published by the Royal Society under the terms of the Creative Commons Attribution License <http://creativecommons.org/licenses/by/4.0/>, which permits unrestricted use, provided the original author and source are credited.

THE ROYAL SOCIETY
PUBLISHING

1. Introduction

Leishmaniasis is a vector borne disease caused by protozoan parasites of genus *Leishmania* and transmitted by phlebotomine sand flies [1]. Visceral leishmaniasis (VL) is the most serious form of the disease and can be fatal in 95% of cases if left untreated [2,3]. VL is responsible for around 500,000 infections and 51,000 deaths per year, worldwide, deeming it second only to malaria in numbers of fatalities [4]. Leishmaniasis is endemic and presents a global health problem in 98 countries [5]. Over 90% of new cases occur in India, Ethiopia, Kenya, Somalia, Sudan, South Sudan, and Brazil [6]. The Indian subcontinent accounts for two-thirds of the total global cases, of which more than 50% occur in the state of Bihar, India [7] where VL is known as “Kala-azar” (Hindi for “black fever”). It has been targeted for elimination with the current goal of reducing the incidence of VL to below 1/10,000 by the year 2020 [8]. The elimination efforts are working [9] although the targets are now being reformulated for 2030 [10].

In the Indian subcontinent, VL is caused by parasites belonging to *Leishmania donovani* complex [8]. It is transmitted from human to human by female sand fly *Phlebotomous argentipes*, without any known animal reservoirs [11]. The sand flies are active and feed during the night with the female activity peaking just before midnight [12]. They normally seek shelter in animal burrows, or other protected areas [13] and thrive in poor housing conditions [14,15]. They are generally weak flyers and usually fly close to the ground in short hops [16]. At the same time, there are indications that they are capable of longer and more sustained flight and may be more exophilic and exophagic than previously reported [17].

In humans, the parasite infects the reticuloendothelial system, causing persistent fever and anemia [6,14] and affecting several internal organs, usually the spleen, liver, and bone marrow [18]. Because the symptoms persist, the individuals typically seek treatment, especially in Bihar where treatment is available [19]. However, the social and cultural stigma linked to VL results in a large percentage of individuals seeking treatment at private rather than public health facilities [20] which results in under-reporting true incidence and prevalence of the disease [2]. After recovery from acute illness, about 5–10% of patients develop a chronic cutaneous form called post kala-azar dermal leishmaniasis (PKDL) [11]. Moreover, a few PKDL patients have had no history of VL [21]. Since PKDL is not a life-threatening condition and the treatment can be very burdensome and unpleasant, many PKDL patients remain untreated [22]. Because of the anthroponotic nature of the transmission of *L. donovani* in the Indian subcontinent [11], the PKDL patients used to be considered reservoirs of infection [23]. At the same time, the role of PKDL and asymptomatic individuals in transmission is still unclear [24].

The control of VL depends on chemotherapy treatment [11], vector control [25], and bite prevention [26]. Human VL vaccines are not yet fully developed although several trials are under way [27,28] and their impact is already modelled [29]. Preventative measures in high risk areas include (1) avoiding sleeping in mud and thatched housing [1], (2) environmental management [30], (3) using insect repellent [31], (4) using insecticide treated netting, ITN, [32–34] and, most recently, (5) treated wall lining [35].

The ITN ownership varies significantly with wealth; almost all of the wealthiest households own an ITN while many of the poorest do not [36]. However, even in relatively poor areas, the use of ITNs was already demonstrated to be very cost-effective [37,38]. Human behavior such as inconsistent use due to hot weather or inadequate education diminishes the effectiveness of ITNs [34,39].

KalaNet, a cluster randomised controlled trial in Nepal and India evaluated the efficacy of long lasting ITNs in the prevention of VL [40]. It has been demonstrated that a cluster-wide distribution of ITNs reduced the vector density [41] but did not reduce the risk of *L. donovani* infection or clinical VL [42]. Other trials in Bangladesh showed that ITNs may reduce the VL incidence rate [35,43]. This apparent contradiction raises the question about the role that ITNs may play in controlling VL [42]. The purpose of this paper is to gain insight into the situation by mathematical modelling.

There are many mathematical models of VL dynamics, see for example [44–47] for recent
reviews. A model of VL transmission at the district-level of Bihar to estimate the basic
reproduction number was created in [2]. Different vector control measures for VL elimination
were considered in [48]. A multi-state Markov model of VL was developed in [49]; an individual-
based, stochastic, stage-structured model of a temperature-driven sand fly population was
developed in [50] who also simulated the effects of the use of drugs administered to cattle on the
vector control. A set of three age-structured model variants based on [51], each with individuals
from a different disease stage being the main contributors to transmission: asymptomatic
individuals, previously immune individuals in whom infection has reactivated, and individuals
with PKDL was created in [52]. The cost effectiveness of different drug treatments was studied in
[53] and [54]. On top of the above mathematical models of anthropolomorphic VL relevant to India,
there are many models of zoonotic VL that incorporate animal reservoirs as well, see for example
[3,4,55,56].

[revised manuscript text omitted]

2, 3, 4 and 5. The parameter values are estimated in the section Model calibration based on
empirical evidence and data from the literature.

The compartments are denoted S (susceptible), I (infected and potentially infectious), R
(recovering), and E (exposed). Subscript F denotes sand flies. For the human compartments we
use the subscript P for PCR-positive, D for DAT-positive, C for cellular immunity (LST-positive),
S for symptomatic, T for treated, U for untreated and L for PKDL. Greek letters stand for the
rates. Lower case roman letters stand for probabilities/proportions of the populations.

(c) Differences between our model and the original model in [51]

Aside from slight differences in the notation, we made the following changes and additions to the
original model from [51].

- (i) We explicitly added a parameter p signifying the protection against sand fly's bites by
using ITNs.
 - (ii) We separated PKDL cases (I_{HL} in [51]) into treated (I_{TL}) and untreated (I_{UL}) cases to
better account for the different duration of the stages and infectivity of the cases.
 - (iii) We assume that the human population size is constant with a birth rate Λ (as opposed
to $\alpha_H N_H$ considered in [51]). Doing so allows the human population to converge to a
constant and we can thus solve for the equilibria of the dynamics.
 - (iv) We keep the exponential growth of sand flies, denoted $\alpha_F N_F$ in [51], but we stipulate
that the growth rate $\alpha_F = \mu_F$ in order to keep the population size constant.
 - (v) We consider the death rate of treated individuals to be $\mu_1 = \mu + d_{T1}\tau_1$ and $\mu_2 = \mu +$
$d_{T2}\tau_2$, instead of $\mu_1 = \mu_2 = \mu + \mu_K + f_T\tau_1$. Specifically, we do not include the death rate
of untreated individuals (μ_K) and we consider the rates different in a different lines of
treatment.

Figure 1. Scheme of the ODE model for VL transmission. Based on [51].

(d) ODEs for the model

The model of the transmission dynamics described above yields the following system of ordinary differential equations.

$$\frac{dS}{dt} = \Lambda + \rho_C R_C - (\mu + \lambda)S \tag{2.3}$$

$$\frac{dI_P}{dt} = \lambda S - (\mu + \gamma_P)I_P \tag{2.4}$$

$$\frac{dI_D}{dt} = \gamma_P I_P - (\mu + \gamma_D)I_D \tag{2.5}$$

$$\frac{dR_D}{dt} = f_D \gamma_D I_D - (\mu + \rho_D)R_D \tag{2.6}$$

$$\frac{dI_S}{dt} = f_S \gamma_D I_D - (\mu_S + \gamma_S)I_S \tag{2.7}$$

$$\frac{dI_{T1}}{dt} = \gamma_S I_S - (\mu_1 + \tau_1)I_{T1} \tag{2.8}$$

$$\frac{dI_{T2}}{dt} = s_1 \tau_1 I_{T1} - (\mu_2 + \tau_2)I_{T2} \tag{2.9}$$

$$\frac{dR_L}{dt} = s_2 \tau_1 I_{T1} + s_4 \tau_2 I_{T2} + f_L \gamma_D I_D - (\mu + \rho_L)R_L \tag{2.10}$$

$$\frac{dI_{TL}}{dt} = p_{TL} \rho_L R_L - (\mu + \tau_3)I_{TL} \tag{2.11}$$

$$\frac{dI_{UL}}{dt} = (1 - p_{TL}) \rho_L R_L - (\mu + \gamma_L)I_{UL} \tag{2.12}$$

$$\frac{dR_T}{dt} = s_3 \tau_1 I_{T1} + s_5 \tau_2 I_{T2} + \tau_3 I_{TL} + \gamma_L I_{UL} - (\mu + \rho_T)R_T \tag{2.13}$$

$$\frac{dR_C}{dt} = \rho_D R_D + \rho_T R_T - (\mu + \rho_C)R_C \tag{2.14}$$

$$\frac{dS_F}{dt} = \mu_F N_F - (\mu_F + \lambda_F)S_F \tag{2.15}$$

$$\frac{dE_F}{dt} = \lambda_F S_F - (\mu_F + \sigma_F)E_F \tag{2.16}$$

$$\frac{dI_F}{dt} = \sigma_F E_F - \mu_F I_F \tag{2.17}$$

3. Analysis

(a) Equilibria of the ODE system

The equilibria of the dynamics (2.3) - (2.17) are found by solving the following system of algebraic
equations.

Table 1. Notation - Compartments

Notation	Meaning
N	Total number of humans
S	Susceptible humans (PCR-, DAT-, LST-)
I_P	Early asymptomatic (PCR+, DAT-, LST-)
I_D	Late asymptomatic (PCR+, DAT+, LST-)
I_S	Symptomatic Kala-azar (PCR+, DAT+, LST-)
I_{T1}	Receiving first line of treatment (PCR+, DAT+, LST-)
I_{T2}	Receiving second line of treatment (PCR+, DAT+, LST-)
I_{TL}	Treated cases with PKDL (PCR+, DAT+, LST-)
I_{TL}	Untreated cases with PKDL (PCR+, DAT+, LST-)
R_{\sim}	Recovering asymptomatic (PCR-, DAT+, LST-)
R_{\sim}	Recovering after treatment (PCR-, DAT+, LST-)
R_C	Recovered (PCR-, DAT-, LST+)
R_L	Cases with dormant infection (PCR-, DAT+, LST-)
N_F	Total number of sand flies ($N_F = n_F \frac{\Delta}{\mu}$)
S_F	Susceptible sand flies
E_F	Exposed sand flies
I_F	Infected sand flies

Table 2. Parameter values - sand flies. Times are in days.

Symbol	Description	Value	Range	Reference
n_F	Number of sand flies per human	3	[2.1, 3.5]	[7, Table S15]
β^{-1}	Duration of feeding cycle	4	[2, 8]	[80,81]
σ_F^{-1}	Sojourn time in latent stage E_F	5	[3, 7]	[82]
μ_F^{-1}	Life expectancy of sand flies	14	[6, 31]	[83]
i_F	Probability that a bite by an infected sand fly infects a susceptible human	1	[0,1]	[51]

$$0 = \Lambda + \rho_C R_C - (\mu + \lambda)S \quad (3.1)$$

$$0 = \lambda S - (\mu + \gamma_P)I_P \quad (3.2)$$

$$0 = \gamma_P I_P - (\mu + \gamma_D)I_D \quad (3.3)$$

$$0 = f_D \gamma_D I_D - (\mu + \rho_D)R_D \quad (3.4)$$

$$0 = f_S \gamma_D I_D - (\mu_S + \gamma_S)I_S \quad (3.5)$$

$$0 = \gamma_S I_S - (\mu_1 + \tau_1)I_{T1} \quad (3.6)$$

$$0 = s_1 \tau_1 I_{T1} - (\mu_2 + \tau_2)I_{T2} \quad (3.7)$$

$$0 = s_2 \tau_1 I_{T1} + s_4 \tau_2 I_{T2} + f_L \gamma_D I_D - (\mu + \rho_L)R_L \quad (3.8)$$

$$0 = p_{TL} \rho_L R_L - (\mu + \tau_3)I_{TL} \quad (3.9)$$

$$0 = (1 - p_{TL}) \rho_L R_L - (\mu + \gamma_L)I_{UL} \quad (3.10)$$

$$0 = s_3 \tau_1 I_{T1} + s_5 \tau_2 I_{T2} + \tau_3 I_{TL} + \gamma_L I_{UL} - (\mu + \rho_T)R_T \quad (3.11)$$

$$0 = \rho_D R_D + \rho_T R_T - (\mu + \rho_C)R_C \quad (3.12)$$

$$0 = \mu_F N_F - (\mu_F + \lambda_F)S_F \quad (3.13)$$

$$0 = \lambda_F S_F - (\mu_F + \sigma_F)E_F \quad (3.14)$$

$$0 = \sigma_F E_F - \mu_F I_F \quad (3.15)$$

Table 3. Parameter values - humans. Times are in months. Rates are per capita per month.

Symbol	Description	Value	Range	Reference
Λ	Recruitment rate	$\frac{0.0182}{12}$	$[\frac{0.0105}{12}, \frac{0.0333}{12}]$	[84]
μ^{-1}	Life expectancy	$70 \cdot 12$	$[50 \cdot 12, 90 \cdot 12]$	[85]
μ_K^{-1}	Life expectancy with untreated symptomatic VL	30	[5, 36]	[2]
γ_P^{-1}	Sojourn time in I_P	5	[4, 6]	[86]
γ_D^{-1}	Sojourn time in I_D	1.13	[0.5, 4]	[86]
γ_S^{-1}	Time between onset of symptoms to the start of treatment	1	[0.5, 2.5]	[87]
γ_L^{-1}	Mean duration of the stage I_{UL}	55.5	$[32, 16 \cdot 12]$	[22]
ρ_C^{-1}	Duration of LST-positivity in R_C	33	[10, 38]	[59]
ρ_D^{-1}	Duration of DAT-positivity in R_D	6	[4, 8]	[88]
ρ_T^{-1}	Duration of DAT-positivity in R_T	6		ρ_D^{-1}
p	Fraction of the population using ITNs	0.7	[0,1]	[32]
f_L	Fraction of I_D who become dormant	$5.5 \cdot 10^{-4}$	$[10^{-4}, 10^{-3}]$	[21,22]
f_S	Fraction of I_D who develop symptomatic VL	0.035	[0.01, 0.15]	[22]
f_D	Fraction of I_D who recover without showing symptoms	0.9384		$1 - (f_S + f_L)$
i_P	Probability that an individual in I_P infects a feeding sand fly	0.024	[0,1]	Estimated
i_D	Probability that an individual in I_D infects a feeding sand fly	0.1	[0,1]	Estimated
i_S	Probability that an individual in I_S infects a feeding sand fly	0.475	[0,1]	Estimated
i_{T1}	Probability that an individual in I_{T1} infects a feeding sand fly	0	[0,1]	[24]
i_{T2}	Probability that an individual in I_{T2} infects a feeding sand fly	0	[0,1]	[24]
i_{TL}	Probability that an individual in I_{TL} infects a feeding sand fly	0	[0,1]	[24]
i_{UL}	Probability that an individual in I_{UL} infects a feeding sand fly	0.4325	[0,1]	Estimated

rsos.royalsocietypublishing.org R. Soc. open sci. 0000000

There are two equilibria; the disease-free equilibrium denoted by

$$E^0 = (S^0, I_P^0, I_D^0, R_D^0, R_C^0, I_S^0, I_{T1}^0, I_{T2}^0, I_{TL}^0, I_{UL}^0, R_T^0, R_C^0, S_F^0, E_F^0, I_F^0) \quad (3.16)$$

and the endemic equilibrium given by

$$E^* = (S^*, I_P^*, I_D^*, R_D^*, I_S^*, I_{T1}^*, I_{T2}^*, R_L^*, I_{TL}^*, I_{UL}^*, R_T^*, R_C^*, S_F^*, E_F^*, I_F^*). \quad (3.17)$$

The system (3.1) - (3.15) will be solved as follows, corresponding to starting at compartment S
 and going downstream. By (3.2), and since $\lambda = (1 - p)\beta i_F I_F$,

$$I_P^* = \frac{\lambda}{\mu + \gamma_P} S^* = (1 - p)\beta i_F \frac{1}{\mu + \gamma_P} S^* I_F^*. \quad (3.18)$$

Similarly, by (3.3)-(3.12), for every compartment $Comp \in \{I_P, I_D, R_D, R_C, I_S, I_{T1}, I_{T2}, I_{TL}, I_{UL}, R_T, R_C\}$
 we have

Table 4. Parameter values - treatment. Times are in months. Costs are in USD (2010). 2010 Exchange rate: 1 USD = 40 INR.

10

Symbol	Description	Value	Range	Reference
τ_1^{-1}	Duration of the first-line treatment	1	[0.75, 1]	[89]
τ_2^{-1}	Duration of the second-line treatment	1	[0.75, 1]	[89]
τ_3^{-1}	Duration of the PKDL treatment	6	[2.5, 7]	[21,90,91]
s_1	Probability of not responding to the first-line treatment	0.06	[0.01, 0.15]	[89]
s_2	Fraction of I_{T1} cases that appear to be recovering but became dormant	0.063	[0.02, 0.1]	[51]
s_3	Probability that the first line of treatment is successful	0.877		$1 - (s_1 + s_2)$
s_4	Fraction of I_{T2} cases that appear to be recovering but became dormant	0.063	[0.02, 0.1]	s_2
s_5	Probability that the second line of treatment is successful	0.937		$1 - s_4$
d_{T1}	Fraction of patients dying due to the first line of treatment	0.04	[0.02, 0.13]	[92]
d_{T2}	Fraction of patients dying due to the second line of treatment	0.04	[0.02, 0.13]	[92]
ρ_L^{-1}	Duration until relapse to PKDL	21	[16, 26]	[21]
p_{TL}	Fraction of PKDL patients that seek treatment	0.5	[0, 1]	[22]
C_{IS}	Cost of being in stage I_S	19	[11, 26]	[93]
C_{IT1}	Cost of first-line treatment	146	[110, 170]	[93], [94]
C_{IT2}	Cost of second-line treatment	146	[110, 170]	[93], [94]
C_{ITL}	Cost of PKDL treatment	349	[290, 410]	[95]
C_{RT}	Cost of being in R_T after treatment	57	[30, 100]	[93]
C_{RL}	Cost of being in R_L after treatment	57		C_{RT}
C_{ITN}	Cost of ITN	3.62	[1.75, 5.50]	[96]

Table 5. Auxiliary notation

Symbol	Description	Equation
λ_F	Force of infection	(2.2)
λ	Force of infection	(2.1)
μ_S^{-1}	Life expectancy in I_S	$(\mu + \mu_K)^{-1}$
μ_1^{-1}	Life expectancy in I_{T1}	$(\mu + d_{T1}\tau_1)^{-1}$
μ_2^{-1}	Life expectancy in I_{T2}	$(\mu + d_{T2}\tau_2)^{-1}$
C_{VL}	Expected cost of getting sick	(4.4)
T_{Comp}	Expected time an individual spends in a compartment $Comp \in \{I_P, \dots, R_C\}$ given it started in compartment I_P	(4.4)
T_{Cycle}	Expected time it takes an individual to become susceptible again (given it started in I_P and conditional on surviving)	(3.36)
T_I	Average time an individual spends as infectious to sand fly (weighted by the infectivity)	(3.33)

$$Comp^* = (1 - p)\beta i_F T_{Comp} S^* I_F^* \tag{3.19}$$

where the quantities T_{Comp} correspond to the expected time an individual spends in a compartment $Comp$ given it started in compartment I_P and are given by

$$T_{I_P} = \frac{1}{\mu + \gamma_P} \tag{3.20}$$

$$T_{I_D} = \frac{\gamma_P}{\mu + \gamma_D} T_{I_P} \tag{3.21}$$

$$T_{R_D} = \frac{f_D \gamma_D}{\mu + \rho_D} T_{I_D} \tag{3.22}$$

$$T_{I_S} = \frac{f_S \gamma_D}{\mu_S + \gamma_S} T_{I_D} \tag{3.23}$$

$$T_{I_{T1}} = \frac{\gamma_S}{\mu_1 + \tau_1} T_{I_S} \tag{3.24}$$

$$T_{I_{T2}} = \frac{s_1 \tau_1}{\mu_2 + \tau_2} T_{I_{T1}} \tag{3.25}$$

$$T_{R_L} = \frac{s_2 \tau_1 T_{I_{T1}} + s_4 \tau_2 T_{I_{T2}} + f_L \gamma_D T_{I_D}}{\mu + \rho_L} \tag{3.26}$$

$$T_{I_{TL}} = \frac{p_{TL} \rho_L}{\mu + \tau_3} T_{R_L} \tag{3.27}$$

$$T_{I_{UL}} = \frac{(1 - p_{TL}) \rho_L}{\mu + \gamma_L} T_{R_L} \tag{3.28}$$

$$T_{R_T} = \frac{s_3 \tau_1 T_{I_{T1}} + s_5 \tau_2 T_{I_{T2}} + \tau_3 T_{I_{TL}} + \gamma_L T_{I_{UL}}}{\mu + \rho_T} \tag{3.29}$$

$$T_{R_C} = \frac{\rho_D T_{R_D} + \rho_T T_{R_T}}{\mu + \rho_C} \tag{3.30}$$

Disease Free Equilibrium

We assume that $I_P^0 = 0$. By (3.3), $I_D^0 = 0$. By (3.4), $R_D^0 = 0$. By (3.5), $I_S^0 = 0$. By (3.6), $I_{T1}^0 = 0$. By (3.7), $I_{T2}^0 = 0$. By (3.8), $R_L^0 = 0$. By (3.9), $I_{TL}^0 = 0$, and, by (3.10), $I_{UL}^0 = 0$. By (3.11), $R_T^0 = 0$. By (3.12), $R_C^0 = 0$. From (3.1), $S^0 = \frac{\Lambda}{\mu}$. By (2.2), $\lambda_F = 0$. Thus, by (3.13), $S_F^0 = N_F = n_F \frac{\Lambda}{\mu}$. Thus, $E_F^0 = 0$ and $I_F^0 = 0$. So our disease free equilibrium is given by

$$E^0 = \left(\frac{\Lambda}{\mu}, 0, 0, 0, 0, 0, 0, 0, 0, 0, 0, 0, 0, 0, n_F \frac{\Lambda}{\mu}, 0, 0 \right). \tag{3.31}$$

Basic reproduction number

The formula for \mathcal{R}_0 can be derived as follows. Assume that there is a single infected person in compartment I_P . As derived in above, that person spends an expected time T_{Comp} in each of the compartments $Comp \in \{I_P, I_D, I_S, I_{T1}, I_{T2}, I_{TL}, I_{UL}\}$. During that time, they expose sand flies at rate $\beta i_X N_F = \beta i_X n_F \frac{\Lambda}{\mu}$. Each of the exposed sand flies becomes infectious with the probability $\sigma_F / (\mu_F + \sigma_F)$. Each sand fly stays infectious for the time μ_F^{-1} and during that time it infects humans at rate $(1 - p)\beta i_F N = (1 - p)\beta i_F \frac{\Lambda}{\mu}$. Putting it all together yields

$$\mathcal{R}_0 = (1 - p)\beta^2 i_F n_F \left(\frac{\Lambda}{\mu} \right)^2 \left(\frac{\sigma_F}{\mu_F + \sigma_F} \right) \left(\frac{1}{\mu_F} \right) T_i \tag{3.32}$$

where

$$T_i = i_P T_{I_P} + i_D T_{I_D} + i_S T_{I_S} + i_{T1} T_{I_{T1}} + i_{T2} T_{I_{T2}} + i_{TL} T_{I_{TL}} + i_{UL} T_{I_{UL}}. \tag{3.33}$$

Endemic Equilibrium

By looking at all human compartments and balancing what is coming in and out (i.e. adding all
the equations (3.1)-(3.12)), we get

$$\begin{aligned} \Lambda = & \mu S^* + \mu I_P^* + \mu I_D^* + \mu R_D^* + \mu S I_S^* + \mu_1 I_{T1}^* + \mu_2 I_{T2}^* \cdots \\ & + \mu R_L^* + \mu I_{TL}^* + \mu I_{UL}^* + \mu R_T^* + \mu R_C^*. \end{aligned} \quad (3.34)$$

By (3.20)-(3.30),

$$\Lambda = \mu(S^* + (1-p)\beta i_F T_{Cycle} S^* I_F^*) \quad (3.35)$$

where

$$\begin{aligned} T_{Cycle} = & T_{IP} + T_{ID} + T_{RD} + \frac{\mu_S}{\mu} T_{IS} + \frac{\mu_1}{\mu} T_{IT1} + \frac{\mu_2}{\mu} T_{IT2} \cdots \\ & + T_{RL} + T_{ITL} + T_{IUL} + T_{RT} + T_{RC}. \end{aligned} \quad (3.36)$$

Solving for S^* yields

$$S^* = \left(\frac{\Lambda}{\mu}\right) \left(\frac{1}{1 + (1-p)\beta i_F T_{Cycle} I_F^*}\right). \quad (3.37)$$

By (2.2),

$$\lambda_F = \beta(i_P I_P^* + i_D I_D^* + i_S I_S^* + i_{T1} I_{T1}^* + i_{T2} I_{T2}^* + i_{TL} I_{TL}^* + i_{UL} I_{UL}^*) \quad (3.38)$$

$$= (1-p)\beta^2 i_F T_I S^* I_F^* \quad (3.39)$$

$$= (1-p)\beta^2 i_F T_I \left(\frac{\Lambda}{\mu}\right) \frac{I_F^*}{1 + (1-p)\beta i_F T_{Cycle} I_F^*}. \quad (3.40)$$

Thus,

$$\frac{I_F^*}{\lambda_F} = \frac{1 + (1-p)\beta i_F T_{Cycle} I_F^*}{(1-p)\beta^2 i_F T_I \left(\frac{\Lambda}{\mu}\right)}. \quad (3.41)$$

From (3.15),

$$E_F^* = I_F^* \frac{\mu_F}{\sigma_F}. \quad (3.42)$$

From (3.14),

$$S_F^* = \frac{\mu_F + \sigma_F}{\lambda_F} E_F^* = \frac{\mu_F + \sigma_F}{\lambda_F} \cdot \frac{\mu_F}{\sigma_F} \cdot I_F^* \quad (3.43)$$

$$= \left(\frac{\mu_F + \sigma_F}{\sigma_F}\right) \cdot \mu_F \cdot \left(\frac{1 + (1-p)\beta i_F T_{Cycle} I_F^*}{(1-p)\beta^2 i_F T_I \left(\frac{\Lambda}{\mu}\right)}\right) \quad (3.44)$$

$$= N_F^* \frac{1 + (1-p)\beta i_F T_{Cycle} I_F^*}{\mathcal{R}_0}. \quad (3.45)$$

Thus,

$$N_F^* = S_F^* + E_F^* + I_F^* \tag{3.46}$$

$$= N_F^* \left(\frac{1 + (1-p)\beta i_F T_{Cycle} I_F^*}{\mathcal{R}_0} \right) + \frac{\mu_F}{\sigma_F} I_F^* + I_F^*. \tag{3.47}$$

and

$$I_F^* = \frac{N_F^* \left(1 - \frac{1}{\mathcal{R}_0} \right)}{(1-p)\beta i_F T_{Cycle} \frac{N_F^*}{\mathcal{R}_0} + \frac{\mu_F}{\sigma_F} + 1}. \tag{3.48}$$

Once I_F^* is calculated from (3.48), we can calculate E_F^* , S_F^* and S^* as

$$E_F^* = I_F^* \frac{\mu_F}{\sigma_F} \tag{3.49}$$

$$S_F^* = \frac{\mu_F + \sigma_F}{\lambda_F} E_F^* \tag{3.50}$$

$$S^* = \left(\frac{\Lambda}{\mu} \right) \left(\frac{1}{1 + (1-p)\beta i_F T_{Cycle} I_F^*} \right). \tag{3.51}$$

Finally, using above, we can also find all the human compartments as below

$$I_P^* = (1-p)\beta i_F T_{IP} S^* I_F^* \tag{3.52}$$

$$I_D^* = (1-p)\beta i_F T_{ID} S^* I_F^* \tag{3.53}$$

$$R_D^* = (1-p)\beta i_F T_{RD} S^* I_F^* \tag{3.54}$$

$$I_S^* = (1-p)\beta i_F T_{IS} S^* I_F^* \tag{3.55}$$

$$I_{T1}^* = (1-p)\beta i_F T_{IT1} S^* I_F^* \tag{3.56}$$

$$I_{T2}^* = (1-p)\beta i_F T_{IT2} S^* I_F^* \tag{3.57}$$

$$R_L^* = (1-p)\beta i_F T_{RL} S^* I_F^* \tag{3.58}$$

$$I_{TL}^* = (1-p)\beta i_F T_{ITL} S^* I_F^* \tag{3.59}$$

$$I_{UL}^* = (1-p)\beta i_F T_{IUL} S^* I_F^* \tag{3.60}$$

$$R_T^* = (1-p)\beta i_F T_{RT} S^* I_F^* \tag{3.61}$$

$$R_C^* = (1-p)\beta i_F T_{RC} S^* I_F^* \tag{3.62}$$

4. Model calibration

[revised manuscript text omitted]
 six years, the cohort of $N = 22699$ individuals should experience $72 \cdot \left(\frac{1}{33}\right) \cdot 0.35 \cdot 22699 \approx 17334$ late asymptomatic cases. Since 813 of those developed KA, we get $f_S = 813/17334 = 0.0469$. Similarly, [22] surveyed 24,814 individuals for KA and PKDL in the past nine years. Of those 1,002 reported KA and 17 had PKDL with no history of KA. This gives an estimate

$$f_S = \frac{1002}{9 \cdot 12 \cdot 24814 \cdot \frac{1}{33} \cdot 0.35} = 0.0353. \quad (4.1)$$

Because [21] and [22] used a much larger sample size than [88] and [88] also notes that incidences were larger in India than in Nepal, we will adopt $f_S = 0.035$ and consider a range from 0.01 to 0.15.

We estimate that the fraction of asymptomatic patients who become dormant to be $f_L = 0.00055$ with the range [0.0001, 0.001]. This will be done in a similar fashion as the estimate for f_S based on data from [21] and [22]. A total of $N = 22699$ individuals were surveyed by [21] for KA and PKDL in the past six years and determined that 813 respondents had KA and eight had PKDL with no history of KA. This yields

$$f_L = \frac{8}{6 \cdot 12 \cdot 22699 \cdot \rho_C \cdot 0.35 - 813} = 0.000484. \quad (4.2)$$

Similarly, [22] surveyed 24,814 individuals for KA and PKDL in the past nine years. Of those 1,002
reported KA and 17 had PKDL with no history of KA. This gives an estimate

$$f_L = \frac{17}{9 \cdot 12 \cdot 24814 \cdot \rho_C \cdot 0.35 - 1002} = 0.00062. \quad (4.3)$$

We note that authors of [51] provide an estimate $f_L = 0.0001$ based on [21] although it seems that
they wanted to use $f_L = 8/22699 = 0.00035$.

(c) Treatment parameters

We will assume that the first line of treatment fails with the probability $s_1 = 0.06$ [89] and that
range is [0.01, 0.12]. Miltefosine is a common and effective first-line treatment in Bihar, India,
with a clinical efficacy of 94% [89]. Studies like [104] give $s_1 = 0.03$ and lower while other studies
such as [105] show that the treatment failures by common treatments can be as high as $s_1 = 0.12$.

The fraction of KA patients who appear to recover under KA first-line treatment but will
develop PKDL (conditional on surviving treatment, $1 - d_{T1}$) is $s_2 = 0.063$ [105]. This is based
on the presentation "Cohort observational study to estimate the cumulative incidence of PKDL
in VL patients treated with three regimens in Bihar (Presenter: Suman Rijal)" that reported on a
cohort of 1622 KA patients treated between 2012 and 2015 and followed up in 2016 and 2017 to
determine the occurrence of PKDL. The cumulative incidence of PKDL was 6.3%, with a PKDL
rate of 4.8%, 5.7% and 9.2% depending on the treatment. Since [51] reported $s_2 = 0.03$, we will
use [0.02, 0.1] for the range.

We will assume the probability to die during the treatment (as a result of the treatment) to
be about $d_{T1} = d_{T2} = 0.04$. This is based on [92, Table 6] who reported the death incidence of
(presumably treated) KA cases as 45.75 per 1000 (compared to a regular mortality rate of 2.9 per
1000 in the state of Bihar for the age group 15-59). The mortality varies greatly by age group
from about 0.02 (5-14 years) to about 0.13 (60+ years). We will thus assume the range to be [0.02,
0.13]. We note that [51] assumed $f_T = 0.05$. Overall, the numbers seem in line with other studies;
[106] reported 27 deaths out of 553 patients (4.8%) treated with sodium antimony gluconate and,
cumulatively, nine studies discussed in [106] report 51 deaths out of 1819 patients (2.8%).

The time from the apparent cure from KA until relapse to PKDL is assumed to be $\rho_L^{-1} = 21$
348 months, with the range 16 to 26 months [21]. This is in line with 19 months reported by [22].

The duration of the first line of treatment is $\tau_1^{-1} = 1$ month, with the range [0.6, 1] months
[89]. The duration varies by the treatment type, but the most common first-line drug used in
Bihar, India is Miltefosine [94].

Similarly, the duration of the second line of treatment is $\tau_2^{-1} = 1$ month, with the range [0.6,
1] months [89]. The most common second-line drug used in Bihar, India is amphotericin B [94].

The duration of the PKDL treatment is $\tau_3^{-1} = 6$ months with range one to seven months.
There are three different kinds of treatments: sodium stibogluconate (SSG), miltefosine (MF) and
amphotericin B. The SSG treatment takes six months (six 30 days long cycles [21]), the MF
treatment takes approximately three to four months [90,91], and amphotericin B treatment takes
three weeks [22]. While the last option seems the fastest, it does not mean that PKDL is cured
within three weeks. In fact [22] followed 30 PKDL cases treated with amphotericin B, and while
reported an improvement in four months, only one case was completely cured. We do not have
additional data but it seems that even in the case of this treatment, the duration of PKDL is six
362 months.

The duration of untreated PKDL, γ_L^{-1} , will be estimated as follows. In [22], the authors
followed 98 PKDL patients that never received treatment and provides estimated resolution
rates: 8% within one year of onset, 34% within two years, and 67% within five years. Fitting the
exponential decay to this data by matlab, see Figure 2, yields $\gamma_L^{-1} = 55.5$ months with the range
of 32 months to 16 years.

The fraction of PKDL patients that receive treatment is estimated as $p_{TL} = 0.5$. This estimate is
based on [22] who reports that out of 185 PKDL cases, 98 did not seek the treatment.

(d) The cost parameters

Once infected, all individuals have to go through I_P and I_D . Those stages are asymptomatic and consequently with no associated costs. The costs appear only if and when an individual experiences symptoms (in I_S), is being treated (I_{T1} , I_{T2} , and I_{TL}) or is recovering after the treatment (in R_T and R_L). The costs are denoted by C_{I_S} , $C_{I_{T1}}$, $C_{I_{T2}}$, $C_{I_{TL}}$, C_{R_T} , and C_{R_L} and the actual values are estimated below.

The cost of being in compartment I_S is estimated as $C_{I_S} = 19$ USD [93], and we will assume the range [11, 26]. The average monthly income in Bihar, India in 2010 was 38 USD per month [93]. It takes about a month to get a diagnosis [87]. During that time, the patient loses 2.14 weeks (about half a month) of work, i.e. the cost is about \$19.

The cost of getting the first line of treatment is $C_{I_{T1}} = 146$ USD [93], and we will assume the range [120, 170]. This includes 127 USD of direct medical costs for one month of treatment [93], and the indirect cost of lost work. For every month of illness, half a month of work is lost (19 USD loss) [93]. This is in line with \$100 direct medical costs used by [94].

The cost of getting the second line of treatment is $C_{I_{T2}} = C_{I_{T1}} = 146$ USD [93]. Both lines of treatment last approximately one month, so the cost of the individual second-treatment will be about the same.

The cost of getting treatment for PKDL is assumed to be $C_{I_{TL}} = 349.00$ USD [95]. We will assume the range [290, 410]. The direct cost of SSG treatment for PKDL costs 179 USD [95]. The treatment lasts about six months and so the loss of productivity during this illness is 170 USD.

The cost of recovering from treatment without a dormant infection is $C_{R_T} = 57$ USD [93], and we will assume the range [60, 110]. During recovery, a patient can only work 3.21 weeks/month on average, instead of the normal 4.29, i.e. losing 25% of the wage per month [93]. Since the average monthly wage is 38 USD and R_T lasts approximately $\rho_T^{-1} = 6$ months, the total loss is about $57 = 0.25 \cdot 38 \cdot 6$.

The cost of recovering from treatment, but with a dormant infection is $C_{R_L} = C_{R_T} = 57$ USD [93]. We assume that even though individuals in this stage do have a dormant infection, the time and the cost needed to recover from treatment will be about the same.

The overall cost of VL infection is thus given by

$$C_{VL} = P(I_P \rightarrow I_S)C_{I_S} + P(I_P \rightarrow I_{T1})C_{I_{T1}} + P(I_P \rightarrow I_{T2})C_{I_{T2}} + P(I_P \rightarrow I_{TL})C_{I_{TL}} + P(I_P \rightarrow R_T|I_{T1} \text{ or } I_{T2} \text{ or } I_{TL})C_{R_T} + P(I_P \rightarrow R_L|I_{T1} \text{ or } I_{T2})C_{R_L} \quad (4.4)$$

where the probabilities $P(I_P \rightarrow C)$ (and $P(I_P \rightarrow C|C')$) of getting to a compartment C (through a compartment C') when currently at a compartment I_P are given by

$$P(I_P \rightarrow I_S) = T_{I_D} f_S \gamma_D \quad (4.5)$$

$$P(I_P \rightarrow I_{T1}) = T_{I_S} \gamma_S \quad (4.6)$$

$$P(I_P \rightarrow I_{T2}) = T_{I_{T1}} s_1 \tau_1 \quad (4.7)$$

$$P(I_P \rightarrow I_{TL}) = T_{R_L} p_{TL} \rho_L \quad (4.8)$$

$$P(I_P \rightarrow R_T|I_{T1} \text{ or } I_{T2} \text{ or } I_{TL}) = T_{I_{T1}} s_3 \tau_1 + T_{I_{T2}} s_5 \tau_2 + T_{I_{TL}} \tau_3 \quad (4.9)$$

$$P(I_P \rightarrow R_L|I_{T1} \text{ or } I_{T2}) = T_{I_{T1}} s_2 \tau_1 + T_{I_{T2}} s_4 \tau_2. \quad (4.10)$$

The cost of ITN is $C_{ITN} = 3.62$ USD [35]. If the households opt to use the impregnated durable wall lining, which seems even more effective than ITNs, the cost grows significantly to about 13.57 USD [35]. At the same time, one could argue that the wall lining protects the whole household, while the ITN protects only those that sleep under it.

Figure 3. Maximum likelihood estimates for i_P , i_D (left) and i_S , i_{UL} and i_F (right). While the numerical estimates for i_S and i_{UL} fluctuates (dotted lines), they follow a linear trend (solid lines).

5. Results

(a) Estimating VL transmission probabilities

Unlike [51], we will assume that individuals receiving treatment cannot infect the sand flies, i.e. $i_{T1} = 0$, $i_{T2} = 0$, and $i_{TL} = 0$. The treated individuals are not in their regular environments but rather at the treatment facility. Consequently, they are likely better protected from sand fly bites. Moreover, even if they are bitten and infect the fly, they are too far from their home for this event to meaningfully contribute to VL transmission in their home/village. This assumption is in agreement with [24] who reported that after receiving treatment, the VL patients are likely less infectious to the sand flies resulting in less transmission to family members during their 18-month follow-up.

The remaining transmission probabilities, specifically i_P , i_D for asymptomatic individuals, i_S and i_{UL} for untreated symptomatic individuals and i_F for the transmission from the sand fly are estimated using the maximum likelihood method by fitting the model to KalaNet trial data [51, Table 1].

The probability that a susceptible human becomes infected when an infected sand fly bites them, i_F , was estimated as very close to 1, see Figure 3. This is consistent with the fact that the parasite can modify the sand fly's feeding apparatus so that the fly feeds more persistently and releases more parasites [99,107].

The estimates for i_{UL} and i_S are more or less linear in p , the proportion of population that used ITNs during the KalaNet trial. The linear regression on the maximum likelihood estimates come out as $i_{UL} = 0.253 + 0.254p$ and $i_S = 0.357 + 0.156p$, i.e. $i_{UL} \approx 0.4325$ and $i_S \approx 0.475$ when $p \approx 0.7$.

The estimates for i_P and i_D are quite sensitive on the p . For small $p \approx 0$, we get $i_D \approx 0.024$ and $i_P \approx 0.004$ which is consistent with [51]. However, for $p \approx 0.7$, $i_D \approx 0.1$ and $i_P \approx 0.024$. If p grows even more, the values of i_D and i_P grow rapidly.

(b) Model validation

The model is validated using KalaNet trial data [51, Table 1]. Our model gives the following prevalences (the KalaNet data are in the parenthesis): 0.7586 (0.76) for $S + R_C$, 0.0984 (0.1) for I_P , 0.0222 (0.02) for i_D , 0.1179 (0.12) for R_D , and 0.0051 (0.005) for I_F .

Figure 4. The expected cost of getting sick if not using ITNs when the population usage is p (solid line) versus the cost of ITNs (dotted line). The optimal use (a Nash equilibrium) occurs when the lines intersect. VL is eliminated when the solid line reaches the x -axis.

(c) ITN use for VL elimination

To find the ITN usage level necessary to achieve the complete elimination of VL, we need to find
 the smallest $p_{HI} \in [0, 1]$ such that when $p \geq p_{HI}$, then $\mathcal{R}_0 \leq 1$. It follows from (3.32) that $\mathcal{R}_0 =$
 $(1 - p)\mathcal{R}_0(0)$ where

$$\mathcal{R}_0(0) = \beta^2 i_F n_F \left(\frac{\Lambda}{\mu}\right)^2 \left(\frac{\sigma_F}{\mu_F + \sigma_F}\right) \left(\frac{1}{\mu_F}\right) T_I \quad (5.1)$$

is the basic reproduction number when nobody is using the protection. Consequently,

$$p_{HI} = \begin{cases} 1 - \frac{1}{\mathcal{R}_0(0)}, & \text{if } \mathcal{R}_0(0) > 1, \\ 0, & \text{otherwise.} \end{cases} \quad (5.2)$$

(d) Optimal use of ITNs

In this section, we will find the optimal proportion of the use of ITNs. We are looking for Nash
 equilibrium - a proportion that, when adopted by the population, no individual has an incentive
 to deviate from their choice. To find a Nash equilibrium value of p , we have to solve

$$C_{ITN} = \frac{\beta i_F I_F^*}{\mu + \beta i_F I_F^*} C_{VL} \quad (5.3)$$

where C_{ITN} is the cost protection, $\frac{\beta i_F I_F^*}{\mu + \beta i_F I_F^*}$ is the probability of getting infected by an infected
 sand fly, and C_{VL} is the expected cost one pays after such an event. Note that (5.3) is an equation
 for p because I_F^* is a function of p . Figure 4 illustrates a graphical solution of (5.3).

Recall that, by (3.48),

$$I_F^* = \frac{N_F^* \left(1 - \frac{1}{\mathcal{R}_0(0)(1-p)}\right)}{(1-p)\beta i_F T_{Cycle} \frac{N_F^*}{\mathcal{R}_0(0)(1-p)} + \frac{\mu_F}{\sigma_F} + 1} \quad (5.4)$$

and in particular I_F^* is decreasing in p and thus $\frac{\beta i_F I_F^*}{\mu + \beta i_F I_F^*}$ is decreasing in p . The maximum value
 of I_F^* is thus

$$I_F^*(0) = \frac{N_F^* \left(1 - \frac{1}{\mathcal{R}_0(0)}\right)}{\beta i_F T_{Cycle} \frac{N_F^*}{\mathcal{R}_0(0)} + \frac{\mu_F}{\sigma_F} + 1} \quad (5.5)$$

attained for $p = 0$ and consequently, (5.3) has a solution only if

$$C_{ITN} \leq \frac{\beta i_F I_F^*(0)}{\mu + \beta i_F I_F^*(0)} C_{VL}. \quad (5.6)$$

It follows from (5.3) that (when (5.6) is true)

$$I_F^* = \frac{\left(\frac{C_{ITN}}{C_{VL}}\right) \mu}{\beta i_F \left(1 - \frac{C_{ITN}}{C_{VL}}\right)}. \quad (5.7)$$

Thus, after algebraic manipulations of (5.4), we get that p_{NE} is given by

$$p_{NE} = 1 - \frac{1}{\mathcal{R}_0(0) \left[1 - \frac{I_F^*}{N_F^*} \left(\beta i_F T_{cycle} \frac{N_F^*}{\mathcal{R}_0(0)} + \frac{\mu_F}{\sigma_F} + 1\right)\right]} \quad (5.8)$$

where I_F^* is given by (5.7).

It follows that $p_{NE} \approx p_{HI}$. For our parameter values, we have $p_{HI} = 0.96211$, $p_{NE} = 0.96206$ with $\mathcal{R}_0(p_{NE}) = 1.0017$. It means that the disease can be almost eliminated by the optimal voluntary use of ITNs alone.

(e) Sensitivity analysis

The sensitivity of the outcomes (ITN use for disease elimination, p_{HI} , and optimal ITN use, p_{NE}) on different parameter values is displayed in Table 6 and shown in Figures 5 and 6. Since $p_{NE} \approx p_{HI}$, only the sensitivity of p_{HI} is shown.

It follows that p_{HI} (and p_{NE}) are not overly sensitive to any parameter. The sensitivity index is at most 0.08 (for μ or λ) or -0.08 (for β^{-1}). It is the second highest for μ_F^{-1} and closely followed by n_F and i_F . It follows that increasing the time between the bites and the reduction of the lifespan or the number of sand flies are the most promising control measure apart from using ITNs.

6. Discussion

We expanded the mathematical model of VL transmission [51] to understand the role of ITN usage in VL dynamics and possible elimination. We calibrated the model based on the data found in the literature. We validated the model on KalaNet data. We concluded that in order to eradicate VL, the ITN usage would have to be about 96%. Our Figure 4 may explain why there was no significant effect of ITN use on VL incidence during the KalaNet trial - the risk of VL infection is essentially a step function with a jump just around the disease elimination level. At the end of the KalaNet trial, the use of ITNs was very high: 91% of the individuals slept more than 80% of the nights under treated nets [42]. Yet, even with such high ITN use, our model predicts that the disease is not eliminated and the risk of infection almost does not change. The difference between 91% and 96% does not seem large, but note that 91% coverage leaves 9% of the population unprotected. This is more than double the unprotected population if the ITNs use is 96% or more.

While our model demonstrates that the elimination of VL by ITN use is possible in theory, the question whether it could be achieved in practice remains open. Residents in VL endemic areas seem to be reasonably aware of role of ITNs in the prevention of VL and other vector borne diseases [38]. It has been demonstrated theoretically [61,76] as well as empirically [62] that individuals behave rationally and that the high cost of protection (relative to the cost of the disease) is often the reason why the protection is not adopted. Yet, giving the ITNs for free during the KalaNet trial achieved only 91% coverage. The ITN use should thus be combined with other intervention methods.

Table 6. The sensitivity index SI_y of a variable y on a parameter x was calculated as $\left(\frac{x}{y}\right) \cdot \left(\frac{\partial y}{\partial x}\right)$, see for example [108]. The numbers were rounded to the three decimal places. Parameters are as specified in Tables 2, 3 and 4. The sensitivity index -0.5 means that a 1% increase of a parameter value x will result in the 0.5% decrease of the variable y .

Parameter	$SI_{p_{HI}}$	$SI_{(p_{HI}-p_{NE})}$
n_F	0.040	-1.968
β^{-1}	-0.079	3.829
σ_F^{-1}	-0.010	0.525
μ_F^{-1}	0.049	-2.542
i_F	0.040	-1.991
Λ	0.079	-3.875
μ^{-1}	0.080	-5.026
μ_K^{-1}	0.000	-0.079
γ_P^{-1}	0.017	-0.735
γ_D^{-1}	0.016	-0.773
γ_S^{-1}	0.002	-0.035
f_L	0.001	-0.063
f_S	0.006	-1.797
ρ_C^{-1}	0.000	0.644
ρ_D^{-1}	0.000	0.121
i_P	0.017	-0.708
i_D	0.016	-0.772
i_S	0.002	-0.114
i_{T1}	0.000	0.000
i_{T2}	0.000	0.000
i_L	0.000	0.000
i_{UL}	0.004	-0.211
τ_1^{-1}	-0.000	0.002
τ_2^{-1}	-0.000	0.000
τ_3^{-1}	0.000	0.000
γ_{UL}^{-1}	0.004	-0.196
s_1	0.000	-0.063
s_2	0.003	-0.231
s_4	0.000	-0.014
d_{T1}	-0.000	0.051
d_{T2}	-0.000	0.003
δ_L^{-1}	-0.000	0.009
s_{TL}	-0.004	0.106
C_{ITN}	0.000	1.687
C_{IT1}	0.000	-0.908

Our model, specifically Figure 5, suggests that increasing β^{-1} , the interval between two sand fly bites, from four days to six or more days can reduce the ITN coverage needed for VL elimination from 96% to 90% or even lower (which falls below 91% observed during the KalaNet trial). The increase of β^{-1} could be achieved by giving the sand flies other biting opportunities. While controversial, this intervention method has been studied for other vector-borne NTDs such as Chagas disease [109]. The impact of the presence of cattle on VL incidence has already been studied separately. The cattle are associated with increased VL risk in some studies and decreased risk in others [32,110], reflecting the complexity of the effects on sand fly abundance, aggregation, feeding behavior and VL incidence [111]. An increase in blood meal opportunities could lead to increase in the sand fly population size, possibly erasing the positive effect of increased biting

rsos.royalsocietypublishing.org R. Soc. open sci. 0000000

Figure 5. Dependence of p_{HI} on different parameter values. Unless varied, the parameter values are as specified in Tables 2, 3, 4. For those parameters, $p_{HI} = 0.96211$, $p_{NE} = 0.96206$.

interval. Consequently, an effective elimination strategy should include multiple interventions at
 the same time - (1) using ITNs to prevent insect bites as much as possible, (2) using cattle and/or
 other animals (that cannot serve as hosts for *L. donovani*) to divert vector bites away from humans
 [4], and (3) using insecticide residual spraying [7] or (4) destroying breeding sites [48] to keep the
 vector population under control.

Data Accessibility. This article has no additional data.

Authors' Contributions. A.K.F.: conceptualization, formal analysis, investigation, and writing - original
 draft

C.P.G.: conceptualization, formal analysis, investigation, software, and writing - original draft

504 J.A.W. - conceptualization, formal analysis, investigation, and writing - original draft

Y.L. - supervision, writing - original draft

506 J.R. - writing - original draft, writing - review and editing, software, methodology, supervision,
 conceptualization, resources

D.T. - writing - original draft, writing - review and editing, methodology, supervision, conceptualization,
 formal analysis, validation, funding acquisition

The order of the authors was determined as follows: The first three authors are undergraduate students in
 alphabetical order. The last three authors are professors in alphabetical order.

Figure 6. Dependence of p_{HI} on different parameter values. Unless varied, the parameter values are as specified in Tables 2, 3, 4. For those parameters, $p_{HI} = 0.96211$, $p_{NE} = 0.96206$.

Competing Interests. There are no competing interests.

Funding. AKE, CPG and JAW were supported by the VCU REU program in mathematics funded by the
 National Security Agency ([https://www.nsa.gov/what-we-do/research/math-sciences-program/
 proposal-guidelines/](https://www.nsa.gov/what-we-do/research/math-sciences-program/proposal-guidelines/)) grant number H98230-20-1-0011 awarded to DT. The funders had no role in study
 design, data collection and analysis, decision to publish, or preparation of the manuscript.

References

1. CDC. 2020 Parasites - Leishmaniasis. [https://www.cdc.gov/parasites/
 leishmaniasis/index.html](https://www.cdc.gov/parasites/leishmaniasis/index.html). Accessed June 29, 2020.
 2. Mubayi A, Castillo-Chavez C, Chowell G, Kribs-Zaleta C, Siddiqui NA, Kumar N, Das P. 2010
 Transmission dynamics and underreporting of Kala-azar in the Indian state of Bihar. *Journal
 of Theoretical Biology* **262**, 177–185.
 3. Zou L, Chen J, Ruan S. 2017 Modeling and analyzing the transmission dynamics of visceral
 leishmaniasis. *Mathematical Biosciences & Engineering* **14**, 1585.
 4. Zhao S, Kuang Y, Wu CH, Ben-Arieh D, Ramalho-Ortigao M, Bi K. 2016 Zoonotic visceral
 leishmaniasis transmission: modeling, backward bifurcation, and optimal control. *Journal of
 Mathematical Biology* **73**, 1525–1560.
 5. Forestier CL. 2013 Imaging host–Leishmania interactions: significance in visceral
 leishmaniasis. *Parasite Immunology* **35**, 256–266.
 6. WHO. 2020 Leishmaniasis: Epidemiological Situation. [https://www.who.int/
 leishmaniasis/burden/en/](https://www.who.int/leishmaniasis/burden/en/). Accessed June 26, 2020.
 7. Kumar V, Mandal R, Das S, Kesari S, Dinesh DS, Pandey K, Das VR, Topno RK, Sharma
 MP, Dasgupta RK. 2020 Kala-azar elimination in a highly-endemic district of Bihar, India:
 A success story. *PLOS Neglected Tropical Diseases* **14**, e0008254.
 8. Sundar S, Singh OP, Chakravarty J. 2018 Visceral leishmaniasis elimination targets in India,
 strategies for preventing resurgence. *Expert review of anti-infective therapy* **16**, 805–812.
 9. Cloots K, Uranw S, Ostyn B, Bhattarai NR, Le Rutte E, Khanal B, Picado A, Chappuis F, Hasker
 E, Karki P. 2020 Impact of the visceral leishmaniasis elimination initiative on *Leishmania
 donovani* transmission in Nepal: a 10-year repeat survey. *The Lancet Global Health* **8**, e237–e243.

- 10. NTD Modelling Consortium Visceral Leishmaniasis Group. 2019 Insights from mathematical
modelling and quantitative analysis on the proposed WHO 2030 targets for visceral
leishmaniasis on the Indian subcontinent. *Gates Open Research* **3**, 1651.
- 11. Singh OP, Singh B, Chakravarty J, Sundar S. 2016 Current challenges in treatment options for
visceral leishmaniasis in India: a public health perspective. *Infectious diseases of poverty* **5**, 1–15.
- 12. Dinesh D, Ranjan A, Palit A, Kishore K, Kar S. 2001 Seasonal and nocturnal landing/biting
behaviour of *Phlebotomus argentipes* (Diptera: Psychodidae). *Annals of Tropical Medicine &*
*Parasitology* **95**, 197–202.
- 13. Feliciangeli M. 2004 Natural breeding places of phlebotomine sandflies. *Medical and Veterinary*
*Entomology* **18**, 71–80.
- 14. Gawade S, Nanaware M, Gokhale R, Adhav P. 2012 Visceral leishmaniasis: A case report. *The*
*Australasian Medical Journal* **5**, 130.
- 15. Younis LG, Kroeger A, Joshi AB, Das ML, Omer M, Singh VK, Gurung CK, Banjara MR.
2020 Housing structure including the surrounding environment as a risk factor for visceral
leishmaniasis transmission in Nepal. *PLoS Neglected Tropical Diseases* **14**, e0008132.
- 16. Goddard J. 2013 *Physician's Guide to Arthropods of Medical Importance*. CRC Press: Taylor &
Francis Group.
- 17. Poché RM, Garlapati R, Elnaïem DEA, Perry D, Poché D. 2012 The role of Palmyra palm trees
(*Borassus flabellifer*) and sand fly distribution in northeastern India. *Journal of Vector Ecology* **37**,
148–153.
- 18. Piscopo TV, Azzopardi CM. 2007 Leishmaniasis. *U.S. National Library of Medicine*.
- 19. Das A, Karthick M, Dwivedi S, Banerjee I, Mahapatra T, Srikantiah S, Chaudhuri I. 2016
Epidemiologic correlates of mortality among symptomatic visceral leishmaniasis cases:
findings from situation assessment in high endemic foci in India. *PLoS Neglected Tropical*
*Diseases* **10**, e0005150.
- 20. Ranjan A, Sur D, Singh VP, Siddique NA, Manna B, Lal CS, Sinha PK, Kishore K, Bhattacharya
SK. 2005 Risk factors for Indian kala-azar. *The American Journal of Tropical Medicine and Hygiene*
**73**, 74–78.
- 21. Rahman KM, Islam S, Rahman MW, Kenah E, Galive CM, Zahid M, Maguire J, Rahman M,
Haque R, Luby SP, Bern C. 2010 Increasing Incidence of Post-Kala-Azar Dermal Leishmaniasis
in a Population-Based Study in Bangladesh. *Clinical Infectious Diseases* **50**, 73–76.
- 22. Islam S, Kenah E, Bhuiyan MAA, Rahman KM, Goodhew B, Ghalib CM, Zahid M, Ozaki
572 M, Rahman M, Haque R. 2013 Clinical and immunological aspects of post-kala-azar dermal
leishmaniasis in Bangladesh. *The American Journal of Tropical Medicine and Hygiene* **89**, 345–353.
- 23. Addy M, Nandy A. 1992 Ten years of kala-azar in west Bengal, Part I. Did post-kala-
azar dermal leishmaniasis initiate the outbreak in 24-Parganas?. *Bulletin of the World Health*
*Organization* **70**, 341–346.
- 24. Das VNR, Pandey RN, Siddiqui NA, Chapman LA, Kumar V, Pandey K, Matlashewski G,
Das P. 2016 Longitudinal study of transmission in households with visceral leishmaniasis,
asymptomatic infections and PKDL in highly endemic villages in Bihar, India. *PLoS Neglected*
*Tropical Diseases* **10**, e0005196.
- 25. Wilson AL, Courtenay O, Kelly-Hope LA, Scott TW, Takken W, Torr SJ, Lindsay SW. 2020 The
importance of vector control for the control and elimination of vector-borne diseases. *PLoS*
*Neglected Tropical Diseases* **14**, e0007831.
- 26. Stockdale L, Newton R. 2013 A review of preventative methods against human leishmaniasis
infection. *PLoS Neglected Tropical Diseases* **7**, e2278.
- 27. Osman M, Mistry A, Keding A, Gabe R, Cook E, Forrester S, Wiggins R, Di Marco S, Colloca
S, Siani L et al.. 2017 A third generation vaccine for human visceral leishmaniasis and post
kala azar dermal leishmaniasis: First-in-human trial of ChAd63-KH. *PLoS Neglected Tropical*
*Diseases* **11**, e0005527.
- 28. Moafi M, Rezvan H, Sherkat R, Taleban R. 2019 Leishmania vaccines entered in clinical trials:
A review of literature. *International journal of preventive medicine* **10**.
- 29. Le Rutte EA, Coffeng LE, Malvolti S, Kaye PM, de Vlas SJ. 2020 The potential impact of human
visceral leishmaniasis vaccines on population incidence. *PLoS Neglected Tropical Diseases* **14**,
e0008468.

- 30. Kumar V, Kesari S, Sinha N, Palit A, Ranjan A, Kishore K, Saran R, Kar S. 1995 Field trial of
an ecological approach for the control of *Phlebotomus argentipes* using mud & lime plaster.. *The*
*Indian Journal of Medical Research* **101**, 154.
- 31. Wasserberg G, Weeks EN, Logan JL, Agneessens J, Stewart SA, Dewhirst S. 2019 Efficacy of
the insect repellent IR3535 on the sand fly *Phlebotomus papatasi* in human volunteers. *Journal*
*of Vector Ecology* **44**.
- 32. Bern C, Joshi AB, Jha SN, Das ML, Hightower A, Thakur G, Bista MB. 2000 Factors associated
with visceral leishmaniasis in Nepal: bed-net use is strongly protective.. *The American Journal*
*of Tropical Medicine and Hygiene* **63**, 184–188.
- 33. Chappuis F, Sundar S, Hailu A, Ghalib H, Rijal S, Peeling RW, Alvar J, Boelaert M. 2007
Visceral leishmaniasis: what are the needs for diagnosis, treatment and control?. *Nature*
*Reviews Microbiology* **5**, 873–882.
- 34. Ostyn B, Vanlerberghe V, Picado A, Dinesh DS, Sundar S, Chappuis F, Rijal S, Dujardin JC,
Coosemans M, Boelaert M. 2008 Vector control by insecticide-treated nets in the fight against
visceral leishmaniasis in the Indian subcontinent, what is the evidence?. *Tropical Medicine*
*International Health* **13**, 1073–1085.
- 35. Mondal D, Das ML, Kumar V, Huda MM, Das P, Ghosh D, Priyanka J, Matlashewski G,
Kroeger A, Upfill-Brown A et al.. 2016 Efficacy, safety and cost of insecticide treated wall
lining, insecticide treated bed nets and indoor wall wash with lime for visceral leishmaniasis
vector control in the Indian sub-continent: a multi-country cluster randomized controlled
trial. *PLoS Neglected Tropical Diseases* **10**, e0004932.
- 36. Vanlerberghe V, Singh S, Paudel I, Ostyn B, Picado A, Sanchez A, Rijal S, Sundar S, Davies
C, Boelaert M. 2010 Determinants of bednet ownership and use in visceral leishmaniasis-
endemic areas of the Indian subcontinent. *Tropical Medicine & International Health* **15**, 60–67.
- 37. Kroeger A, Ordóñez-Gonzalez o, Behrend M, Alvarez G. 1999 Bednet impregnation for
Chagas disease control: a new perspective. *Tropical Medicine and International Health* **4**, 194–198.
- 38. Mishra RN, Singh S, Vanlerberghe V, Sundar S, Boelaert M, Lefevre P. 2010 Lay perceptions of
kala-azar, mosquitoes and bed nets in Bihar, India. *Tropical Medicine & International Health* **15**,
36–41.
- 39. Kroeger A, González M, Ordóñez-González J. 1999 Insecticide-treated materials for malaria
control in Latin America: to use or not to use?. *Transactions of the Royal Society of Tropical*
*Medicine and Hygiene* **93**, 565–570.
- 40. Picado A, Singh SP, Rijal S, Sundar S, Ostyn B, Chappuis F, Uranw S, Gidwani K, Khanal B, Rai
M et al.. 2010a Longlasting insecticidal nets for prevention of *Leishmania donovani* infection
in India and Nepal: paired cluster randomised trial. *BMJ* **341**.
- 41. Picado A, Das ML, Kumar V, Kesari S, Dinesh DS, Roy L, Rijal S, Das P, Rowland M, Sundar
S. 2010b Effect of village-wide use of long-lasting insecticidal nets on visceral Leishmaniasis
vectors in India and Nepal: a cluster randomized trial. *PLoS Neglected Tropical Diseases* **4**, e587.
- 42. Picado A, Ostyn B, Rijal S, Sundar S, Singh SP, Chappuis F, Das ML, Khanal B, Gidwani K,
Hasker E et al.. 2015 Long-lasting insecticidal nets to prevent visceral leishmaniasis in the
Indian subcontinent; methodological lessons learned from a cluster randomised controlled
trial. *PLoS Neglected Tropical Diseases* **9**, e0003597.
- 43. Chowdhury R, Chowdhury V, Faria S, Akter S, Dash AP, Bhattacharya SK, Maheswary
NP, Bern C, Akhter S, Alvar J et al.. 2019 Effect of insecticide-treated bed nets on visceral
leishmaniasis incidence in Bangladesh. A retrospective cohort analysis. *PLoS Neglected Tropical*
*Diseases* **13**, e0007724.
- 44. Rock KS, le Rutte EA, de Vlas SJ, Adams ER, Medley GF, Hollingsworth TD. 2015 Uniting
mathematics and biology for control of visceral leishmaniasis. *Trends in Parasitology* **31**, 251–
259.
- 45. Hirve S, Boelaert M, Matlashewski G, Mondal D, Arana B, Kroeger A, Olliaro P. 2016
Transmission dynamics of visceral leishmaniasis in the Indian subcontinent—a systematic
literature review. *PLoS Neglected Tropical Diseases* **10**, e0004896.
- 46. DebRoy S, Prosper O, Mishoe A, Mubayi A. 2017 Challenges in modeling complexity of
neglected tropical diseases: a review of dynamics of visceral leishmaniasis in resource limited
settings. *Emerging Themes in Epidemiology* **14**, 10.

47. Bi K, Chen Y, Zhao S, Kuang Y, John Wu CH. 2018 Current visceral leishmaniasis research: a
research review to inspire future study. *BioMed Research International* **2018**.
48. Stauch A, Duerr HP, Picado A, Ostyn B, Sundar S, Rijal S, Boelaert M, Dujardin JC, Eichner
11 M. 2014 Model-based investigations of different vector-related intervention strategies to
12 eliminate visceral leishmaniasis on the Indian subcontinent. *PLoS Neglected Tropical Diseases*
**8**, e2810.
49. Chapman LA, Dyson L, Courtenay O, Chowdhury R, Bern C, Medley GF, Hollingsworth
TD. 2015 Quantification of the natural history of visceral leishmaniasis and consequences for
control. *Parasites and Vectors* **8**, 521.
50. Poche DM, Grant WE, Wang HH. 2016 Visceral leishmaniasis on the Indian subcontinent:
modelling the dynamic relationship between vector control schemes and vector life cycles.
*PLoS Neglected Tropical Diseases* **10**.
51. Stauch A, Sarkar RR, Picado A, Ostyn B, Sundar S, Rijal S, Boelaert M, Dujardin JC, Duerr HP.
2011 Visceral leishmaniasis in the Indian subcontinent: modelling epidemiology and control.
*PLoS Neglected Tropical Diseases* **5**.
52. Le Rutte EA, Coffeng LE, Bontje DM, Hasker EC, Postigo JAR, Argaw D, Boelaert MC,
De Vlas SJ. 2016 Feasibility of eliminating visceral leishmaniasis from the Indian subcontinent:
explorations with a set of deterministic age-structured transmission models. *Parasites and*
*Vectors* **9**, 24.
53. Meheus F, Balasegaram M, Olliaro P, Sundar S, Rijal S, Faiz MA, Boelaert M. 2010 Cost-
effectiveness analysis of combination therapies for visceral leishmaniasis in the Indian
subcontinent. *PLoS Neglected Tropical Diseases* **4**, e818.
54. Stauch A, Duerr HP, Dujardin JC, Vanaerschot M, Sundar S, Eichner M. 2012 Treatment of
visceral leishmaniasis: model-based analyses on the spread of antimony-resistant *L. donovani*
in Bihar, India. *PLoS Neglected Tropical Diseases* **6**, e1973.
55. Burattini MN, Coutinho FA, Lopez LF, Massad E. 1998 Modelling the dynamics of
leishmaniasis considering human, animal host and vector populations. *Journal of Biological*
*Systems* **6**, 337–356.
56. Hussaini N, Okuneye K, Gumel AB. 2017 Mathematical analysis of a model for zoonotic
visceral leishmaniasis. *Infectious Disease Modelling* **2**, 455–474.
57. Rabi Das VN, Bimal S, Siddiqui NA, Kumar A, Pandey K, Sinha SK, Topno RK, Mahentesh
48 V, Singh AK, Lal CS. 2020 Conversion of asymptomatic infection to symptomatic visceral
leishmaniasis: A study of possible immunological markers. *PLoS Neglected Tropical Diseases*
**14**, e0008272.
58. Manson-Bahr P, Heisch R, Garnham P et al.. 1959 Studies in Leishmaniasis in East Africa.
IV. The Montenegro Test in Kala-Azar in Kenya.. *Transactions of the Royal Society of Tropical*
*Medicine and Hygiene* **53**, 380–83.
59. Bern C, Amann J, Haque R, Chowdhury R, Ali M, Kurkjian KM, Vaz L, Wagatsuma Y, Breiman
RF, Secor WE et al.. 2006 Loss of leishmanin skin test antigen sensitivity and potency in a
longitudinal study of visceral leishmaniasis in Bangladesh. *The American Journal of Tropical*
*Medicine and Hygiene* **75**, 744–748.
60. Molina R, Jiménez M, García-Martínez J, San Martín JV, Carrillo E, Sánchez C, Moreno J,
Alves F, Alvar J. 2020 Role of asymptomatic and symptomatic humans as reservoirs of visceral
leishmaniasis in a Mediterranean context. *PLoS Neglected Tropical Diseases* **14**, e0008253.
61. Bauch CT, Earn DJ. 2004 Vaccination and the theory of games. *Proceedings of the National*
*Academy of Sciences* **101**, 13391–13394.
62. Maskin E. 1999 Nash equilibrium and welfare optimality. *The Review of Economic Studies* **66**,
23–38.
63. Chang SL, Piraveenan M, Pattison P, Prokopenko M. 2020 Game theoretic modelling of
infectious disease dynamics and intervention methods: a review. *Journal of Biological Dynamics*
14, 57–89.
64. Ibuka Y, Li M, Vietri J, Chapman GB, Galvani AP. 2014 Free-riding behavior in vaccination
decisions: an experimental study. *PLoS One* **9**.

65. Crawford K, Lancaster A, Oh H, Rychtář J. 2015 A voluntary use of insecticide-treated cattle can eliminate African sleeping sickness. *Letters in Biomathematics* **2**, 91–101.
66. Klein SRM, Foster AO, Feagins DA, Rowell JT, Erovenko IV. 2019 Optimal voluntary and mandatory insect repellent usage and emigration strategies to control the chikungunya outbreak on Reunion Island. *Preprint*.
67. Kobe J, Pritchard N, Short Z, Erovenko IV, Rychtář J, Rowell JT. 2018 A Game-Theoretic Model of Cholera with Optimal Personal Protection Strategies. *Bulletin of Mathematical Biology* **80**, 2580–2599.
68. Dorsett C, Oh H, Paulemond ML, Rychtář J. 2016 Optimal repellent usage to combat dengue fever. *Bulletin of Mathematical Biology* **78**, 916–922.
69. Brettin A, Rossi-Goldthorpe R, Weishaar K, Erovenko IV. 2018 Ebola could be eradicated through voluntary vaccination. *Royal Society Open Science* **5**, 171591.
70. Chouhan A, Maiwand S, Ngo M, Putalapattu V, Rychtář J, Taylor D. 2020 Game-theoretical model of retroactive Hepatitis B vaccination in China. *Bulletin of Mathematical Biology* **82**, 80.
71. Scheckelhoff K, Ejaz A, Erovenko IV. 2019 A game-theoretic model of optimal clean equipment usage to prevent hepatitis C among injecting drug users. *Preprint*.
72. Martinez A, Machado J, Sanchez E, Erovenko IV. 2019 Optimal vaccination strategies to reduce endemic levels of meningitis in Africa. *Preprint*.
73. Bankuru SV, Kossol S, Hou W, Mahmoudi P, Rychtář J, Taylor D. 2020 A Game-theoretic Model of Monkeypox to Assess Vaccination Strategies. *PeerJ* **8**, e9272.
74. Cheng E, Gambhirrao N, Patel R, Zhouandai A, Rychtář J, Taylor D. 2020 A game-theoretical analysis of Poliomyelitis vaccination. *Journal of Theoretical Biology* **499**, 110298.
75. Sykes D, Rychtář J. 2015 A game-theoretic approach to valuating toxoplasmosis vaccination strategies. *Theoretical Population Biology* **105**, 33–38.
76. Acosta-Alonzo CB, Erovenko IV, Lancaster A, Oh H, Rychtář J, Taylor D. 2020 High endemic levels of typhoid fever in rural areas of Ghana may stem from optimal voluntary vaccination behavior. *Proceedings of Royal Society A* p. 20200354.
77. Verelst F, Willem L, Beutels P. 2016 Behavioural change models for infectious disease transmission: a systematic review (2010–2015). *Journal of The Royal Society Interface* **13**, 20160820.
78. Das ML, Rowland M, Austin JW, De Lazzari E, Picado A. 2014 Do size and insecticide treatment matter? Evaluation of different nets against *Phlebotomus argentipes*, the vector of visceral leishmaniasis in Nepal. *PLoS One* **9**, e114915.
79. Dinesh DS, Das P, Picado A, Davies C, Speybroeck N, Ostyn B, Boelaert M, Coosemans M. 2008 Long-lasting insecticidal nets fail at household level to reduce abundance of sandfly vector *Phlebotomus argentipes* in treated houses in Bihar (India). *Tropical Medicine & International Health* **13**, 953–958.
80. de Souza Leal MMC, Ovallos FG, de Castro Gomes CM, de Oliveira Lavitschka C, Galati EAB. 2014 Host-biting rate and susceptibility of some suspected vectors to *Leishmania braziliensis*. *Parasites and Vectors* **7**, 1–11.
81. Hartemink N, Vanwambeke SO, Heesterbeek H, Rogers D, Morley D, Pesson B, Davies C, Mahamdallie S, Ready P. 2011 Integrated mapping of establishment risk for emerging vector-borne infections: a case study of canine leishmaniasis in southwest France. *PLoS One* **6**, e20817.
82. Sacks DL, Perkins PV. 1985 Development of Infective Stage Leishmania Promastigotes within Phlebotomine Sand Flies. *The American Journal of Tropical Medicine and Hygiene* **34**, 456–459.
83. Srinivasan R, Panicker K. 1993 Laboratory observations on the biology of the phlebotomid sandfly, *Phlebotomus papatasi* (Scopoli, 1786). *Southeast Asian Journal of Tropical Medicine and Public Health* **24**, 536–536.
84. CIA. 2020 The World Factbook–South Asia: India. <https://www.cia.gov/library/publications/resources/the-world-factbook/geos/in.html>. Accessed June 17, 2020.
85. World Bank. 2020 Life expectancy at birth. https://data.worldbank.org/indicator/SP.DYN.LE00.IN?cid=GPD_10. Accessed April 13, 2020.

- 86. Hailu A, Gramiccia M, Kager P. 2009 Visceral leishmaniasis in Aba-Roba, south-western
Ethiopia: prevalence and incidence of active and subclinical infections. *Annals of Tropical*
*Medicine & Parasitology* **103**, 659–670.
- 87. Kumar A, Saurabh S, Jamil S, Kumar V. 2020 Intensely clustered outbreak of visceral
leishmaniasis (kala-azar) in a setting of seasonal migration in a village of Bihar, India. *BMC*
*Infectious Diseases* **20**, 10.
- 88. Ostyn B, Gidwani K, Khanal B, Picado A, Chappuis F, Singh SP, Rijal S, Sundar S, Boelaert
763 M. 2011 Incidence of symptomatic and asymptomatic *Leishmania donovani* infections in high-
764 endemic foci in India and Nepal: a prospective study. *PLoS Neglected Tropical Diseases* **5**, e1284.
- 89. van Griensven J, Balasegaram M, Meheus F, Alvar J, Lynen L, Boelaert M. 2010 Combination
therapy for visceral leishmaniasis. *The Lancet Infectious Diseases* **10**, 184–194.
- 90. Sundar S, Singh A, Chakravarty J, Rai M. 2015 Efficacy and safety of miltefosine in treatment
of post-kala-azar dermal leishmaniasis. *The Scientific World Journal* **2015**.
- 91. Ghosh S, Das NK, Mukherjee S, Mukhopadhyay D, Barbhuiya JN, Hazra A, Chatterjee M. 2015
Inadequacy of 12-week miltefosine treatment for Indian post-kala-azar dermal leishmaniasis.
*The American Journal of Tropical Medicine and Hygiene* **93**, 767–769.
- 92. Jervis S, Chapman LA, Dwivedi S, Karthick M, Das A, Le Rutte EA, Courtenay O, Medley GF,
Banerjee I, Mahapatra T. 2017 Variations in visceral leishmaniasis burden, mortality and the
pathway to care within Bihar, India. *Parasites and Vectors* **10**, 601.
- 93. Sundar S, Arora R, Singh SP, Boelaert M, Varghese B. 2010 Household cost-of-illness of visceral
leishmaniasis in Bihar, India. *Tropical Medicine & International Health* **15**, 50–54.
- 94. Hasker E, Singh SP, Malaviya P, Singh RP, Shankar R, Boelaert M, Sundar S. 2010 Management
of visceral leishmaniasis in rural primary health care services in Bihar, India. *Tropical Medicine*
*& International Health* **15**, 55–62.
- 95. Ozaki M, Islam S, Rahman KM, Rahman A, Luby SP, Bern C. 2011 Economic Consequences
of Post-Kala-Azar Dermal Leishmaniasis in a Rural Bangladeshi Community. *The American*
*Journal of Tropical Medicine and Hygiene* **85**, 528–534.
- 96. Odomos Naturals. 2020 Dabur Odomos Naturals Mosquito Repellent Spray.
[https://www.amazon.in/Odomos-Naturals-Mosquito-Repellent-Spray/
dp/B01CJVG5TM/ref=sr_1_2?dchild=1&fpw=pantry&keywords=deet&qid=
1593529178&s=pantry&sr=8-2&srs=9574332031](https://www.amazon.in/Odomos-Naturals-Mosquito-Repellent-Spray/dp/B01CJVG5TM/ref=sr_1_2?dchild=1&fpw=pantry&keywords=deet&qid=1593529178&s=pantry&sr=8-2&srs=9574332031). Accessed June 30, 2020.
- 97. Killick-Kendrick R, Rioux J. 2002 Mark-release-recapture of sand flies fed on leishmanial dogs:
the natural life-cycle of *Leishmania infantum* in *Phlebotomus ariasi*. *Parassitologia* **44**, 67–71.
- 98. Guilvard E, Wilkes T, Killick-Kendrick R, Rioux JA. 1980 Ecologie des Leishmanioses dans le
Sud de la France-15. Déroulement des cycles gonotrophiques chez *Phlebotomus ariasi* Tonnoir,
1921 et *Phlebotomus mascittii* Grassi, 1908 en Cévennes. Corollaire épidémiologique. *Annales de*
*Parasitologie Humaine et Comparée* **55**, 659–664.
- 99. Ready PD. 2008 *Leishmania* manipulates sandfly feeding to enhance its transmission. *Trends*
*in Parasitology* **24**, 151–153.
- 100. Rogers ME, Bates PA. 2007 *Leishmania* manipulation of sand fly feeding behavior results in
enhanced transmission. *PLoS Pathogens* **3**, e91.
- 101. Shimozako HJ, Wu J, Massad E. 2017 Mathematical modelling for Zoonotic Visceral
Leishmaniasis dynamics: a new analysis considering updated parameters and notified human
Brazilian data. *Infectious Disease Modelling* **2**, 143–160.
- 102. Le Rutte EA, Chapman LA, Coffeng LE, Jervis S, Hasker EC, Dwivedi S, Karthick M,
Das A, Mahapatra T, Chaudhuri I. 2017 Elimination of visceral leishmaniasis in the Indian
subcontinent: a comparison of predictions from three transmission models. *Epidemics* **18**,
67–80.
- 103. Das V, Siddiqui N, Verma R, Topno R, Singh D, Das S, Ranjan A, Pandey K, Kumar N, Das
P. 2011 Asymptomatic infection of visceral leishmaniasis in hyperendemic areas of Vaishali
district, Bihar, India: a challenge to kala-azar elimination programmes. *Transactions of the Royal*
*Society of Tropical Medicine and Hygiene* **105**, 661–666.
- 104. Sundar S, Sinha PK, Rai M, Verma DK, Nawin K, Alam S, Chakravarty J, Vaillant M, Verma
809 N, Pandey K. 2011 Comparison of short-course multidrug treatment with standard therapy

- for visceral leishmaniasis in India: an open-label, non-inferiority, randomised controlled trial.
*The Lancet* **377**, 477–486.
- 105. Zijlstra EE, Kumar A, Sharma A, Rijal S, Mondal D, Routray S. 2020 Report of the Fifth Post-
Kala-Azar Dermal Leishmaniasis Consortium Meeting, Colombo, Sri Lanka, 14–16 May 2018.
*Parasites Vectors* **13**.
- 106. Ahasan H, Chowdhury M, Azhar M, Rafiqueuddin A, Azad K. 1996 Deaths in visceral
leishmaniasis (Kala-azar) during treatment. *Medical Journal of Malaysia* **51**, 29–32.
- 107. Courtenay O, Peters NC, Rogers ME, Bern C. 2017 Combining epidemiology with basic
biology of sand flies, parasites, and hosts to inform leishmaniasis transmission dynamics and
control. *PLoS Pathogens* **13**, e1006571.
- 108. Arriola L, Hyman JM. 2009 Sensitivity analysis for uncertainty quantification in
mathematical models. In *Mathematical and statistical estimation approaches in epidemiology* , pp.
195–247. Springer.
- 109. Cohen JE, Gürtler RE. 2001 Modeling household transmission of American trypanosomiasis.
*Science* **293**, 694–698.
- 110. Perry D, Dixon K, Garlapati R, Gendernalik A, Poché D, Poché R. 2013 Visceral leishmaniasis
prevalence and associated risk factors in the Saran district of Bihar, India, from 2009 to July of
2011. *The American Journal of Tropical Medicine and Hygiene* **88**, 778–784.
- 111. Bern C, Courtenay O, Alvar J. 2010 Of cattle, sand flies and men: a systematic review of risk
factor analyses for South Asian visceral leishmaniasis and implications for elimination. *PLoS*
*Neglected Tropical Diseases* **4**, e599.

Appendix B

Response to reviewers' comments

We thank the editor(s) and the reviewers for careful reading of the manuscript and for their suggestions and comments. We appreciate all the feedback given to us. We believe we successfully addressed all the comments and issues. Detailed list and responses are below.

Reviewer 1 wrote:

Comments to the Author(s)

This is an interesting and very well written paper about mathematical modeling of individual behavior regarding the use of insecticide-treated nets (ITN) as preventive method against visceral leishmaniasis (VL). The authors present in detail a modification of the mathematical model describing the sand flies-humans dynamics, previously developed by Stauch et al. (2011, PLoS Neglected Tropical Diseases). The authors study the dynamical system within a game-theoretical framework, minimizing the expected cost of using ITN. The main outcome of the model is the proportion of people using ITN that may lead to VL elimination.

Our response: We thank the reviewer for a careful reading of the manuscript and for comments and suggestions for its improvement

Reviewer 1 wrote: The manuscript may benefit from reformulating and/or explaining some modeling choices. For instance, in line 86, the authors mention free-riders but don't mention whether or not they model the free-rider behavior explicitly.

Our response: We added "*incorporated in our model as individuals that do not use ITNs,*" to the sentence

Reviewer 1 wrote: In line 110, the authors say that equation (2.1) may be interpreted as if "ITNs provide a very effective protection", while their model considers perfect protection, instead.

Our response: It was an unfortunate wording on our end. We deleted the sentence and added this just before equation (2.1): *For simplicity, we assume that ITNs offer a perfect protection and thus the force of infection is ...*

We further added paragraph about possible model extension to the discussion where we mention how to include imperfect protection.

Reviewer 1 wrote: Also, the choice of the computation/definition of R_0 could be supported by a reference.

Our response: We added a reference to Anderson and May 1991, a recent review by Delameter et al and van den Driessche 2002

Reviewer 1 wrote:

Assumption (iii) in line 176 could also be supported by a reference. In addition, the authors could contrast this assumption with available data on the population size. If not, they could discuss the limitations of such an assumption.

Our response: We did not find a reference but we added the following “*This change makes the ODE system less sensitive to changes in the birth rate and death rates. In the original model of \cite{stauch2011visceral}, even a small change in α_H , μ_H , or μ_K would destabilize the system and result in either exponential growth or exponential extinction of the population. In our model, a (reasonable) change in Λ , μ_H , or μ_K will cause the system to converge to a potentially different population size, but the population will not go extinct nor grow above any bound. We can also solve for the equilibria of the dynamics.*”

We did not include it in the text but we noticed that Stauch et al (2011) never mentioned the value of α in their manuscript. We suspect that they just picked the value of α_H at the end once all other parameters were fixed. That (and only that) choice of α_H made their ODE system converge to an equilibrium rather than going to 0 or infinity.

Reviewer 1 wrote: Please consider discussing the stability of the equilibria, which may give further insight about the system behavior and thus helping to interpret the results, as well as discussing the results of the KalaNet trial (see line 468).

Our response: We added the following after the calculation of R_e : *It follows from \eqref{eq:IF} that the endemic equilibrium exists only if $R > 1$. While we did not perform a formal stability analysis, the ODE system -- although large -- is quite standard and similar to one considered in \cite{wei2008epidemic}. Consequently, the disease-free equilibrium is globally asymptotically stable if $R \leq 1$, and unstable if $R > 1$. The endemic equilibrium is locally asymptotically stable if $R > 1$.*

Reviewer 1 wrote:

The authors compute the reproduction number for the system subject to a control measure. Therefore, it is the effective reproduction number and not the basic reproduction number. The authors could use the notation R_e , instead of R_0 , where $R_e = R_e(p)$.

Our response: We thank the reviewer for the comment. We changed to R_e as suggested and now call it properly the effective reproduction number instead of basic reproduction number.

Reviewer 1 wrote:

Please recall the parameters values used to produce figure 4, in the caption. Please expand the description of figures 5 and 6 in their captions, including a short description of the parameters set depicted in the x-axes.

Our response: We added the line: *The parameter values are as specified in Tables \ref{tab: Sand Flies}, \ref{tab: Humans}, \ref{tab: Treatment}. to the caption of Figure 4. There already was a similar statement in the caption to Figure 5 and 6. The model has too many parameters to list in the caption.*

Reviewer 1 wrote:

Please ensure notation consistency for numbers of 4 or more digits.

Our response: We adjusted the output of sensitivity analysis to 4 digits. The difference between ITN use needed for complete elimination and the Nash equilibrium coverage (resulting from optimal voluntary use) is quite small and 5 digits were needed to show the difference.

Reviewer 2

Comments to the Author(s)

SUMMARY

My recommendation to the Editor is to accept this manuscript, pending major corrections. Please interpret 'major' in the sense that these suggestions will likely affect the results. I thought the quality of the research and its write-up were of a high standard for this modelling manuscript and would very much like to see its corrected version added to the VL research base.

The findings: The authors have developed a model by amending an existing one and parametrising it with mostly reasonable values and ranges. The two research impacts are:

A) They obtain a threshold estimate for the ITN coverage required for (zero) disease elimination. (labelling this as RI-A)

B) Considering that ITN use is voluntary, they make a sensible and (probably for VL) novel application of game theory to assess if voluntary ITN use alone, influenced by the costs and benefits at the individual level, is sufficient to meet this threshold. (RI-B)

Quality: It is of a high research standard: indicated by the mathematical foundations and clear exposition in the text, strongly informed in model structure and parameters by existing VL research, the rigour of the analysis including validation and sensitivity analysis. Moreover, it is the first manuscript I have reviewed which provides working code and reproducible results! It therefore ticks many best practices of 21st century modelling.

Our response: We thank the reviewer for the kind and encouraging words!

Reviewer wrote:

Potential impact: It also contributes to the ongoing discussion following the unfortunate findings of the Kalanet trial in the previous decade. This is the primary reason why ITNs do not form a part of today's VL control. WHO has launched the NTD 2021-2030 roadmap earlier in 2021 and so this could be an ideal time for research during this receptive policy window. It could reawaken minds and inform future epidemiological and

entomological study designs to definitively answer if ITNs have a contributing role in VL control on the Indian subcontinent. The manuscript could work harder at highlighting this point for the VL policy maker audience.

Our response: We rearranged the discussion and tried to make the point as reviewer suggest.

Reviewer wrote:

To open up this research piece to further discussion and improve its publication impact I suggest releasing the initially submitted draft as a preprint and later interacting with the VL research community at forthcoming conferences such as WorldLeish 2022 which will probably start accepting abstracts later this year, but this of course is at the authors' discretion.

Our response: We will definitely keep the conference in mind although we may not be able to physically travel anywhere in near future. We are more than happy to collaborate and interact with VL community if a remote collaboration is a possibility. We do not have any contacts in VL community and would be very happy if the reviewer could help us share our work.

Reviewer wrote:

Scope of this review: I spent 3 days reviewing the manuscript, code and writing up this review. I did not check the majority of the mathematical formulae at an analytical level. Instead I focussed on code correctness from code lines 1-478, parameter suitability and if this modelling was appropriate for VL.

Conflict of interest: I declare that I have included citations within this review which either myself or my supervisors (T Déirdre Hollingsworth & Lloyd Chapman) are co-authors. The authors are not compelled to add these citations. I included them to provide justification and to encourage the authors why a suggestion is necessary or why it has been raised.

Current major limitations: I believe there are some major bugs in the code that would affect results but these should be quick to fix. Some coding practices may also help improve the confidence in the code correctness. Some parameters may not fit the setting or understanding of the disease the best and I have highlighted the ones most likely to impact results, partly from the authors' helpful sensitivity analysis; this may require a rapid review of the recent literature for parameter updates but again this shouldn't take long. No change to the model structure is necessary. Finally, although the manuscript text itself was painless to read, it could focus better at the main audience—the VL research community. Some of these are detailed below and are also attached on commented files.

BEFORE THE AUTHORS READ THIS REVIEW (OR THE PUBLIC, AFTER PUBLICATION)

My overriding motivation for this peer review is to protect the field from errors in research methods or misinterpretation of results, that could make policy makers make suboptimal decisions in VL programmes that ultimately harm people. The annual

research output for VL modelling is relatively small compared to other diseases, so you can realise that avoidable errors could have more impact in this context than others. I haven't seen the authors' names from previous VL modelling research, so if you are new to VL then welcome to the field! None of the following review is intended to 'put you in your place' nor be patronising, and your future modelling for this neglected tropical disease is very much welcomed. In the spirit of open science some of the reviewers may have declared their names to be made public and in return I hope the authors allow all reviews to be open (for reviewers who allow it).

REVIEW STRUCTURE

This review is structured by the manuscript then code run, and grouped by things essential to impacting the results' values or presentation (major) or affecting the understanding/model improvements. This review is long but I believe in thoroughness and once some of what I suggest is corrected, then I think what you have got here could be really valuable to the VL field.

Our response: We cannot thank the reviewer enough for all the input and hard work that was put in the review. We benefited tremendously from all the comments, suggestions and provided references.

Reviewer wrote:

MANUSCRIPT

MAJOR

Top, most important

- An extensive range of research has been drawn upon to inform parameters selected. Some suggestions have been made from empirical sources for the refinements of some parameters. See mns attachment for more details.

Our response: We list and provide detailed responses to all major comments in the manuscript separately below.

Reviewer wrote:

- "Multiple lab-based studies suggest that female sand flies have fairly short adult lifespans (<20 days) [16] with further reductions when infected [15]" (<https://journals.plos.org/plosntds/article?id=10.1371/journal.pntd.0009033>) therefore it is unrealistic to assume the mortality of I_F in Fig 1 is the same as S_F or E_F (=mu_F).

Our response: We added a subsection with several paragraphs about model limitations and possible extensions and added the fact about varying mortalities as one of several extensions to consider. While the reviewer writes above that "No change to the model structure is necessary" and this addition seems like a minor one, it actually requires a relatively major change in many of the analytical calculations(see also our response about stability re reviewer #1 comment to line 176). Related to several other reviewer's comments, we are more than open to collaborate with the proper set of authors and stakeholders and create an even better model(s) in the future.

Reviewer wrote:

- Making it more policy-friendly: reduce the amount of mathematical formulae by moving formulae (lines 212-223 & 446-452) describing the intermediate mathematical logic to a

Supplementary information as there's quite a few pages to flip through from line 185-225. Please do not understand what I mean, I just mean reduce the amount, the level at which you are communicating is fine and there is no need to dumb things down. Your short-hand notation like Comp on line 197 was clever and for instance Eqns 3.1-3.15 could be represented as a one-liner, something like $dX/dt = 0$ where $X=\{\text{etc}\}$.

Our response: We moved most of the math to the appendix as suggested.

Reviewer wrote: Although the Discussion covers limitations in the findings, model limitations are also necessary.

Our response: We now have a subsection on model limitation and extension.

Reviewer wrote: Also despite the model being validated against previous trial data, what would be the external validity of these findings to say Nepal?

Our response: We added the following to the section on model validation:

To validate the model on another data set, one would have to potentially update the parameter values to properly reflect the time and location of the experiment where the data came from. Then, we can use formulas for endemic equilibrium from the Appendix A{ref{ap: endemic}} to obtain distribution of population across different model compartments.

Reviewer wrote:

- Parts of the Introduction feel a bit encyclopaedic i.e. they demonstrate an excellent understanding of the natural history of disease progression and epidemiology however some of the detail may not be necessary if it is not part of the model. Therefore, streamlining the Introduction would save you several lines.

Our response: We followed reviewer's suggestions in the actual manuscript and removed the zoonotic VL models from the introduction. If we misunderstood, we can insert it back.

Reviewer wrote:

On the other hand, the authors introduce game theory concepts that are likely not to be familiar to the VL audience and elaborating a little further would save the reader having to follow the citations to understand the basic concepts of game theory.

Our response: Connected to comments of Reviewer 1, we added some more about the game theory.

Reviewer wrote:

- Title may need changing depending on the corrected results. The current title for the presented results sounds a bit optimistic given line 478. It also gives the impression that RI-B is the primary finding, plus the use of 'game theory', 'nash equilibrium' keywords alongside, however the abstract doesn't mention it and only mentions RI-A. I think the main body of the mns does give proper precedence to RI-A and then RI-B as a secondary and consequential finding. In fact, I think in the weight of scientific evidence, that RI-B is weaker than RI-A, as quite broad assumptions are applied to achieve RI-B, whereas RI-A is mostly informed from the published literature; this is elaborated on the attached manuscript lines 87-91. The authors may believe their main contribution to the

field is the novel application of RI-B, when in fact I think it could be the robust model developed for RI-A which has the greatest effect.

Our response: We revised the title and the keywords. Based on some other reviewer's suggestions, we added a bit more emphasis on the parameter estimation and in particular the role of asymptomatic individuals.

Reviewer wrote:

Have any other papers you are aware of estimated an ITN coverage threshold for elimination? I can't think of any. If not then this needs to be emphasised more in the mns as a key finding to highlight the contribution your work is making.

Our response: We are not aware of any other work applying math modelling of ITN to VL and we inserted a statement early in the discussion in that effect.

Reviewer wrote: Also, as this study is based in the IN/NP country setting, whereas VL also affects East Africa & South America, it may be worth including the regional setting eg "in Bihar, India" in the title.

Our response: We added as suggested

Reviewer wrote:

MINOR

In no particular order:

- Other than suboptimal ITN coverage, other reasons have not been mentioned that could explain the lack of an effect of the Kalanet trial e.g. from [42] "In theory, LNs could be an effective tool to prevent *L. donovani* transmission, as *P. argentipes* are supposed to bite people indoors while they sleep [14]. However, recent entomological findings in India indicate that *L. donovani* vectors are more exophilic and exophagic than previously reported [15,16]. If *P. argentipes* bite people outdoors (e.g., in the early evening when and where bed nets are not deployed), LNs will have a limited impact on *L. donovani* transmission".

Our response: We added the paragraph about possible model extension in this direction to the subsection "model limitation and possible extensions"

Reviewer wrote:

- On p3 line 40 you state that ITN ownership is associated with wealth however the assumptions on p4 line 78-79 collapses this to a single mean level for everyone. The heterogeneity of affordability for medical costs produces uncertainty that is hidden in the point estimate $p_{\{HI\}}$. Also, the VL susceptibility is bound to reduce with wealth, as malnutrition and certain interior wall types are VL risk factors.

Our response: We added this as another of the model limitations

Reviewer wrote:• Consider adding a PRIME-NTD summary table to your appendix to describe how you have interacted with policymakers. <https://journals.plos.org/plosntds/article?id=10.1371/journal.pntd.0008033>

Our response: We did as suggested.

Reviewer 2 wrote: • Some readers may interpret “elimination” as “elimination as a public health concern” rather than true/zero elimination. Clarifying this at first use would help.

Our response: We thank the reviewer for bringing this to our attention. We do not specifically say 0 cases now or use the phrase complete elimination. At the same time, we realized that the voluntary use of ITNs brings the disease to elimination as a public health concern and explicitly say it too.

Reviewer 2 wrote:

• Note that eradication means zero disease globally so perhaps you mean elimination which is local to a particular region.

Our response: We thank the reviewer for spotting this; this word escaped our attention.

Reviewer 2 wrote:

• I note that all the current authors are from US institutions. Although the authors did an excellent job in summarising the pertinent epidemiology and dynamics, I think the addition of authors with in-country experience would have greatly informed on the real-world conditions in Bihar, India. Please note I am basing the authors in-country experience on their institution location which could be false. Journals have rejected NTD manuscripts from my European colleagues last year because they didn't have authors from the affected continent and I have to agree with this policy. I encourage Editors to challenge this in future and the authors to collaborate with in-region researchers as otherwise research quality could suffer. Also it would deprive upcoming researchers in these regions from contributing to research that affects their home.

Our response: We share reviewer's sentiment. For what it is worth, Jan Rychtar was in Bihar in 2010 (specifically in the vicinity of Patna, Gaya and Bodh Gaya). While he did not work on VL at that time, he was able to experience the conditions of rural India. The nature of this particular project (6 intensive weeks during summer 2020 with many last minute changes due to COVID-19) prevented us from having too extensive collaborations on this.

However, we are very open to collaboration on further VL projects. Profs. Rychtar, Taylor and Lu will be happy to work closely with the reviewer and/or his/her team on VL modelling in the future. In fact, we have another cohort of students coming in June 2021 and, with an appropriate planning, this could give our students an excellent experience according to the best practices described by the reviewer and the provided references.

Reviewer 2 wrote:

CODE RUN

Excellent use of commenting and code structuring in the MATLAB file. Believe it or not I did run it and also read through it to try and find bugs! So, thank you very much for spending the time in making the code understandable; it was not wasted time! Please note a commented PDF of the code is also attached.

Our response: Thank you for spending the time to check and run the code. We are glad the comments were helpful. In the revision, we tried to improve based on your suggestions.

MAJOR

Top, most important:

Reviewer 2 wrote:

- Code lines 60 & 115 incorrect parameter values. See attached. Editor: please don't mark the paper down for these errors. Think how many papers get published without ever releasing their code!

Our response: The values are and were correct. The issue stems from units (years in Matlab versus months in the manuscript). We believe the months are more natural for the manuscript. However, for figures, years are more natural so it was coded in years for Matlab. The comments about conversion were already in the original code. We rewrote $1/70$ to $1/(70*12)*12$ [in fact to $1/(68.7*12)*12$ due to new value as pointed by the reviewer at another place] and $1/40$ to $1/(40*12)*12$. The values did not change but now they are written in a style that is consistent with other parameter values

Reviewer 2 wrote:

- Manuscript figs 2 & 3 were not produced when the code was run so not all the figures are reproducible.

Our response: The commands that called the functions to generate the figures 2 and 3 were indeed left commented out in the submitted code by an overlook. We uncommented them and added a specific disp to the code to make it clearer that the figure are being generated.

Reviewer 2 wrote:

- I am not entirely confident that the code runs as the authors may have intended. This is not just because of the initial errors found on code lines 60 & 115 but because I could not find any unit tests in the code to check the correctness of the code output. This could be improved by simple error checks inserted into functions to check function arguments (especially those called multiple times), and checks on the expected value ranges and data structures from each function return, which is essential in my opinion to be confident in the results. This is set out in further detail in Responsible modelling: Unit testing for infectious disease epidemiology

(<https://www.sciencedirect.com/science/article/pii/S1755436520300451>). I encourage Prof Rychtář to involve the other co-authors in the collective task of pair code review and mentioning this in the Methods to add to the confidence in the code correctness.

Our response: We added a subsection (Numerical implementation) and added a reference to the paper. We included several consistency tests in the code (we more or less had them in previous versions of the code but deleted them in the final submission as we did not realize that they should be an integral part of the code as well. We debugged the code as it was written and constructed (prior the original submission). We often did not end the lines by ";" and inserted plenty of "pause" and manually redid the calculations on the paper/calculators for numerous values to make sure the code runs as intended. We also generated figures that are no longer part of the code just to see that the outputs make sense.

We fully agree with the reviewer that without such tests, the code would be full of bugs. In fact, by reinstating the tests, we found and corrected a small typo in the crucial function for obtaining equilibria.

MINOR

In no particular order:

Reviewer 2 wrote:

- The file runs in one go and outputs all 41 MATLAB figs to screen and also saves as PDFs; this is good. It is unclear however from the screen, which of the figures relate to which part of the code, unless code chunks are run separately. Therefore, can the disp() and saveFig() also include a title to match it to the code please. This caused some inefficiency to the reviewing process.

Our response: We now added the command that displays the file name and links it to the figure to the file name

Reviewer 2 wrote:

- Please could you make your publication, SI and code also available on Zenodo or similar archived repo as some journals have a time limit for archive removal.

Our response: We opted for an open access journal so that the final and reviewed version could be accessible to everyone.

Reviewer 2 wrote:

- Is MATLAB fig1 = manuscript fig 4? If yes then this has been successfully reproduced. The same goes for manuscript figures 5 {7-10,12,15-17,20-27,18-19,11} & 6 {13-14,29-31,33-38}? where the left-to-right order here refer to their top left to top-right then carriage return order in the sub figure arrays in the manuscript. There were some orphaned MATLAB figures generated {28,32,39-41}. Could these be described by a list of figures and caption for just these orphaned figures in a Supplementary information? Otherwise if they were for the authors' own checking then better to comment out their figure/PDF generation when the code is run as a whole.

ENDS=====

Our response: We commented out the sections of the code that do not produce figures for the manuscript. (we also uncommented the sections that produce the figures for mns but were left commented out in the original submission).

Responses to comments in the pdf for the matlab file

- We added a check for the optimization toolbox. The code should now run without it (although parameters values may be a bit off if the user changes some parameters. The transmission probability values were hardcoded in the case the user does not have the toolbox). Proper prompts should be now displaying if the code run without the toolbox (although we were not able to test it since our version has the toolbox)

Reviewer 2 wrote: On mns line 55 you are informed by Stauch on vector control parameters but why are non-vector parameters like f_{HS} also being changed?

Our response: Our understanding of the work of Stauch et al (2011) was that they run the optimization technique to fit their model to KalaNet data and through this they estimated many parameters. We tried to avoid the parametrization by fitting to data as much as possible. Since

we could find the data for f_{HS} (and others) directly, we used the values we found rather than those that were estimated.

Reviewer 2 wrote:

1d is far too short for VL onset to treatment time for this study setting. Also it's not within the range quoted in Table 3. Pls double check that other parameters are within ranges quoted in mns tables.

Our response: We agree that one day is not realistic. We did not use one day but one month as specified in Table 3 and earlier in the code. Reviewer's comment is in the section where we set the parameters to the values used in Stauch et al. They used 1 day and so we had to code it as one day here.

Reviewer 2 wrote: But I thought that earlier you assumed that treated cases are not infectious.

Our response: We agree and we did assume it. Reviewer's comment is in the section where we define parameters as in Stauch, so we had to use what they did

Reviewer 2 wrote: As $\text{sum}(\text{KalaNet})=1.005>1$, does this require normalisation first?

Our response: The human parameters add to 1 and the flies are extra. We agree that the issue was not clear from what we wrote there, so we added a comment to the code to make it a bit clearer

Reviewer 2 wrote: The command window reports this as a local minimum. Would it be worth trying random parameter initial conditions to check that lower local minima don't exist?

Our response: We put back the section of the code where we test exactly for that. We also included a section in the appendix where we describe that we randomly seeded the search with 1000 initial values and searched among all local optima we found. It turned out that starting the search at 0 yields close enough results.

Reviewer 2 wrote: Why not $(1/38*12, 1/10*12)$ as per Table 3 range?

Our response: The reviewer is correct, there was no reason for the values used in the matlab, so we regenerated with values from the table

Responses to comments in the pdf manuscript file

Reviewer wrote: I'm a bit confused with the result of Fig 3 and was wondering if your text in 5a could help. It just seems a bit counterintuitive that for higher p the probability of i_D and i_P Asx humans infecting sandflies is higher.

Our response: We added the following sentence to the end of 5a to clarify

Note that the functions are indeed increasing in β - to achieve the same disease prevalence as measured in KalaNet trial when the population uses higher level of protection β , the disease must be more transmittable.

Reviewer wrote: 14 is a case report. Is there a better source that can more broadly characterise clinical features?

Our response: We agree and this is why we cited two sources. We can eliminate the reference to the case report completely if this is preferable.

Reviewer wrote: Staff from CARE have also reported KA symptoms to oscillate ie a fever that comes and goes over weeks.

Our response: We were not able to locate the information. We will be happy to include it, but could you please provide us with the reference? One of the suggested WHO publications talks about CARE, but it is regarding deaths reports, nothing about symptoms.

Reviewer wrote: You may be interested by this recent Xeno study [https://www.thelancet.com/journals/lanmic/article/PIIS2666-5247\(20\)30166-X/fulltext](https://www.thelancet.com/journals/lanmic/article/PIIS2666-5247(20)30166-X/fulltext) and commentary [https://www.thelancet.com/journals/lanmic/article/PIIS2666-5247\(20\)30222-6/fulltext](https://www.thelancet.com/journals/lanmic/article/PIIS2666-5247(20)30222-6/fulltext) that supports the view that Asx have no role in transmission versus our spatiotemporal analysis that found a 2% relative infectiousness of Asx vs KA was consistent with the data <https://www.pnas.org/content/117/41/25742>.

Our response: We added a relatively detailed analysis of findings from Singh et al 2021. We found that despite the highlights indicating up to 88% transmission from active VL and PKDL cases, only about 10% of bites actually result in parasite transmission. We now discuss this as well as the role of asymptomatic individuals on several places of the paper. In particular, we point out that what Singh et al call asymptomatic cases, Stauch et al (and consequently us) call recovered; i.e., there is no factual difference between their findings that those individuals do not transmit the parasite and our model (that assumed no transmission from recovered classes from the beginning).

Reviewer wrote: Is this relevant though for VL modelling in India?

Our response: We agree that the relevance to India is low but wanted to point out to other models of VL relevant to other regions. We can delete the sentence if needed.

Reviewer wrote: But by what process do individuals receive this information? Surely incidence and ITN coverage are not known by villagers.

Our response: The reviewer is correct. We acknowledged the issue in the section on model limitation.

Reviewer wrote: However this line of reasoning is based on vaccines, most of which confer long-term protection and are a one-off decision based on information available then, whereas ITNs have to be continually used by homeowners and also repaired/replaced while the person has to be continually receiving information on the threat VL poses, since if they don't they then forget about it due to other acute health priorities. For a disease like VL with long incubation periods it may be difficult for people to infer/see the benefit of ITNs for themselves as the time of infection is hidden and onset of symptoms happens 5 months later. Therefore they will have forgotten their poor adherence of ITNs when exposure occurred, so they won't see a clear link between cause and effect.

Our response: We agree with what the reviewer wrote but believe that it stems from our improper introduction of game theory. The game theory assume that individuals are selfish and act in their own self-interest We added this to the intro paragraphs: *In our setting, the game theoretical framework assumes that people will sleep under ITN not to protect others, but to protect themselves. Specifically, people will not start sleeping under ITNs once they learn they have VL, but they may start sleeping under ITN once they learn their neighbor has VL.*

The time lag the reviewer mentions is important though and we added it to the model limitations/possible extension.

Reviewer wrote: Was the code also adapted from theirs or did you write this from scratch?

Our response: We wrote the code from scratch.

Reviewer wrote: This unrealistically assumes 100% net adherence and efficacy.

Our response: We agree and put a paragraph about it in the discussion.

Reviewer wrote: I_A could be more intuitive as a label

Our response: We agree. We tried to adhere to the original Stauch et al (2011) as much as possible. Their notation is perhaps not entirely intuitive, but it has an inner logic and so after various attempts on our side to change it, we reverted mostly to theirs. We just dropped extra subscripts to make our formulas fit on a line as much as possible,

Reviewer wrote: What evidence shows that dormant PKDL or active PKDL cases are LST-ve? Is a weaker version more realistic? ie LST status predicts progression to PKDL but some PKDL cases may still be LST-ve?

Our response: Here we followed Stauch et al (2011) exactly. The only effect of whether PKDL are or are not always LST negative is on the position of I_TL and I_UL (lumped into I_HL in Stauch) in the model diagram. It does not influence any flow to or from the compartments.

Reviewer wrote: Is this true of Miltefosine though? Surely far less toxic than the previous SSG. Our response: we added: *(although a newer drug Miltefosine is far less toxic than previous SSG \citep{den2009developments})*

Reviewer wrote: Tidy diagram with consistent inter-compartment spacing and use of arrows in 8 directions. However the use of white space could be improved to reduce congestion in the I_{T1} area. Worth labelling the two groups for human host and sandfly host. Listing what symbols mean here as a fig1 caption to avoid referring back to p5. Worth stating in the caption that the arrow path annotations represent transition rates. Worth adding a traffic light label to each compartment to aid reader in understanding PCR/DAT/LST transitions. Shade infective stages or KA vs PKDL?

Our response: We did most of the suggestions. If we did not reduce the congestion enough, it was not because we did not want to or did not try, we just did not know how to do it better.

Reviewer wrote: but $f_L + f_S + f_D \neq 1$. See Table 3. Can they be normalised to fix this?

Our response: Thank you so much for catching this. They should add to 1 and they add to 1 in the matlab code (because we define on f_L and f_S and calculate f_D as $1 - f_L - f_S$ as specified in the right column of Table 3. However, the written numerical value of f_D was indeed incorrect and it is fixed now.

Reviewer wrote: extra 3 cols for PCR/DAT/LST would aid reading

Our response: We added the columns to the table as suggested

Reviewer wrote: I_{TL} duplic?

Our response: It was a typo, one is now fixed to I_{UL}

Reviewer wrote: Lifespan? As expectancy is used more for summaries of the lifespan distribution in demography.

Our response: We changed it to Expected lifespan in the light of this suggestion. We kept “life expectancy”, i.e., the average number of years that a newborn is expected to live (if current mortality rates continue to apply) for the corresponding human parameter.

Reviewer wrote: Why not γ_{UL} rather than γ_L for consistency with fig 1?

Our response: We tried to have only one subscript whenever possible because otherwise the formulae did not fit on one line too well. For most compartments and rates it meant to drop “H” from the subscript; here we had to drop U.

Reviewer wrote: Since these are intelligent guesses it may be worth parameter hunting in [https://www.thelancet.com/journals/lanmic/article/PIIS2666-5247\(20\)30166-X/fulltext](https://www.thelancet.com/journals/lanmic/article/PIIS2666-5247(20)30166-X/fulltext) and <https://academic.oup.com/cid/article/69/2/251/5144025>

Our response: We thank the reviewer for the reference. We incorporated the data from these papers into the model calibration (and the paper in general)

Reviewer wrote: Table 6 here from a recent mortality survey will provide better estimates https://www.who.int/docs/default-source/searo/evaluation-reports/independent-assessment-of-kala-azar-elimination-programme-in-india.pdf?sfvrsn=fa0d8baa_2

Our response: We are unable to decipher data from Table 6 and make estimates for mortality if untreated. While Table 6 that a significant number of patients die within a month from the diagnosis, it is not quite clear how long it took to be diagnosed and the report itself states that the mortality is associated with many factors, including late health care seeking. Nevertheless, we included a comment about it in the manuscript (and we thank the reviewer for pointing us in this direction).

Reviewer wrote: Doesn't the Indian govt pay for treatment costs, lost wages and travel costs? Perhaps this recent survey will help answer the reimbursement scheme's effectiveness (<https://journals.plos.org/plosone/article?id=10.1371/journal.pone.0227911>)

Our response: We added the following paragraph: *The Indian government provides a free care to VL patients; however, hoping for a quick cure of seemingly minor illness, patients prefer to access private providers which contributes to high out-of-pocket expenditures \citep{nair2020quality}. VL is often misdiagnosed in private hospitals but patients keep accessing care in private sector until no money is left \citep{nair2020quality}.*

Reviewer wrote: I'm a bit confused with the result of Fig 3 and was wondering if your text in 5a could help. It just seems a bit counterintuitive that for higher p the probability of i_D and i_P Asx humans infecting sandflies is higher.

Our response: We added the following to the end of the section: *Note that the functions are indeed increasing in ρ - to achieve the same disease prevalence as measured in KalaNet trial when the population uses higher level of protection ρ , the disease must be more transmittable. RESPONSE MAY CHANGE BASED ON HOW THE PARAMETERS FOR TRANSITION TURN OUT*

Reviewer wrote: worth adding caveats: people get perfect info, broken nets are replenished

Our response: We added (*provided people get perfect information and broken ITNs are replenished*)

Reviewer wrote: If you're talking about Fig 4, don't you mean cost function?

Our response: We added (which correlates with the mean cost function)

Reviewer wrote: This was for vaccination rather than vector control programmes.

Our response: Correct. We added reference to a recent paper by another group of our students that deals with Chagas and ITNs

<https://journals.plos.org/plosntds/article?rev=2&id=10.1371/journal.pntd.0008833>

Reviewer wrote: sort desc?

Our response: We sorted table 6 in descendent order

Reviewer wrote: split in two for infected/uninfected sandfly

Our response: We added a section on model limitation and possible extension and discuss this mortality split there. The incorporation of the split would result in a complete revision of the analytical formulas and calculations.

Reviewer wrote: [84] only refers to India. Rural Bihar specifically (and temporally closer to KalaNet data) available from 2011 online census. Same for human death rate μ .

Our response: We thank the reviewer for pointing us to the proper place to search and we updated the numbers accordingly to the data found on rural Bihar in online publications from Indian Census to life expectancy 68.7 in rural Bihar and crude birth rate 27.7 per 1000 in rural Bihar.

Reviewer wrote: Acknowledgements: Anyone to mention?

Our response: We added Dr. Wasserberg who pointed us in the right direction re vectors and we also thank the reviewers for their valuable feedback and comments.

Appendix C**ROYAL SOCIETY
OPEN SCIENCE****Mathematical modeling of the use of insecticide treated
nets for elimination of visceral leishmaniasis in Bihar, India**

Journal:	Royal Society Open Science
Manuscript ID	RSOS-201960.R1
Article Type:	Research
Date Submitted by the Author:	30-Mar-2021
Complete List of Authors:	Fortunato, Anna; University of Richmond Glasser, Casey; Virginia Tech Watson, Joy; Virginia State University Lu, Yongjin; Virginia State University Rychtar, Jan; Virginia Commonwealth University, Mathematics and Applied Mathematics Taylor, Dewey; Virginia Commonwealth University, Mathematics and Applied Mathematics
Subject:	Mathematical modelling < MATHEMATICS
Keywords:	Kala-azar, PKDL, Vector-borne diseases, Asymptomatic transmission, Parameter estimation
Subject Category:	Mathematics

Author-supplied statements

Relevant information will appear here if provided.

Ethics

Does your article include research that required ethical approval or permits?:

This article does not present research with ethical considerations

Statement (if applicable):

CUST_IF_YES_ETHICS :No data available.

Data

It is a condition of publication that data, code and materials supporting your paper are made publicly available. Does your paper present new data?:

Yes

Statement (if applicable):

The matlab code supporting this article have been uploaded as part of the supplementary material.

Conflict of interest

I/We declare we have no competing interests

Statement (if applicable):

CUST_STATE_CONFLICT :No data available.

Authors' contributions

This paper has multiple authors and our individual contributions were as below

Statement (if applicable):

A.K.F.: conceptualization, formal analysis, investigation, and writing - original draft

C.P.G: conceptualization, formal analysis, investigation, software, and writing - original draft

J.A.W. - conceptualization, formal analysis, investigation, and writing - original draft

Y.L. - supervision, writing - original draft

J.R. - writing - original draft, writing - review and editing, software, methodology, supervision, conceptualization, resources

D.T. - writing - original draft, writing - review and editing, methodology, supervision, conceptualization, formal analysis, validation, funding acquisition

The order of the authors was determined as follows: The first three authors are undergraduate students in alphabetical order. The last three authors are professors in alphabetical order.

ROYAL SOCIETY OPEN SCIENCE

rsos.royalsocietypublishing.org

Research

Article submitted to journal

Subject Areas:

mathematics, biology, epidemiology

Keywords:

Kala-azar, PKDL, Asymptomatic transmission, Parameter estimation, Vector-borne diseases

Author for correspondence:

Jan Rychtář

e-mail: rychtarj@vcu.edu

[revised manuscript text omitted]

The control of VL depends on chemotherapy treatment [12], vector control [29], bite prevention [30] and active case detection [31]. Human VL vaccines are not yet fully developed although several trials are under way [32,33] and their impact is already modelled [34]. Preventative measures in high risk areas include (1) avoiding sleeping in mud and thatched housing [1], (2) environmental management [35], (3) using insect repellent [36], (4) using insecticide treated netting, ITN, [37–39], (5) indoor residual spraying [7] and, most recently, (6) treated wall lining [40].

The ITN ownership varies significantly with wealth; almost all of the wealthiest households own an ITN while many of the poorest do not [41]. However, even in relatively poor areas, the use of ITNs was already demonstrated to be very cost-effective [42,43]. Human behavior such as inconsistent use due to hot weather or inadequate education diminishes the effectiveness of ITNs [39,44].

KalaNet, a cluster randomised controlled trial in Nepal and India evaluated the efficacy of long lasting ITNs in the prevention of VL [45]. It has been demonstrated that a cluster-wide distribution

[revised manuscript text omitted]

 2, 3, 4 and 5. The parameter values are estimated in the Section 4 based on empirical evidence and
 data from the literature.

The compartments are denoted S (susceptible), I (infected and potentially infectious), R
 (recovering), and E (exposed). Subscript F denotes sand flies. For the human compartments we
 use the subscript P for PCR-positive, D for DAT-positive, C for cellular immunity (LST-positive),
 S for symptomatic, T for treated, U for untreated and L for PKDL. Greek letters stand for the
 rates. Lower case roman letters stand for probabilities/proportions of the populations.

(c) Differences between our model and the original model in [57]

Aside from slight differences in the notation, we made the following changes and additions to the
 original model from [57].

- (i) We explicitly added a parameter p signifying the level of protection against sand fly's
 bites by using ITNs.
- (ii) We added a parameter e for the entomological efficacy of ITNs against the vector.
- (iii) We separated PKDL cases (I_{HL} in [57]) into treated (I_{TL}) and untreated (I_{UL}) cases to
 better account for the different duration of the stages and infectivity of the cases.
- (iv) We assume that the human population size is constant with a birth rate Λ (as opposed to
 $\alpha_H N_H$ considered in [57]). This change makes the ODE system less sensitive to changes
 in the birth rate and death rate. In the original model of [57], even a small change in

Figure 1. Scheme of the ODE model for VL transmission; based on [57]. Majority of the human population is in the asymptomatic VL cycle: individuals are born as susceptible (S) and if bitten by an infected sand fly (I_F) — signified by the red dotted arrow — they progress through early asymptomatic (I_P), late asymptomatic (I_D), recovering asymptomatic (R_D) and recovered (R_C). Recovered individuals lose immunity and eventually become susceptible again. Sand flies are born susceptible (S_F). If they bite an individual in any of the red or brown compartments — signified by the red curly brace — they become exposed (E_F) and eventually infectious (I_F). All human compartments are color-coded based on their PCR, DAT and LST tests as shown on the left; informally, the intensity of the color corresponds to the intensity of the infection which increases from left to right. The solid black arrows represent transitions between compartments.

α_H, μ_H , or μ_K would destabilize the system and result in either exponential growth or
 exponential extinction of the population. In our model, a (reasonable) change in Λ, μ_H ,
 or μ_K will cause the system to converge to an equilibrium with a potentially different
 population size, but the population will not go extinct nor grow above any bound. We
 can also solve for the equilibria of the dynamics.

(v) We keep the exponential growth of sand flies, denoted $\alpha_F N_F$ in [57], but we stipulate that
 $\alpha_F = \mu_F$ in order to keep the population size constant.

(vi) We consider the death rate of treated individuals to be $\mu_1 = \mu + d_{T1}\tau_1$ and $\mu_2 = \mu +$
 $d_{T2}\tau_2$, instead of $\mu_1 = \mu_2 = \mu + \mu_K + f_T\tau_1$. Specifically, we do not include the death rate
 of untreated individuals (μ_K) and we consider the rates different in a different lines of
 treatment.

(d) Numerical implementation

We coded the model in Matlab (version 2020a with optimization toolbox) and made the code
 available in the supplemental information. Following the best practices highlighted in [103], we
 included several tests to ensure correctness of the code. Specifically, we checked that analytical
 and graphical solutions (which were coded independently) yield the same results. We closely
 tracked the code execution to make sure the code runs as expected and values are passed
 correctly from function to function. Throughout the work on this manuscript, the formal analysis
 and numerical code were developed side by side and checked against each other. Independent
 cross-checks on a graphical calculator were also performed.

Table 1. Notation — Compartments

Notation	Meaning	PCR	DAT	LST
N	Total number of humans			
S	Susceptible humans	-	-	-
I_P	Early asymptomatic	+	-	-
I_D	Late asymptomatic	+	+	-
I_S	Symptomatic Kala-azar	+	+	-
I_{T1}	Receiving first line of treatment	+	+	-
I_{T2}	Receiving second line of treatment	+	+	-
I_{TL}	Treated cases with PKDL	+	+	-
I_{UL}	Untreated cases with PKDL	+	+	-
R_D	Recovering asymptomatic	-	+	-
R_T	Recovering after treatment	-	+	-
R_C	Recovered	-	-	+
R_L	Cases with dormant infection	-	+	-
N_F	Total number of sand flies ($N_F = n_F \frac{\Lambda}{\mu}$)			
S_F	Susceptible sand flies			
E_F	Exposed sand flies			
I_F	Infected sand flies			

Table 2. Parameter values — sand flies. Times are in days.

Symbol	Description	Value	Range	Reference
n_F	Number of sand flies per human	3	[2.1, 3.5]	[7, Table S15]
β^{-1}	Duration of feeding cycle	4	[2, 8]	[85,86]
σ_F^{-1}	Sojourn time in latent stage E_F	5	[3, 7]	[87]
μ_F^{-1}	Expected lifespan of sand flies	14	[6, 31]	[88]
i_F	Probability that a bite by an infected sand fly infects a susceptible human	1	[0, 1]	[57]

(e) Model limitations and possible extensions

There are always many ways how to make any model more realistic. We list several possible extensions of our model here.

Infectious sand flies seem to feed more often [104,105]. One may thus need to consider two different bite rates. One for susceptible sand flies (this will play a role in λ_F , i.e., humans infecting sand flies) and another one for sand flies infecting humans (this will play a role in λ , i.e., sand flies infecting humans).

We did not consider any ITN related mortality of the sand flies. This is in agreement with [106], although slight reduction of sand fly prevalence was observed by [46]. We also do not consider increased mortality of infected sand flies [105,107]. To incorporate any of these two, one would have to consider $\mu_F = \mu_F(p)$ for an appropriate increasing function $\mu_F(p)$. Furthermore, the total mortality of the infected sand flies would be $\mu_F(p) + \mu_I(p)$ where $\mu_I(p)$ represents the additional mortality caused by the infection and the fact that infected flies bites faster, i.e. could suffer from an increased ITN related mortality as well. To make these additions work, we would have to assume the birth rate to be Λ_F ; otherwise the sand fly population would not have nontrivial finite equilibrium.

We also did not consider that ITNs offer only imperfect protection. First, smaller mesh size and insecticide treatment increase the net efficacy, but a small number of sand flies were still found

rsos.royalsocietypublishing.org R. Soc. open sci. 0000000

Table 3. Parameter values — humans. Times are in months. Rates are per capita per month.

Symbol	Description	Value	Range	Reference
Λ	Recruitment rate	$\frac{0.0277}{12}$	$[\frac{0.0105}{12}, \frac{0.0333}{12}]$	[89]
μ^{-1}	Life expectancy	$67.8 \cdot 12$	$[50, 90] \cdot 12$	[90]
μ_K^{-1}	Life expectancy with untreated symptomatic VL	30	[5, 36]	[2]
γ_P^{-1}	Sojourn time in I_P	5	[4, 6]	[91]
γ_D^{-1}	Sojourn time in I_D	1.13	[0.5, 4]	[91]

[revised manuscript text omitted]
 six years, the cohort of $N = 22699$ individuals should experience $72 \cdot \left(\frac{1}{33}\right) \cdot 0.35 \cdot 22699 \approx 17334$ late asymptomatic cases. Since 813 of those developed KA, we get $f_S = 813/17334 = 0.0469$. Similarly, [23] surveyed 24,814 individuals for KA and PKDL in the past nine years. Of those 1,002 reported KA and 17 had PKDL with no history of KA. This gives an estimate

$$f_S = \frac{1002}{9 \cdot 12 \cdot 24814 \cdot \frac{1}{33} \cdot 0.35} = 0.0353. \quad (4.1)$$

Because [22] and [23] used a much larger sample size than [93] and [93] also notes that incidences were larger in India than in Nepal, we will adopt $f_S = 0.035$ and consider a range from 0.01 to 0.15.

We estimate that the fraction of asymptomatic patients who become dormant to be $f_L = 0.00055$ with the range [0.0001, 0.001]. This will be done in a similar fashion as the estimate for f_S based on data from [22] and [23]. A total of $N = 22699$ individuals were surveyed by [22] for

[revised manuscript text omitted]

 within three weeks. In fact [23] followed 30 PKDL cases treated with amphotericin B, and while
 reported an improvement in four months, only one case was completely cured. We do not have
 additional data but it seems that even in the case of this treatment, the duration of PKDL is six
 477 months.

The duration of untreated PKDL, γ_L^{-1} , will be estimated as follows. In [23], the authors
 followed 98 PKDL patients that never received treatment and provides estimated resolution
 rates: 8% within one year of onset, 34% within two years, and 67% within five years. Fitting the
 exponential decay to this data by Matlab, see Figure 3, yields $\gamma_L^{-1} = 55.5$ months with the range
 of 32 months to 16 years.

The fraction of PKDL patients that receive treatment is estimated as $p_{TL} = 0.5$. This estimate is
 based on [23] who reports that out of 185 PKDL cases, 98 did not seek the treatment.

(e) The cost parameters

Once infected, all individuals have to go through I_P and I_D . Those stages are asymptomatic
 and consequently with no associated costs. The costs appear only if and when an individual
 experiences symptoms (in I_S), is being treated (I_{T1} , I_{T2} , and I_{TL}) or is recovering after the
 treatment (in R_T and R_L). The costs are denoted by C_{I_S} , $C_{I_{T1}}$, $C_{I_{T2}}$, $C_{I_{TL}}$, C_{R_T} , and C_{R_L} and
 the actual values are estimated below.

The Indian government provides a free care to VL patients; however, hoping for a quick cure
 of seemingly minor illness, patients prefer to access private providers which contributes to high
 out-of-pocket expenditures [111]. VL is often misdiagnosed in private hospitals but patients keep
 accessing care in private sector until no money is left [111].

The cost of being in compartment I_S is estimated as $C_{I_S} = 19$ USD [100], and we will assume
 the range [11, 26]. The average monthly income in Bihar, India in 2010 was 38 USD per month
 [100]. It takes about a month to get a diagnosis [92]. During that time, the patient loses 2.14 weeks
 (about half a month) of work, i.e. the cost is about \$19.

The cost of getting the first line of treatment is $C_{I_{T1}} = 146$ USD [100], and we will assume the
 range [120, 170]. This includes 127 USD of direct medical costs for one month of treatment [100],
 and the indirect cost of lost work. For every month of illness, half a month of work is lost (19 USD
 loss) [100]. This is in line with \$100 direct medical costs used by [101].

The cost of getting the second line of treatment is $C_{I_{T2}} = C_{I_{T1}} = 146$ USD [100]. Both lines of
 treatment last approximately one month, so the cost of the individual second-treatment will be
 about the same.

The cost of getting treatment for PKDL is assumed to be $C_{I_{TL}} = 349.00$ USD [102]. We will
 assume the range [290, 410]. The direct cost of SSG treatment for PKDL costs 179 USD [102]. The
 treatment lasts about six months and so the loss of productivity during this illness is 170 USD.

The cost of recovering from treatment without a dormant infection is $C_{R_T} = 57$ USD [100], and
 we will assume the range [60, 110]. During recovery, a patient can only work 3.21 weeks/month
 on average, instead of the normal 4.29, i.e. losing 25% of the wage per month [100]. Since the
 average monthly wage is 38 USD and R_T lasts approximately $\rho_T^{-1} = 6$ months, the total loss is
 about $57 = 0.25 \cdot 38 \cdot 6$.

The cost of recovering from treatment, but with a dormant infection is $C_{R_L} = C_{R_T} = 57$ USD
 [100]. We assume that even though individuals in this stage do have a dormant infection, the time
 and the cost needed to recover from treatment will be about the same.

The overall cost of VL infection is thus given by

$$C_{VL} = P(I_P \rightarrow I_S)C_{I_S} + P(I_P \rightarrow I_{T1})C_{I_{T1}} + P(I_P \rightarrow I_{T2})C_{I_{T2}} + P(I_P \rightarrow I_{TL})C_{I_{TL}} + \\ P(I_P \rightarrow R_T|I_{T1} \text{ or } I_{T2} \text{ or } I_{TL})C_{R_T} + P(I_P \rightarrow R_L|I_{T1} \text{ or } I_{T2})C_{R_L} \quad (4.4)$$

where the probabilities $P(I_P \rightarrow C)$ (and $P(I_P \rightarrow C|C')$) of getting to a compartment C (through
 a compartment C') when currently at a compartment I_P are given by

$$P(I_P \rightarrow I_S) = T_{IP} f_S \gamma_D \quad (4.5)$$

$$P(I_P \rightarrow I_{T1}) = T_{IS} \gamma_S \quad (4.6)$$

$$P(I_P \rightarrow I_{T2}) = T_{IT1} s_1 \tau_1 \quad (4.7)$$

$$P(I_P \rightarrow I_{TL}) = T_{RL} p_{TL} \rho_L \quad (4.8)$$

$$P(I_P \rightarrow R_T|I_{T1} \text{ or } I_{T2} \text{ or } I_{TL}) = T_{IT1} s_3 \tau_1 + T_{IT2} s_5 \tau_2 + T_{ITL} \tau_3 \quad (4.9)$$

$$P(I_P \rightarrow R_L|I_{T1} \text{ or } I_{T2}) = T_{IT1} s_2 \tau_1 + T_{IT2} s_4 \tau_2. \quad (4.10)$$

The cost of ITN is $C_{ITN} = 3.62$ USD [40]. If the households opt to use the impregnated durable
 wall lining, which seems even more effective than ITNs, the cost grows significantly to about 13.57
 522 USD [40]. At the same time, one could argue that the wall lining protects the whole household,
 while the ITN protects only those that sleep under it.

5. Results

[revised manuscript text omitted]
(0) \left[1 - \frac{I_F^*}{N_F^*} \left(\beta i_F T_{\text{cycle}} \frac{N_F^*}{\mathcal{R}_e(0)} + \frac{\mu_F}{\sigma_F} + 1 \right) \right]} \tag{5.4}$$

583 where I_F^* is given by (A 5).

584 It follows that $p_{NE} \approx p_{HI}$. For our parameter values, we have $p_{HI} = 0.
[revised manuscript text omitted]

MP, Dasgupta RK. 2020 Kala-azar elimination in a highly-endemic district of Bihar, India:
A success story. *PLOS Neglected Tropical Diseases* **14**, e0008254.
- 8. Sundar S, Singh OP, Chakravarty J. 2018 Visceral leishmaniasis elimination targets in India,
strategies for preventing resurgence. *Expert review of anti-infective therapy* **16**, 805–812.
- 9. Cloots K, Uranw S, Ostyn B, Bhattarai NR, Le Rutte E, Khanal B, Picado A, Chappuis F, Hasker
E, Karki P. 2020 Impact of the visceral leishmaniasis elimination initiative on *Leishmania
donovani* transmission in Nepal: a 10-year repeat survey. *The Lancet Global Health* **8**, e237–e243.
- 10. NTD Modelling Consortium Visceral Leishmaniasis Group. 2019 Insights from mathematical
modelling and quantitative analysis on the proposed WHO 2030 targets for visceral
leishmaniasis on the Indian subcontinent. *Gates Open Research* **3**, 1651.

- 11. WHO. 2020 Ending the neglect to attain the sustainable development goals: a road map for
neglected tropical diseases 2021–2030. Technical report World Health Organization.
- 12. Singh OP, Singh B, Chakravarty J, Sundar S. 2016 Current challenges in treatment options for
visceral leishmaniasis in India: a public health perspective. *Infectious diseases of poverty* **5**, 1–15.
- 13. Dinesh D, Ranjan A, Palit A, Kishore K, Kar S. 2001 Seasonal and nocturnal landing/biting
behaviour of *Phlebotomus argentipes* (Diptera: Psychodidae). *Annals of Tropical Medicine &*
*Parasitology* **95**, 197–202.
- 14. Feliciangeli M. 2004 Natural breeding places of phlebotomine sandflies. *Medical and Veterinary*
*Entomology* **18**, 71–80.
- 15. Gawade S, Nanaware M, Gokhale R, Adhav P. 2012 Visceral leishmaniasis: A case report. *The*
*Australasian Medical Journal* **5**, 130.
- 16. Younis LG, Kroeger A, Joshi AB, Das ML, Omer M, Singh VK, Gurung CK, Banjara MR.
2020 Housing structure including the surrounding environment as a risk factor for visceral
leishmaniasis transmission in Nepal. *PLoS Neglected Tropical Diseases* **14**, e0008132.
- 17. Goddard J. 2013 *Physician's Guide to Arthropods of Medical Importance*. CRC Press: Taylor &
Francis Group.
- 18. Poché RM, Garlapati R, Elnaiem DEA, Perry D, Poché D. 2012 The role of Palmyra palm trees
(*Borassus flabellifer*) and sand fly distribution in northeastern India. *Journal of Vector Ecology* **37**,
148–153.
- 19. Piscopo TV, Azzopardi CM. 2007 Leishmaniasis. *U.S. National Library of Medicine*.
- 20. Das A, Karthick M, Dwivedi S, Banerjee I, Mahapatra T, Srikantiah S, Chaudhuri I. 2016
Epidemiologic correlates of mortality among symptomatic visceral leishmaniasis cases:
findings from situation assessment in high endemic foci in India. *PLoS Neglected Tropical*
*Diseases* **10**, e0005150.
- 21. Ranjan A, Sur D, Singh VP, Siddique NA, Manna B, Lal CS, Sinha PK, Kishore K, Bhattacharya
SK. 2005 Risk factors for Indian kala-azar. *The American Journal of Tropical Medicine and Hygiene*
**73**, 74–78.
- 22. Rahman KM, Islam S, Rahman MW, Kenah E, Galive CM, Zahid M, Maguire J, Rahman M,
Haque R, Luby SP, Bern C. 2010 Increasing Incidence of Post-Kala-Azar Dermal Leishmaniasis
in a Population-Based Study in Bangladesh. *Clinical Infectious Diseases* **50**, 73–76.
- 23. Islam S, Kenah E, Bhuiyan MAA, Rahman KM, Goodhew B, Ghalib CM, Zahid M, Ozaki
730 M, Rahman M, Haque R. 2013 Clinical and immunological aspects of post-kala-azar dermal
leishmaniasis in Bangladesh. *The American Journal of Tropical Medicine and Hygiene* **89**, 345–353.
- 24. Addy M, Nandy A. 1992 Ten years of kala-azar in west Bengal, Part I. Did post-kala-
azar dermal leishmaniasis initiate the outbreak in 24-Parganas?. *Bulletin of the World Health*
*Organization* **70**, 341–346.
- 25. Das VNR, Pandey RN, Siddiqui NA, Chapman LA, Kumar V, Pandey K, Matlashewski G,
Das P. 2016 Longitudinal study of transmission in households with visceral leishmaniasis,
asymptomatic infections and PKDL in highly endemic villages in Bihar, India. *PLoS Neglected*
*Tropical Diseases* **10**, e0005196.
- 26. Burza S, Mahajan R, Sanz MG, Sunyoto T, Kumar R, Mitra G, Lima MA. 2014a HIV and
visceral leishmaniasis coinfection in Bihar, India: an underrecognized and underdiagnosed
threat against elimination. *Clinical infectious diseases* **59**, 552–555.
- 27. Burza S, Mahajan R, Sinha PK, van Griensven J, Pandey K, Lima MA, Sanz MG, Sunyoto T,
Kumar S, Mitra G et al.. 2014b Visceral leishmaniasis and HIV co-infection in Bihar, India:
long-term effectiveness and treatment outcomes with liposomal amphotericin B (AmBisome).
*PLoS Negl Trop Dis* **8**, e3053.
- 28. Akuffo H, Costa C, van Griensven J, Burza S, Moreno J, Herrero M. 2018 New insights into
leishmaniasis in the immunosuppressed. *PLoS neglected tropical diseases* **12**, e0006375.
- 29. Wilson AL, Courtenay O, Kelly-Hope LA, Scott TW, Takken W, Torr SJ, Lindsay SW. 2020 The
importance of vector control for the control and elimination of vector-borne diseases. *PLoS*
*Neglected Tropical Diseases* **14**, e0007831.
- 30. Stockdale L, Newton R. 2013 A review of preventative methods against human leishmaniasis
infection. *PLoS Neglected Tropical Diseases* **7**, e2278.

31. Das VNR, Pandey RN, Pandey K, Singh V, Kumar V, Matlashewski G, Das P. 2014 Impact of ASHA training on active case detection of visceral leishmaniasis in Bihar, India. *PLoS Negl Trop Dis* **8**, e2774.
32. Osman M, Mistry A, Keding A, Gabe R, Cook E, Forrester S, Wiggins R, Di Marco S, Colloca S, Siani L et al.. 2017 A third generation vaccine for human visceral leishmaniasis and post kala azar dermal leishmaniasis: First-in-human trial of ChAd63-KH. *PLoS Neglected Tropical Diseases* **11**, e0005527.
33. Moafi M, Rezvan H, Sherkat R, Taleban R. 2019 Leishmania vaccines entered in clinical trials: A review of literature. *International journal of preventive medicine* **10**.
34. Le Rutte EA, Coffeng LE, Malvolti S, Kaye PM, de Vlas SJ. 2020 The potential impact of human visceral leishmaniasis vaccines on population incidence. *PLoS Neglected Tropical Diseases* **14**, e0008468.
35. Kumar V, Kesari S, Sinha N, Palit A, Ranjan A, Kishore K, Saran R, Kar S. 1995 Field trial of an ecological approach for the control of *Phlebotomus argentipes* using mud & lime plaster. *The Indian Journal of Medical Research* **101**, 154.
36. Wasserberg G, Weeks EN, Logan JL, Agneessens J, Stewart SA, Dewhirst S. 2019 Efficacy of the insect repellent IR3535 on the sand fly *Phlebotomus papatasi* in human volunteers. *Journal of Vector Ecology* **44**.
37. Bern C, Joshi AB, Jha SN, Das ML, Hightower A, Thakur G, Bista MB. 2000 Factors associated with visceral leishmaniasis in Nepal: bed-net use is strongly protective. *The American Journal of Tropical Medicine and Hygiene* **63**, 184–188.
38. Chappuis F, Sundar S, Hailu A, Ghalib H, Rijal S, Peeling RW, Alvar J, Boelaert M. 2007 Visceral leishmaniasis: what are the needs for diagnosis, treatment and control?. *Nature Reviews Microbiology* **5**, 873–882.
39. Ostyn B, Vanlerberghe V, Picado A, Dinesh DS, Sundar S, Chappuis F, Rijal S, Dujardin JC, Coosemans M, Boelaert M. 2008 Vector control by insecticide-treated nets in the fight against visceral leishmaniasis in the Indian subcontinent, what is the evidence?. *Tropical Medicine International Health* **13**, 1073–1085.
40. Mondal D, Das ML, Kumar V, Huda MM, Das P, Ghosh D, Priyanka J, Matlashewski G, Kroeger A, Uppill-Brown A et al.. 2016 Efficacy, safety and cost of insecticide treated wall lining, insecticide treated bed nets and indoor wall wash with lime for visceral leishmaniasis vector control in the Indian sub-continent: a multi-country cluster randomized controlled trial. *PLoS Neglected Tropical Diseases* **10**, e0004932.
41. Vanlerberghe V, Singh S, Paudel I, Ostyn B, Picado A, Sanchez A, Rijal S, Sundar S, Davies C, Boelaert M. 2010 Determinants of bednet ownership and use in visceral leishmaniasis-endemic areas of the Indian subcontinent. *Tropical Medicine & International Health* **15**, 60–67.
42. Kroeger A, Ordóñez-Gonzalez o, Behrend M, Alvarez G. 1999 Bednet impregnation for Chagas disease control: a new perspective. *Tropical Medicine and International Health* **4**, 194–198.
43. Mishra RN, Singh S, Vanlerberghe V, Sundar S, Boelaert M, Lefevre P. 2010 Lay perceptions of kala-azar, mosquitoes and bed nets in Bihar, India. *Tropical Medicine & International Health* **15**, 36–41.
44. Kroeger A, González M, Ordóñez-González J. 1999 Insecticide-treated materials for malaria control in Latin America: to use or not to use?. *Transactions of the Royal Society of Tropical Medicine and Hygiene* **93**, 565–570.
45. Picado A, Singh SP, Rijal S, Sundar S, Ostyn B, Chappuis F, Uranw S, Gidwani K, Khanal B, Rai M et al.. 2010a Longlasting insecticidal nets for prevention of *Leishmania donovani* infection in India and Nepal: paired cluster randomised trial. *BMJ* **341**.
46. Picado A, Das ML, Kumar V, Kesari S, Dinesh DS, Roy L, Rijal S, Das P, Rowland M, Sundar S. 2010b Effect of village-wide use of long-lasting insecticidal nets on visceral Leishmaniasis vectors in India and Nepal: a cluster randomized trial. *PLoS Neglected Tropical Diseases* **4**, e587.
47. Picado A, Ostyn B, Rijal S, Sundar S, Singh SP, Chappuis F, Das ML, Khanal B, Gidwani K, Hasker E et al.. 2015 Long-lasting insecticidal nets to prevent visceral leishmaniasis in the Indian subcontinent; methodological lessons learned from a cluster randomised controlled trial. *PLoS Neglected Tropical Diseases* **9**, e0003597.

48. Chowdhury R, Chowdhury V, Faria S, Akter S, Dash AP, Bhattacharya SK, Maheswary
NP, Bern C, Akhter S, Alvar J et al. 2019 Effect of insecticide-treated bed nets on visceral
leishmaniasis incidence in Bangladesh. A retrospective cohort analysis. *PLoS Neglected Tropical*
*Diseases* **13**, e0007724.
- 11 811 49. Rock KS, le Rutte EA, de Vlas SJ, Adams ER, Medley GF, Hollingsworth TD. 2015 Uniting
mathematics and biology for control of visceral leishmaniasis. *Trends in Parasitology* **31**, 251–
259.
- 13 814 50. Hirve S, Boelaert M, Matlashewski G, Mondal D, Arana B, Kroeger A, Olliaro P. 2016
Transmission dynamics of visceral leishmaniasis in the Indian subcontinent—a systematic
literature review. *PLoS Neglected Tropical Diseases* **10**, e0004896.
- 16 817 51. DebRoy S, Prosper O, Mishoe A, Mubayi A. 2017 Challenges in modeling complexity of
818 neglected tropical diseases: a review of dynamics of visceral leishmaniasis in resource limited
settings. *Emerging Themes in Epidemiology* **14**, 10.
- 20 820 52. Bi K, Chen Y, Zhao S, Kuang Y, John Wu CH. 2018 Current visceral leishmaniasis research: a
821 research review to inspire future study. *BioMed Research International* **2018**.
- 22 822 53. Stauch A, Duerr HP, Picado A, Ostyn B, Sundar S, Rijal S, Boelaert M, Dujardin JC, Eichner
823 M. 2014 Model-based investigations of different vector-related intervention strategies to
824 eliminate visceral leishmaniasis on the Indian subcontinent. *PLoS Neglected Tropical Diseases*
**8**, e2810.
- 26 826 54. Chapman LA, Dyson L, Courtenay O, Chowdhury R, Bern C, Medley GF, Hollingsworth
TD. 2015 Quantification of the natural history of visceral leishmaniasis and consequences for
control. *Parasites and Vectors* **8**, 521.
- 55. Poche DM, Grant WE, Wang HH. 2016 Visceral leishmaniasis on the Indian subcontinent:
modelling the dynamic relationship between vector control schemes and vector life cycles.
*PLoS Neglected Tropical Diseases* **10**.
- 56. Chapman LA, Spencer SE, Pollington TM, Jewell CP, Mondal D, Alvar J, Hollingsworth
TD, Cameron MM, Bern C, Medley GF. 2020 Inferring transmission trees to guide targeting
of interventions against visceral leishmaniasis and post-kala-azar dermal leishmaniasis.
*Proceedings of the National Academy of Sciences* **117**, 25742–25750.
- 57. Stauch A, Sarkar RR, Picado A, Ostyn B, Sundar S, Rijal S, Boelaert M, Dujardin JC, Duerr HP.
2011 Visceral leishmaniasis in the Indian subcontinent: modelling epidemiology and control.
*PLoS Neglected Tropical Diseases* **5**.
- 58. Le Rutte EA, Coffeng LE, Bontje DM, Hasker EC, Postigo JAR, Argaw D, Boelaert MC,
De Vlas SJ. 2016 Feasibility of eliminating visceral leishmaniasis from the Indian subcontinent:
explorations with a set of deterministic age-structured transmission models. *Parasites and*
*Vectors* **9**, 24.
- 59. Meheus F, Balasegaram M, Olliaro P, Sundar S, Rijal S, Faiz MA, Boelaert M. 2010 Cost-
effectiveness analysis of combination therapies for visceral leishmaniasis in the Indian
subcontinent. *PLoS Neglected Tropical Diseases* **4**, e818.
- 60. Stauch A, Duerr HP, Dujardin JC, Vanaerschot M, Sundar S, Eichner M. 2012 Treatment of
visceral leishmaniasis: model-based analyses on the spread of antimony-resistant *L. donovani*
in Bihar, India. *PLoS Neglected Tropical Diseases* **6**, e1973.
- 61. Rabi Das VN, Bimal S, Siddiqui NA, Kumar A, Pandey K, Sinha SK, Topno RK, Mahentesh
850 V, Singh AK, Lal CS. 2020 Conversion of asymptomatic infection to symptomatic visceral
leishmaniasis: A study of possible immunological markers. *PLoS Neglected Tropical Diseases*
**14**, e0008272.
- 62. Manson-Bahr P, Heisch R, Garnham P et al.. 1959 Studies in Leishmaniasis in East Africa.
IV. The Montenegro Test in Kala-Azar in Kenya.. *Transactions of the Royal Society of Tropical*
*Medicine and Hygiene* **53**, 380–83.
- 63. Bern C, Amann J, Haque R, Chowdhury R, Ali M, Kurkjian KM, Vaz L, Wagatsuma Y, Breiman
RF, Secor WE et al.. 2006 Loss of leishmanin skin test antigen sensitivity and potency in a
longitudinal study of visceral leishmaniasis in Bangladesh. *The American Journal of Tropical*
*Medicine and Hygiene* **75**, 744–748.

64. Molina R, Jiménez M, García-Martínez J, San Martín JV, Carrillo E, Sánchez C, Moreno J, Alves F, Alvar J. 2020 Role of asymptomatic and symptomatic humans as reservoirs of visceral leishmaniasis in a Mediterranean context. *PLoS Neglected Tropical Diseases* **14**, e0008253.
65. Bauch CT, Earn DJ. 2004 Vaccination and the theory of games. *Proceedings of the National Academy of Sciences* **101**, 13391–13394.
66. Maskin E. 1999 Nash equilibrium and welfare optimality. *The Review of Economic Studies* **66**, 23–38.
67. Chang SL, Piraveenan M, Pattison P, Prokopenko M. 2020 Game theoretic modelling of infectious disease dynamics and intervention methods: a review. *Journal of Biological Dynamics* **14**, 57–89.
68. Ibuka Y, Li M, Vietri J, Chapman GB, Galvani AP. 2014 Free-riding behavior in vaccination decisions: an experimental study. *PLoS One* **9**.
69. Crawford K, Lancaster A, Oh H, Rychtář J. 2015 A voluntary use of insecticide-treated cattle can eliminate African sleeping sickness. *Letters in Biomathematics* **2**, 91–101.
70. Klein SRM, Foster AO, Feagins DA, Rowell JT, Erovenko IV. 2020 Optimal voluntary and mandatory insect repellent usage and emigration strategies to control the chikungunya outbreak on Reunion Island. *PeerJ* **8**, e10151.
71. Kobe J, Pritchard N, Short Z, Erovenko IV, Rychtář J, Rowell JT. 2018 A Game-Theoretic Model of Cholera with Optimal Personal Protection Strategies. *Bulletin of Mathematical Biology* **80**, 2580–2599.
72. Dorsett C, Oh H, Paulemond ML, Rychtář J. 2016 Optimal repellent usage to combat dengue fever. *Bulletin of Mathematical Biology* **78**, 916–922.
73. Brettin A, Rossi-Goldthorpe R, Weishaar K, Erovenko IV. 2018 Ebola could be eradicated through voluntary vaccination. *Royal Society Open Science* **5**, 171591.
74. Chouhan A, Maiwand S, Ngo M, Putalapattu V, Rychtář J, Taylor D. 2020 Game-theoretical model of retroactive Hepatitis B vaccination in China. *Bulletin of Mathematical Biology* **82**, 80.
75. Scheckelhoff K, Ejaz A, Erovenko IV. 2019 A game-theoretic model of optimal clean equipment usage to prevent hepatitis C among injecting drug users. *Preprint*.
76. Martinez A, Machado J, Sanchez E, Erovenko IV. 2019 Optimal vaccination strategies to reduce endemic levels of meningitis in Africa. *Preprint*.
77. Bankuru SV, Kossol S, Hou W, Mahmoudi P, Rychtář J, Taylor D. 2020 A Game-theoretic Model of Monkeypox to Assess Vaccination Strategies. *PeerJ* **8**, e9272.
78. Cheng E, Gambhirrao N, Patel R, Zhouwandai A, Rychtář J, Taylor D. 2020 A game-theoretical analysis of Poliomyelitis vaccination. *Journal of Theoretical Biology* **499**, 110298.
79. Sykes D, Rychtář J. 2015 A game-theoretic approach to valuating toxoplasmosis vaccination strategies. *Theoretical Population Biology* **105**, 33–38.
80. Verelst F, Willem L, Beutels P. 2016 Behavioural change models for infectious disease transmission: a systematic review (2010–2015). *Journal of The Royal Society Interface* **13**, 20160820.
81. Acosta-Alonzo CB, Erovenko IV, Lancaster A, Oh H, Rychtář J, Taylor D. 2020 High endemic levels of typhoid fever in rural areas of Ghana may stem from optimal voluntary vaccination behavior. *Proceedings of Royal Society A* p. 20200354.
82. Han CY, Issa H, Rychtář J, Taylor D, Umana N. 2020 A voluntary use of insecticide treated nets can stop the vector transmission of Chagas disease. *PLoS Neglected Tropical Diseases* **14**, e0008833.
83. Behrend MR, Basáñez MG, Hamley JI, Porco TC, Stolk WA, Walker M, de Vlas SJ, Consortium NM. 2020 Modelling for policy: the five principles of the Neglected Tropical Diseases Modelling Consortium. *PLoS Neglected Tropical Diseases* **14**, e0008033.
84. den Boer ML, Alvar J, Davidson RN, Ritmeijer K, Balasegaram M. 2009 Developments in the treatment of visceral leishmaniasis. *Expert opinion on emerging drugs* **14**, 395–410.
85. de Souza Leal MMC, Ovallos FG, de Castro Gomes CM, de Oliveira Lavitschka C, Galati EAB. 2014 Host-biting rate and susceptibility of some suspected vectors to *Leishmania braziliensis*. *Parasites and Vectors* **7**, 1–11.
86. Hartemink N, Vanwambeke SO, Heesterbeek H, Rogers D, Morley D, Pesson B, Davies C, Mahamdallie S, Ready P. 2011 Integrated mapping of establishment risk for emerging vector-borne infections: a case study of canine leishmaniasis in southwest France. *PLoS One* **6**, e20817.

87. Sacks DL, Perkins PV. 1985 Development of Infective Stage Leishmania Promastigotes within Phlebotomine Sand Flies. *The American Journal of Tropical Medicine and Hygiene* **34**, 456–459.
88. Srinivasan R, Panicker K. 1993 Laboratory observations on the biology of the phlebotomid sandfly, *Phlebotomus papatasi* (Scopoli, 1786). *Southeast Asian Journal of Tropical Medicine and Public Health* **24**, 536–536.
89. Census India. 2016 Estimates of fertility indicators. https://censusindia.gov.in/vital_statistics/SRS_Report_2016/7.Chap_3-Fertility_Indicators-2016.pdf. Accessed March 16, 2021.
90. Census India. 2015 Abridged life tables 2010-14. https://www.censusindia.gov.in/Vital_Statistics/SRS_Life_Table/2.Analysis_2010-14.pdf. Accessed March 16, 2021.
91. Hailu A, Gramiccia M, Kager P. 2009 Visceral leishmaniasis in Aba-Roba, south-western Ethiopia: prevalence and incidence of active and subclinical infections. *Annals of Tropical Medicine & Parasitology* **103**, 659–670.
92. Kumar A, Saurabh S, Jamil S, Kumar V. 2020 Intensely clustered outbreak of visceral leishmaniasis (kala-azar) in a setting of seasonal migration in a village of Bihar, India. *BMC Infectious Diseases* **20**, 10.
93. Ostyn B, Gidwani K, Khanal B, Picado A, Chappuis F, Singh SP, Rijal S, Sundar S, Boelaert M. 2011 Incidence of symptomatic and asymptomatic *Leishmania donovani* infections in high-endemic foci in India and Nepal: a prospective study. *PLoS Neglected Tropical Diseases* **5**, e1284.
94. Singh OP, Tiwary P, Kushwaha AK, Singh SK, Singh DK, Lawyer P, Rowton E, Chaubey R, Singh AK, Rai TK et al. 2021 Xenodiagnosis to evaluate the infectiousness of humans to sandflies in an area endemic for visceral leishmaniasis in Bihar, India: a transmission-dynamics study. *The Lancet Microbe* **2**, e23–e31.
95. Mondal D, Bern C, Ghosh D, Rashid M, Molina R, Chowdhury R, Nath R, Ghosh P, Chapman LA, Alim A. 2019 Quantifying the infectiousness of post-kala-azar dermal leishmaniasis toward sand flies. *Clinical infectious diseases* **69**, 251–258.
96. van Griensven J, Balasegaram M, Meheus F, Alvar J, Lynen L, Boelaert M. 2010 Combination therapy for visceral leishmaniasis. *The Lancet Infectious Diseases* **10**, 184–194.
97. Sundar S, Singh A, Chakravarty J, Rai M. 2015 Efficacy and safety of miltefosine in treatment of post-kala-azar dermal leishmaniasis. *The Scientific World Journal* **2015**.
98. Ghosh S, Das NK, Mukherjee S, Mukhopadhyay D, Barbhuiya JN, Hazra A, Chatterjee M. 2015 Inadequacy of 12-week miltefosine treatment for Indian post-kala-azar dermal leishmaniasis. *The American Journal of Tropical Medicine and Hygiene* **93**, 767–769.
99. Jervis S, Chapman LA, Dwivedi S, Karthick M, Das A, Le Rutte EA, Courtenay O, Medley GF, Banerjee I, Mahapatra T. 2017 Variations in visceral leishmaniasis burden, mortality and the pathway to care within Bihar, India. *Parasites and Vectors* **10**, 601.
100. Sundar S, Arora R, Singh SP, Boelaert M, Varghese B. 2010 Household cost-of-illness of visceral leishmaniasis in Bihar, India. *Tropical Medicine & International Health* **15**, 50–54.
101. Hasker E, Singh SP, Malaviya P, Singh RP, Shankar R, Boelaert M, Sundar S. 2010 Management of visceral leishmaniasis in rural primary health care services in Bihar, India. *Tropical Medicine & International Health* **15**, 55–62.
102. Ozaki M, Islam S, Rahman KM, Rahman A, Luby SP, Bern C. 2011 Economic Consequences of Post-Kala-Azar Dermal Leishmaniasis in a Rural Bangladeshi Community. *The American Journal of Tropical Medicine and Hygiene* **85**, 528–534.
103. Lucas TC, Pollington TM, Davis EL, Hollingsworth TD. 2020 Responsible modelling: Unit testing for infectious disease epidemiology. *Epidemics* p. 100425.
104. Ready PD. 2008 *Leishmania* manipulates sandfly feeding to enhance its transmission. *Trends in Parasitology* **24**, 151–153.
105. Rogers ME, Bates PA. 2007 *Leishmania* manipulation of sand fly feeding behavior results in enhanced transmission. *PLoS Pathogens* **3**, e91.
106. Dinesh DS, Das P, Picado A, Davies C, Speybroeck N, Ostyn B, Boelaert M, Coosemans M. 2008 Long-lasting insecticidal nets fail at household level to reduce abundance of sandfly vector *Phlebotomus argentipes* in treated houses in Bihar (India). *Tropical Medicine & International Health* **13**, 953–958.

107. Carmichael S, Powell B, Hoare T, Walrad PB, Pitchford JW. 2021 Variable bites and dynamic
populations; new insights in Leishmania transmission. *PLoS Neglected Tropical Diseases* **15**,
e0009033.
- 108. Das ML, Rowland M, Austin JW, De Lazzari E, Picado A. 2014 Do size and insecticide
treatment matter? Evaluation of different nets against *Phlebotomus argentipes*, the vector of
visceral leishmaniasis in Nepal. *PLoS One* **9**, e114915.
- 109. Schenkel K, Rijal S, Koirala S, Koirala S, Vanlerberghe V, Van der Stuyft P, Gramiccia M,
Boelaert M. 2006 Visceral leishmaniasis in southeastern Nepal: A cross-sectional survey on
*Leishmania donovani* infection and its risk factors. *Tropical Medicine and International Health* **11**,
1792–1799.
- 110. Valero NNH, Uriarte M. 2020 Environmental and socioeconomic risk factors associated with
visceral and cutaneous leishmaniasis: a systematic review. *Parasitology research* **119**, 365–384.
- 111. Nair M, Kumar P, Pandey S, Kazmi S, Moreto-Planas L, Ranjan A, Burza S. 2020 Quality of life
perceptions amongst patients co-infected with Visceral Leishmaniasis and HIV: A qualitative
study from Bihar, India. *PloS One* **15**, e0227911.
- 112. Singh A, Chakraborty S, Roy TK. 2008 Village size in India: How relevant is it in the context
of development?. *Asian Population Studies* **4**, 111–134.
- 113. Anderson RM, May RM. 1992 *Infectious diseases of humans: dynamics and control*. Oxford
university press.
- 114. Delamater PL, Street EJ, Leslie TF, Yang YT, Jacobsen KH. 2019 Complexity of the basic
reproduction number (R_0). *Emerging infectious diseases* **25**, 1.
- 115. van den Driessche P, Watmough J. 2002 Reproduction numbers and sub-threshold endemic
equilibria for compartmental models of disease transmission. *Mathematical Biosciences* **180**, 29–
48.
- 116. Wei HM, Li XZ, Martcheva M. 2008 An epidemic model of a vector-borne disease with direct
transmission and time delay. *Journal of Mathematical Analysis and Applications* **342**, 895–908.
- 117. Killick-Kendrick R, Rioux J. 2002 Mark-release-recapture of sand flies fed on leishmanial
dogs: the natural life-cycle of *Leishmania infantum* in *Phlebotomus ariasi*. *Parassitologia* **44**, 67–71.
- 118. Guilvard E, Wilkes T, Killick-Kendrick R, Rioux JA. 1980 Ecologie des Leishmanioses dans le
Sud de la France-15. Déroulement des cycles gonotrophiques chez *Phlebotomus ariasi* Tonnoir,
1921 et *Phlebotomus mascittii* Grassi, 1908 en Cévennes. Corollaire épidémiologique. *Annales de*
*Parasitologie Humaine et Comparée* **55**, 659–664.
- 119. Hussaini N, Okuneye K, Gumel AB. 2017 Mathematical analysis of a model for zoonotic
visceral leishmaniasis. *Infectious Disease Modelling* **2**, 455–474.
- 120. Shimozako HJ, Wu J, Massad E. 2017 Mathematical modelling for Zoonotic Visceral
Leishmaniasis dynamics: a new analysis considering updated parameters and notified human
Brazilian data. *Infectious Disease Modelling* **2**, 143–160.
- 121. CIA. 2020 The World Factbook–South Asia: India. [https://www.cia.gov/library/
publications/resources/the-world-factbook/geos/in.html](https://www.cia.gov/library/publications/resources/the-world-factbook/geos/in.html). Accessed June 17,
2020.
- 122. World Bank. 2020 Life expectancy at birth. [https://data.worldbank.org/
indicator/SP.DYN.LE00.IN?cid=GPD_10](https://data.worldbank.org/indicator/SP.DYN.LE00.IN?cid=GPD_10). Accessed April 13, 2020.
- 123. Le Rutte EA, Chapman LA, Coffeng LE, Jervis S, Hasker EC, Dwivedi S, Karthick M,
Das A, Mahapatra T, Chaudhuri I. 2017 Elimination of visceral leishmaniasis in the Indian
subcontinent: a comparison of predictions from three transmission models. *Epidemics* **18**,
67–80.
- 124. Das V, Siddiqui N, Verma R, Topno R, Singh D, Das S, Ranjan A, Pandey K, Kumar N, Das

[revised manuscript text omitted]
^* \left(1 - \frac{1}{\mathcal{R}_e(0)}\right)}{\beta i_F T_{\text{Cycle}} \frac{N_F^*}{\mathcal{R}_e(0)} + \frac{\mu_F}{\sigma_F} + 1} \quad (\text{A } 3)$$

attained for $p = 0$ and consequently, (5.3) has a solution only if

$$C_{\text{ITN}} \leq \frac{\beta i_F I_F^*(0)}{\mu + \beta i_F I_F^*(0)} C_{\text{VL}}. \quad (\text{A } 4)$$

It follows from (5.3) that (when (A 4) is true)

$$I_F^* = \frac{\left(\frac{C_{\text{ITN}}}{C_{\text{VL}}}\right) \mu}{\beta i_F \left(1 - \frac{C_{\text{ITN}}}{C_{\text{VL}}}\right)}. \quad (\text{A } 5)$$

Thus, after algebraic manipulations of (A 2), we get that p_{NE} is given by

$$p_{\text{NE}} = 1 - \frac{1}{\mathcal{R}_e(0) \left[1 - \frac{I_F^*}{N_F^*} \left(\beta i_F T_{\text{Cycle}} \frac{N_F^*}{\mathcal{R}_e(0)} + \frac{\mu_F}{\sigma_F} + 1\right)\right]} \quad (\text{A } 6)$$

where I_F^* is given by (A 5).

D. Sensitivity Analysis

Table 6. The sensitivity index SI_y of a variable y on a parameter x was calculated as $\left(\frac{x}{y}\right) \cdot \left(\frac{\partial y}{\partial x}\right)$ [134]. The numbers were rounded to the three decimal places. Parameters are as specified in Tables 2, 3 and 4. The sensitivity index -0.5 means that a 1% increase of a parameter value x will result in the 0.5% decrease of the variable y .

Parameter	$SI_{p_{HI}}$	$SI_{(p_{HI}-p_{NE})}$
Λ	0.0846	-3.9272
μ^{-1}	0.0846	-5.0223
μ_F^{-1}	0.0620	-3.1936
i_F	0.0425	-1.9916
n_F	0.0424	-1.9688
i_D	0.0201	-0.8375
γ_P^{-1}	0.0192	-0.7946
γ_D^{-1}	0.0188	-0.8742
i_P	0.0148	-0.5655
f_S	0.0031	-1.7764
i_{UL}	0.0024	-0.1131
γ_{UL}^{-1}	0.0020	-0.0938
s_2	0.0018	-0.1683
i_S	0.0012	-0.0557
γ_S^{-1}	0.0011	0.0216
f_L	0.0005	-0.0460
μ_K^{-1}	0.0001	-0.0708
s_1	0.0001	-0.0692
s_4	0.0001	-0.0099
ρ_C^{-1}	0	0.6465
ρ_D^{-1}	0	0.1220
i_{T1}	0	0.0000
i_{T2}	0	0.0000
i_L	0	0.0000
τ_1^{-1}	0	0.0014
τ_2^{-1}	0	0.0001
τ_3^{-1}	0	0.0003
d_{T2}	0	0.0025
C_{ITN}	0	1.8081
C_{IT1}	0	-1.0668
d_{T1}	-0.0001	0.0455
ρ_L^{-1}	-0.0001	0.0069
p_{TL}	-0.0024	-0.0040
σ_F^{-1}	-0.0111	0.5252
β^{-1}	-0.0849	3.8384

Figure 8. Dependence of p_{HI} on different parameter values. Unless varied, the parameter values are as specified in Tables 2, 3, 4. For those parameters, $p_{HI} = 0.95963$, $p_{NE} = 0.95956$.

Figure 9. Dependence of p_{HI} on different parameter values. Unless varied, the parameter values are as specified in Tables 2, 3, 4. For those parameters, $p_{HI} = 0.95963$, $p_{NE} = 0.95956$.

37
royalsocietypublishing.org R. Soc. open sci. 0000000

Table 7. Policy-Relevant Items for Reporting Models in Epidemiology of Neglected Tropical Diseases (PRIME-NTD) Summary Table as specified in [83].

Principle	What has been done to satisfy the principle?	Where in the manuscript is this described?
1. Stakeholder engagement	We did not directly engage the stakeholder. However, we based our model on [57] and extensive VL literature describing KalaNet trial.	Sections 1 and 2
2. Complete model documentation	We provided a detailed model description. Implemented the model numerically in Matlab with ample comments.	Section 2. Supplementary information.
3. Complete description of data used	We calibrated our model on data available in the literature. We detailed procedures of how we extracted parameter values that were not found directly. We derived VL transmission probabilities.	Section 4 Section 4 Section 5 (a)
4. Communicating uncertainty	We performed sensitivity analysis based on [134].	Section 5 (e), Table 6, Figures 8 and 9 and the Matlab code
5. Testable model outcomes	We predicted that asymptomatic individuals (PCR+, DAT+) can transmit VL and their infectiousness is at about 50% of the level of untreated symptomatic individuals. We gave formula for the minimal ITN coverage needed to achieve complete elimination of VL and predicted that 96% ITN coverage is needed. We also predicted that voluntary use of ITN should yield elimination of VL as public concern.	Section 5 (a) and (b) Section 5 (c) Section 5 (d) with mathematical details in Appendix C

Response to reviewers' comments

We thank the editor(s) and the reviewers for careful reading of the manuscript and for their suggestions and comments. We appreciate all the feedback given to us. We believe we successfully addressed all the comments and issues. Detailed list and responses are below.

Reviewer 1 wrote:

Comments to the Author(s)

This is an interesting and very well written paper about mathematical modeling of individual behavior regarding the use of insecticide-treated nets (ITN) as preventive method against visceral leishmaniasis (VL). The authors present in detail a modification of the mathematical model describing the sand flies-humans dynamics, previously developed by Stauch et al. (2011, PLoS Neglected Tropical Diseases). The authors study the dynamical system within a game-theoretical framework, minimizing the expected cost of using ITN. The main outcome of the model is the proportion of people using ITN that may lead to VL elimination.

Our response: We thank the reviewer for a careful reading of the manuscript and for comments and suggestions for its improvement

Reviewer 1 wrote: The manuscript may benefit from reformulating and/or explaining some modeling choices. For instance, in line 86, the authors mention free-riders but don't mention whether or not they model the free-rider behavior explicitly.

Our response: We added "*incorporated in our model as individuals that do not use ITNs,*" to the sentence

Reviewer 1 wrote: In line 110, the authors say that equation (2.1) may be interpreted as if "ITNs provide a very effective protection", while their model considers perfect protection, instead.

Our response: It was an unfortunate wording on our end. We deleted the sentence and added this just before equation (2.1): *For simplicity, we assume that ITNs offer a perfect protection and thus the force of infection is ...*

We further added paragraph about possible model extension to the discussion where we mention how to include imperfect protection.

Reviewer 1 wrote: Also, the choice of the computation/definition of R_0 could be supported by a reference.

Our response: We added a reference to Anderson and May 1991, a recent review by Delameter et al and van den Driessche 2002

Reviewer 1 wrote:

Assumption (iii) in line 176 could also be supported by a reference. In addition, the authors could contrast this assumption with available data on the population size. If not, they could discuss the limitations of such an assumption.

Our response: We did not find a reference but we added the following “*This change makes the ODE system less sensitive to changes in the birth rate and death rates. In the original model of \cite{stauch2011visceral}, even a small change in α_H , μ_H , or μ_K would destabilize the system and result in either exponential growth or exponential extinction of the population. In our model, a (reasonable) change in Λ , μ_H , or μ_K will cause the system to converge to a potentially different population size, but the population will not go extinct nor grow above any bound. We can also solve for the equilibria of the dynamics.*”

We did not include it in the text but we noticed that Stauch et al (2011) never mentioned the value of alpha in their manuscript. We suspect that they just picked the value of alpha_H at the end once all other parameters were fixed. That (and only that) choice of alpha_H made their ODE system converge to an equilibrium rather than going to 0 or infinity.

Reviewer 1 wrote: Please consider discussing the stability of the equilibria, which may give further insight about the system behavior and thus helping to interpret the results, as well as discussing the results of the KalaNet trial (see line 468).

Our response: We added the following after the calculation of R_e : *It follows from \eqref{eq:IF} that the endemic equilibrium exists only if $R > 1$. While we did not perform a formal stability analysis, the ODE system -- although large -- is quite standard and similar to one considered in \cite{wei2008epidemic}. Consequently, the disease-free equilibrium is globally asymptotically stable if $R \leq 1$, and unstable if $R > 1$. The endemic equilibrium is locally asymptotically stable if $R > 1$.*

Reviewer 1 wrote:

The authors compute the reproduction number for the system subject to a control measure. Therefore, it is the effective reproduction number and not the basic reproduction number. The authors could use the notation R_e , instead of R_0 , where $R_e = R_e(p)$.

Our response: We thank the reviewer for the comment. We changed to R_e as suggested and now call it properly the effective reproduction number instead of basic reproduction number.

Reviewer 1 wrote:

Please recall the parameters values used to produce figure 4, in the caption. Please expand the description of figures 5 and 6 in their captions, including a short description of the parameters set depicted in the x-axes.

Our response: We added the line: *The parameter values are as specified in Tables \ref{tab: Sand Flies}, \ref{tab: Humans}, \ref{tab: Treatment}.* to the caption of Figure 4. There already was a similar statement in the caption to Figure 5 and 6. The model has too many parameters to list in the caption.

Reviewer 1 wrote:

Please ensure notation consistency for numbers of 4 or more digits.

Our response: We adjusted the output of sensitivity analysis to 4 digits. The difference between ITN use needed for complete elimination and the Nash equilibrium coverage (resulting from optimal voluntary use) is quite small and 5 digits were needed to show the difference.

Reviewer 2

Comments to the Author(s)

SUMMARY

My recommendation to the Editor is to accept this manuscript, pending major corrections. Please interpret 'major' in the sense that these suggestions will likely affect the results. I thought the quality of the research and its write-up were of a high standard for this modelling manuscript and would very much like to see its corrected version added to the VL research base.

The findings: The authors have developed a model by amending an existing one and parametrising it with mostly reasonable values and ranges. The two research impacts are:

A) They obtain a threshold estimate for the ITN coverage required for (zero) disease elimination. (labelling this as RI-A)

B) Considering that ITN use is voluntary, they make a sensible and (probably for VL) novel application of game theory to assess if voluntary ITN use alone, influenced by the costs and benefits at the individual level, is sufficient to meet this threshold. (RI-B)

Quality: It is of a high research standard: indicated by the mathematical foundations and clear exposition in the text, strongly informed in model structure and parameters by existing VL research, the rigour of the analysis including validation and sensitivity analysis. Moreover, it is the first manuscript I have reviewed which provides working code and reproducible results! It therefore ticks many best practices of 21st century modelling.

Our response: We thank the reviewer for the kind and encouraging words!

Reviewer wrote:

Potential impact: It also contributes to the ongoing discussion following the unfortunate findings of the Kalanet trial in the previous decade. This is the primary reason why ITNs do not form a part of today's VL control. WHO has launched the NTD 2021-2030 roadmap earlier in 2021 and so this could be an ideal time for research during this receptive policy window. It could reawaken minds and inform future epidemiological and entomological study designs to definitively answer if ITNs have a contributing role in VL

control on the Indian subcontinent. The manuscript could work harder at highlighting this
point for the VL policy maker audience.

**Our response:** We rearranged the discussion and tried to make the point as reviewer
suggest.

**Reviewer wrote:**

To open up this research piece to further discussion and improve its publication impact I
suggest releasing the initially submitted draft as a preprint and later interacting with the
VL research community at forthcoming conferences such as WorldLeish 2022 which will
probably start accepting abstracts later this year, but this of course is at the authors'
discretion.

**Our response:** We will definitely keep the conference in mind although we may not be
able to physically travel anywhere in near future. We are more than happy to collaborate
and interact with VL community if a remote collaboration is a possibility. We do not have
any contacts in VL community and would be very happy if the reviewer could help us
share our work.

**Reviewer wrote:**

Scope of this review: I spent 3 days reviewing the manuscript, code and writing up this
review. I did not check the majority of the mathematical formulae at an analytical level.
Instead I focussed on code correctness from code lines 1-478, parameter suitability and
if this modelling was appropriate for VL.

Conflict of interest: I declare that I have included citations within this review which either
myself or my supervisors (T Déirdre Hollingsworth & Lloyd Chapman) are co-authors.
The authors are not compelled to add these citations. I included them to provide
justification and to encourage the authors why a suggestion is necessary or why it has
been raised.

Current major limitations: I believe there are some major bugs in the code that would
affect results but these should be quick to fix. Some coding practices may also help
improve the confidence in the code correctness. Some parameters may not fit the
setting or understanding of the disease the best and I have highlighted the ones most
likely to impact results, partly from the authors' helpful sensitivity analysis; this may
require a rapid review of the recent literature for parameter updates but again this
shouldn't take long. No change to the model structure is necessary. Finally, although
the manuscript text itself was painless to read, it could focus better at the main
audience—the VL research community. Some of these are detailed below and are also
attached on commented files.

**BEFORE THE AUTHORS READ THIS REVIEW (OR THE PUBLIC, AFTER**
**PUBLICATION)**

My overriding motivation for this peer review is to protect the field from errors in
research methods or misinterpretation of results, that could make policy makers make
suboptimal decisions in VL programmes that ultimately harm people. The annual
research output for VL modelling is relatively small compared to other diseases, so you

can realise that avoidable errors could have more impact in this context than others. I
haven't seen the authors' names from previous VL modelling research, so if you are
new to VL then welcome to the field! None of the following review is intended to 'put you
in your place' nor be patronising, and your future modelling for this neglected tropical
disease is very much welcomed. In the spirit of open science some of the reviewers
may have declared their names to be made public and in return I hope the authors allow
all reviews to be open (for reviewers who allow it).

REVIEW STRUCTURE

This review is structured by the manuscript then code run, and grouped by things
essential to impacting the results' values or presentation (major) or affecting the
understanding/model improvements. This review is long but I believe in thoroughness
and once some of what I suggest is corrected, then I think what you have got here could
be really valuable to the VL field.

**Our response:** We cannot thank the reviewer enough for all the input and hard work
that was put in the review. We benefited tremendously from all the comments,
suggestions and provided references.

**Reviewer wrote:**

MANUSCRIPT

MAJOR

Top, most important

• An extensive range of research has been drawn upon to inform parameters selected.
Some suggestions have been made from empirical sources for the refinements of some
parameters. See mns attachment for more details.

**Our response:** We list and provide detailed responses to all major comments in the
manuscript separately below.

**Reviewer wrote:**

• "Multiple lab-based studies suggest that female sand flies have fairly short adult
lifespans (<20 days) [16] with further reductions when infected [15]"
(<https://journals.plos.org/plosntds/article?id=10.1371/journal.pntd.0009033>) therefore it
is unrealistic to assume the mortality of I_F in Fig 1 is the same as S_F or E_F (=mu_F).

**Our response:** We added a subsection with several paragraphs about model limitations
and possible extensions and added the fact about varying mortalities as one of several
extensions to consider. While the reviewer writes above that "No change to the model
structure is necessary" and this addition seems like a minor one, it actually requires a
relatively major change in many of the analytical calculations(see also our response
about stability re reviewer #1 comment to line 176). Related to several other reviewer's
comments, we are more than open to collaborate with the proper set of authors and
stakeholders and create an even better model(s) in the future. 50

**Reviewer wrote:**

• Making it more policy-friendly: reduce the amount of mathematical formulae by moving
formulae (lines 212-223 & 446-452) describing the intermediate mathematical logic to a
Supplementary information as there's quite a few pages to flip through from line 185-

225. Please do not understand what I mean, I just mean reduce the amount, the level at
which you are communicating is fine and there is no need to dumb things down. Your
short-hand notation like Comp on line 197 was clever and for instance Eqns 3.1-3.15
could be represented as a one-liner, something like $dX/dt = 0$ where $X=\{\text{etc}\}$.

**Our response:** We moved most of the math to the appendix as suggested.

**Reviewer wrote:** Although the Discussion covers limitations in the findings, model
limitations are also necessary.

**Our response:** We now have a subsection on model limitation and extension.

**Reviewer wrote:** Also despite the model being validated against previous trial data,
what would be the external validity of these findings to say Nepal?

**Our response:** We added the following to the section on model validation:

*To validate the model on another data set, one would have to potentially update the*
*parameter values to properly reflect the time and location of the experiment where the*
*data came from. Then, we can use formulas for endemic equilibrium from the Appendix*
*A\ref{ap: endemic} to obtain distribution of population accross different model*
*compartments.*

**Reviewer wrote:**

• Parts of the Introduction feel a bit encyclopaedic i.e. they demonstrate an excellent
understanding of the natural history of disease progression and epidemiology however
some of the detail may not be necessary if it is not part of the model. Therefore,
streamlining the Introduction would save you several lines.

**Our response:** We followed reviewer's suggestions in the actual manuscript and
removed the zoonotic VL models from the introduction. If we misunderstood, we can
insert it back.

**Reviewer wrote:**

On the other hand, the authors introduce game theory concepts that are likely not to be
familiar to the VL audience and elaborating a little further would save the reader having
to follow the citations to understand the basic concepts of game theory.

**Our response:** Connected to comments of Reviewer 1, we added some more about the
game theory.

**Reviewer wrote:**

• Title may need changing depending on the corrected results. The current title for the
presented results sounds a bit optimistic given line 478. It also gives the impression that
RI-B is the primary finding, plus the use of 'game theory', 'nash equilibrium' keywords
alongside, however the abstract doesn't mention it and only mentions RI-A. I think the
main body of the mns does give proper precedence to RI-A and then RI-B as a
secondary and consequential finding. In fact, I think in the weight of scientific evidence,
that RI-B is weaker than RI-A, as quite broad assumptions are applied to achieve RI-B,
whereas RI-A is mostly informed from the published literature; this is elaborated on the
attached manuscript lines 87-91. The authors may believe their main contribution to the
field is the novel application of RI-B, when in fact I think it could be the robust model
developed for RI-A which has the greatest effect.

**Our response:** We revised the title and the keywords. Based on some other reviewer's
suggestions, we added a bit more emphasis on the parameter estimation and in
particular the role of asymptomatic individuals.

**Reviewer wrote:**

Have any other papers you are aware of estimated an ITN coverage threshold for
elimination? I can't think of any. If not then this needs to be emphasised more in the
mns as a key finding to highlight the contribution your work is making.

**Our response:** We are not aware of any other work applying math modelling of ITN to
VL and we inserted a statement early in the discussion in that effect.

**Reviewer wrote:** Also, as this study is based in the IN/NP country setting, whereas VL
also affects East Africa & South America, it may be worth including the regional setting
eg "in Bihar, India" in the title.

**Our response:** We added as suggested

**Reviewer wrote:**

MINOR

In no particular order:

• Other than suboptimal ITN coverage, other reasons have not been mentioned that
could explain the lack of an effect of the Kalanet trial e.g. from [42] "In theory, LNs could
be an effective tool to prevent *L. donovani* transmission, as *P. argentipes* are supposed
to bite people indoors while they sleep [14]. However, recent entomological findings in
India indicate that *L. donovani* vectors are more exophilic and exophagic than
previously reported [15,16]. If *P. argentipes* bite people outdoors (e.g., in the early
evening when and where bed nets are not deployed), LNs will have a limited impact on
*L. donovani* transmission".

**Our response:** We added the paragraph about possible model extension in this
direction to the subsection "model limitation and possible extensions"

**Reviewer wrote:**

• On p3 line 40 you state that ITN ownership is associated with wealth however the
assumptions on p4 line 78-79 collapses this to a single mean level for everyone. The
heterogeneity of affordability for medical costs produces uncertainty that is hidden in the
point estimate p_{HI} . Also, the VL susceptibility is bound to reduce with wealth, as
malnutrition and certain interior wall types are VL risk factors.

**Our response:** We added this as another of the model limitations

**Reviewer wrote:** • Consider adding a PRIME-NTD summary table to your appendix to
describe how you have interacted with
policymakers. <https://journals.plos.org/plosntds/article?id=10.1371/journal.pntd.0008033>

**Our response:** We did as suggested.

**Reviewer 2 wrote:** • Some readers may interpret “elimination” as “elimination as a
public health concern” rather than true/zero elimination. Clarifying this at first use would
help.

**Our response:** We thank the reviewer for bringing this to our attention. We t specifically
say 0 cases now or use the phrase complete elimination. At the same time, we realized
that the voluntary use of ITNs brings the disease to elimination as a public health
concern and explicitly say it too.

**Reviewer 2 wrote:**

• Note that eradication means zero disease globally so perhaps you mean elimination
which is local to a particular region.

**Our response:** We thank the reviewer for spotting this; this word escaped our attention.

**Reviewer 2 wrote:**

• I note that all the current authors are from US institutions. Although the authors did an
excellent job in summarising the pertinent epidemiology and dynamics, I think the
addition of authors with in-country experience would have greatly informed on the real-
world conditions in Bihar, India. Please note I am basing the authors in-country
experience on their institution location which could be false. Journals have rejected NTD
manuscripts from my European colleagues last year because they didn't have authors
from the affected continent and I have to agree with this policy. I encourage Editors to
challenge this in future and the authors to collaborate with in-region researchers as
otherwise research quality could suffer. Also it would deprive upcoming researchers in
these regions from contributing to research that affects their home.

**Our response:** We share reviewer's sentiment. For what it is worth, Jan Rychtar was in
Bihar in 2010 (specifically in the vicinity of Patna, Gaya and Bodh Gaya). While he did
not work on VL at that time, he was able to experience the conditions of rural India. The
nature of this particular project (6 intensive weeks during summer 2020 with many last
minute changes due to COVID-19) prevented us from having too extensive
collaborations on this.

However, we are very open to collaboration on further VL projects. Profs. Rychtar,
Taylor and Lu will be happy to work closely with the reviewer and/or his/her team on VL
modelling in the future. In fact, we have another cohort of students coming in June 2021
and, with an appropriate planning, this could give our students an excellent experience
according to the best practices described by the reviewer and the provided references.

**Reviewer 2 wrote:**

**CODE RUN**

Excellent use of commenting and code structuring in the MATLAB file. Believe it or not I
did run it and also read through it to try and find bugs! So, thank you very much for
spending the time in making the code understandable; it was not wasted time! Please
note a commented PDF of the code is also attached.

**Our response:** Thank you for spending the time to check and run the code. We are
glad the comments were helpful. In the revision, we tried to improve based on your
suggestions.

MAJOR

Top, most important:

**Reviewer 2 wrote:**

• Code lines 60 & 115 incorrect parameter values. See attached. Editor: please don't
mark the paper down for these errors. Think how many papers get published without
ever releasing their code!

**Our response:** The values are and were correct. The issue stems from units (years in
Matlab versus months in the manuscript). We believe the months are more natural for
the manuscript. However, for figures, years are more natural so it was coded in years
for Matlab. The comments about conversion were already in the original code. We
rewrote $1/70$ to $1/(70*12)*12$ [in fact to $1/(68.7*12)*12$ due to new value as pointed by
the reviewer at another place] and $1/40$ to $1/(40*12)*12$. The values did not change but
now they are written in a style that is consistent with other parameter values

**Reviewer 2 wrote:**

• Manuscript figs 2 & 3 were not produced when the code was run so not all the figures
are reproducible.

**Our response:** The commands that called the functions to generate the figures 2 and 3
were indeed left commented out in the submitted code by an overlook. We
uncommented them and added a specific disp to the code to make it clearer that the
figure are being generated.

**Reviewer 2 wrote:**

• I am not entirely confident that the code runs as the authors may have intended. This
is not just because of the initial errors found on code lines 60 & 115 but because I could
not find any unit tests in the code to check the correctness of the code output. This
could be improved by simple error checks inserted into functions to check function
arguments (especially those called multiple times), and checks on the expected value
ranges and data structures from each function return, which is essential in my opinion to
be confident in the results. This is set out in further detail in Responsible modelling: Unit
testing for infectious disease epidemiology

(<https://www.sciencedirect.com/science/article/pii/S1755436520300451>). I encourage
Prof Rychtář to involve the other co-authors in the collective task of pair code review
and mentioning this in the Methods to add to the confidence in the code correctness.

**Our response:** We added a subsection (Numerical implementation) and added a
reference to the paper. We included several consistency tests in the code (we more or
less had them in previous versions of the code but deleted them in the final submission
as we did not realize that they should be an integral part of the code as well. We
debugged the code as it was written and constructed (prior the original submission).
We often did not end the lines by “;” and inserted plenty of “pause” and manually redid
the calculations on the paper/calculators for numerous values to make sure the code
runs as intended. We also generated figures that are no longer part of the code just to
see that the outputs make sense.

We fully agree with the reviewer that without such tests, the code would be full of bugs.
In fact, by reinstating the tests, we found and corrected a small typo in the crucial
function for obtaining equilibria.

MINOR

In no particular order:

**Reviewer 2 wrote:**

• The file runs in one go and outputs all 41 MATLAB figs to screen and also saves as
PDFs; this is good. It is unclear however from the screen, which of the figures relate to
which part of the code, unless code chunks are run separately. Therefore, can the disp()
and saveFig() also include a title to match it to the code please. This caused some
inefficiency to the reviewing process.

**Our response:** We now added the command that displays the file name and links it to
the figure to the file name

**Reviewer 2 wrote:**

• Please could you make your publication, SI and code also available on Zenodo or
similar archived repo as some journals have a time limit for archive removal.

**Our response:** We opted for an open access journal so that the final and reviewed
version could be accessible to everyone.

**Reviewer 2 wrote:**

• Is MATLAB fig1 = manuscript fig 4? If yes then this has been successfully reproduced.
The same goes for manuscript figures 5 {7-10,12,15-17,20-27,18-19,11} & 6 {13-14,29-
31,33-38}? where the left-to-right order here refer to their top left to top-right then
carriage return order in the sub figure arrays in the manuscript. There were some
orphaned MATLAB figures generated {28,32,39-41}. Could these be described by a list
of figures and caption for just these orphaned figures in a Supplementary information?
Otherwise if they were for the authors' own checking then better to comment out their
figure/PDF generation when the code is run as a whole.

ENDS=====

**Our response:** We commented out the sections of the code that do not produce figures for the
manuscript. (we also uncommented the sections that produce the figures for mns but were left
commented out in the original submission).

Responses to comments in the pdf for the matlab file

- - We added a check for the optimization toolbox. The code should now run without it
(although parameters values may be a bit off if the user changes some parameters. The
transmission probability values were hardcoded in the case the user does not have the
toolbox). Proper prompts should be now displaying if the code run without the toolbox
(although we were not able to test it since our version has the toolbox)

**Reviewer 2 wrote:** On mns line 55 you are informed by Stauch on vector control parameters
but why are non-vector parameters like f_{HS} also being changed?

**Our response:** Our understanding of the work of Stauch et al (2011) was that they run the
optimization technique to fit their model to KalaNet data and through this they estimated many
parameters. We tried to avoid the parametrization by fitting to data as much as possible. Since

we could find the data for f_{HS} (and others) directly, we used the values we found rather than
those that were estimated.

**Reviewer 2 wrote:**

1d is far too short for VL onset to treatment time for this study setting. Also it's not within the
range quoted in Table 3. Pls double check that other parameters are within ranges quoted in
mns tables.

**Our response:** We agree that one day is not realistic. We did not use one day but one month as
specified in Table 3 and earlier in the code. Reviewer's comment is in the section where we set
the parameters to the values used in Stauch et al. They used 1 day and so we had to code it as
one day here.

**Reviewer 2 wrote:** But I thought that earlier you assumed that treated cases are not infectious.

**Our response:** We agree and we did assume it. Reviewer's comment is in the section where we
define parameters as in Stauch, so we had to use what they did

**Reviewer 2 wrote:** As $\text{sum}(\text{KalaNet})=1.005>1$, does this require normalisation first?

**Our response:** The human parameters add to 1 and the flies are extra. We agree that the issue
was not clear from what we wrote there, so we added a comment to the code to make it a bit
clearer

**Reviewer 2 wrote:** The command window reports this as a local minimum. Would it be worth
trying random parameter initial conditions to check that lower local minima don't exist?

**Our response:** We put back the section of the code where we test exactly for that. We also
included a section in the appendix where we describe that we randomly seeded the search with
1000 initial values and searched among all local optima we found. It turned out that starting the
search at 0 yields close enough results. 35

**Reviewer 2 wrote:** Why not $(1/38*12, 1/10*12)$ as per Table 3 range?

**Our response:** The reviewer is correct, there was no reason for the values used in the matlab,
so we regenerated with values from the table

**Responses to comments in the pdf manuscript file**

**Reviewer wrote:** I'm a bit confused with the result of Fig 3 and was wondering if your text in 5a
could help. It just seems a bit counterintuitive that for higher p the probability of i_D and i_P
Asx humans infecting sandflies is higher.

**Our response:** We added the following sentence to the end of 5a to clarify

Note that the functions are indeed increasing in $\$p\$$ - to achieve the same disease prevalence
as measured in KalaNet trial when the population uses higher level of protection $\$p\$$, the
disease must be more transmittable.

**Reviewer wrote:** 14 is a case report. Is there a better source that can more broadly characterise
clinical features?

**Our response:** We agree and this is why we cited two sources. We can eliminate the reference
to the case report completely if this is preferable.

**Reviewer wrote:** Staff from CARE have also reported KA symptoms to oscillate ie a fever that
comes and goes over weeks.

**Our response:** We were not able to locate the information. We will be happy to include it, but
could you please provide us with the reference? One of the suggested WHO publications talks
about CARE, but it is regarding deaths reports, nothing about symptoms.

**Reviewer wrote:** You may be interested by this recent Xeno study
[https://www.thelancet.com/journals/lanmic/article/PIIS2666-5247\(20\)30166-X/fulltext](https://www.thelancet.com/journals/lanmic/article/PIIS2666-5247(20)30166-X/fulltext) and
commentary [https://www.thelancet.com/journals/lanmic/article/PIIS2666-5247\(20\)30222-](https://www.thelancet.com/journals/lanmic/article/PIIS2666-5247(20)30222-6/fulltext)
[6/fulltext](https://www.thelancet.com/journals/lanmic/article/PIIS2666-5247(20)30222-6/fulltext) that supports the view that Asx have no role in transmission versus our
spatiotemporal analysis that found a 2% relative infectiousness of Asx vs KA was consistent with
the data <https://www.pnas.org/content/117/41/25742>.

**Our response:** We added a relatively detailed analysis of findings from Singh et al 2021. We
found that despite the highlights indicating up to 88% transmission from active VL and PKDL
cases, only about 10% of bites actually result in parasite transmission. We now discuss this as
well as the role of asymptomatic individuals on several places of the paper. In particular, we
point out that what Singh et al call asymptomatic cases, Stauch et al (and consequently us) call
recovered; i.e., there is no factual difference between their findings that those individuals do
not transmit the parasite and our model (that assumed no transmission from recovered classes
from the beginning).

**Reviewer wrote:** Is this relevant though for VL modelling in India?

**Our response:** We agree that the relevance to India is low but wanted to point out to other
models of VL relevant to other regions. We can delete the sentence if needed.

**Reviewer wrote:** But by what process do individuals receive this information? Surely incidence
and ITN coverage are not known by villagers.

**Our response:** The reviewer is correct. We acknowledged the issue in the section on model
limitation.

**Reviewer wrote:** However this line of reasoning is based on vaccines, most of which confer
long-term protection and are a one-off decision based on information available then, whereas
ITNs have to be continually used by homeowners and also repaired/replaced while the person
has to be continually receiving information on the threat VL poses, since if they don't they then
forget about it due to other acute health priorities. For a disease like VL with long incubation
periods it may be difficult for people to infer/see the benefit of ITNs for themselves as the time
of infection is hidden and onset of symptoms happens 5 months later. Therefore they will have
forgotten their poor adherence of ITNs when exposure occurred, so they won't see a clear link
between cause and effect.

**Our response:** We agree with what the reviewer wrote but believe that it stems from our
improper introduction of game theory. The game theory assume that individuals are selfish and
act in their own self-interest We added this to the intro paragraphs: *In our setting, the game*
*theoretical framework assumes that people will sleep under ITN not to protect others, but to*
*protect themselves. Specifically, people will not start sleeping under ITNs once they learn they*
*have VL, but they may start sleeping under ITN once they learn their neighbor has VL.*

The time lag the reviewer mentions is important though and we added it to the model
limitations/possible extension.

**Reviewer wrote:** Was the code also adapted from theirs or did you write this from scratch?

**Our response:** We wrote the code from scratch.

**Reviewer wrote:** This unrealistically assumes 100% net adherence and efficacy.

**Our response:** We agree and put a paragraph about it in the discussion.

**Reviewer wrote:** I_A could be more intuitive as a label

**Our response:** We agree. We tried to adhere to the original Stauch et al (2011) as much as
possible. Their notation is perhaps not entirely intuitive, but it has an inner logic and so after
various attempts on our side to change it, we reverted mostly to theirs. We just dropped extra
subscripts to make our formulas fit on a line as much as possible,

**Reviewer wrote:** What evidence shows that dormant PKDL or active PKDL cases are LST-ve? Is a
weaker version more realistic? ie LST status predicts progression to PKDL but some PKDL cases
may still be LST-ve?

**Our response:** Here we followed Stauch et al (2011) exactly. The only effect of whether PKDL
are or are not always LST negative is on the position of I_TL and I_UL (lumped into I_HL in
Stauch) in the model diagram. It does not influence any flow to or from the compartments.

**Reviewer wrote:** Is this true of Miltefosine though? Surely far less toxic than the previous SSG.
**Our response:** we added: *(although a newer drug Miltefosine is far less toxic than previous SSG*
*\citep{den2009developments})*

**Reviewer wrote:** Tidy diagram with consistent inter-compartment spacing and use of arrows in
8 directions. However the use of white space could be improved to reduce congestion in the
I_{T1} area. Worth labelling the two groups for human host and sandfly host. Listing what
symbols mean here as a fig1 caption to avoid referring back to p5. Worth stating in the caption
that the arrow path annotations represent transition rates. Worth adding a traffic light label to
each compartment to aid reader in understanding PCR/DAT/LST transitions. Shade infective
stages or KA vs PKDL?

**Our response:** We did most of the suggestions. If we did not reduce the congestion enough, it
was not because we did not want to or did not try, we just did not know how to do it better.

**Reviewer wrote:** but $f_L + f_S + f_D \neq 1$. See Table 3. Can they be normalised to fix this?

**Our response:** Thank you so much for catching this. They should add to 1 and they add to 1 in
the matlab code (because we define on f_L and f_S and calculate f_D as $1 - f_L - f_S$ as specified in
the right column of Table 3. However, the written numerical value of f_D was indeed incorrect
and it is fixed now.

**Reviewer wrote:** extra 3 cols for PCR/DAT/LST would aid reading

**Our response:** We added the columns to the table as suggested

**Reviewer wrote:** I_{TL} duplic?

**Our response:** It was a typo, one is now fixed to I_{UL}

**Reviewer wrote:** Lifespan? As expectancy is used more for summaries of the lifespan
distribution in demography.

**Our response:** We changed it to Expected lifespan in the light of this suggestion. We kept “life
expectancy”, i.e., the average number of years that a newborn is expected to live (if current
mortality rates continue to apply) for the corresponding human parameter.

**Reviewer wrote:** Why not γ_{UL} rather than γ_L for consistency with fig 1?

**Our response:** We tried to have only one subscript whenever possible because otherwise the
formulae did not fit on one line too well. For most compartments and rates it meant to drop
“H” from the subscript; here we had to drop U.

**Reviewer wrote:** Since these are intelligent guesses it may be worth parameter hunting in
[https://www.thelancet.com/journals/lanmic/article/PIIS2666-5247\(20\)30166-X/fulltext](https://www.thelancet.com/journals/lanmic/article/PIIS2666-5247(20)30166-X/fulltext) and
<https://academic.oup.com/cid/article/69/2/251/5144025>

**Our response:** We thank the reviewer for the reference. We incorporated the data from these
papers into the model calibration (and the paper in general)

**Reviewer wrote:** Table 6 here from a recent mortality survey will provide better estimates

[https://www.who.int/docs/default-source/searo/evaluation-reports/independent-assessment-](https://www.who.int/docs/default-source/searo/evaluation-reports/independent-assessment-of-kala-azar-elimination-programme-in-india.pdf?sfvrsn=fa0d8baa_2)
[of-kala-azar-elimination-programme-in-india.pdf?sfvrsn=fa0d8baa_2](https://www.who.int/docs/default-source/searo/evaluation-reports/independent-assessment-of-kala-azar-elimination-programme-in-india.pdf?sfvrsn=fa0d8baa_2)

**Our response:** We are unable to decipher data from Table 6 and make estimates for mortality if
untreated. While Table 6 that a significant number of patients die within a month from the
diagnosis, it is not quite clear how long it took to be diagnosed and the report itself states that
the mortality is associated with many factors, including late health care seeking. Nevertheless,
we included a comment about it in the manuscript (and we thank the reviewer for pointing us
in this direction).

**Reviewer wrote:** Doesn't the Indian govt pay for treatment costs, lost wages and travel costs?
Perhaps this recent survey will help answer the reimbursement scheme's effectiveness
(<https://journals.plos.org/plosone/article?id=10.1371/journal.pone.0227911>)

**Our response:** We added the following paragraph: *The Indian government provides a free care*
*to VL patients; however, hoping for a quick cure of seemingly minor illness, patients prefer to*
*access private providers which contributes to high out-of-pocket expenditures*
*\citep{nair2020quality}. VL is often misdiagnosed in private hospitals but patients keep*
*accessing care in private sector until no money is left \citep{nair2020quality}.*

**Reviewer wrote:** I'm a bit confused with the result of Fig 3 and was wondering if your text in 5a
could help. It just seems a bit counterintuitive that for higher p the probability of i_D and i_P
Asx humans infecting sandflies is higher.

**Our response:** We added the following to the end of the section: *Note that the functions are*
*indeed increasing in p - to achieve the same disease prevalence as measured in KalaNet trial*
*when the population uses higher level of protection p , the disease must be more*
*transmittable. RESPONSE MAY CHANGE BASED ON HOW THE PARAMETERS FOR TRANSITION*
*TURN OUT*

**Reviewer wrote:** worth adding caveats: people get perfect info, broken nets are replenished

**Our response:** We added (*provided people get perfect information and broken ITNs are*
*replenished*)

**Reviewer wrote:** If you're talking about Fig 4, don't you mean cost function?

**Our response:** We added (which correlates with the mean cost function)

**Reviewer wrote:** This was for vaccination rather than vector control programmes.

**Our response:** Correct. We added reference to a recent paper by another group of our students
that deals with Chagas and ITNs

<https://journals.plos.org/plosntds/article?rev=2&id=10.1371/journal.pntd.0008833>

**Reviewer wrote:** sort desc?

**Our response:** We sorted table 6 in descendent order

**Reviewer wrote:** split in two for infected/uninfected sandfly

**Our response:** We added a section on model limitation and possible extension and discuss this
mortality split there. The incorporation of the split would result in a complete revision of the
analytical formulas and calculations.

**Reviewer wrote:** [84] only refers to India. Rural Bihar specifically (and temporally closer to
Kalanet data) available from 2011 online census. Same for human death rate μ .

**Our response:** We thank the reviewer for pointing us to the proper place to search and we
updated the numbers accordingly to the data found on rural Bihar in online publications from
Indian Census to life expectancy 68.7 in rural Bihar and crude birth rate 27.7 per 1000 in rural
Bihar. 53

**Reviewer wrote:** Acknowledgements: Anyone to mention?

Our response: We added Dr. Wasserberg who pointed us in the right direction re vectors and we also thank the reviewers for their valuable feedback and comments.

ROYAL SOCIETY
OPEN SCIENCE

rsos.royalsocietypublishing.org

Research

Article submitted to journal

Subject Areas:

mathematics, biology, epidemiology

Keywords:

Kala-azar, PKDL, Asymptomatic
transmission, Parameter estimation,
Vector-borne diseases

Author for correspondence:

Jan Rychtář
e-mail: rychtarj@vcu.edu

~~A voluntary~~ Mathematical
modeling of the use of
insecticide treated nets ~~can~~
~~help eliminate~~ for elimination
of visceral leishmaniasis in
Bihar, India

[revised manuscript text omitted]

The control of VL depends on chemotherapy treatment [13], vector control [30], ~~and~~ bite prevention [31] and active case detection [32]. Human VL vaccines are not yet fully developed although several trials are under way [33,34] and their impact is already modelled [35]. Preventative measures in high risk areas include (1) avoiding sleeping in mud and thatched housing [1], (2) environmental management [36], (3) using insect repellent [37], (4) using insecticide treated netting, ITN, [38–40], (5) indoor residual spraying [8] and, most recently, (56) treated wall lining [41].

The ITN ownership varies significantly with wealth; almost all of the wealthiest households own an ITN while many of the poorest do not [42]. However, even in relatively poor areas, the use of ITNs was already demonstrated to be very cost-effective [43,44]. Human behavior such as inconsistent use due to hot weather or inadequate education diminishes the effectiveness of ITNs [40,45].

KalaNet, a cluster randomised controlled trial in Nepal and India evaluated the efficacy of long lasting ITNs in the prevention of VL [46]. It has been demonstrated that a cluster-wide

distribution of ITNs reduced the vector density [47] but did not reduce the risk of *L. donovani*
infection or clinical VL [48]. Other trials in Bangladesh showed that ITNs may reduce the VL
incidence rate [41,49]. The use of ITNs was also recommended for PKDL and VL-HIV patients
[10]. This apparent contradiction raises the question about the role that ITNs may play in
controlling VL [48]. The purpose of this paper is to gain insight into the situation by mathematical
~~modelling~~ modeling.

There are many mathematical models of VL dynamics, see for example [50–53] for recent
reviews. A model of VL transmission at the district-level of Bihar to estimate the basic
reproduction number was created in [2]. Different vector control measures for VL elimination
were considered in [54]. A multi-state Markov model of VL was developed in [55]; an individual-
based, stochastic, ~~stage-structured-compartmental~~ model of a temperature-driven sand fly
population was developed in [56] who also simulated the effects of the use of drugs administered
to cattle on the vector control. [57] developed methods to analyze longitudinal spatial incidence
data on VL and PKDL. A set of three age-structured model variants based on [58], each
with individuals from a different disease stage being the main contributors to transmission:
asymptomatic individuals, previously immune individuals in whom infection has reactivated,
and individuals with PKDL was created in [59]. The cost effectiveness of different drug treatments
was studied in [60] and [61]. ~~On top of the above mathematical models of anthropomorphic VL~~
~~relevant to India, there are many models of zoonotic VL that incorporate animal reservoirs as~~
~~well, see for example [3,4,62,63].-~~

[revised manuscript text omitted]

 1, 2, 3 and 4. The parameter values are estimated in the section Model calibration Section 4 based
 on empirical evidence and data from the literature.

The compartments are denoted S (susceptible), I (infected and potentially infectious), R
 (recovering), and E (exposed). Subscript F denotes sand flies. For the human compartments we
 use the subscript P for PCR-positive, D for DAT-positive, C for cellular immunity (LST-positive),
 S for symptomatic, T for treated, U for untreated and L for PKDL. Greek letters stand for the
 rates. Lower case roman letters stand for probabilities/proportions of the populations.

(c) Differences between our model and the original model in [58]

Aside from slight differences in the notation, we made the following changes and additions to the
 original model from [58].

Figure 1. Scheme of the ODE model for VL transmission—Based; based on [58]. Majority of the human population is in the asymptomatic VL cycle: individuals are born as susceptible (S) and if bitten by an infected sand fly (I_F) — signified by the red dotted arrow — they progress through early asymptomatic (I_P), late asymptomatic (I_D), recovering asymptomatic (R_D) and recovered (R_C). Recovered individuals lose immunity and eventually become susceptible again. Sand flies are born susceptible (S_F). If they bite an individual in any of the red or brown compartments — signified by the red curly brace — they become exposed (E_F) and eventually infectious (I_F). All human compartments are color-coded based on their PCR, DAT and LST tests as shown on the left; informally, the intensity of the color corresponds to the intensity of the infection which increases from left to right. The solid black arrows represent transitions between compartments.

6
rsos.royalsocietypublishing.org R. Soc. open sci. 0000000

- (i) We explicitly added a parameter p signifying the level of protection against sand fly's bites by using ITNs.
- (ii) We added a parameter e for the entomological efficacy of ITNs against the vector.
- (iii) We separated PKDL cases (I_{HL} in [58]) into treated (I_{TL}) and untreated (I_{UL}) cases to better account for the different duration of the stages and infectivity of the cases.
- (iv) We assume that the human population size is constant with a birth rate Λ (as opposed to $\alpha_H N_H$ considered in [58]). Doing so allows the human population. This change makes the ODE system less sensitive to changes in the birth rate and death rate. In the original model of [58], even a small change in α_H , μ_H , or μ_K would destabilize the system and result in either exponential growth or exponential extinction of the population. In our model, a (reasonable) change in Λ , μ_H , or μ_K will cause the system to converge to a constant and we can thus an equilibrium with a potentially different population size, but the population will not go extinct nor grow above any bound. We can also solve for the equilibria of the dynamics.
- (v) We keep the exponential growth of sand flies, denoted $\alpha_F N_F$ in [58], but we stipulate that the growth rate $\alpha_F = \mu_F$ in order to keep the population size constant.
- (vi) We consider the death rate of treated individuals to be $\mu_1 = \mu + d_{T1}\tau_1$ and $\mu_2 = \mu + d_{T2}\tau_2$, instead of $\mu_1 = \mu_2 = \mu + \mu_K + f_T\tau_1$. Specifically, we do not include the death rate of untreated individuals (μ_K) and we consider the rates different in a different lines of treatment.

Table 1. Parameter values ~~—~~ sand flies. Times are in days.

Symbol	Description	Value	Range	Reference
n_F n_F	Number of sand flies per human	3	2.1, 3.5[2.1, 3.5]	[8, Table S15]
β^{-1}	Duration of feeding cycle	4	[2, 8]	[90,91]
σ_F^{-1}	Sojourn time in latent stage E_F E_F	5	[3, 7]	[92]
ϱ_F^{-1}				
μ_F^{-1}	Life expectancy Expected lifespan of sand flies	14	[6, 31]	[93]
μ_F^{-1}				
i_F	Probability that a bite by an infected sand fly infects a susceptible human	1	0.1 [0, 1]	[58]

(d) Numerical implementation

We coded the model in Matlab (version 2020a with optimization toolbox) and made the code available in the supplemental information. Following the best practices highlighted in [111], we included several tests to ensure correctness of the code. Specifically, we checked that analytical and graphical solutions (which were coded independently) yield the same results. We closely tracked the code execution to make sure the code runs as expected and values are passed correctly from function to function. Throughout the work on this manuscript, the formal analysis and numerical code were developed side by side and checked against each other. Independent cross-checks on a graphical calculator were also performed.

(e) Model limitations and possible extensions

There are always many ways how to make any model more realistic. We list several possible extensions of our model here.

Infectious sand flies seem to feed more often [112,113]. One may thus need to consider two different bite rates. One for susceptible sand flies (this will play a role in λ_F , i.e., humans infecting sand flies) and another one for sand flies infecting humans (this will play a role in λ , i.e., sand flies infecting humans).

3. Analysis

We did not consider any ITN related mortality of the sand flies. This is in agreement with [89], although slight reduction of sand fly prevalence was observed by [47]. We also do not consider increased mortality of infected sand flies [113,114]. To incorporate any of these two, one would have to consider $\mu_F = \mu_F(p)$ for an appropriate increasing function $\mu_F(p)$. Furthermore, the total mortality of the infected sand flies would be $\mu_F(p) + \mu_I(p)$ where $\mu_I(p)$ represents the additional mortality caused by the infection and the fact that infected flies bites faster, i.e. could suffer from an increased ITN related mortality as well. To make these additions work, we would have to assume the birth rate to be Λ_F ; otherwise the sand fly population would not have nontrivial finite equilibrium.

[revised manuscript text omitted]

the fly got inside anyway, or the bite occurred outside). Further changes would have to be made in (2.2) in the calculation of the force of infection for the sand flies. We would have to properly account for two distinct scenarios depending whether the fly bites an infected individual who does not use ITNs, or whether it bites an individual despite them using ITNs. The first scenario accounts for $\frac{1-p}{1-pe}$ cases, the second one for the remaining $\frac{p(1-e)}{1-pe}$ cases. Moreover, in the second scenario, we would need to add one more factor $(1 - e)$ to stand for a second "failure" of the ITN.

The equilibria of the dynamics — are found by solving the following system of algebraic equations. Recent results of [57] suggest that incorporating spatial structure into the model would greatly increase its realism.

Table 4. Auxiliary notation

Symbol	Description	Equation
λ_F λ_F	Force of infection (humans infecting vectors)	(2.2)
λ	Force of infection (vectors infecting humans)	(2.1)
μ_S^{-1}	Life expectancy in I_S	$(\mu + \mu_K)^{-1}$
μ_1^{-1}	Life expectancy in I_{T1}	$(\mu + d_{T1}\tau_1)^{-1}$
μ_2^{-1}	Life expectancy in I_{T2}	$(\mu + d_{T2}\tau_2)^{-1}$
C_{VL}	Expected cost of getting sick	(4.4)
T_{Comp}	Expected time an individual spends in a compartment $Comp \in \{I_P, \dots, R_C\}$ given it started in compartment I_P	(a)-(a)
T_{Cycle}	Expected time it takes an individual to become susceptible again (given it started in I_P and conditional on surviving)	(A 43)
T_i	Average time an individual spends as infectious to sand fly (weighted by the infectivity)	(3.3)

$$0 = \Lambda + \rho_C R_C - (\mu + \lambda) S$$

$$0 = \lambda S - (\mu + \gamma_P) I_P$$

$$0 = \gamma_P I_P - (\mu + \gamma_D) I_D$$

$$0 = f_D \gamma_D I_D - (\mu + \rho_D) R_D$$

$$0 = f_S \gamma_D I_D - (\mu_S + \gamma_S) I_S$$

$$0 = \gamma_S I_S - (\mu_1 + \tau_1) I_{T1}$$

$$0 = s_1 \tau_1 I_{T1} - (\mu_2 + \tau_2) I_{T2}$$

$$0 = s_2 \tau_1 I_{T1} + s_4 \tau_2 I_{T2} + f_L \gamma_D I_D - (\mu + \rho_L) R_L$$

$$0 = p_{TL} \rho_L R_L - (\mu + \tau_3) I_{TL}$$

$$0 = (1 - p_{TL}) \rho_L R_L - (\mu + \gamma_L) I_{UL}$$

$$0 = s_3 \tau_1 I_{T1} + s_5 \tau_2 I_{T2} + \tau_3 I_{TL} + \gamma_L I_{UL} - (\mu + \rho_T) R_T$$

$$0 = \rho_D R_D + \rho_T R_T - (\mu + \rho_C) R_C$$

$$0 = \mu_F N_F - (\mu_F + \lambda_F) S_F$$

$$0 = \lambda_F S_F - (\mu_F + \sigma_F) E_F$$

$$0 = \sigma_F E_F - \mu_F I_F$$

There are two equilibria; the disease-free equilibrium denoted by-

$$E^0 = (S^0, I_P^0, I_D^0, R_D^0, R_C^0, I_S^0, I_{T1}^0, I_{T2}^0, I_{TL}^0, I_{UL}^0, R_T^0, R_L^0, S_F^0, E_F^0, I_F^0)$$

and the endemic equilibrium given by-

$$E^* = (S^*, I_P^*, I_D^*, R_D^*, I_S^*, I_{T1}^*, I_{T2}^*, R_L^*, I_{TL}^*, I_{UL}^*, R_T^*, R_C^*, S_F^*, E_F^*, I_F^*).$$

Also, there are several limitations of the game-theoretical framework, particularly the assumption
 that all individuals are provided with the same information and use it in the same (and rational)
 way to assess costs and risks [68]. First, ITN ownership is associated with wealth and the cost
 of ITNs is often prohibitive for the poor [42]. Moreover, VL susceptibility is bound to reduce
 with wealth, as malnutrition and certain interior wall types are VL risk factors [115,116]. One
 would have to account for this by considering non-homogeneous population. Second, VL has
 a long incubation period during which individuals are asymptomatic yet may be contagious.
 This makes the risk assessment prone to errors as people may think that there is no or very
 low risk of contracting VL (and thus stop using ITNs) while in reality someone in their vicinity
 may already be asymptomatic. The risk assessment is further complicated by the fact that even
 professionals often misdiagnose acute symptomatic VL cases [117] and by the social and cultural
 stigma associated with VL [22].

The system – will be solved as follows, corresponding to starting at compartment S and going
 downstream. By -, and since $\lambda = (1 - p)\beta i_F I_F$, Finally, unlike vaccines that offer a long lasting
 protection after just one decision, ITNs have to be used repeatedly. In principle, individuals
 decide every night whether or not to use ITNs. This dynamic decision process is not captured
 in our model. However, we hypothesize that coupled with long incubation periods of VL and
 especially PKDL, the seemingly optimal behavior of using ITNs only if acute VL and PKDL cases
 reach above certain threshold, could lead to periodic spikes and drops of VL and PKDL cases.
 This is illustrated in Figure ?? . We can see that soon after the ITN coverage is relaxed, the early
 asymptomatic, I_P , and symptomatic VL cases are on the rise, followed by a rise of asymptomatic
 recovered cases, R_D , and only a very small and slow rise of late asymptomatic cases, I_D . Once
 the population starts to use more ITNs again, I_P declines almost immediately, soon followed by a
 decline of I_D and symptomatic VL cases and a trailing decline of asymptomatic recovered cases.
 At the same time, even as ITN coverage is on the rise, there is a slow but steady increase of PKDL
 cases that act as a reservoir of infections and can restart the cycle again. Overall, the importance
 of understanding of the consequences of dynamical decisions warrant future studies.

$$I_P^* = \frac{\lambda}{\mu + \gamma_P} S^* = (1 - p)\beta i_F \frac{1}{\mu + \gamma_P} S^* I_F^*.$$

Similarly, by -, for every compartment $Comp \in \{I_P, I_D, R_D, R_C, I_S, I_{T1}, I_{T2}, I_{TL}, I_{UL}, R_T, R_C\}$
 we have-

**Disease Free Equilibrium**

We assume that $I_P^0 = 0$. By -, $I_D^0 = 0$. By -, $R_D^0 = 0$. By -, $R_C^0 = 0$. By -, $I_S^0 = 0$. By -, $I_{T1}^0 = 0$. By -, $I_{T2}^0 = 0$. By -, $R_L^0 = 0$.
 By -, $I_{TL}^0 = 0$.

**3. Analysis**

The detailed analysis is shown in Appendix A. Here we present only the summary. There are two
 possible equilibria; (1) the disease-free equilibrium with all humans and sand flies susceptible in
 $S^0 = \frac{\Lambda}{\mu}$ and $S_F^0 = \nu_F \frac{\Lambda}{\mu}$, by -, $I_{UL}^0 = 0$. By -, $R_T^0 = 0$. By -, $R_C^0 = 0$. From -, $S^0 = \frac{\Lambda}{\mu}$. By -, $\lambda_F = 0$. Thus,
 by -, $S_F^0 = N_F = n_F \frac{\Lambda}{\mu}$. Thus, $E_F^0 = 0$ and $I_F^0 = 0$. So our disease free equilibrium is given by-

and (2) the endemic equilibrium given by

$$E_-^{0*} = \frac{\Lambda}{\mu} (S^*, I_P^*, I_D^*, I_{D'}^*, I_S^*, I_{T1}^*, I_{T2}^*, I_{R_L}^*, I_{T_L}^*, I_{U_L}^*, I_{R_T}^*, I_{R_C}^*, n_F \frac{\Lambda}{\mu} S_F^*, E_F^*, I_{U_F}^*) \quad (3.1)$$

The explicit formulas are given in the Appendix A(c).

Basic reproduction number

When the ITNs usage is p , the effective reproduction number, $\mathcal{R}_e(p)$, is the average number of new infections caused by a single infected individual [119,120]. The formula for $\mathcal{R}_e = \mathcal{R}_e(p)$ can be derived as follows using the next-generation matrix method [121] but also directly as shown below.

Assume that there is a single infected person in compartment I_P . As derived in above, that person spends an expected time $T_{Comp} = T_{Comp}$ in each of the compartments $Comp \in \{I_P, I_D, I_S, I_{T1}, I_{T2}, I_{T_L}, I_{U_L}\}$. During that time, they $Comp \in \{I_P, I_D, I_S, I_{T1}, I_{T2}, I_{T_L}, I_{U_L}\}$ the formulas for T_{Comp} are given in ((a))-(a). During the time T_{Comp} , the infectious individuals expose sand flies at rate $\beta i_X N_F = \beta i_X n_F \frac{\Lambda}{\mu} \beta i_X N_F = \beta i_X n_F \frac{\Lambda}{\mu}$. Each of the exposed sand flies becomes infectious with the probability $\frac{\sigma_F}{(\mu_F + \sigma_F)} \frac{\sigma_F}{(\mu_F + \sigma_F)}$. Each sand fly stays infectious for the time $\frac{1}{\mu_F} = \frac{1}{\mu_F}$ and during that time it infects humans at rate $(1-p)\beta i_F N = (1-p)\beta i_F \frac{\Lambda}{\mu}$. Putting it all together yields

$$\mathcal{R}_e(p) = (1-p)\beta i_F n_{FF} \left(\frac{\Lambda}{\mu}\right)^2 \left(\frac{\sigma_F}{\mu_F + \sigma_F} \frac{\sigma_F}{\mu_F + \sigma_F} \right) \left(\frac{1}{\mu_F} \frac{1}{\mu_F} \right) T_1 \quad (3.2)$$

where

$$T_1 = i_P T_{I_P} + i_D T_{I_D} + i_S T_{I_S} + i_{T1} T_{I_{T1}} + i_{T2} T_{I_{T2}} + i_{T_L} T_{I_{T_L}} + i_{U_L} T_{I_{U_L}}. \quad (3.3)$$

Endemic Equilibrium

By looking at all human compartments and balancing what is coming in and out (i.e. adding all the equations), we get It follows from (A 55) that the endemic equilibrium exists only if $\mathcal{R}_e(p) > 1$. While we did not perform a formal stability analysis, the ODE system – although large – is quite standard and similar to one considered in [122]. Consequently, the disease-free equilibrium is globally asymptotically stable if $\mathcal{R}_e(p) < 1$, and unstable if $\mathcal{R}_e(p) > 1$. The endemic equilibrium is locally asymptotically stable if $\mathcal{R}_e(p) > 1$.

$$\Lambda = \mu S^* + \mu I_P^* + \mu I_D^* + \mu R_D^* + \mu_S I_S^* + \mu_1 I_{T1}^* + \mu_2 I_{T2}^* \cdots + \mu R_L^* + \mu I_{T_L}^* + \mu I_{U_L}^* + \mu R_T^* + \mu R_C^*.$$

By

$$\Lambda = \mu (S^* + (1-p)\beta i_F T_{Cycle} S^* I_F^*)$$

where

$$T_{Cycle} = T_{I_P} + T_{I_D} + T_{R_D} + \frac{\mu_S}{\mu} T_{I_S} + \frac{\mu_1}{\mu} T_{I_{T1}} + \frac{\mu_2}{\mu} T_{I_{T2}} \cdots + T_{R_L} + T_{I_{T_L}} + T_{I_{U_L}} + T_{R_T} + T_{R_C}.$$

Solving for S^* yields

$$S^* = \left(\frac{\Lambda}{\mu}\right) \left(\frac{1}{1 + (1-p)\beta i_F T_{Cycle} I_F^*} \right).$$

By

$$\begin{aligned} \lambda_F &= \beta(i_P I_P^* + i_D I_D^* + i_S I_S^* + i_{T1} I_{T1}^* + i_{T2} I_{T2}^* + i_{TL} I_{TL}^* + i_{UL} I_{UL}^*) \\ &= (1-p)\beta^2 i_F T_1 S^* I_F^* \\ &= (1-p)\beta^2 i_F T_1 \left(\frac{\Lambda}{\mu}\right) \frac{I_F^*}{1 + (1-p)\beta i_F T_{Cycle} I_F^*}. \end{aligned}$$

Thus,

$$\frac{I_F^*}{\lambda_F} = \frac{1 + (1-p)\beta i_F T_{Cycle} I_F^*}{(1-p)\beta^2 i_F T_1 \left(\frac{\Lambda}{\mu}\right)}.$$

From,

$$E_F^* = I_F^* \frac{\mu_F}{\sigma_F}.$$

From,

$$\begin{aligned} S_F^* &= \frac{\mu_F + \sigma_F}{\lambda_F} E_F^* = \frac{\mu_F + \sigma_F}{\lambda_F} \cdot \frac{\mu_F}{\sigma_F} \cdot I_F^* \\ &= \left(\frac{\mu_F + \sigma_F}{\sigma_F}\right) \cdot \mu_F \cdot \left(\frac{1 + (1-p)\beta i_F T_{Cycle} I_F^*}{(1-p)\beta^2 i_F T_1 \left(\frac{\Lambda}{\mu}\right)}\right) \\ &= N_F^* \frac{1 + (1-p)\beta i_F T_{Cycle} I_F^*}{\mathcal{R}_e}. \end{aligned}$$

Thus,

$$\begin{aligned} N_F^* &= S_F^* + E_F^* + I_F^* \\ &= N_F^* \left(\frac{1 + (1-p)\beta i_F T_{Cycle} I_F^*}{\mathcal{R}_e}\right) + \frac{\mu_F}{\sigma_F} I_F^* + I_F^*. \end{aligned}$$

and

$$I_F^* = \frac{N_F^* \left(1 - \frac{1}{\mathcal{R}_e}\right)}{(1-p)\beta i_F T_{Cycle} \frac{N_F^*}{\mathcal{R}_e} + \frac{\mu_F}{\sigma_F} + 1}.$$

Once I_F^* is calculated from, we can calculate E_F^* , S_F^* and S^* as

$$\begin{aligned} E_F^* &= I_F^* \frac{\mu_F}{\sigma_F} \\ S_F^* &= \frac{\mu_F + \sigma_F}{\lambda_F} E_F^* \\ S^* &= \left(\frac{\Lambda}{\mu} \right) \left(\frac{1}{1 + (1-p)\beta i_F T_{\text{Cycle}} I_F^*} \right). \end{aligned}$$

Finally, using above, we can also find all the human compartments as below-

$$I_P^* = (1-p)\beta i_F T_{I_P} S^* I_F^*$$

$$I_D^* = (1-p)\beta i_F T_{I_D} S^* I_F^*$$

$$R_D^* = (1-p)\beta i_F T_{R_D} S^* I_F^*$$

$$I_S^* = (1-p)\beta i_F T_{I_S} S^* I_F^*$$

$$I_{T1}^* = (1-p)\beta i_F T_{I_{T1}} S^* I_F^*$$

$$I_{T2}^* = (1-p)\beta i_F T_{I_{T2}} S^* I_F^*$$

$$R_L^* = (1-p)\beta i_F T_{R_L} S^* I_F^*$$

$$I_{TL}^* = (1-p)\beta i_F T_{I_{TL}} S^* I_F^*$$

$$I_{UL}^* = (1-p)\beta i_F T_{I_{UL}} S^* I_F^*$$

$$R_T^* = (1-p)\beta i_F T_{R_T} S^* I_F^*$$

$$R_C^* = (1-p)\beta i_F T_{R_C} S^* I_F^*$$

4. Model calibration

(a) Sand fly parameters

The number of sand flies per human is $n_F = 3$ with range 2.1 – 3.5 [8, Table S15]. This is in line with the India KalaNet site: 938 sand flies (94.2% of which were *P. argentipes*) from 325 households [47]. We note that this is different than the estimated value $n_F = 5.27$ with the range of 3.45 – 9.90 used by [58].

The duration of the feeding cycle is $\beta^{-1} = 4$ days as in [58] with the range two to eight days [90]. Sand flies normally take one blood meal per oviposition cycle [91,123,124] and the cycle usually takes four days [90] although infectious sand flies seem to feed more often [112,113]. We note that other models such as [3,63,125] used $\beta^{-1} = 10$ days based on [91] who estimated β^{-1} between six and thirty days based on experimental results of [123] who seemed to let the flies *P. ariasi* oviposit for six or more days. While the literature varies on the actual length, it all agrees on the fact that the flies bite only once per oviposition cycle.

The sojourn time in the latent stage E_F is $\sigma_F^{-1} = 5$ days, with the range three to seven days [92].

The life expectancy of sand flies is $\mu_F^{-1} = 14$ days with the range six to thirty one days [93].

Figure 3. Estimates-Left: data from [98] fitted to $y = 28e^{-0.8845t}$ which yields $\rho_D^{-1} = 1.13$ months. Center: data from [66] fitted to $y = 1.0008e^{-0.36176t}$ which yields an estimate for γ_D (left) ρ_C^{-1} as 0.36176^{-1} years, p_C (center) and γ_L (right) i.e., 33 months. Right: data from [24] fitted to $y = 1.0441e^{-0.
[revised manuscript text omitted]
 six years, the cohort of $N = 22699$ individuals should experience $72 \cdot \left(\frac{1}{33}\right) \cdot 0.35 \cdot 22699 \approx 17334$ late asymptomatic cases. Since 813 of those developed KA, we get $f_S = 813/17334 = 0.0469$. Similarly, [24] surveyed 24,814 individuals for KA and PKDL in the past nine years. Of those 1,002 reported KA and 17 had PKDL with no history of KA. This gives an estimate

$$f_S = \frac{1002}{9 \cdot 12 \cdot 24814 \cdot \frac{1}{33} \cdot 0.35} = 0.0353. \quad (4.1)$$

Because [23] and [24] used a much larger sample size than [100] and [100] also notes that incidences were larger in India than in Nepal, we will adopt $f_S = 0.035$ and consider a range from 0.01 to 0.15.

We estimate that the fraction of asymptomatic patients who become dormant to be $f_L = 0.00055$ with the range [0.0001, 0.001]. This will be done in a similar fashion as the estimate for f_S based on data from [23] and [24]. A total of $N = 22699$ individuals were surveyed by [23] for KA and PDKL in the past six 
[revised manuscript text omitted]

 510 within three weeks. In fact [24] followed 30 PKDL cases treated with amphotericin B, and while
 511 27 reported an improvement in four months, only one case was completely cured. We do not have
 512 additional data but it seems that even in the case of this treatment, the duration of PKDL is six
 513 months.

514 The duration of untreated PKDL, γ_L^{-1} , will be estimated as follows. In [24], the authors
 515 followed 98 PKDL patients that never received treatment and provides estimated resolution
 516 rates: 8% within one year of onset, 34% within two years, and 67% within five years. Fitting the
 517 exponential decay to this data by matlabMatlab, see Figure 3, yields $\gamma_L^{-1} = 55.5$ months with the
 518 range of 32 months to 16 years.

519 The fraction of PKDL patients that receive treatment is estimated as $p_{TL} = 0.5$. This estimate is
 520 based on [24] who reports that out of 185 PKDL cases, 98 did not seek the treatment.

521 (e) The cost parameters

522 Once infected, all individuals have to go through I_P and I_D . Those stages are asymptomatic
 523 and consequently with no associated costs. The costs appear only if and when an individual
 524 experiences symptoms (in I_S), is being treated (I_{T1} , I_{T2} , and I_{TL}) or is recovering after the
 525 treatment (in R_T and R_L). The costs are denoted by C_{I_S} , $C_{I_{T1}}$, $C_{I_{T2}}$, $C_{I_{TL}}$, C_{R_T} , and C_{R_L} and
 526 the actual values are estimated below.

The Indian government provides a free care to VL patients; however, hoping for a quick cure of seemingly minor illness, patients prefer to access private providers which contributes to high out-of-pocket expenditures [117]. VL is often misdiagnosed in private hospitals but patients keep accessing care in private sector until no money is left [117].

The cost of being in compartment I_S is estimated as $C_{I_S} = 19$ USD [107], and we will assume the range [11, 26]. The average monthly income in Bihar, India in 2010 was 38 USD per month [107]. It takes about a month to get a diagnosis [99]. During that time, the patient loses 2.14 weeks (about half a month) of work, i.e. the cost is about \$19.

The cost of getting the first line of treatment is $C_{I_{T1}} = 146$ USD [107], and we will assume the range [120, 170]. This includes 127 USD of direct medical costs for one month of treatment [107], and the indirect cost of lost work. For every month of illness, half a month of work is lost (19 USD loss) [107]. This is in line with \$100 direct medical costs used by [108].

The cost of getting the second line of treatment is $C_{I_{T2}} = C_{I_{T1}} = 146$ USD [107]. Both lines of treatment last approximately one month, so the cost of the individual second-treatment will be about the same.

The cost of getting treatment for PKDL is assumed to be $C_{I_{TL}} = 349.00$ USD [109]. We will assume the range [290, 410]. The direct cost of SSG treatment for PKDL costs 179 USD [109]. The treatment lasts about six months and so the loss of productivity during this illness is 170 USD.

The cost of recovering from treatment without a dormant infection is $C_{R_T} = 57$ USD [107], and we will assume the range [60, 110]. During recovery, a patient can only work 3.21 weeks/month on average, instead of the normal 4.29, i.e. losing 25% of the wage per month [107]. Since the average monthly wage is 38 USD and R_T lasts approximately $\rho_T^{-1} = 6$ months, the total loss is about $57 = 0.25 \cdot 38 \cdot 6$.

The cost of recovering from treatment, but with a dormant infection is $C_{R_L} = C_{R_T} = 57$ USD [107]. We assume that even though individuals in this stage do have a dormant infection, the time and the cost needed to recover from treatment will be about the same.

The overall cost of VL infection is thus given by

$$C_{VL} = P(I_P \rightarrow I_S)C_{I_S} + P(I_P \rightarrow I_{T1})C_{I_{T1}} + P(I_P \rightarrow I_{T2})C_{I_{T2}} + P(I_P \rightarrow I_{TL})C_{I_{TL}} + P(I_P \rightarrow R_T|I_{T1} \text{ or } I_{T2} \text{ or } I_{TL})C_{R_T} + P(I_P \rightarrow R_L|I_{T1} \text{ or } I_{T2})C_{R_L} \quad (4.4)$$

where the probabilities $P(I_P \rightarrow C)$ (and $P(I_P \rightarrow C|C')$) of getting to a compartment C (through a compartment C') when currently at a compartment I_P are given by

$$P(I_P \rightarrow I_S) = T_{ID} f_S \gamma_D \quad (4.5)$$

$$P(I_P \rightarrow I_{T1}) = T_{IS} \gamma_S \quad (4.6)$$

$$P(I_P \rightarrow I_{T2}) = T_{I_{T1}} s_1 \tau_1 \quad (4.7)$$

$$P(I_P \rightarrow I_{TL}) = T_{R_L} p_{TL} \rho_L \quad (4.8)$$

$$P(I_P \rightarrow R_T|I_{T1} \text{ or } I_{T2} \text{ or } I_{TL}) = T_{I_{T1}} s_3 \tau_1 + T_{I_{T2}} s_5 \tau_2 + T_{I_{TL}} \tau_3 \quad (4.9)$$

$$P(I_P \rightarrow R_L|I_{T1} \text{ or } I_{T2}) = T_{I_{T1}} s_2 \tau_1 + T_{I_{T2}} s_4 \tau_2. \quad (4.10)$$

The cost of ITN is $C_{ITN} = 3.62$ USD [41]. If the households opt to use the impregnated durable wall lining, which seems even more effective than ITNs, the cost grows significantly to about 13.57 USD [41]. At the same time, one could argue that the wall lining protects the whole household, while the ITN protects only those that sleep under it.

5. Results

(a) Estimating VL transmission probabilities

Unlike [58], we will assume that individuals receiving treatment cannot infect the sand flies, i.e. $i_{T1} = 0, i_{T2} = 0$, and $i_{TL} = 0$. The treated individuals are not in their regular environments but rather at the treatment facility. Consequently, they are likely better protected from sand fly bites. Moreover, even if they are bitten and infect the fly, they are too far from their home for this event to meaningfully contribute to VL transmission in their home/village. This assumption is in agreement with [26] who reported that after receiving treatment, the VL patients are likely less infectious to the sand flies resulting in less transmission to family members during their 18-month follow-up.

The remaining transmission probabilities, specifically

(a) The role of asymptomatic individuals in VL transmissions

The transmission probabilities i_P, i_D for asymptomatic individuals, i_S and i_{UL} for untreated symptomatic individuals and i_F for the transmission from the sand fly to humans were estimated using the maximum likelihood method by fitting built-in Matlab function to fit the model to KalaNet trial data [58, Table 1] as described in Appendix B.

The probability that a susceptible human becomes infected when by an infected sand fly bites them, i_F , was estimated as very close to 1, $i_F \approx 1$; see Figure 4. This high transmission probability is consistent with the fact that the parasite can modify the sand fly's feeding apparatus so that the fly feeds more persistently and releases more parasites [112,132].

The estimates for i_{UL} and i_S are more or less linear in i_P and i_D depend on p , the proportion of population that used ITNs during the KalaNet trial. The linear regression on the maximum likelihood estimates come out as $i_{UL} = 0.253 + 0.254p$ ITN coverage during KalaNet trial as well as the transmission probabilities from untreated symptomatic individuals, i_S and i_{UL} . However, Figure 4 demonstrates that, for reasonable values of p , i_S and $i_S = 0.357 + 0.156p$, i.e. $i_{UL} \approx 0.4325$ and $i_S \approx 0.475$ when $p \approx 0.7$.

The i_{UL} , the estimates for i_P and i_D are quite sensitive on the p . i_{UL} are fairly stable.

For small $p \approx 0$, we get $i_D \approx 0.024$, $i_D \approx 0.02$ and $i_P \approx 0.004$ which is consistent with [58]. However, for $p \approx 0.7$, $i_D \approx 0.1$ and $i_P \approx 0.024$ we get $i_D \approx 0.05$ and $i_P \approx 0.01$. If p grows even more, the values of i_D and i_P grow rapidly as well. Note that the estimates for i_P and i_D are indeed increasing in p – to achieve the same disease prevalence as measured in KalaNet trial when the population uses higher level of protection p , the disease must be more transmittable.

At the same time, as the probability of transmission from an untreated symptomatic individuals increases, both i_P and i_D decrease, i.e. the asymptomatic individuals become less important if the symptomatic individuals transmit the parasites very likely. Nevertheless, even if symptomatic individuals transmit the disease with 100% probability, the role of late asymptomatic individuals is still not negligible.

When i_S and i_{UL} are around 0.1 as recently measured by [101], we get $i_P = 0.01$ and $i_D = 0.05$. This means that late asymptomatic individuals (PCR positive, DAT positive and LST negative) are roughly 50% as important to VL transmission as untreated symptomatic VL and PKDL cases.

(b) Model validation

The model is validated using KalaNet trial data [58, Table 1]. Our model gives the following prevalences (the KalaNet data are in the parenthesis): 0.7586 – 0.7599 (0.76) for $S + R_C$, 0.0984 – 0.0979 (0.1) for I_P , 0.0222 – 0.0221 (0.02) for i_D , 0.1179 – 0.1173 (0.12) for R_D , and 0.0051 – 0.0108 (0.005) for I_F – I_F .

To validate the model on another data set, one would have to potentially update the parameter values to properly reflect the time and location of the experiment where the data came from. Then, we can use formulas for endemic equilibrium from the Appendix A(c) to obtain distribution of population across different model compartments.

We note that while not impossible, it is hard to make KalaNet trial data, our model and the model from [58] consistent without asymptomatic transmissions. Specifically, if the PCR positive,

Figure 4. Maximum likelihood estimates for i_P (left), i_D (leftcenter) and i_S, i_{UL} and i_F (right). While the numerical estimates for i_S and i_{UL} fluctuates. Top row: $i_S = i_{UL} = 0.1$ while the ITN coverage p varies. The transmission probabilities increase (dotted lines) as the protection level increases, they follow a linear trend (solid lines) the disease must be more transmittable to have the same disease prevalence. Bottom row: $p = 0.7$ while $i_S = i_{UL}$ varies. As the probability of symptomatic transmission increases, the role of asymptomatic transmissions decreases while the role of sand flies remains constant.

DAT positive, and LST negative asymptomatic individuals in compartment I_D cannot infect
 the sand flies, the model can still predict prevalences of 0.0952 for I_P , 0.0215 for I_D , 0.114
 for R_D , and 0.009 for I_F which is in a reasonable agreement with KalaNet data; however, all
 of this can be achieved only under very unrealistic assumption that symptomatic individuals
 transmit the parasite 100% of the time. In fact, with much more realistic values of $i_S \approx i_{UL} \approx 0.1$ as
 estimated in Section 4 from [101], the population would be in disease-free equilibrium. Without

Figure 5. Prevalence of I_P (left), I_D (center) and I_F (right) as a function of transmission probability from the untreated symptomatic individuals if the asymptomatic individuals do not transmit parasites at all. The dotted lines corresponds to data from KalaNet trial.

asymptomatic transmissions, one would need $i_S > 0.18$ and $i_{UL} > 0.18$ for VL to become endemic.
This is illustrated in Figure 5.

(c) Minimal ITN use coverage needed for VL elimination

To find the ITN usage level necessary to achieve the complete elimination of VL, we need to find
the smallest $p_{HI} \in [0, 1]$ such that when $p \geq p_{HI}$, then $\mathcal{R}_e \leq 1$. It follows from (3.2) that
$\mathcal{R}_e = (1-p)\mathcal{R}_e(0)$ where $\mathcal{R}_e(p) = (1-p)\mathcal{R}_e(0)$ where

$$623 \mathcal{R}_e(0) = \beta^2 i_F n_{FF} \left(\frac{\Lambda}{\mu} \right)^2 \left(\frac{\sigma_F}{\mu_F + \sigma_F} \frac{\sigma_F}{\mu_F + \sigma_F} \right) \left(\frac{1}{\mu_F} \frac{1}{\mu_F} \right) T_i \quad (5.1)$$

is the basic effective reproduction number when nobody is using the protection. Consequently,

$$625 p_{HI} = \begin{cases} 1 - \frac{1}{\mathcal{R}_e(0)}, & \text{if } \mathcal{R}_e(0) > 1, \\ 0, & \text{otherwise.} \end{cases} \quad (5.2)$$

For the parameter values as specified in Tables 1, 2, 3, $p_{HI} = 0.95963$, i.e., one needs just under
96% ITN coverage for a complete VL elimination.

(d) Optimal voluntary use of ITNs

In this section, we will find the optimal proportion of the use of ITNs. We are looking for Nash
equilibrium - a proportion that, when adopted by the population, no individual has an incentive
to deviate from their choice. To find a Nash equilibrium value of p , we have to solve

$$632 C_{ITN} = \frac{\beta i_F I_F^*}{\mu + \beta i_F I_F^*} \frac{\beta i_F I_F^*}{\mu + \beta i_F I_F^*} C_{VL} \quad (5.3)$$

where C_{ITN} is the cost protection, $\frac{\beta i_F I_F^*}{\mu + \beta i_F I_F^*} - \frac{\beta i_F I_F^*}{\mu + \beta i_F I_F^*}$ is the probability of getting infected by an
 infected sand fly, and C_{VL} is the expected cost one pays after such an event. Note that (5.3) is an
 equation for p because $I_F^* - I_F^*$ is a function of p . Figure 6 illustrates a graphical solution of (5.3).

The expected cost of getting sick if not using ITNs when the population usage is p (solid line)
 versus the cost of ITNs (dotted line). The optimal use (a Nash equilibrium) occurs when the lines
 intersect. VL is eliminated when the solid line reaches the x -axis. Recall that, by After algebraic
 manipulations shown in the Appendix C,

$$I_F^* = \frac{N_F^* \left(1 - \frac{1}{\mathcal{R}_e(0)(1-p)}\right)}{(1-p)\beta i_F T_{\text{Cycle}} \frac{N_F^*}{\mathcal{R}_e(0)(1-p)} + \frac{\mu_F}{\sigma_F} + 1}$$

and in particular I_F^* is decreasing in p and thus $\frac{\beta i_F I_F^*}{\mu + \beta i_F I_F^*}$ is decreasing in p . The maximum value
 of I_F^* is thus

$$I_F^*(0) = \frac{N_F^* \left(1 - \frac{1}{\mathcal{R}_e(0)}\right)}{\beta i_F T_{\text{Cycle}} \frac{N_F^*}{\mathcal{R}_e(0)} + \frac{\mu_F}{\sigma_F} + 1}$$

attained for $p=0$ and consequently, has a solution only if

$$C_{ITN} \leq \frac{\beta i_F I_F^*(0)}{\mu + \beta i_F I_F^*(0)} C_{VL}.$$

It follows from that (when is true)

$$I_F^* = \frac{\left(\frac{C_{ITN}}{C_{VL}}\right) \mu}{\beta i_F \left(1 - \frac{C_{ITN}}{C_{VL}}\right)}.$$

Thus, after algebraic manipulations of , we get that p_{NE} is given by

$$p_{NE} = 1 - \frac{1}{\mathcal{R}_e(0) \left[1 - \frac{I_F^*}{N_F^*} \left(\beta i_F T_{\text{Cycle}} \frac{N_F^*}{\mathcal{R}_e(0)} + \frac{\mu_F}{\sigma_F} + 1\right)\right]} \frac{1}{\mathcal{R}_e(0) \left[1 - \frac{I_F^*}{N_F^*} \left(\beta i_F T_{\text{Cycle}} \frac{N_F^*}{\mathcal{R}_e(0)} + \frac{\mu_F}{\sigma_F} + 1\right)\right]} \tag{5.4}$$

where $I_F^* - I_F^*$ is given by (A 5).

It follows that $p_{NE} \approx p_{HI}$. For our parameter values, we have $p_{HI} = 0.96211, p_{NE} = 0.96206$ with
 $\mathcal{R}_e(p_{NE}) = 1.0017, p_{HI} = 0.95963, p_{NE} = 0.95956$ with $\mathcal{R}_e(p_{NE}) = 1.0018$. It means that the disease
 can be almost eliminated by the optimal voluntary use of ITNs alone (provided people get perfect
 information and broken ITNs are replenished). In fact, with the voluntary use of ITNs, the disease
 would become eliminated as a public health problem.

(e) Sensitivity analysis

The sensitivity of the outcomes (ITN use for disease elimination, p_{HI} , and optimal ITN use, p_{NE}) on
 different parameter values is displayed in Table 5 and shown in Figures 8 and 9. Since $p_{NE} \approx p_{HI}$,
 only the sensitivity of p_{HI} is shown.

It follows that p_{HI} (and p_{NE}) are not overly sensitive to any parameter. The sensitivity index is
 at most 0.08-0.0846 (for μ or Λ) or -0.08 -0.0849 (for β^{-1}). It is the second highest for $\mu_F^{-1} - \mu_F^{-1}$
 and closely followed by $n_F - n_F$ and i_F . Sensitivity to other parameters is between -0.011 and 0.02.

It follows that increasing the time between the bites and the reduction of the lifespan or the
 number of sand flies are the most promising control measure apart from using ITNs.

Figure 6. The expected cost of getting sick if not using ITNs when the population usage is p (solid line) versus the cost of ITNs (dotted line). The optimal use (a Nash equilibrium) occurs when the lines intersect. VL is eliminated when the solid line reaches the x -axis. The parameter values are as specified in Tables 1, 2, 3.

The sensitivity index SI_y of a variable y on a parameter x was calculated as $\left(\frac{x}{y}\right) \cdot \left(\frac{\partial y}{\partial x}\right)$, see for example [133]. The numbers were rounded to the three decimal places. Parameters are as specified in Tables 1, 2 and 3. The sensitivity index -0.5 means that a 1% increase of a parameter value x will result in the 0.5% decrease of the variable y .

Parameter n_F 0.040 -1.968 β^{-1} -0.079 3.829 σ_F^{-1} -0.010 0.525 μ_F^{-1} 0.049 -2.542 i_F 0.040 -1.991 -0.079 -3.875 μ^{-1} 0.080 -5.026 μ_K^{-1} 0.000 -0.079 γ_P^{-1} 0.017 -0.735 γ_D^{-1} 0.016 -0.773 γ_S^{-1} 0.002 -0.035 f_L 0.001 -0.063 f_S 0.006 -1.797 p_C^{-1} 0.000 0.644 p_D^{-1} 0.000 0.121 i_P 0.017 -0.708 i_D 0.016 -0.772 i_S 0.002 -0.114 i_{T1} 0.000 0.000 i_{T2} 0.000 0.000 i_L 0.000 0.000 i_{UL} 0.004 -0.211 τ_1^{-1} 0.000 0.002 τ_2^{-1} 0.000 0.000 τ_3^{-1} 0.000 0.000 γ_{UL}^{-1} 0.004 -0.196 s_1 0.000 -0.063 s_2 0.003 -0.231 s_4 0.000 -0.014 d_{T1} 0.000 0.051 d_{T2} 0.000 0.003 δ_L^{-1} 0.000 0.009 s_{TL} 0.004 0.106 C_{ITN} 0.000 1.687 C_{T1} 0.000 -0.908

[revised manuscript text omitted]

MP, Dasgupta RK. 2020 Kala-azar elimination in a highly-endemic district of Bihar, India:
A success story. *PLOS Neglected Tropical Diseases* **14**, e0008254.
- 9. Sundar S, Singh OP, Chakravarty J. 2018 Visceral leishmaniasis elimination targets in India,
strategies for preventing resurgence. *Expert review of anti-infective therapy* **16**, 805–812.
- 10. Cloots K, Uranw S, Ostyn B, Bhattarai NR, Le Rutte E, Khanal B, Picado A, Chappuis F, Hasker
E, Karki P. 2020 Impact of the visceral leishmaniasis elimination initiative on *Leishmania
donovani* transmission in Nepal: a 10-year repeat survey. *The Lancet Global Health* **8**, e237–e243.
- 11. NTD Modelling Consortium Visceral Leishmaniasis Group. 2019 Insights from mathematical
modelling and quantitative analysis on the proposed WHO 2030 targets for visceral
leishmaniasis on the Indian subcontinent. *Gates Open Research* **3**, 1651.
- 12. WHO. 2020 Ending the neglect to attain the sustainable development goals: a road map for
neglected tropical diseases 2021–2030. Technical report World Health Organization.
- 13. Singh OP, Singh B, Chakravarty J, Sundar S. 2016 Current challenges in treatment options for
visceral leishmaniasis in India: a public health perspective. *Infectious diseases of poverty* **5**, 1–15.
- 14. Dinesh D, Ranjan A, Palit A, Kishore K, Kar S. 2001 Seasonal and nocturnal landing/biting
behaviour of *Phlebotomus argentipes* (Diptera: Psychodidae). *Annals of Tropical Medicine &
Parasitology* **95**, 197–202.

15. Feliciangeli M. 2004 Natural breeding places of phlebotomine sandflies. *Medical and Veterinary*
*Entomology* **18**, 71–80.
- 16. Gawade S, Nanaware M, Gokhale R, Adhav P. 2012 Visceral leishmaniasis: A case report. *The*
*Australasian Medical Journal* **5**, 130.
- 17. Younis LG, Kroeger A, Joshi AB, Das ML, Omer M, Singh VK, Gurung CK, Banjara MR.
2020 Housing structure including the surrounding environment as a risk factor for visceral
leishmaniasis transmission in Nepal. *PLoS Neglected Tropical Diseases* **14**, e0008132.
- 18. Goddard J. 2013 *Physician's Guide to Arthropods of Medical Importance*. CRC Press: Taylor &
Francis Group.
- 19. Poché RM, Garlapati R, Elnaïem DEA, Perry D, Poché D. 2012 The role of Palmyra palm trees
(*Borassus flabellifer*) and sand fly distribution in northeastern India. *Journal of Vector Ecology* **37**,
148–153.
- 20. Piscopo TV, Azzopardi CM. 2007 Leishmaniasis. *U.S. National Library of Medicine*.
- 21. Das A, Karthick M, Dwivedi S, Banerjee I, Mahapatra T, Srikantiah S, Chaudhuri I. 2016
Epidemiologic correlates of mortality among symptomatic visceral leishmaniasis cases:
findings from situation assessment in high endemic foci in India. *PLoS Neglected Tropical*
*Diseases* **10**, e0005150.
- 22. Ranjan A, Sur D, Singh VP, Siddique NA, Manna B, Lal CS, Sinha PK, Kishore K, Bhattacharya
SK. 2005 Risk factors for Indian kala-azar. *The American Journal of Tropical Medicine and Hygiene*
**73**, 74–78.
- 23. Rahman KM, Islam S, Rahman MW, Kenah E, Galive CM, Zahid M, Maguire J, Rahman M,
Haque R, Luby SP, Bern C. 2010 Increasing Incidence of Post-Kala-Azar Dermal Leishmaniasis
in a Population-Based Study in Bangladesh. *Clinical Infectious Diseases* **50**, 73–76.
- 24. Islam S, Kenah E, Bhuiyan MAA, Rahman KM, Goodhew B, Ghalib CM, Zahid M, Ozaki
813 M, Rahman M, Haque R. 2013 Clinical and immunological aspects of post-kala-azar dermal
leishmaniasis in Bangladesh. *The American Journal of Tropical Medicine and Hygiene* **89**, 345–353.
- 25. Addy M, Nandy A. 1992 Ten years of kala-azar in west Bengal, Part I. Did post-kala-
azar dermal leishmaniasis initiate the outbreak in 24-Parganas?. *Bulletin of the World Health*
*Organization* **70**, 341–346.
- 26. Das VNR, Pandey RN, Siddiqui NA, Chapman LA, Kumar V, Pandey K, Matlashewski G,
Das P. 2016 Longitudinal study of transmission in households with visceral leishmaniasis,
asymptomatic infections and PKDL in highly endemic villages in Bihar, India. *PLoS Neglected*
*Tropical Diseases* **10**, e0005196.
- 27. Burza S, Mahajan R, Sanz MG, Sunyoto T, Kumar R, Mitra G, Lima MA. 2014a HIV and
visceral leishmaniasis coinfection in Bihar, India: an underrecognized and underdiagnosed
threat against elimination. *Clinical infectious diseases* **59**, 552–555.
- 28. Burza S, Mahajan R, Sinha PK, van Griensven J, Pandey K, Lima MA, Sanz MG, Sunyoto T,
Kumar S, Mitra G et al.. 2014b Visceral leishmaniasis and HIV co-infection in Bihar, India:
long-term effectiveness and treatment outcomes with liposomal amphotericin B (AmBisome).
*PLoS Negl Trop Dis* **8**, e3053.
- 29. Akuffo H, Costa C, van Griensven J, Burza S, Moreno J, Herrero M. 2018 New insights into
leishmaniasis in the immunosuppressed. *PLoS neglected tropical diseases* **12**, e0006375.
- 30. Wilson AL, Courtenay O, Kelly-Hope LA, Scott TW, Takken W, Torr SJ, Lindsay SW. 2020 The
importance of vector control for the control and elimination of vector-borne diseases. *PLoS*
*Neglected Tropical Diseases* **14**, e0007831.
- 31. Stockdale L, Newton R. 2013 A review of preventative methods against human leishmaniasis
infection. *PLoS Neglected Tropical Diseases* **7**, e2278.
- 32. Das VNR, Pandey RN, Pandey K, Singh V, Kumar V, Matlashewski G, Das P. 2014 Impact of
ASHA training on active case detection of visceral leishmaniasis in Bihar, India. *PLoS Negl*
*Trop Dis* **8**, e2774.
- 33. Osman M, Mistry A, Keding A, Gabe R, Cook E, Forrester S, Wiggins R, Di Marco S, Colloca
S, Siani L et al.. 2017 A third generation vaccine for human visceral leishmaniasis and post
kala azar dermal leishmaniasis: First-in-human trial of ChAd63-KH. *PLoS Neglected Tropical*
*Diseases* **11**, e0005527.

- 34. Moafi M, Rezvan H, Sherkat R, Taleban R. 2019 Leishmania vaccines entered in clinical trials:
A review of literature. *International journal of preventive medicine* **10**.
- 35. Le Rutte EA, Coffeng LE, Malvolti S, Kaye PM, de Vlas SJ. 2020 The potential impact of human
visceral leishmaniasis vaccines on population incidence. *PLoS Neglected Tropical Diseases* **14**,
e0008468.
- 36. Kumar V, Kesari S, Sinha N, Palit A, Ranjan A, Kishore K, Saran R, Kar S. 1995 Field trial of
an ecological approach for the control of *Phlebotomus argentipes* using mud & lime plaster.. *The*
*Indian Journal of Medical Research* **101**, 154.
- 37. Wasserberg G, Weeks EN, Logan JL, Agneessens J, Stewart SA, Dewhirst S. 2019 Efficacy of
the insect repellent IR3535 on the sand fly *Phlebotomus papatasi* in human volunteers. *Journal*
*of Vector Ecology* **44**.
- 38. Bern C, Joshi AB, Jha SN, Das ML, Hightower A, Thakur G, Bista MB. 2000 Factors associated
with visceral leishmaniasis in Nepal: bed-net use is strongly protective.. *The American Journal*
*of Tropical Medicine and Hygiene* **63**, 184–188.
- 39. Chappuis F, Sundar S, Hailu A, Ghalib H, Rijal S, Peeling RW, Alvar J, Boelaert M. 2007
Visceral leishmaniasis: what are the needs for diagnosis, treatment and control?. *Nature*
*Reviews Microbiology* **5**, 873–882.
- 40. Ostyn B, Vanlerberghe V, Picado A, Dinesh DS, Sundar S, Chappuis F, Rijal S, Dujardin JC,
Coosemans M, Boelaert M. 2008 Vector control by insecticide-treated nets in the fight against
visceral leishmaniasis in the Indian subcontinent, what is the evidence?. *Tropical Medicine*
*International Health* **13**, 1073–1085.
- 41. Mondal D, Das ML, Kumar V, Huda MM, Das P, Ghosh D, Priyanka J, Matlashewski G,
Kroeger A, Uphill-Brown A et al.. 2016 Efficacy, safety and cost of insecticide treated wall
lining, insecticide treated bed nets and indoor wall wash with lime for visceral leishmaniasis
vector control in the Indian sub-continent: a multi-country cluster randomized controlled
trial. *PLoS Neglected Tropical Diseases* **10**, e0004932.
- 42. Vanlerberghe V, Singh S, Paudel I, Ostyn B, Picado A, Sanchez A, Rijal S, Sundar S, Davies
C, Boelaert M. 2010 Determinants of bednet ownership and use in visceral leishmaniasis-
endemic areas of the Indian subcontinent. *Tropical Medicine & International Health* **15**, 60–67.
- 43. Kroeger A, Ordóñez-Gonzalez o, Behrend M, Alvarez G. 1999 Bednet impregnation for
Chagas disease control: a new perspective. *Tropical Medicine and International Health* **4**, 194–198.
- 44. Mishra RN, Singh S, Vanlerberghe V, Sundar S, Boelaert M, Lefevre P. 2010 Lay perceptions of
kala-azar, mosquitoes and bed nets in Bihar, India. *Tropical Medicine & International Health* **15**,
36–41.
- 45. Kroeger A, González M, Ordóñez-González J. 1999 Insecticide-treated materials for malaria
control in Latin America: to use or not to use?. *Transactions of the Royal Society of Tropical*
*Medicine and Hygiene* **93**, 565–570.
- 46. Picado A, Singh SP, Rijal S, Sundar S, Ostyn B, Chappuis F, Uranw S, Gidwani K, Khanal B, Rai
M et al.. 2010a Longlasting insecticidal nets for prevention of *Leishmania donovani* infection
in India and Nepal: paired cluster randomised trial. *BMJ* **341**.
- 47. Picado A, Das ML, Kumar V, Kesari S, Dinesh DS, Roy L, Rijal S, Das P, Rowland M, Sundar
S. 2010b Effect of village-wide use of long-lasting insecticidal nets on visceral Leishmaniasis
vectors in India and Nepal: a cluster randomized trial. *PLoS Neglected Tropical Diseases* **4**, e587.
- 48. Picado A, Ostyn B, Rijal S, Sundar S, Singh SP, Chappuis F, Das ML, Khanal B, Gidwani K,
Hasker E et al.. 2015 Long-lasting insecticidal nets to prevent visceral leishmaniasis in the
Indian subcontinent; methodological lessons learned from a cluster randomised controlled
trial. *PLoS Neglected Tropical Diseases* **9**, e0003597.
- 49. Chowdhury R, Chowdhury V, Faria S, Akter S, Dash AP, Bhattacharya SK, Maheswary
NP, Bern C, Akhter S, Alvar J et al.. 2019 Effect of insecticide-treated bed nets on visceral
leishmaniasis incidence in Bangladesh. A retrospective cohort analysis. *PLoS Neglected Tropical*
*Diseases* **13**, e0007724.
- 50. Rock KS, le Rutte EA, de Vlas SJ, Adams ER, Medley GF, Hollingsworth TD. 2015 Uniting
mathematics and biology for control of visceral leishmaniasis. *Trends in Parasitology* **31**, 251–
259.

- 51. Hirve S, Boelaert M, Matlashewski G, Mondal D, Arana B, Kroeger A, Olliaro P. 2016
Transmission dynamics of visceral leishmaniasis in the Indian subcontinent—a systematic
literature review. *PLoS Neglected Tropical Diseases* **10**, e0004896.
- 52. DebRoy S, Prosper O, Mishoe A, Mubayi A. 2017 Challenges in modeling complexity of
neglected tropical diseases: a review of dynamics of visceral leishmaniasis in resource limited
settings. *Emerging Themes in Epidemiology* **14**, 10.
- 53. Bi K, Chen Y, Zhao S, Kuang Y, John Wu CH. 2018 Current visceral leishmaniasis research: a
research review to inspire future study. *BioMed Research International* **2018**.
- 54. Stauch A, Duerr HP, Picado A, Ostyn B, Sundar S, Rijal S, Boelaert M, Dujardin JC, Eichner
906 M. 2014 Model-based investigations of different vector-related intervention strategies to
907 eliminate visceral leishmaniasis on the Indian subcontinent. *PLoS Neglected Tropical Diseases*
**8**, e2810.
- 55. Chapman LA, Dyson L, Courtenay O, Chowdhury R, Bern C, Medley GF, Hollingsworth
TD. 2015 Quantification of the natural history of visceral leishmaniasis and consequences for
control. *Parasites and Vectors* **8**, 521.
- 56. Poche DM, Grant WE, Wang HH. 2016 Visceral leishmaniasis on the Indian subcontinent:
modelling the dynamic relationship between vector control schemes and vector life cycles.
*PLoS Neglected Tropical Diseases* **10**.
- 57. Chapman LA, Spencer SE, Pollington TM, Jewell CP, Mondal D, Alvar J, Hollingsworth
TD, Cameron MM, Bern C, Medley GF. 2020 Inferring transmission trees to guide targeting
of interventions against visceral leishmaniasis and post-kala-azar dermal leishmaniasis.
*Proceedings of the National Academy of Sciences* **117**, 25742–25750.
- 58. Stauch A, Sarkar RR, Picado A, Ostyn B, Sundar S, Rijal S, Boelaert M, Dujardin JC, Duerr HP.
2011 Visceral leishmaniasis in the Indian subcontinent: modelling epidemiology and control.
*PLoS Neglected Tropical Diseases* **5**.
- 59. Le Rutte EA, Coffeng LE, Bontje DM, Hasker EC, Postigo JAR, Argaw D, Boelaert MC,
De Vlas SJ. 2016 Feasibility of eliminating visceral leishmaniasis from the Indian subcontinent:
explorations with a set of deterministic age-structured transmission models. *Parasites and*
*Vectors* **9**, 24.
- 60. Meheus F, Balasegaram M, Olliaro P, Sundar S, Rijal S, Faiz MA, Boelaert M. 2010 Cost-
effectiveness analysis of combination therapies for visceral leishmaniasis in the Indian
subcontinent. *PLoS Neglected Tropical Diseases* **4**, e818.
- 61. Stauch A, Duerr HP, Dujardin JC, Vanaerschot M, Sundar S, Eichner M. 2012 Treatment of
visceral leishmaniasis: model-based analyses on the spread of antimony-resistant *L. donovani*
in Bihar, India. *PLoS Neglected Tropical Diseases* **6**, e1973.
- 62. Burattini MN, Coutinho FA, Lopez LF, Massad E. 1998 Modelling the dynamics of
leishmaniasis considering human, animal host and vector populations. *Journal of Biological*
*Systems* **6**, 337–356.
- 63. Hussaini N, Okuneye K, Gumel AB. 2017 Mathematical analysis of a model for zoonotic
visceral leishmaniasis. *Infectious Disease Modelling* **2**, 455–474.
- 64. Rabi Das VN, Bimal S, Siddiqui NA, Kumar A, Pandey K, Sinha SK, Topno RK, Mahentesh
938 V, Singh AK, Lal CS. 2020 Conversion of asymptomatic infection to symptomatic visceral
leishmaniasis: A study of possible immunological markers. *PLoS Neglected Tropical Diseases*
**14**, e0008272.
- 65. Manson-Bahr P, Heisch R, Garnham P et al.. 1959 Studies in Leishmaniasis in East Africa.
IV. The Montenegro Test in Kala-Azar in Kenya.. *Transactions of the Royal Society of Tropical*
*Medicine and Hygiene* **53**, 380–83.
- 66. Bern C, Amann J, Haque R, Chowdhury R, Ali M, Kurkjian KM, Vaz L, Wagatsuma Y, Breiman
RF, Secor WE et al.. 2006 Loss of leishmanin skin test antigen sensitivity and potency in a
longitudinal study of visceral leishmaniasis in Bangladesh. *The American Journal of Tropical*
*Medicine and Hygiene* **75**, 744–748.
- 67. Molina R, Jiménez M, García-Martínez J, San Martín JV, Carrillo E, Sánchez C, Moreno J,
Alves F, Alvar J. 2020 Role of asymptomatic and symptomatic humans as reservoirs of visceral
leishmaniasis in a Mediterranean context. *PLOS Neglected Tropical Diseases* **14**, e0008253.

68. Bauch CT, Earn DJ. 2004 Vaccination and the theory of games. *Proceedings of the National Academy of Sciences* **101**, 13391–13394.
69. Maskin E. 1999 Nash equilibrium and welfare optimality. *The Review of Economic Studies* **66**, 23–38.
70. Chang SL, Piraveenan M, Pattison P, Prokopenko M. 2020 Game theoretic modelling of infectious disease dynamics and intervention methods: a review. *Journal of Biological Dynamics* **14**, 57–89.
71. Ibuka Y, Li M, Vietri J, Chapman GB, Galvani AP. 2014 Free-riding behavior in vaccination decisions: an experimental study. *PLoS One* **9**.
72. Crawford K, Lancaster A, Oh H, Rychtář J. 2015 A voluntary use of insecticide-treated cattle can eliminate African sleeping sickness. *Letters in Biomathematics* **2**, 91–101.
73. Klein SRM, Foster AO, Feagins DA, Rowell JT, Erovenko IV. 2020 Optimal voluntary and mandatory insect repellent usage and emigration strategies to control the chikungunya outbreak on Reunion Island. *PeerJ* **8**, e10151.
74. Kobe J, Pritchard N, Short Z, Erovenko IV, Rychtář J, Rowell JT. 2018 A Game-Theoretic Model of Cholera with Optimal Personal Protection Strategies. *Bulletin of Mathematical Biology* **80**, 2580–2599.
75. Dorsett C, Oh H, Paulemond ML, Rychtář J. 2016 Optimal repellent usage to combat dengue fever. *Bulletin of Mathematical Biology* **78**, 916–922.
76. Brettin A, Rossi-Goldthorpe R, Weishaar K, Erovenko IV. 2018 Ebola could be eradicated through voluntary vaccination. *Royal Society Open Science* **5**, 171591.
77. Chouhan A, Maiwand S, Ngo M, Putalapattu V, Rychtář J, Taylor D. 2020 Game-theoretical model of retroactive Hepatitis B vaccination in China. *Bulletin of Mathematical Biology* **82**, 80.
78. Scheckelhoff K, Ejaz A, Erovenko IV. 2019 A game-theoretic model of optimal clean equipment usage to prevent hepatitis C among injecting drug users. *Preprint*.
79. Martinez A, Machado J, Sanchez E, Erovenko IV. 2019 Optimal vaccination strategies to reduce endemic levels of meningitis in Africa. *Preprint*.
80. Bankuru SV, Kossol S, Hou W, Mahmoudi P, Rychtář J, Taylor D. 2020 A Game-theoretic Model of Monkeypox to Assess Vaccination Strategies. *PeerJ* **8**, e9272.
81. Cheng E, Gambhirrao N, Patel R, Zhouandai A, Rychtář J, Taylor D. 2020 A game-theoretical analysis of Poliomyelitis vaccination. *Journal of Theoretical Biology* **499**, 110298.
82. Sykes D, Rychtář J. 2015 A game-theoretic approach to valuating toxoplasmosis vaccination strategies. *Theoretical Population Biology* **105**, 33–38.
83. Acosta-Alonzo CB, Erovenko IV, Lancaster A, Oh H, Rychtář J, Taylor D. 2020 High endemic levels of typhoid fever in rural areas of Ghana may stem from optimal voluntary vaccination behavior. *Proceedings of Royal Society A* p. 20200354.
84. Verelst F, Willem L, Beutels P. 2016 Behavioural change models for infectious disease transmission: a systematic review (2010–2015). *Journal of The Royal Society Interface* **13**, 20160820.
85. Han CY, Issa H, Rychtář J, Taylor D, Umana N. 2020 A voluntary use of insecticide treated nets can stop the vector transmission of Chagas disease. *PLoS Neglected Tropical Diseases* **14**, e0008833.
86. Behrend MR, Basáñez MG, Hamley JI, Porco TC, Stolk WA, Walker M, de Vlas SJ, Consortium NM. 2020 Modelling for policy: the five principles of the Neglected Tropical Diseases Modelling Consortium. *PLoS Neglected Tropical Diseases* **14**, e0008033.
87. Das ML, Rowland M, Austin JW, De Lazzari E, Picado A. 2014 Do size and insecticide treatment matter? Evaluation of different nets against *Phlebotomus argentipes*, the vector of visceral leishmaniasis in Nepal. *PLoS One* **9**, e114915.
88. den Boer ML, Alvar J, Davidson RN, Ritmeijer K, Balasegaram M. 2009 Developments in the treatment of visceral leishmaniasis. *Expert opinion on emerging drugs* **14**, 395–410.
89. Dinesh DS, Das P, Picado A, Davies C, Speybroeck N, Ostyn B, Boelaert M, Coosemans M. 2008 Long-lasting insecticidal nets fail at household level to reduce abundance of sandfly vector *Phlebotomus argentipes* in treated houses in Bihar (India). *Tropical Medicine & International Health* **13**, 953–958.

- 90. de Souza Leal MMC, Ovallos FG, de Castro Gomes CM, de Oliveira Lavitschka C, Galati EAB.
2014 Host-biting rate and susceptibility of some suspected vectors to *Leishmania braziliensis*.
*Parasites and Vectors* **7**, 1–11.
- 91. Hartemink N, Vanwambeke SO, Heesterbeek H, Rogers D, Morley D, Pesson B, Davies C,
Mahamdallie S, Ready P. 2011 Integrated mapping of establishment risk for emerging vector-
borne infections: a case study of canine leishmaniasis in southwest France. *PLoS One* **6**, e20817.
- 92. Sacks DL, Perkins PV. 1985 Development of Infective Stage *Leishmania* Promastigotes within
Phlebotomine Sand Flies. *The American Journal of Tropical Medicine and Hygiene* **34**, 456–459.
- 93. Srinivasan R, Panicker K. 1993 Laboratory observations on the biology of the phlebotomid
sandfly, *Phlebotomus papatasi* (Scopoli, 1786). *Southeast Asian Journal of Tropical Medicine and*
*Public Health* **24**, 536–536.
- 94. CIA. 2020 The World Factbook–South Asia: India. [https://www.cia.gov/library/
publications/resources/the-world-factbook/geos/in.html](https://www.cia.gov/library/publications/resources/the-world-factbook/geos/in.html). Accessed June 17,
2020.
- 95. Census India. 2016 Estimates of fertility indicators. [https://censusindia.gov.in/
vital_statistics/SRS_Report_2016/7.Chap_3-Fertility_Indicators-2016.
pdf](https://censusindia.gov.in/vital_statistics/SRS_Report_2016/7.Chap_3-Fertility_Indicators-2016.pdf). Accessed March 16, 2021.
- 96. World Bank. 2020 Life expectancy at birth. [https://data.worldbank.org/indicator/
SP.DYN.LE00.IN?cid=GPD_10](https://data.worldbank.org/indicator/SP.DYN.LE00.IN?cid=GPD_10). Accessed April 13, 2020.
- 97. Census India. 2015 Abridged life tables 2010-14. [https://www.censusindia.gov.in/
Vital_Statistics/SRS_Life_Table/2.Analysis_2010-14.pdf](https://www.censusindia.gov.in/Vital_Statistics/SRS_Life_Table/2.Analysis_2010-14.pdf). Accessed March
16, 2021.
- 98. Hailu A, Gramiccia M, Kager P. 2009 Visceral leishmaniasis in Aba-Roba, south-western
Ethiopia: prevalence and incidence of active and subclinical infections. *Annals of Tropical
Medicine & Parasitology* **103**, 659–670.
- 99. Kumar A, Saurabh S, Jamil S, Kumar V. 2020 Intensely clustered outbreak of visceral
leishmaniasis (kala-azar) in a setting of seasonal migration in a village of Bihar, India. *BMC
Infectious Diseases* **20**, 10.
- 100. Ostyn B, Gidwani K, Khanal B, Picado A, Chappuis F, Singh SP, Rijal S, Sundar S, Boelaert
1034 M. 2011 Incidence of symptomatic and asymptomatic *Leishmania donovani* infections in high-
1035 endemic foci in India and Nepal: a prospective study. *PLoS Neglected Tropical Diseases* **5**, e1284.
- 101. Singh OP, Tiwary P, Kushwaha AK, Singh SK, Singh DK, Lawyer P, Rowton E, Chaubey
R, Singh AK, Rai TK et al.. 2021 Xenodiagnosis to evaluate the infectiousness of humans
to sandflies in an area endemic for visceral leishmaniasis in Bihar, India: a transmission-
dynamics study. *The Lancet Microbe* **2**, e23–e31.
- 102. Mondal D, Bern C, Ghosh D, Rashid M, Molina R, Chowdhury R, Nath R, Ghosh P, Chapman
LA, Alim A. 2019 Quantifying the infectiousness of post-kala-azar dermal leishmaniasis
toward sand flies. *Clinical infectious diseases* **69**, 251–258.
- 103. van Griensven J, Balasegaram M, Meheus F, Alvar J, Lynen L, Boelaert M. 2010 Combination
therapy for visceral leishmaniasis. *The Lancet Infectious Diseases* **10**, 184–194.
- 104. Sundar S, Singh A, Chakravarty J, Rai M. 2015 Efficacy and safety of miltefosine in treatment
of post-kala-azar dermal leishmaniasis. *The Scientific World Journal* **2015**.
- 105. Ghosh S, Das NK, Mukherjee S, Mukhopadhyay D, Barbhuiya JN, Hazra A, Chatterjee
1048 M. 2015 Inadequacy of 12-week miltefosine treatment for Indian post-kala-azar dermal
leishmaniasis. *The American Journal of Tropical Medicine and Hygiene* **93**, 767–769.
- 106. Jervis S, Chapman LA, Dwivedi S, Karthick M, Das A, Le Rutte EA, Courtenay O, Medley
GF, Banerjee I, Mahapatra T. 2017 Variations in visceral leishmaniasis burden, mortality and
the pathway to care within Bihar, India. *Parasites and Vectors* **10**, 601.
- 107. Sundar S, Arora R, Singh SP, Boelaert M, Varghese B. 2010 Household cost-of-illness of
visceral leishmaniasis in Bihar, India. *Tropical Medicine & International Health* **15**, 50–54.
- 108. Hasker E, Singh SP, Malaviya P, Singh RP, Shankar R, Boelaert M, Sundar S. 2010
Management of visceral leishmaniasis in rural primary health care services in Bihar, India.
*Tropical Medicine & International Health* **15**, 55–62.

- 109. Ozaki M, Islam S, Rahman KM, Rahman A, Luby SP, Bern C. 2011 Economic Consequences
of Post-Kala-Azar Dermal Leishmaniasis in a Rural Bangladeshi Community. *The American*
*Journal of Tropical Medicine and Hygiene* **85**, 528–534.
- 110. Odomos Naturals. 2020 Dabur Odomos Naturals Mosquito Repellent Spray.
[https://www.amazon.in/Odomos-Naturals-Mosquito-Repellent-Spray/](https://www.amazon.in/Odomos-Naturals-Mosquito-Repellent-Spray/dp/B01CJVG5TM/ref=sr_1_2?dchild=1&fpw=pantry&keywords=deet&qid=1593529178&s=pantry&sr=8-2&srs=9574332031)
[dp/B01CJVG5TM/ref=sr_1_2?dchild=1&fpw=pantry&keywords=deet&qid=](https://www.amazon.in/Odomos-Naturals-Mosquito-Repellent-Spray/dp/B01CJVG5TM/ref=sr_1_2?dchild=1&fpw=pantry&keywords=deet&qid=1593529178&s=pantry&sr=8-2&srs=9574332031)
[1593529178&s=pantry&sr=8-2&srs=9574332031](https://www.amazon.in/Odomos-Naturals-Mosquito-Repellent-Spray/dp/B01CJVG5TM/ref=sr_1_2?dchild=1&fpw=pantry&keywords=deet&qid=1593529178&s=pantry&sr=8-2&srs=9574332031). Accessed June 30, 2020.
- 111. Lucas TC, Pollington TM, Davis EL, Hollingsworth TD. 2020 Responsible modelling: Unit
testing for infectious disease epidemiology. *Epidemics* p. 100425.
- 112. Ready PD. 2008 Leishmania manipulates sandfly feeding to enhance its transmission. *Trends*
*in Parasitology* **24**, 151–153.
- 113. Rogers ME, Bates PA. 2007 Leishmania manipulation of sand fly feeding behavior results in
enhanced transmission. *PLoS Pathogens* **3**, e91.
- 114. Carmichael S, Powell B, Hoare T, Walrad PB, Pitchford JW. 2021 Variable bites and dynamic
populations; new insights in Leishmania transmission. *PLoS Neglected Tropical Diseases* **15**,
e0009033.
- 115. Schenkel K, Rijal S, Koirala S, Koirala S, Vanlerberghe V, Van der Stuyft P, Gramiccia M,
Boelaert M. 2006 Visceral leishmaniasis in southeastern Nepal: A cross-sectional survey on
*Leishmania donovani* infection and its risk factors. *Tropical Medicine and International Health* **11**,
1792–1799.
- 116. Valero NNH, Uriarte M. 2020 Environmental and socioeconomic risk factors associated with
visceral and cutaneous leishmaniasis: a systematic review. *Parasitology research* **119**, 365–384.
- 117. Nair M, Kumar P, Pandey S, Kazmi S, Moreto-Planas L, Ranjan A, Burza S. 2020 Quality of life
perceptions amongst patients co-infected with Visceral Leishmaniasis and HIV: A qualitative
study from Bihar, India. *PloS One* **15**, e0227911.
- 118. Singh A, Chakraborty S, Roy TK. 2008 Village size in India: How relevant is it in the context
of development?. *Asian Population Studies* **4**, 111–134.
- 119. Anderson RM, May RM. 1992 *Infectious diseases of humans: dynamics and control*. Oxford
university press.
- 120. Delamater PL, Street EJ, Leslie TF, Yang YT, Jacobsen KH. 2019 Complexity of the basic
reproduction number (R_0). *Emerging infectious diseases* **25**, 1.
- 121. van den Driessche P, Watmough J. 2002 Reproduction numbers and sub-threshold endemic
equilibria for compartmental models of disease transmission. *Mathematical Biosciences* **180**, 29–
48.
- 122. Wei HM, Li XZ, Martcheva M. 2008 An epidemic model of a vector-borne disease with direct
transmission and time delay. *Journal of Mathematical Analysis and Applications* **342**, 895–908.
- 123. Killick-Kendrick R, Rioux J. 2002 Mark-release-recapture of sand flies fed on leishmanial
dogs: the natural life-cycle of *Leishmania infantum* in *Phlebotomus ariasi*. *Parassitologia* **44**, 67–71.
- 124. Guilvard E, Wilkes T, Killick-Kendrick R, Rioux JA. 1980 Ecologie des Leishmanioses dans le
Sud de la France-15. Déroulement des cycles gonotrophiques chez *Phlebotomus ariasi* Tonnoir,
1921 et *Phlebotomus mascittii* Grassi, 1908 en Cévennes. Corollaire épidémiologique. *Annales de*
*Parasitologie Humaine et Comparée* **55**, 659–664.
- 125. Shimozako HJ, Wu J, Massad E. 2017 Mathematical modelling for Zoonotic Visceral
Leishmaniasis dynamics: a new analysis considering updated parameters and notified human
Brazilian data. *Infectious Disease Modelling* **2**, 143–160.
- 126. Le Rutte EA, Chapman LA, Coffeng LE, Jervis S, Hasker EC, Dwivedi S, Karthick M,
Das A, Mahapatra T, Chaudhuri I. 2017 Elimination of visceral leishmaniasis in the Indian
subcontinent: a comparison of predictions from three transmission models. *Epidemics* **18**,
67–80.
- 127. Das V, Siddiqui N, Verma R, Topno R, Singh D, Das S, Ranjan A, Pandey K, Kumar N, Das

[revised manuscript text omitted]
^* \left(1 - \frac{1}{\mathcal{R}_e(0)}\right)}{\beta i_F T_{\text{Cycle}} \frac{N_F^*}{\mathcal{R}_e(0)} + \frac{\mu_F}{\sigma_F} + 1} \tag{A 3}$$

attained for $p = 0$ and consequently (5.3) has a solution only if

$$C_{ITN} \leq \frac{\beta i_F I_F^*(0)}{\mu + \beta i_F I_F^*(0)} C_{VL}. \tag{A 4}$$

It follows from (5.3) that (when (A 4) is true)

$$I_F^* = \frac{\left(\frac{C_{ITN}}{C_{VL}}\right) \mu}{\beta i_F \left(1 - \frac{C_{ITN}}{C_{VL}}\right)}. \tag{A 5}$$

Thus, after algebraic manipulations of (A 2), we get that p_{NE} is given by

$$p_{NE} = 1 - \frac{1}{\mathcal{R}_e(0) \left[1 - \frac{I_F^*}{N_F^*} \left(\beta i_F T_{\text{Cycle}} \frac{N_F^*}{\mathcal{R}_e(0)} + \frac{\mu_F}{\sigma_F} + 1\right)\right]} \tag{A 6}$$

where I_F^* is given by (A 5).

D. Sensitivity Analysis

Table 5. The sensitivity index SI_y of a variable y on a parameter x was calculated as $\left(\frac{x}{y}\right) \cdot \left(\frac{\partial y}{\partial x}\right)$ [133]. The numbers were rounded to the three decimal places. Parameters are as specified in Tables 1, 2 and 3. The sensitivity index -0.5 means that a 1% increase of a parameter value x will result in the 0.5% decrease of the variable y .

Parameter	$SI_{p_{HI}}$	$SI_{(p_{HI}-p_{NE})}$
A	0.0846	-3.9272
μ^{-1}	0.0846	-5.0223
μ_F^{-1}	0.0620	-3.1936
i_F	0.0425	-1.9916
n_F	0.0424	-1.9688
i_D	0.0201	-0.8375
γ_P^{-1}	0.0192	-0.7946
γ_D^{-1}	0.0188	-0.8742
i_P	0.0148	-0.5655
f_S	0.0031	-1.7764
i_{UL}	0.0024	-0.1131
γ_{UL}^{-1}	0.0020	-0.0938
s_2	0.0018	-0.1683
i_S	0.0012	-0.0557
γ_S^{-1}	0.0011	0.0216
f_U	0.0005	-0.0460
μ_K^{-1}	0.0001	-0.0708
s_1	0.0001	-0.0692
s_4	0.0001	-0.0099
ρ_C^{-1}	0	0.6465
ρ_D^{-1}	0	0.1220
i_{T1}	0	0.0000
i_{T2}	0	0.0000
i_U	0	0.0000
T_1^{-1}	0	0.0014
T_2^{-1}	0	0.0001
T_3^{-1}	0	0.0003
d_{T2}	0	0.0025
C_{ITN}	0	1.8081
$C_{I_{FA}}$	0	-1.0668
d_{T1}	-0.0001	0.0455
ρ_U^{-1}	-0.0001	0.0069
ρ_{TL}	-0.0024	-0.0040
σ_F^{-1}	-0.0111	0.5252
β^{-1}	-0.0849	3.8384

Figure 9. Dependence of p_{HI} on different parameter values. Unless varied, the parameter values are as specified in Tables 1, 2, 3. For those parameters, $p_{HI} = 0.95963$, $p_{NE} = 0.95956$.

Table 6. Policy-Relevant Items for Reporting Models in Epidemiology of Neglected Tropical Diseases (PRIME-NTD) Summary Table as specified in [86].

Principle	What has been done to satisfy the principle?	Where in the manuscript is this described?
1. Stakeholder engagement	We did not directly engage the stakeholder. However, we based our model on [58] and extensive VL literature describing KalaNet trial.	Sections 1 and 2
2. Complete model documentation	We provided a detailed model description.	Section 2.
	Implemented the model numerically in Matlab with ample comments.	Supplementary information.
3. Complete description of data used	We calibrated our model on data available in the literature.	Section 4
	We detailed procedures of how we extracted parameter values that were not found directly.	Section 4
	We derived VL transmission probabilities.	Section (a) (a)
4. Communicating uncertainty	We performed sensitivity analysis based on [133].	Section (a) (e), Table 5, Figures 8 and 9 and the Matlab code
5. Testable model outcomes	We predicted that asymptomatic individuals (PCR+, DAT+) can transmit VL and their infectiousness is at about 50% of the level of untreated symptomatic individuals.	Section (a) (a) and (b)
	We gave formula for the minimal ITN coverage needed to achieve complete elimination of VL and predicted that 96% ITN coverage is needed.	Section (a) (c)
	We also predicted that voluntary use of ITN should yield elimination of VL as public concern.	Section (a) (d) with mathematical details in Appendix C

Appendix D

Dear Editor & Authors,

I recommend that this article be accepted pending minor corrections. I won't be needed to do a third review. The Authors constructively engaged in the peer review process and I am satisfied with their responses to my comments; some of my previous comments were valid and some of them not, and that is fine. I understand that the first three authors are undergraduates and they should feel very proud of this manuscript which is of a research standard in my opinion. It is also a credit to their supervisors who supported them in this endeavour.

This is a very rigorous and careful work and they have clearly been up-front and honest about its limitations. Due to the state they leave their code and explain their methods, will encourage others to build-on these analyses, which is only going to be good for the science.

I encourage the Editors to reach out to VL experts (from a policy level and working in vector control programmes/sandfly entomology) to write a commentary on this piece. A key question now for the field is *does this work change the current thinking on vector control applied to Bihar? If not, why not? Is there something lacking in this work and what further research could remedy it? Or are there reasons that would make it infeasible?* Possible contacts who may be interested are:

- SPEAK India partnership
- NVBDCP
- WHO SEARO
- VCRC

When I quote line numbers I refer to the original ones introduced by the authors rather than the additional added by RSOS that repeat on a pagewise basis. When I do refer to page numbers I refer to the author's response PDF that is the first 49 pages of 97.

Please note the other attached PDF(RSOS-201960.R1_Proof_hi_TMP) includes the majority of my response.

Timothy M Pollington.

Appendix E

Responses to the reviewers' comments

In response to several comments by reviewer 2, we made several structural changes of the manuscript. Most notably we

- Added subsection headings (and slightly reshuffled the text) to the introduction
- Renamed original "Discussion" section into "Conclusion"
- Added a discussion section. The bulk of the section contains the original discussion of the model but we added several other parts based on reviewers' comments.

We also did one thorough read through and corrected minor typos and misalignments.

Specific responses:

Reviewer 1 wrote:

Please consider commenting further the (in)stability of the system equilibria in the Discussion section. In lines 292-296, the authors mentioned explicitly the relation between the effective reproduction number and the endemic and disease-free states. While this concept may be well-known for the modeler readers, the remainder may be appreciated by the readers with less modeling background.

Our response: We added the following italic sentence to the first paragraph of the Conclusion section:
... ITN usage would have to be about 96% or more. *At such an ITN coverage, the effective reproduction number is less than 1 and the disease-free equilibrium is stable.*

Reviewer 1 wrote:

However, I believe it still remains to be discussed how the stability of the system changes when the game interacts with the ODE system.

In that regard, the authors may consider discussing further sections 5c and 5d, and how their results interact. The game being coupled to the ODE system, the ITN usage becomes a function of the other parameters, notably the costs. However, the authors made the modeling choice of working with fixed costs to discuss the current epidemic situation. Hence, the outcomes of the model have not been read from a broader perspective of the dynamical system. This is not a problem, provided that is carefully discussed and mentioned as a limitation of the study.

Our response: We added the following to the discussion:

For simplicity, we focused our analysis on equilibria of the ODE system. This approach had several limitations and disadvantages. As already mentioned above, it precludes us from incorporating seasonal sand fly dynamics. Similarly, the model cannot not properly capture dynamical changes in pricing or availability of ITNs or drugs as was recently happening due to COVID-19 \citep{weiss2021indirect}. Most importantly, to understand and model the final stages of VL elimination, one should consider not only the equilibria but also how long it takes to reach an equilibrium. One also needs to explicitly model the interactions of the potential dynamical ITNs coverage and the disease dynamics.

This new text is now followed by "Unlike vaccines that offer a long lasting protection after just one decision ..." and figures showing the dynamical interplay between the ODEs and the vaccination coverage that was originally buried deep in the middle of the manuscript. We believe that together, the part with the original, address the reviewer's point.

Reviewer 1 wrote:

For instance, the behavior around $R_e = 1$ and beyond (in the sense of $p > p_{NE}$) might be important to interpret to discuss the study results, and might also shed light into the results of the

KalaNet trial. Indeed, the closer you get to $R_e=1$, the longer it takes to get to the equilibrium (and so, the harder it takes to reach elimination), which seems to be the situation described in lines 584 and 585.

Our response: We tried to address it together with the previous point and included “, *one should consider not only the equilibria but also how long it takes to reach an equilibrium.* “ in the discussion.

Reviewer 1 wrote:

Moreover, the disease-free state (reached when $p > p_{NE}$) may lose its stability when the ODE system is coupled to the game, which may impact the interpretation of elimination and thus, the discussion around elimination.

Our response: We agree and the Figure 6 illustrates this (the figure was originally buried in the manuscript but it is indeed better placed in the discussion section)

Reviewer 1 wrote:

In agreement with comments from Reviewer 2, ensuring that the difference between elimination (as in $R_e < 1$) and elimination as a public health concern is clear is a priority; as well as ensuring that the purpose of the study is reflected in the title, the abstract and the discussion.

Our response: We carefully reviewed the manuscript and believe that we now make the distinction in all places.

Reviewer 2:

We had hard time extracting the comments from the pdf but we believe we addressed the points the reviewer raised.

Most importantly, we

- added few sentences to the intro about KalaNet, and that it led to the end of use of ITNs.
- Added subsections to the introduction
- Cleared the confusion between patients and cases by calling patients only those individuals that are receiving treatment
- Made slight update to Figure 1 as suggested
- Moved the “limitation of the model “ into the discussion section
- Put USD (and replaced \$ by USD if applicable) to all costs

Appendix F

Dear editor,

Thank you for your encouraging decision and for the opportunity to do one more round of minor revisions.

We have addressed the issues raised by the reviewers. Specifics are provided in the separate file.

Sincerely,

Jan Rychtar